# OKRidge: Scalable Optimal k-Sparse Ridge Regression

**Jiachang Liu, Sam Rosen, Chudi Zhong, Cynthia Rudin**
Duke University
{jiachang.liu, sam.rosen, chudi.zhong}@duke.edu,
cynthia@cs.duke.edu

## Abstract

We consider an important problem in scientific discovery, namely identifying sparse governing equations for nonlinear dynamical systems. This involves solving sparse ridge regression problems to provable optimality in order to determine which terms drive the underlying dynamics. We propose a fast algorithm, OKRidge, for sparse ridge regression, using a novel lower bound calculation involving, first, a saddle point formulation, and from there, either solving (i) a linear system or (ii) using an ADMM-based approach, where the proximal operators can be efficiently evaluated by solving another linear system and an isotonic regression problem. We also propose a method to warm-start our solver, which leverages a beam search. Experimentally, our methods attain provable optimality with run times that are orders of magnitude faster than those of the existing MIP formulations solved by the commercial solver Gurobi.

## 1 Introduction

We are interested in identifying sparse and interpretable governing differential equations arising from nonlinear dynamical systems. These are scientific machine learning problems whose solution involves sparse linear regression. Specifically, these problems require the *exact* solution of sparse regression problems, with the most basic being sparse ridge regression:

$$\min_{\boldsymbol{\beta}} \|\boldsymbol{y} - \boldsymbol{X}\boldsymbol{\beta}\|_2^2 + \lambda_2 \|\boldsymbol{\beta}\|_2^2 \text{ subject to } \|\boldsymbol{\beta}\|_0 \leq k, \tag{1}$$

where $k$ specifies the number of nonzero coefficients for the model. This formulation is general, but in the case of nonlinear dynamical systems, the outcome $y$ is a derivative (usually time or spatial) of each dimension $x$. Here, we assume that the practitioner has included the true variables, along with many other possibilities, and is looking to determine which terms (which transformations of the variables) are real and which are not. This problem is NP-hard [49], and is more challenging in the presence of highly correlated features. Selection of correct features is vital in this context, as many solutions may give good results on training data, but will quickly deviate from the true dynamics when extrapolating past the observed data due to the chaotic nature of complex dynamical systems.

Both heuristic and optimal algorithms have been proposed to solve these problems. Heuristic methods include greedy sequential adding of features [25, 16, 54, 22] or ensemble [31] methods. These methods are fast, but often get stuck in local minima, and there is no way to assess solution quality due to the lack of a lower bound on performance. Optimal methods provide an alternative, but are slow since they must prove optimality. MIOSR [9], a mixed-integer programming (MIP) approach, has been able to certify optimality of solutions given enough time. Slow solvers cause difficulty in performing cross-validation on $\lambda_2$ (the $\ell_2$ regularization coefficient) and $k$ (sparsity level).

We aim to solve sparse ridge regression to certifiable optimality, but in a fraction of the run time. We present a fast branch-and-bound (BnB) formulation, OKRidge. A crucial challenge is obtaining a

37th Conference on Neural Information Processing Systems (NeurIPS 2023).

tight and feasible lower bound for each node in the BnB tree. It is possible to calculate the lower bound via the SOS1 [9], big-M [10], or the perspective formulations (also known as the rotated second-order cone constraints) [32, 4, 59]; the mixed-integer problems can then be solved by a MIP solver. However, these formulations do not consider the special mathematical structure of the regression problem. To calculate a lower bound more efficiently, we first propose a new saddle point formulation for the relaxed sparse ridge regression problem. Based on the new saddle point formulation, we propose two novel methods to calculate the lower bound. The first method is extremely efficient and relies on solving only a linear system of equations. The second method is based on ADMM and can tighten the lower bound given by the first method. Together, these methods give us a tight lower bound, used to prune nodes and provide a small optimality gap. Additionally, we propose a method based on beam-search [58] to get a near-optimal solution quickly, which can be a starting point for both our algorithm and other MIP formulations. Unlike previous methods, our method uses a dynamic programming approach so that previous solutions in the BnB tree can be used while exploring the current node, giving a massive speedup. In summary, our contributions are:

(1) We develop a highly efficient customized branch-and-bound framework for achieving optimality in $k$-sparse ridge regression, using a novel lower bound calculation and heuristic search.
(2) To compute the lower bound, we introduce a new saddle point formulation, from which we derive two efficient methods (one based on solving a linear system and the other on ADMM).
(3) Our warm-start method is based on beam-search and implemented in a dynamic programming fashion, avoiding redundant calculations. We prove that our warm-start method is an approximation algorithm with an exponential factor tighter than previous work.

On benchmarks, OKRidge certifies optimality orders of magnitude faster than the commercial solver Gurobi. For dynamical systems, our method outperforms the state-of-the-art certifiable method by finding superior solutions, particularly in high-dimensional feature spaces.

## 2   Preliminary: Dual Formulation via the Perspective Function

There is an extensive literature on this topic, and a longer review of related work is in Appendix A. If we ignore the constant term $\boldsymbol{y}^T \boldsymbol{y}$, we can rewrite the loss objective in Equation (1) as:

$$\mathcal{L}_{\text{ridge}}(\boldsymbol{\beta}) := \boldsymbol{\beta}^T \boldsymbol{X}^T \boldsymbol{X} \boldsymbol{\beta} - 2\boldsymbol{y}^T \boldsymbol{X} \boldsymbol{\beta} + \lambda_2 \sum_{j=1}^{p} \beta_j^2, \tag{2}$$

with $p$ as the number of features. We are interested in the following optimization problem:

$$\min_{\boldsymbol{\beta}} \mathcal{L}_{\text{ridge}}(\boldsymbol{\beta}) \text{ s.t. } (1 - z_j)\beta_j = 0, \ \sum_{j=1}^{p} z_j \le k, \ z_j \in \{0, 1\}, \tag{3}$$

where $k$ is the number of nonzero coefficients. With the sparsity constraint, the problem is NP-hard. The constraint $(1-z_j)\beta_j$ in Problem (3) can be reformulated with the SOS1, big-M, or the perspective formulation (with quadratic cone constraints), which then can be solved by a MIP solver. Since commercial solvers do not exploit the special structure of the problem, we develop a customized branch-and-bound framework.

For any function $f(a)$, the perspective function is $g(a, b) := bf(\frac{a}{b})$ for the domain $b > 0$ [32, 34, 26] and $g(a, b) = 0$ otherwise. Applying to $f(a) = a^2$, we obtain another function $g(a, b) = \frac{a^2}{b}$. As shown by [4], replacing the loss term $\beta_j^2$ and constraint $(1 - z_j)\beta_j = 0$ with the perspective formula $\beta_j^2/z_j$ in Problem (3) would not change the optimal solution. By the Fenchel conjugate [4], $g(\cdot, \cdot)$ can be rewritten as $g(a, b) = \max_c ac - \frac{c^2}{4}b$. If we define a new perspective loss as:

$$\mathcal{L}_{\text{ridge}}^{\text{Fenchel}}(\boldsymbol{\beta}, \boldsymbol{z}, \boldsymbol{c}) := \boldsymbol{\beta}^T \boldsymbol{X}^T \boldsymbol{X} \boldsymbol{\beta} - 2\boldsymbol{y}^T \boldsymbol{X} \boldsymbol{\beta} + \lambda_2 \sum_{j=1}^{p} \left( \beta_j c_j - \frac{c_j^2}{4} z_j \right), \tag{4}$$

then we can reformulate Problem (3) as:

$$\min_{\boldsymbol{\beta}, \boldsymbol{z}} \max_{\boldsymbol{c}} \mathcal{L}_{\text{ridge}}^{\text{Fenchel}}(\boldsymbol{\beta}, \boldsymbol{z}, \boldsymbol{c}) \text{ s.t. } \sum_{j=1}^{p} z_j \le k, \ z_j \in \{0, 1\}. \tag{5}$$

If we relax the binary constraint $\{0, 1\}$ to the interval $[0, 1]$ and swap $\max$ and $\min$ (no duality gap, as pointed by [4]), we obtain the dual formulation for the convex relaxation of Problem (5):

$$\max_{\boldsymbol{c}} \min_{\boldsymbol{\beta}, \boldsymbol{z}} \mathcal{L}_{\text{ridge}}^{\text{Fenchel}}(\boldsymbol{\beta}, \boldsymbol{z}, \boldsymbol{c}) \text{ s.t. } \sum_{j=1}^{p} z_j \leq k, \ z_j \in [0, 1]. \tag{6}$$

While [4] uses the perspective formulation for safe feature screening, we use it to calculate a lower bound for Problem (3). However, directly solving the maxmin problem is computationally challenging. In Section 3.1, we propose two methods to achieve this in an efficient way.

## 3 Methodology

We propose a custom BnB framework to solve Problem (3). We have 3 steps to process each node in the BnB tree. First, we calculate a lower bound of the node, using two algorithms proposed in the next subsection. If the lower bound exceeds or equals the current best solution, we have proven that it does not lead to any optimal solution, so we prune the node. Otherwise, we go to Step 2, where we perform beam-search to find a near-optimal solution. In Step 3, we use the solution from Step 2 and propose a branching strategy to create new nodes in the BnB tree. We continue until reaching the optimality gap tolerance. Below, we elaborate on each step. In Appendix E, we provide visual illustrations of BnB and beam search as well as the complete pseudocodes of our algorithms.

### 3.1 Lower Bound Calculation

**Tight Saddle Point Formulation**

We first rewrite Equation (2) with a new hyperparameter $\lambda$:

$$\mathcal{L}_{\text{ridge}-\lambda}(\boldsymbol{\beta}, \boldsymbol{z}) := \boldsymbol{\beta}^T \boldsymbol{Q}_\lambda \boldsymbol{\beta} - 2\boldsymbol{y}^T \boldsymbol{X} \boldsymbol{\beta} + (\lambda_2 + \lambda) \sum_{j=1}^{p} \beta_j^2, \tag{7}$$

where $\boldsymbol{Q}_\lambda := \boldsymbol{X}^T \boldsymbol{X} - \lambda \boldsymbol{I}$. We restrict $\lambda \in [0, \lambda_{\min}(\boldsymbol{X}^T \boldsymbol{X})]$, where $\lambda_{\min}(\cdot)$ denotes the minimum eigenvalue of a matrix. We see that $\boldsymbol{Q}_\lambda$ is positive semidefinite, so the first term remains convex. This trick is related to the optimal perspective formulation [62, 28, 37], but we set the diagonal matrix $\text{diag}(\boldsymbol{d})$ in [28] to be $\lambda \boldsymbol{I}$. We call this trick the eigen-perspective formulation. The optimal perspective formulation requires solving semidefinite programming (SDP) problems, which have been shown not scalable to high dimensions [28], and MI-SDP is not supported by Gurobi.

Solving Problem (3) is equivalent to solving the following problem:

$$\min_{\boldsymbol{\beta}, \boldsymbol{z}} \mathcal{L}_{\text{ridge}-\lambda}(\boldsymbol{\beta}, \boldsymbol{z}) \text{ s.t. } (1 - z_j)\beta_j = 0, \ \sum_{j=1}^{p} z_j \leq k, \ z_j \in \{0, 1\}. \tag{8}$$

We get a continuous relaxation of Problem (3) if we relax $\{0, 1\}$ to $[0, 1]$.

We can now define a new loss analogous to the loss defined in Equation (4):

$$\mathcal{L}_{\text{ridge}-\lambda}^{\text{Fenchel}}(\boldsymbol{\beta}, \boldsymbol{z}, \boldsymbol{c}) := \boldsymbol{\beta}^T \boldsymbol{Q}_\lambda \boldsymbol{\beta} - 2\boldsymbol{y}^T \boldsymbol{X} \boldsymbol{\beta} + (\lambda_2 + \lambda) \sum_{j=1}^{p} (\beta_j c_j - \frac{c_j^2}{4} z_j). \tag{9}$$

Then, the dual formulation analogous to Problem (6) is:

$$\max_{\boldsymbol{c}} \min_{\boldsymbol{\beta}, \boldsymbol{z}} \mathcal{L}_{\text{ridge}-\lambda}^{\text{Fenchel}}(\boldsymbol{\beta}, \boldsymbol{z}, \boldsymbol{c}) \text{ s.t. } \sum_{j=1}^{p} z_j \leq k, \ z_j \in [0, 1]. \tag{10}$$

Solving Problem (10) provides us with a lower bound to Problem (8). More importantly, this lower bound becomes tighter as $\lambda$ increases. This novel formulation is the starting point for our work.

We next propose a reparametrization trick to simplify the optimization problem above. For the inner optimization problem in Problem (10), given any $\boldsymbol{c}$, the optimality condition for $\boldsymbol{\beta}$ is (take the gradient with respect to $\boldsymbol{\beta}$ and set the gradient to $\boldsymbol{0}$):

$$\boldsymbol{c} = \frac{2}{\lambda_2 + \lambda}(\boldsymbol{X}^T \boldsymbol{y} - \boldsymbol{Q}_\lambda \boldsymbol{\beta}). \tag{11}$$

Inspired by this optimality condition, we have the following theorem:

**Theorem 3.1.** *If we reparameterize $\boldsymbol{c} = \frac{2}{\lambda_2+\lambda}(\boldsymbol{X}^T\boldsymbol{y} - \boldsymbol{Q}_\lambda\boldsymbol{\gamma})$ with a new parameter $\boldsymbol{\gamma}$, then Problem (10) is equivalent to the following saddle point optimization problem:*

$$\max_{\boldsymbol{\gamma}} \min_{\boldsymbol{z}} \mathcal{L}^{saddle}_{ridge-\lambda}(\boldsymbol{\gamma}, \boldsymbol{z}) \ \ s.t. \ \sum_{j=1}^{p} z_j \leq k, \ \ z_j \in [0,1], \tag{12}$$

*where* $\quad \mathcal{L}^{saddle}_{ridge-\lambda}(\boldsymbol{\gamma}, \boldsymbol{z}) := -\boldsymbol{\gamma}^T\boldsymbol{Q}_\lambda\boldsymbol{\gamma} - \frac{1}{\lambda_2+\lambda}(\boldsymbol{X}^T\boldsymbol{y} - \boldsymbol{Q}_\lambda\boldsymbol{\gamma})^T diag(\boldsymbol{z})(\boldsymbol{X}^T\boldsymbol{y} - \boldsymbol{Q}_\lambda\boldsymbol{\gamma}), \tag{13}$

*and* $diag(\boldsymbol{z})$ *is a diagonal matrix with* $\boldsymbol{z}$ *on the diagonal.*

To our knowledge, this is the first time this formulation is given. Solving the saddle point formulation to optimality in Problem (12) gives us a tight lower bound. However, this is computationally hard.

Our insight is that we can solve Problem (12) approximately while still obtaining a feasible lower bound. Let us define a new function $h(\boldsymbol{\gamma})$ as short-hand for the inner minimization in Problem (12):

$$h(\boldsymbol{\gamma}) = \min_{\boldsymbol{z}} \mathcal{L}^{saddle}_{ridge-\lambda}(\boldsymbol{\gamma}, \boldsymbol{z}) \ \ \text{s.t.} \ \sum_{j=1}^{p} z_j \leq k, \ \ z_j \in [0,1]. \tag{14}$$

For any arbitrary $\boldsymbol{\gamma} \in \mathbb{R}^p$, $h(\boldsymbol{\gamma})$ is a valid lower bound for Problem (3). We should choose $\boldsymbol{\gamma}$ such that this lower bound $h(\boldsymbol{\gamma})$ is tight. Below, we provide two efficient methods to calculate such a $\boldsymbol{\gamma}$.

**Fast Lower Bound Calculation**

First, we provide a fast way to choose $\boldsymbol{\gamma}$. The choice of $\boldsymbol{\gamma}$ is motivated by the following theorem:

**Theorem 3.2.** *The function $h(\boldsymbol{\gamma})$ defined in Equation (14) is lower bounded by*

$$h(\boldsymbol{\gamma}) \geq -\boldsymbol{\gamma}^T\boldsymbol{Q}_\lambda\boldsymbol{\gamma} - \frac{1}{\lambda_2+\lambda}\|\boldsymbol{X}^T\boldsymbol{y} - \boldsymbol{Q}_\lambda\boldsymbol{\gamma}\|_2^2. \tag{15}$$

*Furthermore, the right-hand size of Equation (15) is maximized if $\boldsymbol{\gamma} = \hat{\boldsymbol{\gamma}} = \operatorname{argmin}_{\boldsymbol{\alpha}} \mathcal{L}_{\text{ridge}}(\boldsymbol{\alpha})$, where in this case, $h(\boldsymbol{\gamma})$ evaluated at $\hat{\boldsymbol{\gamma}}$ becomes*

$$h(\hat{\boldsymbol{\gamma}}) = \mathcal{L}_{\text{ridge}}(\hat{\boldsymbol{\gamma}}) + (\lambda_2+\lambda)SumBottom_{p-k}(\{\hat{\gamma}_j^2\}), \tag{16}$$

*where $SumBottom_{p-k}(\cdot)$ denotes the summation of the smallest $p-k$ terms of a given set.*

Here we provide an intuitive explanation of why $h(\hat{\boldsymbol{\gamma}})$ is a valid lower bound. Note that the ridge regression loss is strongly convex. Assuming that the strongly convex parameter is $\mu$ (see Appendix B), by the strong convexity property, we have that for any $\boldsymbol{\gamma} \in \mathbb{R}^p$,

$$\mathcal{L}_{\text{ridge}}(\boldsymbol{\gamma}) \geq \mathcal{L}_{\text{ridge}}(\hat{\boldsymbol{\gamma}}) + \nabla\mathcal{L}_{\text{ridge}}(\hat{\boldsymbol{\gamma}})^T(\boldsymbol{\gamma} - \hat{\boldsymbol{\gamma}}) + \frac{\mu}{2}\|\boldsymbol{\gamma} - \hat{\boldsymbol{\gamma}}\|_2^2. \tag{17}$$

Because $\hat{\boldsymbol{\gamma}}$ minimizes $\mathcal{L}_{\text{ridge}}(\cdot)$, we have $\nabla\mathcal{L}_{\text{ridge}}(\hat{\boldsymbol{\gamma}}) = \boldsymbol{0}$. For the $k$-sparse vector $\boldsymbol{\gamma}$ with $\|\boldsymbol{\gamma}\|_0 \leq k$, the minimum for the right-hand side of Inequality (17) can be achieved if $\gamma_j = \hat{\gamma}_j$ for the top $k$ terms of $\hat{\gamma}_l^2$'s. This ensures the bound applies for all $k$-sparse $\boldsymbol{\gamma}$. Thus, the $k$-sparse ridge regression loss is lower-bounded by

$$\mathcal{L}_{\text{ridge}}(\boldsymbol{\gamma}) \geq \mathcal{L}_{\text{ridge}}(\hat{\boldsymbol{\gamma}}) + \frac{\mu}{2}SumBottom_{p-k}(\{\hat{\gamma}_j^2\})$$

for $\boldsymbol{\gamma} \in \mathbb{R}^p$ with $\|\boldsymbol{\gamma}\|_0 \leq k$. For ridge regression, the strong convexity $\mu$ parameter can be chosen from $[0, 2(\lambda_2 + \lambda_{\min}(\boldsymbol{X}^T\boldsymbol{X}))]$. If we let $\mu = 2(\lambda_2 + \lambda)$, we obtain $h(\hat{\boldsymbol{\gamma}})$ in Theorem 3.2.

The lower bound $h(\hat{\boldsymbol{\gamma}})$ can be calculated extremely efficiently by solving the ridge regression problem (solving the linear system $(\boldsymbol{X}^T\boldsymbol{X} + \lambda_2\boldsymbol{I})\boldsymbol{\gamma} = \boldsymbol{X}^T\boldsymbol{y}$ for $\boldsymbol{\gamma}$) and adding the extra $p-k$ terms. However, this bound is not the tightest we can achieve. In the next subsection, we discuss how to apply ADMM to maximize $h(\boldsymbol{\gamma})$ further based on Equation (14).

**Tight Lower Bound via ADMM**

Let us define $\boldsymbol{p} := \boldsymbol{X}^T\boldsymbol{y} - \boldsymbol{Q}_\lambda\boldsymbol{\gamma}$. Starting from Problem (12), if we minimize $\boldsymbol{z}$ in the inner optimization under the constraints $\sum_{j=1}^{p} z_j \leq k$ and $z_j \in [0, 1]$ for $\forall j$, we have $z_j = 1$ for the top $k$ terms of $p_j^2$ and $z_j = 0$ otherwise. Then, Problem (12) can be reformulated as follows:

$$-\min_{\boldsymbol{\gamma}} \left(F(\boldsymbol{\gamma}) + G(\boldsymbol{p})\right) \;\; \text{s.t.} \;\; \boldsymbol{Q}_\lambda\boldsymbol{\gamma} + \boldsymbol{p} = \boldsymbol{X}^T\boldsymbol{y}, \tag{18}$$

where $F(\boldsymbol{\gamma}) := \boldsymbol{\gamma}^T\boldsymbol{Q}_\lambda\boldsymbol{\gamma}$ and $G(\boldsymbol{p}) := \frac{1}{\lambda_2+\lambda}SumTop_k(\{p_j^2\})$. The solution to this problem is a dense vector that can be used to provide a lower bound on the original $k$-sparse problem. This problem can be solved by the alternating direction method of multipliers (ADMM) [17]. Here, we apply the iterative algorithm with the scaled dual variable $\boldsymbol{q}$ [33]:

$$\boldsymbol{\gamma}^{t+1} = \operatorname*{argmin}_{\boldsymbol{\gamma}} F(\boldsymbol{\gamma}) + \frac{\rho}{2}\|\boldsymbol{Q}_\lambda\boldsymbol{\gamma} + \boldsymbol{p}^t - \boldsymbol{X}^T\boldsymbol{y} + \boldsymbol{q}^t\|_2^2 \tag{19}$$

$$\boldsymbol{\theta}^{t+1} = 2\alpha\boldsymbol{Q}_\lambda\boldsymbol{\gamma}^{t+1} - (1 - 2\alpha)(\boldsymbol{p}^t - \boldsymbol{X}^T\boldsymbol{y}) \tag{20}$$

$$\boldsymbol{p}^{t+1} = \operatorname*{argmin}_{\boldsymbol{p}} G(\boldsymbol{p}) + \frac{\rho}{2}\|\boldsymbol{\theta}^{t+1} + \boldsymbol{p} - \boldsymbol{X}^T\boldsymbol{y} + \boldsymbol{q}^t\|_2^2 \tag{21}$$

$$\boldsymbol{q}^{t+1} = \boldsymbol{q}^t + \boldsymbol{\theta}^{t+1} + \boldsymbol{p}^{t+1} - \boldsymbol{X}^T\boldsymbol{y}, \tag{22}$$

where $\alpha$ is the relaxation factor, and $\rho$ is the step size.

It is known that ADMM suffers from slow convergence when the step size is not properly chosen. According to [33], to ensure the optimized linear convergence rate bound factor, we can pick $\alpha = 1$ and $\rho = \frac{2}{\sqrt{\lambda_{\max}(\boldsymbol{Q}_\lambda)\lambda_{\min>0}(\boldsymbol{Q}_\lambda)}}$[1], where $\lambda_{\max}(\cdot)$ denotes the largest eigenvalue of a matrix, and $\lambda_{\min>0}(\cdot)$ denotes the smallest positive eigenvalue of a matrix.

Having settled the choices for the relaxation factor $\alpha$ and the step size $\rho$, we are left with the task of solving Equation (19) and Equation (21) (also known as evaluating the proximal operators [52]). Interestingly, Equation (19) can be evaluated by solving a linear system while Equation (21) can be evaluated by recasting the problem as an isotonic regression problem.

**Theorem 3.3.** *Let $F(\boldsymbol{\gamma}) = \boldsymbol{\gamma}^T\boldsymbol{Q}_\lambda\boldsymbol{\gamma}$ and $G(\boldsymbol{p}) = \frac{1}{\lambda_2+\lambda}SumTop_k(\{p_j^2\})$. Then the solution for the problem $\boldsymbol{\gamma}^{t+1} = \operatorname{argmin}_{\boldsymbol{\gamma}} F(\boldsymbol{\gamma}) + \frac{\rho}{2}\|\boldsymbol{Q}_\lambda\boldsymbol{\gamma} + \boldsymbol{p}^t - \boldsymbol{X}^T\boldsymbol{y} + \boldsymbol{q}^t\|_2^2$ is*

$$\boldsymbol{\gamma}^{t+1} = \left(\frac{2}{\rho}\boldsymbol{I} + \boldsymbol{Q}_\lambda\right)^{-1}\left(\boldsymbol{X}^T\boldsymbol{y} - \boldsymbol{p}^t - \boldsymbol{q}^t\right). \tag{23}$$

*Furthermore, let $\boldsymbol{a} = \boldsymbol{X}^T\boldsymbol{y} - \boldsymbol{\theta}^{t+1} - \boldsymbol{q}^t$ and $\mathcal{J}$ be the indices of the top $k$ terms of $\{|a_j|\}$. The solution for the problem $\boldsymbol{p}^{t+1} = \operatorname{argmin}_{\boldsymbol{p}} G(\boldsymbol{p}) + \frac{\rho}{2}\|\boldsymbol{\theta}^{t+1} + \boldsymbol{p} - \boldsymbol{X}^T\boldsymbol{y} + \boldsymbol{q}^t\|_2^2$ is $p_j^{t+1} = sign(a_j)\cdot\hat{v}_j$,*

*where*
$$\hat{\boldsymbol{v}} = \operatorname*{argmin}_{\boldsymbol{v}} \sum_{j=1}^{p} w_j(v_j - b_j)^2 \;\; s.t. \;\; v_i \geq v_l \;\; if \;\; |a_i| \geq |a_l| \tag{24}$$

$$w_j = \begin{cases} 1 & \text{if } j \notin \mathcal{J} \\ 1 + \frac{2}{\rho(\lambda_2+\lambda)} & \text{otherwise} \end{cases}, \qquad b_j = \frac{|a_j|}{w_j}.$$

*Problem (24) is an isotonic regression problem and can be efficiently solved in linear time [12, 21].*

## 3.2 Beam-Search as a Heuristic

After finishing the lower bound calculation in Section 3.1, we next explain how to quickly reduce the upper bound in the BnB tree. We discuss how to add features, keep good solutions, and use dynamic programming to improve efficiency. Lastly, we give a theoretical guarantee on the quality of our solution.

---

[1] [33] also considers matrix preconditioning when computing the step size, but this is computationally expensive when the number of features is large, so we ignore matrix rescaling by letting $\boldsymbol{E}$ be the identity matrix in Section VI Subsection A of [33].

Starting from the vector $\mathbf{0}$, we add one coordinate at a time into our support until we reach a solution with support size $k$. At each iteration, we pick the coordinate that results in the largest decrease in the ridge regression loss while keeping coefficients in the existing support fixed:

$$j^* \in \underset{j}{\arg\min} \ \underset{\alpha}{\min} \ \mathcal{L}_{\text{ridge}}(\boldsymbol{\beta} + \alpha \boldsymbol{e}_j) \quad \Longleftrightarrow \quad j^* \in \underset{j}{\arg\max} \ \frac{(\nabla_j \mathcal{L}_{\text{ridge}}(\boldsymbol{\beta}))^2}{\|\boldsymbol{X}_{:j}\|_2^2 + \lambda_2}, \quad (25)$$

where $\boldsymbol{X}_{:j}$ denotes the $j$-th column of $\boldsymbol{X}$, and the right-hand side uses an analytical solution for the line-search for $\alpha$. This is similar to the sparse-simplex algorithm [6]. However, after adding a feature, we adjust the coefficients restricted on the new support by minimizing the ridge regression loss.

The above idea does not handle highly correlated features well. Once a feature is added, it cannot be removed [61]. To alleviate this problem, we use beam-search [58, 43], keeping the best $B$ solutions at each stage of support expansion:

$$j^* \in \arg \text{BottomB}_j (\underset{\alpha}{\min} \ \mathcal{L}_{\text{ridge}}(\boldsymbol{\beta} + \alpha \boldsymbol{e}_j)), \quad (26)$$

where $j^* \in \arg \text{BottomB}_j$ means $j^*$ belongs to the set of solutions whose loss is one of the B smallest losses. Afterwards, we finetune the solution on the newly expanded support and choose the best B solutions for the next stage of support expansion. A visual illustration of beam search can be found in Figure 6 in Appendix E, which also contains the detailed algorithm.

Although many methods have been proposed for sparse ridge regression, none of them have been designed with the BnB tree structure in mind. Our approach is to take advantage of the search history of past nodes to speed up the search process for a current node. To achieve this, we follow a dynamic programming approach by saving the solutions of already explored support sets. Therefore, whenever we need to adjust coefficients on the new support during beam search, we can simply retrieve the coefficients from the history if a support has been explored in the past. Essentially, we trade memory space for computational efficiency.

### 3.2.1 Provable Guarantee

Lastly, using similar methods to [29], we quantify the gap between our found heuristic solution $\hat{\boldsymbol{\beta}}$ and the optimal solution $\boldsymbol{\beta}^*$ in Theorem 3.4. Compared with Theorem 5 in [29], we improve the factor in the exponent from $\frac{m_{2k}}{M_{2k}}$ to $\frac{m_{2k}}{M_1}$ (since $M_1 \leq M_{2k}$, where $M_1$ and $M_{2k}$ are defined in [29]).

**Theorem 3.4.** *Let us define a $k$-sparse vector pair domain to be $\Omega_k := \{(\boldsymbol{x}, \boldsymbol{y}) \in \mathbb{R}^p \times \mathbb{R}^p : \|\boldsymbol{x}\|_0 \leq k, \|\boldsymbol{y}\|_0 \leq k, \|\boldsymbol{x} - \boldsymbol{y}\|_0 \leq k\}$. Any $M_1$ satisfying $f(\boldsymbol{y}) \leq f(\boldsymbol{x}) + \nabla f(\boldsymbol{x})^T(\boldsymbol{y} - \boldsymbol{x}) + \frac{M_1}{2}\|\boldsymbol{y} - \boldsymbol{x}\|_2^2$ for all $(\boldsymbol{x}, \boldsymbol{y}) \in \Omega_1$ is called a restricted smooth parameter with support size 1, and any $m_{2k}$ satisfying $f(\boldsymbol{y}) \geq f(\boldsymbol{x}) + \nabla f(\boldsymbol{x})^T(\boldsymbol{y} - \boldsymbol{x}) + \frac{m_{2k}}{2}\|\boldsymbol{y} - \boldsymbol{x}\|_2^2$ for all $(\boldsymbol{x}, \boldsymbol{y}) \in \Omega_{2k}$ is called a restricted strongly convex parameter with support size $2k$. If $\hat{\boldsymbol{\beta}}$ is our heuristic solution by the beam-search method, and $\boldsymbol{\beta}^*$ is the optimal solution, then:*

$$\mathcal{L}_{ridge}(\boldsymbol{\beta}^*) \leq \mathcal{L}_{ridge}(\hat{\boldsymbol{\beta}}) \leq (1 - e^{-m_{2k}/M_1})\mathcal{L}_{ridge}(\boldsymbol{\beta}^*). \quad (27)$$

### 3.3 Branching and Queuing

**Branching:** The most common branching techniques include most-infeasible branching and strong branching [2, 1, 15, 7]. However, these two techniques require having fractional values for the binary variables $z_j$'s, which we do not compute in our framework. Instead, we propose a new branching strategy based on our heuristic solution $\hat{\boldsymbol{\beta}}$: we branch on the coordinate whose coefficient, if set to 0, would result in the largest increase in the ridge regression loss $\mathcal{L}_{\text{ridge}}$ (See Appendix E for details):

$$j^* = \underset{j}{\arg\max} \ \mathcal{L}_{\text{ridge}}(\hat{\boldsymbol{\beta}} - \hat{\beta}_j \boldsymbol{e}_j). \quad (28)$$

The intuition is that the coordinate with the largest increase in $\mathcal{L}_{\text{ridge}}$ potentially plays a significant role, so we want to fix such a coordinate as early as possible in the BnB tree.

**Queuing:** Besides the branching strategy, we need a queue to pick a node to explore among newly created nodes. Here, we use a breadth-first approach, evaluating nodes in the order they are created.

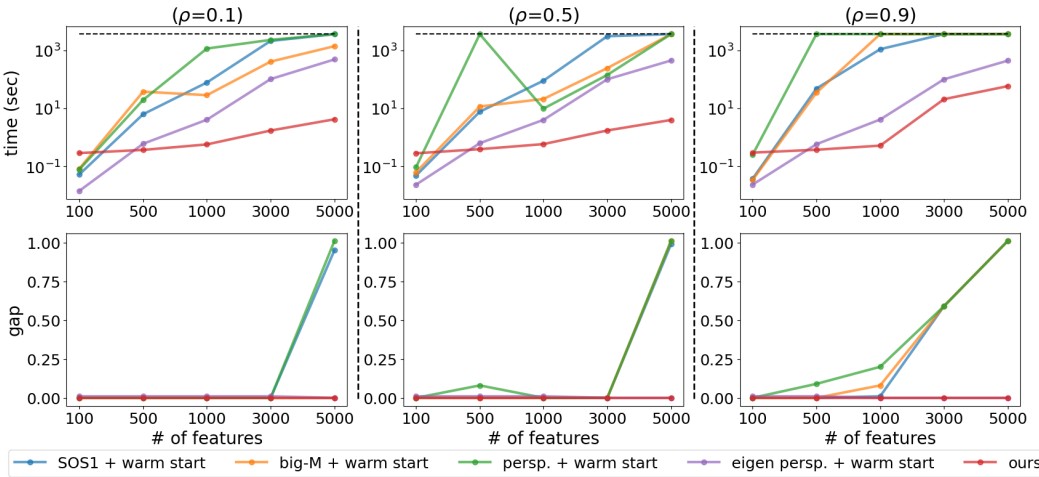

Figure 1: Comparison of running time (top row) and optimality gap (bottom row) between our method and baselines, varying the number of features, for three correlation levels $\rho = 0.1, 0.5, 0.9$ ($n = 100000, k = 10$). Time is on the log scale. Our method is generally orders of magnitude faster than other approaches. Our method achieves the smallest optimality gap, especially when the feature correlation $\rho$ is high.

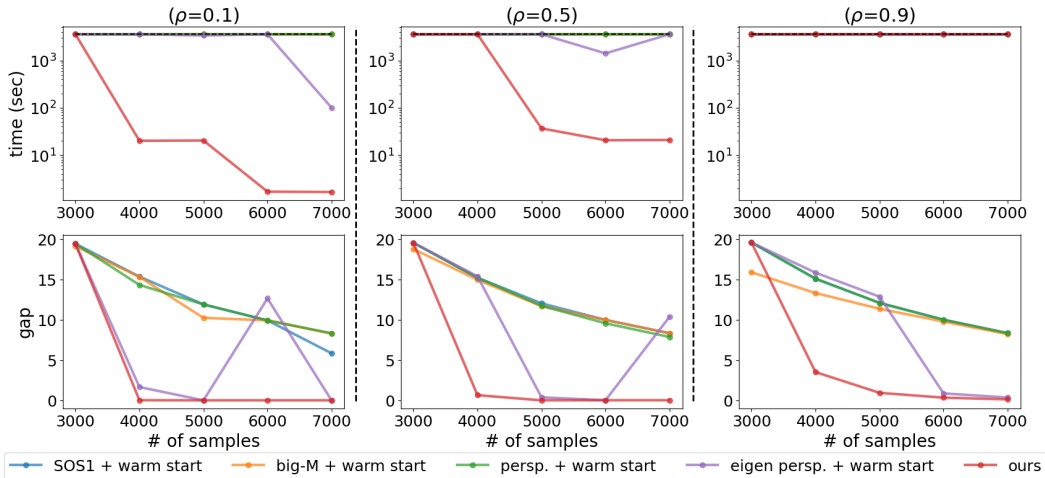

Figure 2: Comparison of running time (top row) and optimality gap (bottom row) between our method and baselines, varying sample sizes, for three correlation levels $\rho = 0.1, 0.5, 0.9$ ($p = 3000, k = 10$). Time is on the log scale. When $\rho = 0.1$ and $\rho = 0.5$, OKRidge is generally orders of magnitude faster than other approaches. In the case $\rho = 0.9$, we achieve the smallest optimality gap as shown in the bottom row.

## 4 Experiments

We test the effectiveness of our OKRidge on synthetic benchmarks and sparse identification of nonlinear dynamical systems (SINDy)[19]. Our main focus is: assessing how well our proposed lower bound calculation speeds up certification (Section 4.1), and evaluating solution quality of OKRidge on challenging applications (Section 4.2). Additional extensive experiments are in Appendix G and H. Our algorithms are written in Python. Any improvements we see over commercial MIP solvers, which are coded in C/C++, are solely due to our specialized algorithms.

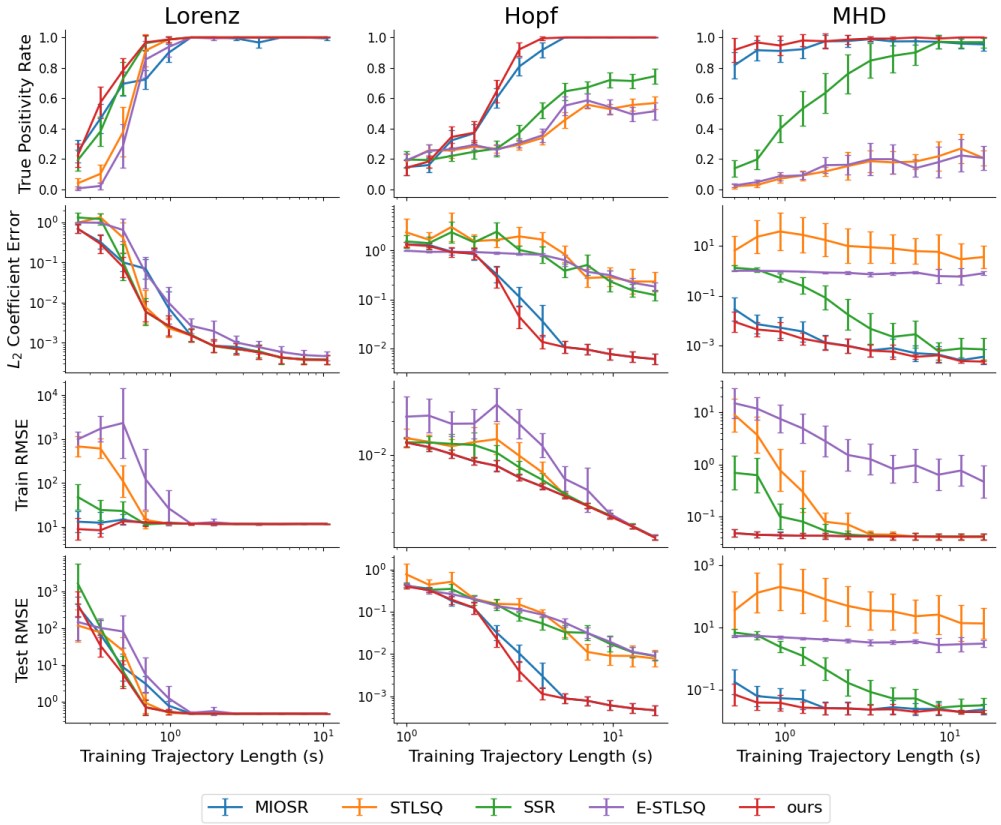

Figure 3: Results on discovering sparse differential equations. On various metrics, OKRidge outperforms all other methods, including MIOSR which uses a commercial (proprietary) MIP solver.

### 4.1 Assessing How Well Our Proposed Lower Bound Calculation Speeds Up Certification

Here, we demonstrate the speed of OKRidge for certifying optimality compared to existing MIPs solved by Gurobi [35]. We set a 1-hour time limit and an optimality gap of relative tolerance $10^{-4}$.

We use a value of 0.001 for $\lambda_2$. Our 4 baselines include MIPs with SOS1, big-M ($M = 50$ to prevent cutting off optimal solutions), perspective [4], and eigen-perspective formulations ($\lambda = \lambda_{\min}(\boldsymbol{X}^T\boldsymbol{X})$) [28]. In the main text, we use plots to present the results. In Appendix G, we present the results in tables. Additionally, in Appendix G, we conduct perturbation studies on $\lambda_2$ ($\lambda_2 = 0.1$ and $\lambda_2 = 10$) and $M$ ($M = 20$ and $M = 5$). Finally, still in Appendix G, we also compare OKRidge with other MIPs including the MOSEK solver [3], SubsetSelectionCIO [11], and L0BNB [39].

Similar to the data generation process in [11, 48], we first sample $x_i \in \mathbb{R}^p$ from a Gaussian distribution $\mathcal{N}(\boldsymbol{0}, \Sigma)$ with mean 0 and covariance matrix $\Sigma$, where $\Sigma_{ij} = \rho^{|i-j|}$. Variable $\rho$ controls the feature correlation. Then, we create the coefficient vector $\boldsymbol{\beta}^*$ with $k$ nonzero entries, where $\boldsymbol{\beta}_j^* = 1$ if $j \bmod (p/k) = 0$. Next, we construct the prediction $y_i = x_i^T\boldsymbol{\beta}^* + \epsilon_i$, where $\boldsymbol{\epsilon}_i \overset{i.i.d.}{\sim} \mathcal{N}(0, \frac{\|X\boldsymbol{\beta}^*\|_2^2}{\text{SNR}})$. SNR stands for signal-to-noise ratio (SNR), and we choose SNR to be 5 in all our experiments.

In the first setting, we fix the number of samples with $n = 100000$ and vary the number of features $p \in \{100, 500, 1000, 3000, 5000\}$ and correlation levels $\rho \in \{0.1, 0.5, 0.9\}$ (See Appendix G for $\rho = 0.3$ and $\rho = 0.7$). We warm-started the MIP solvers by our beam-search solutions. The results can be seen in Figure 1. From both figures, we see that **OKRidge outperforms all existing MIPs solved by Gurobi, usually by orders of magnitude.**

In the second setting, we fix the number of features to $p = 3000$ and vary the number of samples $n \in \{3000, 4000, 5000, 6000, 7000\}$ and the correlation levels $\rho \in \{0.1, 0.5, 0.9\}$ (see Appendix G for $\rho = 0.3$ and $\rho = 0.7$). As in the first setting, we also warm-started the MIP solvers by our beam-search solutions. The results are in Figure 2. When $n$ is close to $p$ or the correlation is high

($\rho = 0.9$), no methods can finish within the 1-hour time limit, but **OKRidge prunes the search space well and achieves the smallest optimality gap. When $n$ becomes larger in the case of $\rho = 0.1$ and $\rho = 0.5$, OKRidge runs orders of magnitude faster than all baselines**.

## 4.2 Evaluating Solution Quality of OKRidge on Challenging Applications

On previous synthetic benchmarks, many heuristics (including our beam search method) can find the optimal solution without branch-and-bound. In this subsection, we work on more challenging scenarios (sparse identification of differential equations). We replicate the experiments in [9] using three dynamical systems from the PySINDy library [27, 42]: Lorenz System, Hopf Bifurcation, and magnetohydrodynamical (MHD) model [24]. The Lorenz System is a 3-D system with the nonlinear differential equations:

$$dx/dt = -\sigma x + \sigma y, \qquad dy/dt = \rho x - y - xz, \qquad dz/dt = xy - \beta z$$

where we use standard parameters $\sigma = 10, \beta = 8/3, \rho = 28$. The true sparsities for each dimension are $(2, 3, 2)$. The Hopf Bifurcation is a 2-D system with nonlinear differential equations:

$$dx/dt = \mu x + \omega y - Ax^3 - Axy^2, \qquad dy/dt = -\omega x + \mu y - Ax^2y - Ay^3$$

where we use the standard parameters $\mu = -0.05, \omega = 1, A = 1$. The true sparsities for each dimension are $(4, 4)$. Finally, the MHD is a 6-D system with the nonlinear differential equations:

$$dV_1/dt = 4V_2V_3 - 4B_2B_3, \quad dV_2/dt = -7V_1V_3 + 7B_1B_2, \quad dV_3/dt = 3V_1V_2 - 3B_1B_2,$$
$$dB_1/dt = 2B_3V_2 - 2V_3B_2, \quad dB_2/dt = 5V_3B_1 - 5B_3V_1, \quad dB_3/dt = 9V_1B_2 - 9B_1V_2.$$

The true sparsities for each dimension are $(2, 2, 2, 2, 2, 2)$.

We use all monomial features (candidate functions) up to 5th order interactions. This results in 56 functions for the Lorentz System, 21 for Hopf Bifurcation, and 462 for MHD. Due to the high-order interaction terms, features are highly correlated, resulting in poor performance of heuristic methods.

### 4.2.1 Baselines and Experimental Setup

In addition to MIOSR (which relies on the SOS1 formulation), we also compare with three common baselines in the SINDy literature: STLSQ [54], SSR [16], and E-STLSQ [31]. The baseline SR3 [25] is not included since the previous literature [9] shows it performs poorly. We compare OKRidge with other baselines using the SINDy library [27, 42]. We follow the experimental setups in [9] for model selection, hyperparameter choices, and evaluation metrics (please see Appendix F for details). In Appendix H, we provide additional experiments on Gurobi with different MIP formulations and comparing with more heuristic baselines.

### 4.2.2 Results

Figure 3 displays the results. **OKRidge (red curves) outperforms all baselines, including MIOSR (blue curves), across evaluation metrics.** On the Lorenz System, all methods recover the true feature support when the training trajectory is long enough. When the training trajectory length is short, i.e., the left part of each subplot, (or equivalently, when the number of samples is small), OKRidge performs uniformly better than all other baselines. On the Hopf Bifurcation, all heuristic methods fail to recover the true support, resulting in poor performance. On the final MHD, OKRidge maintains the top performance and outperforms MIOSR on the true positivity rate. This demonstrates the effectiveness of OKRidge, which incurs lower runtimes and yields better metric scores under high-dimensional settings. The highest runtimes are incurred for the MHD (with 462 candidate functions/features), which are shown in Figure 4.

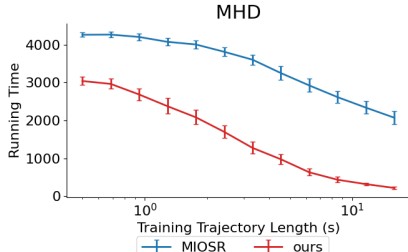

Figure 4: Running time comparison between OKRidge and MIOSR on the MHD system with 462 candidate functions. OKRidge is significantly faster than the previous state of the art.

**Limitations of OKRidge** When the feature dimension is low (under 100s), Gurobi can solve the problem to optimality faster than OKRidge. This is observed on the synthetic benchmarks ($p = 100$) and also on the Hopf Bifurcation ($p = 21$). Since Gurobi is a commercial proprietary solver, we cannot inspect the details of its sophisticated implementation. Gurobi may resort to an enumeration/brute-force approach, which could be faster than spending time to calculate lower bounds in the BnB tree. This being said, OKRidge is still competitive with Gurobi in the low-dimensional setting, and OKRidge scales favorably in high-dimensional settings.

## 5 Conclusion

We presented a method for optimal sparse ridge regression that leverages a novel tight lower bound on the objective. We showed that the method is both faster and more accurate than existing approaches for learning differential equations – a key problem in scientific discovery. This tool (unlike its main competitor) does not require proprietary software with expensive licenses and can have a significant impact on various regression applications.

## Code Availability

Implementations of OKRidge discussed in this paper are available at `https://github.com/jiachangliu/OKRidge`.

## Acknowledgements

The authors gratefully acknowledge funding support from grants NSF IIS-2130250, NSF-NRT DGE-2022040, NSF OAC-1835782, DOE DE-SC0023194, and NIH/NIDA R01 DA054994. The authors would also like to thank the anonymous reviewers for their insightful comments.

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

# Appendix to OKRidge Scalable Optimal k-Sparse Ridge Regression

## Table of Contents

# A   Related Work

**Sparse Identification of Nonlinear Dynamical Systems:** The Sparse Identification of Nonlinear Dynamical Systems (SINDy) framework [19] has been widely adopted for discovering dynamical systems from observed data. The basic framework consists of approximating derivatives, picking a library of features based on positional data, and performing sparse regression on the resulting design matrix. SINDy has expanded to solving PDEs [54, 46] or implicit equations [41], noisy or low data settings [31, 47], and constrained problems [25, 9]. The basic framework can be summarized as a standard regression problem:

$$\dot{X} = \Theta(X)\Xi, \tag{29}$$

where the goal is to find $\Xi = [\beta_1, \ldots, \beta_d]$, with $d$ being the number of dimensions of the dynamical system, $X$ being the observed data, and $\Theta$ is a map from observed data to candidate functions. The problem is typically solved independently across dimensions, yielding $d$ sparse regression problems.

Many methods in the SINDy framework solve regression problems with a greedy backwards selection approach. These methods have been shown to outperform LASSO [25] and Orthogonal Matching Pursuit [16] in learning dynamical systems. The backwards selection approach typically solves a non-sparse regression problem, and then either removes all small coefficients below a threshold, or remove one feature at a time until a desired sparsity level is reached [54]. Although these methods have performed well with reasonable run times, they struggle to identify true dynamics as the number of candidate features increases.

Advances in MIP formulations have led to solving SINDy problems with greater success and ability to verify optimality [9]; however, they tend not to scale as well. The method we present also verifies optimality and is much faster than the MIP formula, particularly for larger number of features $p$.

**Heuristic Methods:** Greedy methods aim to solve the problem to near-optimality, with few guarantees. One direction is greedy pursuit (i.e., forward selection), where one coordinate at a time is added until the required support size is reached [56, 13, 50, 51, 22, 29]. Another direction is iterative thresholding, where gradient descent steps alternate with projection steps to satisfy the support size requirement [14, 40, 11, 59, 57]. Other approaches include randomized rounding on the solution of the boolean-relaxed problem [53], swapping features [61, 6, 38, 63, 44], or solving a smaller problem optimally on a working set [60]. These heuristic solutions may greatly improve the computational speed of MIP solvers when used as warm starts. Typically, the heuristic algorithm starts from scratch each time that it runs, which is slow when running it repeatedly throughout the BnB tree. Our insight is that the search history of heuristic methods in previously solved nodes can be used to speed up the heuristic method in the current node.

**Optimal Methods and Lower Bound Calculation:** In order to certify optimality for this NP-hard problem, we need to perform branch-and-bound and calculate the lower bound for each node in the BnB tree. For the lower bound calculation, early works include the SOS1 formulation or the big-M formulation [10, 9]. Both formulations can be implemented in a commercial solver. However, the SOS1 formulation is not scalable to high dimensions, while the big-M formulation is sensitive to the hyperparameter "M" used to balance scalability and correctness. To circumvent this problem, SubsetSelectionCIO [11] formulate the least-squares term using the Fenchedl duality with dual variables. After the formulation, SubsetSelectionCIO applies callbacks to add cutting planes to get a lower bound. Although OKRidge also applies the Fenchel duality, we apply the Fenchel duality on the $\ell_2$ regularization term. We are solving the problem in the feature space, while SubsetSelectionCIO solves the problem in the sample space. As pointed in [39], the branch-and-cut method in SubsetSelectionCIO [11] runs slow when the $\ell_2$ regularization is small. Recently, the perspective formulation [32, 34, 62, 28, 4, 59, 5, 37] of the $\ell_2$ term has also been used. Through convex relaxation, the lower bound can be obtained by solving a quadratically constrained quadratic program (QCQP) with rotated second-order cone constraints. However, the conic formulation is still computationally intensive for large-scale datasets and has difficulty attaining optimal solutions. Our work builds upon the perspective formulation, but we propose an efficient way to calculate the lower bound through Fenchel duality. Another line of works focuses on optimal perspective formulation [62, 28, 37]. This requires solving a semidefinite programming (SDP) problem at each node, which has been shown not scalable to high dimensions [28]. What is more, MI-SDP is not supported by Gurobi. Our work is related to the optimal perspective formulation, but we set the

diagonal matrix $\mathrm{diag}(\boldsymbol{d}) = \lambda_{\min}(\boldsymbol{X}^T\boldsymbol{X})\boldsymbol{I}$. We call this the eigen-perspective formulation. Although this is not the optimal choice, it is a good approximation and is supported by Gurobi through the QCQP formulation as discussed above. Lastly, there is a recent work called l0bnb [39], which also implements a customized branch-and-bound framework without the commercial solvers. However, l0bnb is solving the $l_0$-regularized problem, which is not the same as the $\ell_0$-constrained problem. To get a solution with the specified sparsity, l0bnb needs to solve the problem multiple times with different $\ell_0$ regularizations.

# B  Background Concepts

**Definition B.1** (**Strong Convexity** [20]). A function $f : \mathbb{R}^p \to \mathbb{R}$ is strongly convex with parameter $\alpha > 0$ if it satisfies the following subgradient inequality:

$$f(\boldsymbol{y}) \geq f(\boldsymbol{x}) + \nabla f(\boldsymbol{x})^T (\boldsymbol{y} - \boldsymbol{x}) + \frac{\alpha}{2} \|\boldsymbol{y} - \boldsymbol{x}\|_2^2 \tag{30}$$

for $\boldsymbol{x}, \boldsymbol{y} \in \mathbb{R}^p$. If the gradient $\nabla f$ does not exist but the subgradient exists, then we have

$$f(\boldsymbol{y}) \geq f(\boldsymbol{x}) + \boldsymbol{g}^T (\boldsymbol{y} - \boldsymbol{x}) + \frac{\alpha}{2} \|\boldsymbol{y} - \boldsymbol{x}\|_2^2 \tag{31}$$

for $\boldsymbol{x}, \boldsymbol{y} \in \mathbb{R}^p$, where $\boldsymbol{g} \in \partial f$ and $\partial f$ is the subdifferential of $f$.

**Definition B.2** (**Smoothness** [20]). A continuously differentiable function $f : \mathbb{R}^p \to \mathbb{R}$ is smooth with parameter $\beta > 0$ if the gradeint $\nabla f$ is $\beta$-Lipschitz. Mathematically, this means that:

$$\|\nabla f(\boldsymbol{x}) - \nabla f(\boldsymbol{y})\|_2 \leq \beta \|\boldsymbol{x} - \boldsymbol{y}\|_2 \tag{32}$$

for $\boldsymbol{x}, \boldsymbol{y} \in \mathbb{R}^p$.

**Lemma B.3** (**Smoothness Property** [20]). *Let $f : \mathbb{R}^p \to \mathbb{R}$ be a $\beta$-smooth function. Then, we have the following sandwich property ($f(\boldsymbol{y})$ is "sandwiched" between two quadratic functions):*

$$f(\boldsymbol{x}) + \nabla f(\boldsymbol{x})^T (\boldsymbol{y} - \boldsymbol{x}) + \frac{\beta}{2} \|\boldsymbol{y} - \boldsymbol{x}\|_2^2 \geq f(\boldsymbol{y}) \geq f(\boldsymbol{x}) + \nabla f(\boldsymbol{x})^T (\boldsymbol{y} - \boldsymbol{x}) - \frac{\beta}{2} \|\boldsymbol{y} - \boldsymbol{x}\|_2^2 \tag{33}$$

*for $\boldsymbol{x}, \boldsymbol{y} \in \mathbb{R}^p$.*

**Definition B.4** (**$k$-sparse Pair Domain** [29]). A domain $\Omega_k \subset \mathbb{R}^p \times \mathbb{R}^p$ is a $k$-sparse vector pair domain if it contains all pairs of $k$-sparse vectors that differ in at most $k$ entries, *i.e*,

$$\Omega_k := \{(\boldsymbol{x}, \boldsymbol{y}) \in \mathbb{R}^p \times \mathbb{R}^p : \|\boldsymbol{x}\|_0 \leq k, \|y\|_0 \leq k, \|\boldsymbol{x} - \boldsymbol{y}\| \leq k\}. \tag{34}$$

**Definition B.5** (**Restricted Strong Convexity** [29]). A function $f : \mathbb{R}^p \to \mathbb{R}$ is restricted strongly convex with parameter $\alpha > 0$ on the $k$-sparse vector pair domain $\Omega_k$ if it satisfies the following inequality:

$$f(\boldsymbol{y}) \geq f(\boldsymbol{x}) + \nabla f(\boldsymbol{x})^T (\boldsymbol{y} - \boldsymbol{x}) + \frac{\alpha}{2} \|\boldsymbol{y} - \boldsymbol{x}\|_2^2 \tag{35}$$

for $(\boldsymbol{x}, \boldsymbol{y}) \in \Omega_k$.

**Definition B.6** (**Restricted Smoothness** [29]). A function $f : \mathbb{R}^p \to \mathbb{R}$ is restrictedly smooth with parameter $\beta > 0$ on the $k$-sparse vector pair domain $\Omega_k$ if it satisfies the following inequality:

$$f(\boldsymbol{x}) + \nabla f(\boldsymbol{x})^T (\boldsymbol{y} - \boldsymbol{x}) + \frac{\beta}{2} \|\boldsymbol{y} - \boldsymbol{x}\|_2^2 \geq f(\boldsymbol{y}) \tag{36}$$

for $(\boldsymbol{x}, \boldsymbol{y}) \in \Omega_k$.

**Lemma B.7** (**Gradient of Ridge Regression**). *At $\boldsymbol{\beta} \in \mathbb{R}^p$, the gradient of our ridge regression function is given by:*

$$\nabla \mathcal{L}_{\text{ridge}}(\boldsymbol{\beta}) = 2\boldsymbol{X}^T (\boldsymbol{X}\boldsymbol{\beta} - \boldsymbol{y}) + 2\lambda_2 \boldsymbol{\beta} \tag{37}$$

*for $\boldsymbol{\beta} \in \mathbb{R}^p$.*

*Proof.*

$$\begin{aligned}
\nabla \mathcal{L}_{\text{ridge}}(\boldsymbol{\beta}) &= \nabla(\boldsymbol{\beta}^T \boldsymbol{X}^T \boldsymbol{X} \boldsymbol{\beta} - 2\boldsymbol{y}^X \boldsymbol{X} \boldsymbol{\beta} + \lambda_2 \boldsymbol{\beta}^T \boldsymbol{\beta}) \\
&= \nabla(\boldsymbol{\beta}^T \boldsymbol{X}^T \boldsymbol{X} \boldsymbol{\beta}) - \nabla(2\boldsymbol{y}^T \boldsymbol{X} \boldsymbol{\beta}) + \lambda_2 \nabla(\boldsymbol{\beta}^T \boldsymbol{\beta}) \\
&= 2\boldsymbol{X}^T \boldsymbol{X} \boldsymbol{\beta} - 2\boldsymbol{X}^T \boldsymbol{y} + 2\lambda_2 \boldsymbol{\beta} \\
&= 2\boldsymbol{X}^T (\boldsymbol{X} \boldsymbol{\beta} - \boldsymbol{y}) + 2\lambda_2 \boldsymbol{\beta}
\end{aligned}$$

$\square$

# C Derivations and Proofs

## C.1 Proof of Theorem 3.1

**Theorem 3.1.** *If we reparameterize $\boldsymbol{c} = \frac{2}{\lambda_2 + \lambda}(\boldsymbol{X}^T \boldsymbol{y} - \boldsymbol{Q}_\lambda \boldsymbol{\gamma})$ with a new parameter $\boldsymbol{\gamma}$, then Problem (10) is equivalent to the following saddle point optimization problem:*

$$\max_{\boldsymbol{\gamma}} \min_{\boldsymbol{z}} \mathcal{L}_{ridge-\lambda}^{saddle}(\boldsymbol{\gamma}, \boldsymbol{z}) \ \ s.t. \ \sum_{j=1}^{p} z_j \leq k, \ \ z_j \in [0, 1], \tag{38}$$

*where* $\quad \mathcal{L}_{ridge-\lambda}^{saddle}(\boldsymbol{\gamma}, \boldsymbol{z}) := -\boldsymbol{\gamma}^T \boldsymbol{Q}_\lambda \boldsymbol{\gamma} - \frac{1}{\lambda_2 + \lambda}(\boldsymbol{X}^T \boldsymbol{y} - \boldsymbol{Q}_\lambda \boldsymbol{\gamma})^T diag(\boldsymbol{z})(\boldsymbol{X}^T \boldsymbol{y} - \boldsymbol{Q}_\lambda \boldsymbol{\gamma}),$ (39)

*and $diag(\boldsymbol{z})$ is a diagonal matrix with $\boldsymbol{z}$ on the diagonal.*

Before we give the proof for Theorem 3.1, we first state one useful lemma below:

**Lemma C.1.** *For an optimal solution $\boldsymbol{c}^*$ to Problem (40), there exists some $\boldsymbol{\beta}^*$ such that $\boldsymbol{c}^* = \frac{2}{\lambda_2 + \lambda}(\boldsymbol{X}^T \boldsymbol{y} - \boldsymbol{Q}_\lambda \boldsymbol{\beta}^*)$.*

Without using this lemma, there is some concern with reparametrizing $\boldsymbol{c}$ with a new parameter $\boldsymbol{\gamma}$ when $\lambda = \lambda_{\min}(\boldsymbol{X}^T \boldsymbol{X})$. In this case, $\boldsymbol{X}^T \boldsymbol{X} - \lambda \boldsymbol{I}$ becomes singular and our reparametrization trick could miss some subspace. This issue of having $\boldsymbol{X}^T \boldsymbol{X} - \lambda \boldsymbol{I}$ being singular would lead to the concern that Problem (39) is a relaxation of Problem (40), not equivalent. Fortunately, due to this lemma, the subspace which we miss via the reparametrization trick will not prevent Problem (39) from achieving the same optimal value as Problem (40).

We now prove this lemma by the method of contradiction.

*Proof.* Suppose the optimal solution $\boldsymbol{c}$ to Problem (40) cannot be written by Equation (41). Through the rank-nullity theorem in linear algebra, we can find $\boldsymbol{d}_1 \in col(\boldsymbol{Q}_\lambda)$ and $d_2 \in ker(\boldsymbol{Q}_\lambda), \boldsymbol{d}_2 \neq \boldsymbol{0}$ so that $\boldsymbol{c} = \frac{2}{\lambda_2 + \lambda}(\boldsymbol{X}^T \boldsymbol{y} - \boldsymbol{d}_1 + \boldsymbol{d}_2)$, where $col(\cdot)$ denotes the column space of a matrix and $ker(\cdot)$ denotes the kernel of a matrix. If we let $\boldsymbol{\beta} = \alpha \boldsymbol{d}_2$ for some real value $\alpha$, then the inner optimization of Problem (40) (ignore the last term involving $z_j$'s for now since $\boldsymbol{\beta}$ and $\boldsymbol{z}$ are separable in the objective function) becomes

$$\begin{aligned}
\mathcal{L}_{ridge-\lambda}^{Fenchel}(\boldsymbol{\beta}, \boldsymbol{z}, \boldsymbol{c}) &= \boldsymbol{\beta}^T \boldsymbol{Q}_\lambda \boldsymbol{\beta} - ((\lambda_2 + \lambda)\boldsymbol{c} - 2\boldsymbol{X}^T \boldsymbol{y})^T \boldsymbol{\beta} \\
&= \boldsymbol{\beta}^T \boldsymbol{Q}_\lambda \boldsymbol{\beta} - ((\lambda_2 + \lambda)\frac{2}{\lambda_2 + \lambda}(\boldsymbol{X}^T \boldsymbol{y} - \boldsymbol{d}_1 + \boldsymbol{d}_2) - 2\boldsymbol{X}^T \boldsymbol{y})^T \boldsymbol{\beta} \\
&= \boldsymbol{\beta}^T \boldsymbol{Q}_\lambda \boldsymbol{\beta} - (2(\boldsymbol{X}^T \boldsymbol{y} - \boldsymbol{d}_1 + \boldsymbol{d}_2) - 2\boldsymbol{X}^T \boldsymbol{y})^T \boldsymbol{\beta} \\
&= \boldsymbol{\beta}^T \boldsymbol{Q}_\lambda \boldsymbol{\beta} + 2(\boldsymbol{d}_1 - \boldsymbol{d}_2)^T \boldsymbol{\beta} \\
&= \boldsymbol{\beta}^T \boldsymbol{Q}_\lambda \boldsymbol{\beta} + 2\boldsymbol{d}_1^T \boldsymbol{\beta} - 2\boldsymbol{d}_2^T \boldsymbol{\beta}.
\end{aligned}$$

Because $\boldsymbol{\beta} = \alpha \boldsymbol{d}_2$, we have $\boldsymbol{\beta} \in ker(\boldsymbol{Q}_\lambda)$ and $\boldsymbol{\beta} \perp \boldsymbol{d}_1$. Thus, the first two terms become $\boldsymbol{0}$, and we are left with $\mathcal{L}_{ridge-\lambda}^{Fenchel} = -2\boldsymbol{d}_2^T \boldsymbol{\beta} = -2\alpha \|\boldsymbol{d}_2\|_2^2$. For the inner optimization of Problem (40), because we want to minimize over $\boldsymbol{\beta}$, we can let $\alpha \to \infty$, and we will get $\mathcal{L}_{ridge-\lambda}^{Fenchel} \to -\infty$. This is obviously not the optimal value for Problem (40) (a simple counter-example is to let $\boldsymbol{c} = \frac{2}{\lambda_2 + \lambda}\boldsymbol{X}^T \boldsymbol{y}$, and we can easily show that $\min_{\boldsymbol{\beta}, \boldsymbol{z}} \mathcal{L}_{ridge-\lambda}^{Fenchel}(\boldsymbol{\beta}, \boldsymbol{z}, \boldsymbol{c})$ is finite). Therefore, $\boldsymbol{d}_2$ must be $\boldsymbol{0}$, and the optimal solution $\boldsymbol{c}$ to Problem (40) must be represented by some $\boldsymbol{\beta}$ through the equation $\boldsymbol{c} = \frac{2}{\lambda_2 + \lambda}(\boldsymbol{X}^T \boldsymbol{y} - \boldsymbol{Q}_\lambda \beta)$. $\quad\square$

Now that we have this lemma, we can prove Theorem 3.1 rigorously. We provide two alternative proofs so that readers can understand the logic from different angles. The first proof is based on using the lemma directly; the second proof is based on the sandwich argument.

### Proof 1

*Proof.* For notational convenience, let us use $\mathcal{D}_z := \{\boldsymbol{z} \mid \sum_{j=1}^{p} z_j \leq k, z_j \in [0, 1]\}$ as a shorthand for the domain of $\boldsymbol{z}$.

Recall that Problem (10) is the following optimization problem:

$$\max_{\boldsymbol{c}} \min_{\boldsymbol{\beta}, \boldsymbol{z}} \mathcal{L}_{\text{ridge}-\lambda}^{\text{Fenchel}}(\boldsymbol{\beta}, \boldsymbol{z}, \boldsymbol{c}) \tag{40}$$

$$\text{subject to} \quad \sum_{j=1}^{p} z_j \leq k, \quad z_j \in [0, 1].$$

where $\mathcal{L}_{\text{ridge}-\lambda}^{\text{Fenchel}}(\boldsymbol{\beta}, \boldsymbol{z}, \boldsymbol{c}) = \boldsymbol{\beta}^T \boldsymbol{Q}_\lambda \boldsymbol{\beta} - 2\boldsymbol{y}^T \boldsymbol{X} \boldsymbol{\beta} + (\lambda_2 + \lambda) \sum_{j=1}^{p} (\beta_j c_j - \frac{c_j^2}{4}) z_j$.

For any fixed $\boldsymbol{c}$, if we take the derivative of $\mathcal{L}_{\text{ridge}-\lambda}^{\text{Fenchel}}(\boldsymbol{\beta}, \boldsymbol{z}, \boldsymbol{c})$ with respect to $\boldsymbol{\beta}$ and set the derivative to $\boldsymbol{0}$, we obtain the optimality condition for $\boldsymbol{\beta}$:

$$\boldsymbol{c} = \frac{2}{\lambda_2 + \lambda} [\boldsymbol{X}^T \boldsymbol{y} - (\boldsymbol{X}^T \boldsymbol{X} - \lambda \boldsymbol{I}) \boldsymbol{\beta}] \tag{41}$$

If we reparametrize $\boldsymbol{c}$ with a new parameter $\boldsymbol{\gamma}$ and let $\boldsymbol{c} = \frac{2}{\lambda_2+\lambda}[\boldsymbol{X}^T \boldsymbol{y} - (\boldsymbol{X}^T \boldsymbol{X} - \lambda \boldsymbol{I})\boldsymbol{\gamma}]$, then $\boldsymbol{\beta} = \boldsymbol{\gamma}$ satisfies the optimality condition for $\boldsymbol{\beta}$ in Equation (41). Therefore, whenever we reparametrize $\boldsymbol{c}$ with $\boldsymbol{c} = \frac{2}{\lambda_2+\lambda}[\boldsymbol{X}^T \boldsymbol{y} - (\boldsymbol{X}^T \boldsymbol{X} - \lambda \boldsymbol{I})\boldsymbol{\gamma}]$, we can just let $\boldsymbol{\beta} = \boldsymbol{\gamma}$, and this ensures us that we can achieve the inner minimum of Equation (40) with respect to $\boldsymbol{\beta}$.

Then, we can reduce Problem (40) to Problem (38) in the following way:

$$\max_{\boldsymbol{c}} \min_{\boldsymbol{\beta}, \boldsymbol{z}} \mathcal{L}_{\text{ridge}-\lambda}^{\text{Fenchel}}(\boldsymbol{\beta}, \boldsymbol{z}, \boldsymbol{c}) \qquad \textit{\# subject to } \sum_{j=1}^{p} z_j \leq k, z_j \in [0, 1]$$

$$= \max_{\boldsymbol{c}} \min_{\boldsymbol{z} \in \mathcal{D}_z, \boldsymbol{\beta}} \mathcal{L}_{\text{ridge}-\lambda}^{\text{Fenchel}}(\boldsymbol{\beta}, \boldsymbol{z}, \boldsymbol{c}) \qquad \textit{\# use } \mathcal{D}_z \textit{ as the shorhand for the domain of } \boldsymbol{z}$$

$$= \max_{\boldsymbol{c}} \min_{\boldsymbol{z} \in \mathcal{D}_z, \boldsymbol{\beta}} \boldsymbol{\beta}^T \boldsymbol{Q}_\lambda \boldsymbol{\beta} - 2\boldsymbol{y}^T \boldsymbol{X} \boldsymbol{\beta} + (\lambda_2 + \lambda) \sum_{j=1}^{p} (\beta_j c_j - \frac{c_j^2}{4} z_j)$$

$$\textit{\# plug in the details for } \mathcal{L}_{\text{ridge}-\lambda}^{\text{Fenchel}}(\boldsymbol{\beta}, \boldsymbol{z}, \boldsymbol{c})$$

$$= \max_{\boldsymbol{c}} \min_{\boldsymbol{z} \in \mathcal{D}_z} \boldsymbol{\gamma}^T \boldsymbol{Q}_\lambda \boldsymbol{\gamma} - 2\boldsymbol{y}^T \boldsymbol{X} \boldsymbol{\gamma} + (\lambda_2 + \lambda) \sum_{j=1}^{p} (\gamma_j c_j - \frac{c_j^2}{4} z_j)$$

$$\textit{\# letting } \boldsymbol{\beta} = \boldsymbol{\gamma} \textit{ achieves the inner minimum for } \boldsymbol{\beta}$$

$$= \max_{\boldsymbol{c}} \min_{\boldsymbol{z} \in \mathcal{D}_z} \boldsymbol{\gamma}^T \boldsymbol{Q}_\lambda \boldsymbol{\gamma} - 2\boldsymbol{y}^T \boldsymbol{X} \boldsymbol{\gamma} + (\lambda_2 + \lambda)(\boldsymbol{\gamma}^T \boldsymbol{c} - \frac{1}{4} \boldsymbol{c}^T diag(\boldsymbol{z}) \boldsymbol{c})$$

$$\textit{\# write the summation in vector forms; } diag(\boldsymbol{z}) \textit{ denotes a diagonal matrix with } \boldsymbol{z} \textit{ on the diagonal}$$

$$= \max_{\boldsymbol{\gamma}} \min_{\boldsymbol{z} \in \mathcal{D}_z} \boldsymbol{\gamma}^T \boldsymbol{Q}_\lambda \boldsymbol{\gamma} - 2\boldsymbol{y}^T \boldsymbol{X} \boldsymbol{\gamma} + (\lambda_2 + \lambda) \left( \boldsymbol{\gamma}^T \frac{2}{\lambda_2 + \lambda} (\boldsymbol{X}^T \boldsymbol{y} - \boldsymbol{Q}_\lambda \boldsymbol{\gamma}) \right.$$

$$\left. - \frac{1}{4} \frac{2}{\lambda_2 + \lambda} (\boldsymbol{X}^T \boldsymbol{y} - \boldsymbol{Q}_\lambda \boldsymbol{\gamma})^T diag(\boldsymbol{z}) \frac{2}{\lambda_2 + \lambda} (\boldsymbol{X}^T \boldsymbol{y} - \boldsymbol{Q}_\lambda \boldsymbol{\gamma}) \right)$$

$$\textit{\# apply the reparametrization } \boldsymbol{c} = \frac{2}{\lambda_2+\lambda}(\boldsymbol{X}^T \boldsymbol{y} - \boldsymbol{Q}_\lambda \boldsymbol{\gamma}).$$

$$\textit{\# remember that due to Lemma C.1, we will not miss the optimal solution } \boldsymbol{c}^*$$

$$= \max_{\boldsymbol{\gamma}} \min_{\boldsymbol{z} \in \mathcal{D}_z} \boldsymbol{\gamma}^T \boldsymbol{Q}_\lambda \boldsymbol{\gamma} - 2\boldsymbol{y}^T \boldsymbol{X} \boldsymbol{\gamma} + 2\boldsymbol{\gamma}^T (\boldsymbol{X}^T \boldsymbol{y} - \boldsymbol{Q}_\lambda \boldsymbol{\gamma}) - \frac{1}{\lambda_2 + \lambda} (\boldsymbol{X}^T \boldsymbol{y} - \boldsymbol{Q}_\lambda \boldsymbol{\gamma})^T diag(\boldsymbol{z}) (\boldsymbol{X}^T \boldsymbol{y} - \boldsymbol{Q}_\lambda \boldsymbol{\gamma})$$

$$\textit{\# distribute } \lambda_2 + \lambda \textit{ inside the parentheses}$$

$$= \max_{\boldsymbol{\gamma}} \min_{\boldsymbol{z} \in \mathcal{D}_z} \boldsymbol{\gamma}^T \boldsymbol{Q}_\lambda \boldsymbol{\gamma} - 2\boldsymbol{y}^T \boldsymbol{X} \boldsymbol{\gamma} + 2\boldsymbol{\gamma}^T \boldsymbol{X}^T \boldsymbol{y} - 2\boldsymbol{\gamma}^T \boldsymbol{Q}_\lambda \boldsymbol{\gamma} - \frac{1}{\lambda_2 + \lambda} (\boldsymbol{X}^T \boldsymbol{y} - \boldsymbol{Q}_\lambda \boldsymbol{\gamma})^T diag(\boldsymbol{z}) (\boldsymbol{X}^T \boldsymbol{y} - \boldsymbol{Q}_\lambda \boldsymbol{\gamma})$$

$$\textit{\# distribute } 2\boldsymbol{\gamma} \textit{ inside the parentheses}$$

$$= \max_{\boldsymbol{\gamma}} \min_{\boldsymbol{z} \in \mathcal{D}_z} -\boldsymbol{\gamma}^T \boldsymbol{Q}_\lambda \boldsymbol{\gamma} - \frac{1}{\lambda_2 + \lambda} (\boldsymbol{X}^T \boldsymbol{y} - \boldsymbol{Q}_\lambda \boldsymbol{\gamma})^T diag(\boldsymbol{z}) (\boldsymbol{X}^T \boldsymbol{y} - \boldsymbol{Q}_\lambda \boldsymbol{\gamma})$$

$$\textit{\# simplify the linear algebra}$$

$$= \max_{\boldsymbol{\gamma}} \min_{\boldsymbol{z}} -\boldsymbol{\gamma}^T \boldsymbol{Q}_\lambda \boldsymbol{\gamma} - \frac{1}{\lambda_2 + \lambda} (\boldsymbol{X}^T \boldsymbol{y} - \boldsymbol{Q}_\lambda \boldsymbol{\gamma})^T diag(\boldsymbol{z}) (\boldsymbol{X}^T \boldsymbol{y} - \boldsymbol{Q}_\lambda \boldsymbol{\gamma})$$

$$\textit{\# subject to } \sum_{j=1}^{p} z_j \leq k, z_j \in [0, 1]$$

$$= \max_{\boldsymbol{\gamma}} \min_{\boldsymbol{z}} \mathcal{L}^{\text{saddle}}_{\text{ridge}-\lambda}(\boldsymbol{\gamma}, \boldsymbol{z})$$

*# subject to $\sum_{j=1}^{p} z_j \le k, z_j \in [0,1]$; apply the definition of $\mathcal{L}^{saddle}_{ridge-\lambda}(\boldsymbol{\gamma}, \boldsymbol{z})$*

This completes the derivation for Theorem 3.1.

$\square$

### Proof 2

*Proof.* We will show $\max_{\boldsymbol{c}} \min_{\boldsymbol{\beta}, \boldsymbol{z}} \mathcal{L}^{\text{Fenchel}}_{\text{ridge}-\lambda}(\boldsymbol{\beta}, \boldsymbol{z}, \boldsymbol{c}) \le \max_{\boldsymbol{\gamma}} \min_{\boldsymbol{z}} \mathcal{L}^{\text{saddle}}_{\text{ridge}-\lambda}(\boldsymbol{\gamma}, \boldsymbol{z})$ and $\max_{\boldsymbol{c}} \min_{\boldsymbol{\beta}, \boldsymbol{z}} \mathcal{L}^{\text{Fenchel}}_{\text{ridge}-\lambda}(\boldsymbol{\beta}, \boldsymbol{z}, \boldsymbol{c}) \ge \max_{\boldsymbol{\gamma}} \min_{\boldsymbol{z}} \mathcal{L}^{\text{saddle}}_{\text{ridge}-\lambda}(\boldsymbol{\gamma}, \boldsymbol{z})$ (both under the constrains $\sum_{j=1}^{p} z_j \le k, z_j \in [0,1]$) and therefore establish the equivalence of optimal values.

For the first inequality, let $\boldsymbol{\beta}^*, \boldsymbol{c}^*$ be the optimal solution to Problem (40) with the relation $\boldsymbol{c}^* = \frac{2}{\lambda_2+\lambda}(\boldsymbol{X}^T\boldsymbol{y} - \boldsymbol{Q}_\lambda\boldsymbol{\beta}^*)$ (this is always true because of Lemma C.1). Furthermore, let $\boldsymbol{\gamma}^* = \boldsymbol{\beta}^*$. Then we have

$$\max_{\boldsymbol{c}} \min_{\boldsymbol{\beta}, \boldsymbol{z}} \mathcal{L}^{\text{Fenchel}}_{\text{ridge}-\lambda}(\boldsymbol{\beta}, \boldsymbol{z}, \boldsymbol{c}) = \min_{\boldsymbol{z}} \mathcal{L}^{\text{Fenchel}}_{\text{ridge}-\lambda}(\boldsymbol{\beta}^*, \boldsymbol{z}, \boldsymbol{c}^*) = \min_{\boldsymbol{z}} \mathcal{L}^{\text{saddle}}_{\text{ridge}-\lambda}(\boldsymbol{\gamma}^*, \boldsymbol{z}) \le \max_{\boldsymbol{\gamma}} \min_{\boldsymbol{z}} \mathcal{L}^{\text{saddle}}_{\text{ridge}-\lambda}(\boldsymbol{\gamma}, \boldsymbol{z})$$

The derivation for the second equality roughly follows the derivation in the first proof we gave.

For the second inequality, let $\hat{\boldsymbol{\gamma}}$ be the optimal solution to Problem (38). Let $\hat{\boldsymbol{c}} = \frac{2}{\lambda_2+\lambda}(\boldsymbol{X}^T\boldsymbol{y} - \boldsymbol{Q}_\lambda\hat{\boldsymbol{\gamma}})$. Furthermore, let $\hat{\boldsymbol{\beta}} = \hat{\boldsymbol{\gamma}}$ (notice that this $\hat{\boldsymbol{\beta}}$ satisfies the optimality condition of Problem (40)). Then we have

$$\max_{\boldsymbol{\gamma}} \min_{\boldsymbol{z}} \mathcal{L}^{\text{saddle}}_{\text{ridge}-\lambda}(\boldsymbol{\gamma}, \boldsymbol{z}) = \min_{\boldsymbol{z}} \mathcal{L}^{\text{saddle}}_{\text{ridge}-\lambda}(\hat{\boldsymbol{\gamma}}, \boldsymbol{z})$$

$$= \min_{\boldsymbol{z}} \mathcal{L}^{\text{Fenchel}}_{\text{ridge}-\lambda}(\hat{\boldsymbol{\beta}}, \boldsymbol{z}, \hat{\boldsymbol{c}})$$

$$= \min_{\boldsymbol{\beta}, \boldsymbol{z}} \mathcal{L}^{\text{Fenchel}}_{\text{ridge}-\lambda}(\boldsymbol{\beta}, \boldsymbol{z}, \hat{\boldsymbol{c}})$$

$$\le \max_{\boldsymbol{c}} \min_{\boldsymbol{\beta}, \boldsymbol{z}} \mathcal{L}^{\text{Fenchel}}_{\text{ridge}-\lambda}(\boldsymbol{\beta}, \boldsymbol{z}, \boldsymbol{c}).$$

Similarly, the derivation for the second equality roughly follows the derivation (but this time in reverse order) in our first proof.

Using these two inequalities, we can establish the equivalence of optimal values between Problem (40) and Problem (38), *i.e.*, $\max_{\boldsymbol{c}} \min_{\boldsymbol{\beta}, \boldsymbol{z}} \mathcal{L}^{\text{Fenchel}}_{\text{ridge}-\lambda}(\boldsymbol{\beta}, \boldsymbol{z}, \boldsymbol{c}) = \max_{\boldsymbol{\gamma}} \min_{\boldsymbol{z}} \mathcal{L}^{\text{saddle}}_{\text{ridge}-\lambda}(\boldsymbol{\gamma}, \boldsymbol{z})$. $\square$

### C.2 Proof of Theorem 3.2

**Theorem 3.2.** *The function $h(\boldsymbol{\gamma})$ defined in Equation (14) is lower bounded by*

$$h(\boldsymbol{\gamma}) \ge -\boldsymbol{\gamma}^T \boldsymbol{Q}_\lambda \boldsymbol{\gamma} - \frac{1}{\lambda_2 + \lambda} \|\boldsymbol{X}^T\boldsymbol{y} - \boldsymbol{Q}_\lambda\boldsymbol{\gamma}\|_2^2. \tag{42}$$

*Furthermore, the right-hand size of Equation (42) is maximized if $\boldsymbol{\gamma} = \hat{\boldsymbol{\gamma}} = \text{argmin}_{\boldsymbol{\alpha}} \mathcal{L}_{ridge}(\boldsymbol{\alpha})$, where in this case, $h(\boldsymbol{\gamma})$ on the left-hand side of Equation (42) becomes*

$$h(\hat{\boldsymbol{\gamma}}) = \mathcal{L}_{\text{ridge}}(\hat{\boldsymbol{\gamma}}) + (\lambda_2 + \lambda) \text{SumBottom}_{p-k}(\{\hat{\gamma}_j^2\}),$$

*where $\text{SumBottom}_{p-k}(\cdot)$ denotes the summation of the smallest $p-k$ terms of a given set.*

*Proof.* We first derive Inequality (42).

$$h(\boldsymbol{\gamma}) = \min_{\boldsymbol{z} \in \mathcal{D}_{\boldsymbol{z}}} \mathcal{L}^{\text{saddle}}_{\text{ridge}-\lambda}(\boldsymbol{\gamma}, \boldsymbol{z}) \qquad \text{\color{green}\# $\mathcal{D}_{\boldsymbol{z}} := \{\boldsymbol{z} \mid \sum_{j=1}^{p} z_j \le k, z_j \in [0,1]\}$}$$

$$= \min_{\boldsymbol{z} \in \mathcal{D}_{\boldsymbol{z}}} -\boldsymbol{\gamma}^T \boldsymbol{Q}_\lambda \boldsymbol{\gamma} - \frac{1}{\lambda_2 + \lambda}(\boldsymbol{X}^T\boldsymbol{y} - \boldsymbol{Q}_\lambda\boldsymbol{\gamma})^T diag(\boldsymbol{z})(\boldsymbol{X}^T\boldsymbol{y} - \boldsymbol{Q}_\lambda\boldsymbol{\gamma})$$

*# apply the definition of $\mathcal{L}^{saddle}_{ridge-\lambda}(\boldsymbol{\gamma}, \boldsymbol{z})$*

$$= \min_{\boldsymbol{z} \in \mathcal{D}_{\boldsymbol{z}}} -\boldsymbol{\gamma}^T \boldsymbol{Q}_\lambda \boldsymbol{\gamma} - \frac{1}{\lambda_2 + \lambda}\boldsymbol{d}^T diag(\boldsymbol{z})\boldsymbol{d} \qquad \text{\color{green}\# let $\boldsymbol{d} := (\boldsymbol{X}^T\boldsymbol{y} - \boldsymbol{Q}_\lambda\boldsymbol{\gamma})$}$$

$$= \min_{\boldsymbol{z} \in \mathcal{D}_{\boldsymbol{z}}} -\boldsymbol{\gamma}^T \boldsymbol{Q}_\lambda \boldsymbol{\gamma} - \frac{1}{\lambda_2 + \lambda} \sum_{j=1}^{p} d_j^2 z_j \quad \text{\# use summation notation instead of matrix notation}$$

$$\geq -\boldsymbol{\gamma}^T \boldsymbol{Q}_\lambda \boldsymbol{\gamma} - \frac{1}{\lambda_2 + \lambda} \sum_{j=1}^{p} d_j^2$$

$$\text{\# notice that } z_j \in [0, 1]; \text{ letting } z_j = 1 \text{ for each } j \text{ gives us a lower bound}$$

$$= -\boldsymbol{\gamma}^T \boldsymbol{Q}_\lambda \boldsymbol{\gamma} - \frac{1}{\lambda_2 + \lambda} \boldsymbol{d}^T \boldsymbol{d} \quad \text{\# use matrix notation in stead of summation notation}$$

$$= -\boldsymbol{\gamma}^T \boldsymbol{Q}_\lambda \boldsymbol{\gamma} - \frac{1}{\lambda_2 + \lambda} (\boldsymbol{X}^T \boldsymbol{y} - \boldsymbol{Q}_\lambda \boldsymbol{\gamma})^T (\boldsymbol{X}^T \boldsymbol{y} - \boldsymbol{Q}_\lambda \boldsymbol{\gamma}). \quad \text{\# apply } \boldsymbol{d} := (\boldsymbol{X}^T \boldsymbol{y} - \boldsymbol{Q}_\lambda \boldsymbol{\gamma})$$

This completes the derivation for Inequality (42).

The right-hand side of Inequality (42) is maximized if the gradient with respect to $\boldsymbol{\gamma}$ is $\boldsymbol{0}$.

The gradient can be derived and simplified as:

$$\frac{\partial}{\partial \boldsymbol{\gamma}} \left( -\boldsymbol{\gamma}^T \boldsymbol{Q}_\lambda \boldsymbol{\gamma} - \frac{1}{\lambda_2 + \lambda} (\boldsymbol{X}^T \boldsymbol{y} - \boldsymbol{Q}_\lambda \boldsymbol{\gamma})^T (\boldsymbol{X}^T \boldsymbol{y} - \boldsymbol{Q}_\lambda \boldsymbol{\gamma}) \right)$$

$$\text{\# gradient of the right-hand side of Inequality (42)}$$

$$= \frac{\partial}{\partial \boldsymbol{\gamma}} \left( -\boldsymbol{\gamma}^T \boldsymbol{Q}_\lambda \boldsymbol{\gamma} - \frac{1}{\lambda_2 + \lambda} (\boldsymbol{y}^T \boldsymbol{X} \boldsymbol{X}^T \boldsymbol{y} - 2 \boldsymbol{y}^T \boldsymbol{X} \boldsymbol{Q}_\lambda \boldsymbol{\gamma} + \boldsymbol{\gamma}^T \boldsymbol{Q}_\lambda^T \boldsymbol{Q}_\lambda \boldsymbol{\gamma}) \right)$$

$$\text{\# expand the quadratic term}$$

$$= \left( -2 \boldsymbol{Q}_\lambda \boldsymbol{\gamma} - \frac{1}{\lambda_2 + \lambda} (-2 \boldsymbol{Q}_\lambda^T \boldsymbol{X}^T \boldsymbol{y} + 2 \boldsymbol{Q}_\lambda^T \boldsymbol{Q}_\lambda \boldsymbol{\gamma}) \right)$$

$$\text{\# apply the differential operator to each term}$$

$$= \left( -2 \boldsymbol{Q}_\lambda^T \boldsymbol{\gamma} - \frac{1}{\lambda_2 + \lambda} (-2 \boldsymbol{Q}_\lambda^T \boldsymbol{X}^T \boldsymbol{y} + 2 \boldsymbol{Q}_\lambda^T \boldsymbol{Q}_\lambda \boldsymbol{\gamma}) \right)$$

$$\text{\# change the first term from } \boldsymbol{Q}_\lambda \boldsymbol{\gamma} \text{ to } \boldsymbol{Q}_\lambda^T \boldsymbol{\gamma} \text{ because } \boldsymbol{Q}_\lambda = \boldsymbol{X}^T \boldsymbol{X} - \lambda \boldsymbol{I} \text{ is symmetric}$$

$$= -\frac{2}{\lambda_2 + \lambda} \boldsymbol{Q}_\lambda^T \left( (\lambda_2 + \lambda) \boldsymbol{\gamma} - \boldsymbol{X}^T \boldsymbol{y} + \boldsymbol{Q}_\lambda \boldsymbol{\gamma} \right)$$

$$\text{\# pull the common factor } -\frac{2}{\lambda_2 + \lambda} \boldsymbol{Q}_\lambda^T \text{ out of the parentheses}$$

$$= -\frac{2}{\lambda_2 + \lambda} \boldsymbol{Q}_\lambda^T \left( (\lambda_2 + \lambda) \boldsymbol{\gamma} - \boldsymbol{X}^T \boldsymbol{y} + (\boldsymbol{X}^T \boldsymbol{X} - \lambda \boldsymbol{I}) \boldsymbol{\gamma} \right)$$

$$\text{\# substitute } \boldsymbol{Q}_\lambda^T = \boldsymbol{X}^T \boldsymbol{X} - \lambda \boldsymbol{I} \text{ for the last term}$$

$$= -\frac{2}{\lambda_2 + \lambda} \boldsymbol{Q}_\lambda^T \left( (\lambda_2 + \lambda) \boldsymbol{I} \boldsymbol{\gamma} - \boldsymbol{X}^T \boldsymbol{y} + (\boldsymbol{X}^T \boldsymbol{X} - \lambda \boldsymbol{I}) \boldsymbol{\gamma} \right)$$

$$\text{\# add an identity matrix in the first term}$$

$$= -\frac{2}{\lambda_2 + \lambda} \boldsymbol{Q}_\lambda^T \left( (-\boldsymbol{X}^T \boldsymbol{y} + (\boldsymbol{X}^T \boldsymbol{X} + \lambda_2 \boldsymbol{I}) \boldsymbol{\gamma}) \right). \quad \text{\# add an identity matrix in the first term}$$

Therefore, if we set $(\boldsymbol{X}^T \boldsymbol{X} + \lambda_2 \boldsymbol{I}) \boldsymbol{\gamma} = \boldsymbol{X}^T \boldsymbol{y}$, then the gradient of the right-hand side of Inequality (42) will be $\boldsymbol{0}$, and the optimality condition is achieved. However, this means that $\boldsymbol{\gamma} = (\boldsymbol{X}^T \boldsymbol{X} + \lambda_2 \boldsymbol{I})^{-1} \boldsymbol{X}^T \boldsymbol{y}$, which gives us the minimizer of the ridge regression loss $\mathcal{L}_{\text{ridge}}(\cdot)$.

Lastly, we show that when $\hat{\boldsymbol{\gamma}} = \text{argmin}_{\boldsymbol{\gamma}} \mathcal{L}_{\text{ridge}}(\boldsymbol{\gamma})$, $h(\hat{\boldsymbol{\gamma}}) = \mathcal{L}_{\text{ridge}}(\hat{\boldsymbol{\gamma}}) + (\lambda_2 + \lambda) SumBottom_{p-k}(\{\hat{\gamma}_j^2\})$.

First, let us derive a useful property. We claim that when $\hat{\boldsymbol{\gamma}} = \text{argmin}_{\boldsymbol{\gamma}} \mathcal{L}_{\text{ridge}}(\boldsymbol{\gamma})$, $\boldsymbol{X}^T \boldsymbol{y} - \boldsymbol{Q}_\lambda \hat{\boldsymbol{\gamma}} = (\lambda_2 + \lambda) \hat{\boldsymbol{\gamma}}$. Below, we prove this by showing that when we subtract the two terms, we get $\boldsymbol{0}$.

$$\boldsymbol{X}^T \boldsymbol{y} - \boldsymbol{Q}_\lambda \hat{\boldsymbol{\gamma}} - (\lambda_2 + \lambda) \hat{\boldsymbol{\gamma}}$$

$$= \boldsymbol{X}^T \boldsymbol{y} - (\boldsymbol{X}^T \boldsymbol{X} - \lambda \boldsymbol{I}) \hat{\boldsymbol{\gamma}} - (\lambda_2 + \lambda) \hat{\boldsymbol{\gamma}} \qquad \text{\# substitute } \boldsymbol{Q}_\lambda = \boldsymbol{X}^T \boldsymbol{X} - \lambda \boldsymbol{I}$$

$$= \boldsymbol{X}^T \boldsymbol{y} - (\boldsymbol{X}^T \boldsymbol{X} - \lambda \boldsymbol{I}) \hat{\boldsymbol{\gamma}} - (\lambda_2 + \lambda) \boldsymbol{I} \hat{\boldsymbol{\gamma}} \qquad \text{\# add the identity matrix } \boldsymbol{I}$$

$$= \boldsymbol{X}^T \boldsymbol{y} - (\boldsymbol{X}^T \boldsymbol{X} + \lambda_2 \boldsymbol{I}) \hat{\boldsymbol{\gamma}} \qquad \text{\# aggregate the relevant terms in front of } \hat{\boldsymbol{\gamma}}$$

$$=\boldsymbol{X}^T\boldsymbol{y} - (\boldsymbol{X}^T\boldsymbol{X} + \lambda_2\boldsymbol{I})(\boldsymbol{X}^T\boldsymbol{X} + \lambda_2\boldsymbol{I})^{-1}\boldsymbol{X}^T\boldsymbol{y} \qquad \textcolor{green}{\# \, \hat{\boldsymbol{\gamma}} = \mathrm{argmin}_{\boldsymbol{\gamma}} \, \mathcal{L}_{\mathrm{ridge}}(\boldsymbol{\gamma}) = \boldsymbol{X}^T\boldsymbol{X} + \lambda_2\boldsymbol{I}}$$

$$=\boldsymbol{X}^T\boldsymbol{y} - \boldsymbol{X}^T\boldsymbol{y} \qquad\qquad\qquad\qquad\qquad\qquad \textcolor{green}{\# \, cancel \, out \, \boldsymbol{X}^T\boldsymbol{X} + \lambda_2\boldsymbol{I}}$$

$$=\boldsymbol{0}.$$

Therefore, we have $\boldsymbol{X}^T\boldsymbol{y} - \boldsymbol{Q}_\lambda\hat{\boldsymbol{\gamma}} = (\lambda_2 + \lambda)\hat{\boldsymbol{\gamma}}$ when $\hat{\boldsymbol{\gamma}} = \mathrm{argmin}_{\boldsymbol{\gamma}} \, \mathcal{L}_{\mathrm{ridge}}(\boldsymbol{\gamma})$. With this property, we can derive the final simplified formula for $h(\hat{\boldsymbol{\gamma}})$:

$$h(\hat{\boldsymbol{\gamma}}) = \min_{\boldsymbol{z}\in\mathcal{D}_{\boldsymbol{z}}} \mathcal{L}_{\mathrm{ridge}-\lambda}^{\mathrm{saddle}}(\hat{\boldsymbol{\gamma}}, \boldsymbol{z}) \qquad\qquad \textcolor{green}{\# \, \mathcal{D}_{\boldsymbol{z}} := \{\boldsymbol{z} \mid \sum_{j=1}^p z_j \le k, z_j \in [0,1]\}}$$

$$= \min_{\boldsymbol{z}\in\mathcal{D}_{\boldsymbol{z}}} -\hat{\boldsymbol{\gamma}}^T\boldsymbol{Q}_\lambda\hat{\boldsymbol{\gamma}} - \frac{1}{\lambda_2 + \lambda}(\boldsymbol{X}^T\boldsymbol{y} - \boldsymbol{Q}_\lambda\hat{\boldsymbol{\gamma}})^T diag(\boldsymbol{z})(\boldsymbol{X}^T\boldsymbol{y} - \boldsymbol{Q}_\lambda\hat{\boldsymbol{\gamma}})$$

$$\textcolor{green}{\# \, apply \, the \, definition \, of \, \mathcal{L}_{\mathrm{ridge}-\lambda}^{\mathrm{saddle}}(\hat{\boldsymbol{\gamma}}, \boldsymbol{z})}$$

$$= \min_{\boldsymbol{z}\in\mathcal{D}_{\boldsymbol{z}}} -\hat{\boldsymbol{\gamma}}^T\boldsymbol{Q}_\lambda\hat{\boldsymbol{\gamma}} - \frac{1}{\lambda_2 + \lambda}(\lambda_2 + \lambda)\hat{\boldsymbol{\gamma}}^T diag(\boldsymbol{z})(\lambda_2 + \lambda)\hat{\boldsymbol{\gamma}}$$

$$\textcolor{green}{\# \, apply \, the \, property \, \boldsymbol{X}^T\boldsymbol{y} - \boldsymbol{Q}_\lambda\hat{\boldsymbol{\gamma}} = (\lambda_2 + \lambda)\hat{\boldsymbol{\gamma}}}$$

$$= \min_{\boldsymbol{z}\in\mathcal{D}_{\boldsymbol{z}}} -\hat{\boldsymbol{\gamma}}^T\boldsymbol{Q}_\lambda\hat{\boldsymbol{\gamma}} - (\lambda_2 + \lambda)\hat{\boldsymbol{\gamma}}^T diag(\boldsymbol{z})\hat{\boldsymbol{\gamma}} \qquad\qquad \textcolor{green}{\# \, cancel \, out \, \lambda_2 + \lambda}$$

$$= \min_{\boldsymbol{z}\in\mathcal{D}_{\boldsymbol{z}}} -\hat{\boldsymbol{\gamma}}^T(\boldsymbol{X}^T\boldsymbol{X} - \lambda\boldsymbol{I})\hat{\boldsymbol{\gamma}} - (\lambda_2 + \lambda)\hat{\boldsymbol{\gamma}}^T diag(\boldsymbol{z})\hat{\boldsymbol{\gamma}} \qquad \textcolor{green}{\# \, apply \, \boldsymbol{Q}_\lambda = \boldsymbol{X}^T\boldsymbol{X} - \lambda\boldsymbol{I}}$$

$$= \min_{\boldsymbol{z}\in\mathcal{D}_{\boldsymbol{z}}} -\hat{\boldsymbol{\gamma}}^T(\boldsymbol{X}^T\boldsymbol{X} - \lambda\boldsymbol{I})\hat{\boldsymbol{\gamma}} - (\lambda_2 + \lambda)\sum_{j=1}^p \hat{\gamma}_j^2 z_j$$

$$\textcolor{green}{\# \, use \, summation \, instead \, of \, matrix \, notation}$$

$$= -\hat{\boldsymbol{\gamma}}^T(\boldsymbol{X}^T\boldsymbol{X} - \lambda\boldsymbol{I})\hat{\boldsymbol{\gamma}} - (\lambda_2 + \lambda)\max_{\boldsymbol{z}\in\mathcal{D}_{\boldsymbol{z}}}\sum_{j=1}^p \hat{\gamma}_j^2 z_j$$

$$\textcolor{green}{\# \, move \, \min(\cdot) \, inside \, the \, negative \, sign \, and \, convert \, it \, to \, \max(\cdot)}$$

$$= -\hat{\boldsymbol{\gamma}}^T(\boldsymbol{X}^T\boldsymbol{X} - \lambda\boldsymbol{I})\hat{\boldsymbol{\gamma}} - (\lambda_2 + \lambda)TopSum_k\{\hat{\gamma}_j^2\}$$

$$\textcolor{green}{\# \, TopSum_k(\cdot) \, denotes \, summation \, over \, the \, top \, k \, terms}$$

$$= -\hat{\boldsymbol{\gamma}}^T(\boldsymbol{X}^T\boldsymbol{X} - \lambda\boldsymbol{I})\hat{\boldsymbol{\gamma}} - (\lambda_2 + \lambda)(\sum_{j=1}^p \hat{\gamma}_j^2 - BottomSum_{p-k}\{\hat{\gamma}_j^2\})$$

$$\textcolor{green}{\# \, BottomSum_{p-k}(\cdot) \, denotes \, summation \, over \, the \, bottom \, p-k \, terms}$$

$$= -\hat{\boldsymbol{\gamma}}^T(\boldsymbol{X}^T\boldsymbol{X} - \lambda\boldsymbol{I})\hat{\boldsymbol{\gamma}} + (\lambda_2 + \lambda)BottomSum_{p-k}\{\hat{\gamma}_j^2\} - (\lambda_2 + \lambda)\sum_{j=1}^p \hat{\gamma}_j^2$$

$$\textcolor{green}{\# \, distribute \, into \, the \, parentheses}$$

$$= -\hat{\boldsymbol{\gamma}}^T(\boldsymbol{X}^T\boldsymbol{X} - \lambda\boldsymbol{I})\hat{\boldsymbol{\gamma}} + (\lambda_2 + \lambda)BottomSum_{p-k}\{\hat{\gamma}_j^2\} - (\lambda_2 + \lambda)\hat{\boldsymbol{\gamma}}^T\hat{\boldsymbol{\gamma}}.$$

$$\textcolor{green}{\# \, use \, matrix \, notation \, instead \, of \, summation \, notation}$$

We note that $-\hat{\boldsymbol{\gamma}}^T(\boldsymbol{X}^T\boldsymbol{X} - \lambda\boldsymbol{I})\hat{\boldsymbol{\gamma}} - (\lambda_2 + \lambda)\hat{\boldsymbol{\gamma}}^T\hat{\boldsymbol{\gamma}} = \mathcal{L}_{\mathrm{ridge}}(\hat{\boldsymbol{\gamma}})$. To show this, we subtract the two terms and prove that the subtraction is equal to 0 below:

$$-\hat{\boldsymbol{\gamma}}^T(\boldsymbol{X}^T\boldsymbol{X} - \lambda\boldsymbol{I})\hat{\boldsymbol{\gamma}} - (\lambda_2 + \lambda)\hat{\boldsymbol{\gamma}}^T\hat{\boldsymbol{\gamma}} - \mathcal{L}_{\mathrm{ridge}}(\hat{\boldsymbol{\gamma}})$$

$$= -\hat{\boldsymbol{\gamma}}^T(\boldsymbol{X}^T\boldsymbol{X} - \lambda\boldsymbol{I})\hat{\boldsymbol{\gamma}} - (\lambda_2 + \lambda)\hat{\boldsymbol{\gamma}}^T\hat{\boldsymbol{\gamma}} - (\hat{\boldsymbol{\gamma}}^T\boldsymbol{X}^T\boldsymbol{X}\hat{\boldsymbol{\gamma}} - 2\boldsymbol{y}^T\boldsymbol{X}\hat{\boldsymbol{\gamma}} + \lambda_2\hat{\boldsymbol{\gamma}}^T\hat{\boldsymbol{\gamma}})$$

$$\textcolor{green}{\# \, plug \, in \, the \, formula \, for \, \mathcal{L}_{\mathrm{ridge}}(\hat{\boldsymbol{\gamma}})}$$

$$= -2\left(\hat{\boldsymbol{\gamma}}^T(\boldsymbol{X}^T\boldsymbol{X} + \lambda_2\boldsymbol{I})\hat{\boldsymbol{\gamma}} - \boldsymbol{y}^T\boldsymbol{X}\hat{\boldsymbol{\gamma}}\right) \qquad\qquad \textcolor{green}{\# \, simplify \, the \, linear \, algebra}$$

$$= -2\left(\hat{\boldsymbol{\gamma}}^T(\boldsymbol{X}^T\boldsymbol{X} + \lambda_2\boldsymbol{I})(\boldsymbol{X}^T\boldsymbol{X} + \lambda_2\boldsymbol{I})^{-1}\boldsymbol{X}^T\boldsymbol{y} - \boldsymbol{y}^T\boldsymbol{X}\hat{\boldsymbol{\gamma}}\right)$$

$$\textcolor{green}{\# \, plug \, in \, the \, equation \, \hat{\boldsymbol{\gamma}} = (\boldsymbol{X}^T\boldsymbol{X} + \lambda_2\boldsymbol{I})^{-1}\boldsymbol{X}^T\boldsymbol{y} \, because \, \hat{\boldsymbol{\gamma}} \, is \, the \, minimizer \, of \, \mathcal{L}_{\mathrm{ridge}}(\cdot)}$$

$$= -2\left(\hat{\boldsymbol{\gamma}}^T\boldsymbol{X}^T\boldsymbol{y} - \boldsymbol{y}^T\boldsymbol{X}\hat{\boldsymbol{\gamma}}\right) \qquad\qquad\qquad \textcolor{green}{\# \, cancel \, out \, \boldsymbol{X}^T\boldsymbol{X} + \lambda_2\boldsymbol{I}}$$

$$= 0.$$

Using the property that $-\hat{\boldsymbol{\gamma}}^T(\boldsymbol{X}^T\boldsymbol{X} - \lambda\boldsymbol{I})\hat{\boldsymbol{\gamma}} - (\lambda_2 + \lambda)\hat{\boldsymbol{\gamma}}^T\hat{\boldsymbol{\gamma}} = \mathcal{L}_{\mathrm{ridge}}(\hat{\boldsymbol{\gamma}})$, we have:

$$h(\hat{\boldsymbol{\gamma}}) = -\hat{\boldsymbol{\gamma}}^T(\boldsymbol{X}^T\boldsymbol{X} - \lambda\boldsymbol{I})\hat{\boldsymbol{\gamma}} - (\lambda_2 + \lambda)\hat{\boldsymbol{\gamma}}^T\hat{\boldsymbol{\gamma}} + (\lambda_2 + \lambda)BottomSum_{p-k}\{\hat{\gamma}_j^2\}$$

$$=\mathcal{L}_{\text{ridge}}(\hat{\boldsymbol{\gamma}}) + (\lambda_2 + \lambda)BottomSum_{p-k}\{\hat{\gamma}_j^2\}.$$

This completes the proof for Theorem 3.2.

$\square$

## C.3 Proof of Theorem 3.3

**Theorem 3.3.** *Let $F(\boldsymbol{\gamma}) = \boldsymbol{\gamma}^T \boldsymbol{Q}_\lambda \boldsymbol{\gamma}$ and $G(\boldsymbol{p}) = \frac{1}{\lambda_2 + \lambda} SumTop_k(\{p_j^2\})$. Then the solution for the problem $\boldsymbol{\gamma}^{t+1} = \text{argmin}_{\boldsymbol{\gamma}} F(\boldsymbol{\gamma}) + \frac{\rho}{2}\|\boldsymbol{Q}_\lambda \boldsymbol{\gamma} + \boldsymbol{p}^t - \boldsymbol{X}^T \boldsymbol{y} + \boldsymbol{q}^t\|_2^2$ is*

$$\boldsymbol{\gamma}^{t+1} = (\frac{2}{\rho}\boldsymbol{I} + \boldsymbol{Q}_\lambda)^{-1}(\boldsymbol{X}^T \boldsymbol{y} - \boldsymbol{p}^t - \boldsymbol{q}^t). \tag{43}$$

*Furthermore, let $\boldsymbol{a} = \boldsymbol{X}^T \boldsymbol{y} - \boldsymbol{\theta}^{t+1} - \boldsymbol{q}^t$ and $\mathcal{J}$ be the indices of the top k terms of $\{|a_j|\}$. The solution for the problem $\boldsymbol{p}^{t+1} = \text{argmin}_{\boldsymbol{p}} G(\boldsymbol{p}) + \frac{\rho}{2}\|\boldsymbol{\theta}^{t+1} + \boldsymbol{p} - \boldsymbol{X}^T \boldsymbol{y} + \boldsymbol{q}^t\|_2^2$ is $p_j^{t+1} = sign(a_j) \cdot \hat{v}_j$,*

*where*
$$\hat{\boldsymbol{v}} = \text{argmin}_{\boldsymbol{v}} \sum_{j=1}^{p} w_j (v_j - b_j)^2 \text{ s.t. } v_i \geq v_l \text{ if } |a_i| \geq |a_l| \tag{44}$$

$$w_j = \begin{cases} 1 & \text{if } j \notin \mathcal{J} \\ 1 + \frac{2}{\rho(\lambda_2 + \lambda)} & \text{otherwise} \end{cases}, \qquad b_j = \frac{|a_j|}{w_j}.$$

*Problem (44) is an isotonic regression problem and can be efficiently solved in linear time [12, 21].*

*Proof.* **Part I**. We first show why Equation (43) is the analytic solution to the first optimization problem $\boldsymbol{\gamma}^{t+1} = \text{argmin}_{\boldsymbol{\gamma}} F(\boldsymbol{\gamma}) + \frac{\rho}{2}\|\boldsymbol{Q}_\lambda \boldsymbol{\gamma} + \boldsymbol{p}^t - \boldsymbol{X}^T \boldsymbol{y} + \boldsymbol{q}^t\|_2^2$.

The loss function we want to minimize is (writing out $F(\boldsymbol{\gamma}) = \boldsymbol{\gamma}^T \boldsymbol{Q}_\lambda \boldsymbol{\gamma}$ explicitly)

$$\boldsymbol{\gamma}^T \boldsymbol{Q}_\lambda \boldsymbol{\gamma} + \frac{\rho}{2}\|\boldsymbol{Q}_\lambda \boldsymbol{\gamma} + \boldsymbol{p}^t - \boldsymbol{X}^T \boldsymbol{y} + \boldsymbol{q}^t\|_2^2$$
$$=\boldsymbol{\gamma}^T \boldsymbol{Q}_\lambda \boldsymbol{\gamma} + \frac{\rho}{2}(\boldsymbol{\gamma}^T \boldsymbol{Q}_\lambda^T \boldsymbol{Q}_\lambda \boldsymbol{\gamma} + 2(\boldsymbol{p}^t - \boldsymbol{X}^T \boldsymbol{y} + \boldsymbol{q}^t)^T \boldsymbol{Q}_\lambda \boldsymbol{\gamma} + \|\boldsymbol{p}^t - \boldsymbol{X}^T \boldsymbol{y} + \boldsymbol{q}^t\|_2^2).$$
*# write out the square term as a sum of three terms*

This loss function is minimized if we take the gradient with respect to $\boldsymbol{\gamma}$ and set it to $\boldsymbol{0}$:

$$\boldsymbol{0} = 2\boldsymbol{Q}_\lambda \boldsymbol{\gamma} + \frac{\rho}{2}(2\boldsymbol{Q}_\lambda^T \boldsymbol{Q}_\lambda \boldsymbol{\gamma} + 2\boldsymbol{Q}_\lambda^T(\boldsymbol{p}^t - \boldsymbol{X}^T \boldsymbol{y} + \boldsymbol{q}^t))$$
*# take the gradient w.r.t. $\boldsymbol{\gamma}$ and set it to $\boldsymbol{0}$; ignore the last term because it is a constant*
$$= 2\boldsymbol{Q}_\lambda \boldsymbol{\gamma} + \rho(\boldsymbol{Q}_\lambda^T \boldsymbol{Q}_\lambda \boldsymbol{\gamma} + \boldsymbol{Q}_\lambda^T(\boldsymbol{p}^t - \boldsymbol{X}^T \boldsymbol{y} + \boldsymbol{q}^t)) \qquad \text{\# distribute 2 inside parentheses}$$
$$= 2\boldsymbol{Q}_\lambda^T \boldsymbol{\gamma} + \rho(\boldsymbol{Q}_\lambda^T \boldsymbol{Q}_\lambda \boldsymbol{\gamma} + \boldsymbol{Q}_\lambda^T(\boldsymbol{p}^t - \boldsymbol{X}^T \boldsymbol{y} + \boldsymbol{q}^t)) \qquad \text{\# } \boldsymbol{Q}_\lambda \text{ is symmetric}$$
$$= \boldsymbol{Q}_\lambda^T(2\boldsymbol{\gamma} + \rho(\boldsymbol{Q}_\lambda \boldsymbol{\gamma} + (\boldsymbol{p}^t - \boldsymbol{X}^T \boldsymbol{y} + \boldsymbol{q}^t))) \qquad \text{\# take out the common factor } \boldsymbol{Q}_\lambda^T$$
$$= \boldsymbol{Q}_\lambda^T(2\boldsymbol{I}\boldsymbol{\gamma} + \rho\boldsymbol{Q}_\lambda \boldsymbol{\gamma} + \rho(\boldsymbol{p}^t - \boldsymbol{X}^T \boldsymbol{y} + \boldsymbol{q}^t)) \qquad \text{\# multiply by identity matrix } \boldsymbol{I}$$
$$= \boldsymbol{Q}_\lambda^T((2\boldsymbol{I} + \rho\boldsymbol{Q}_\lambda)\boldsymbol{\gamma} + \rho(\boldsymbol{p}^t - \boldsymbol{X}^T \boldsymbol{y} + \boldsymbol{q}^t)) \qquad \text{\# aggregate the coefficient for } \boldsymbol{\gamma}$$
$$= \rho\boldsymbol{Q}_\lambda^T((\frac{2}{\rho}\boldsymbol{I} + \boldsymbol{Q}_\lambda)\boldsymbol{\gamma} + (\boldsymbol{p}^t - \boldsymbol{X}^T \boldsymbol{y} + \boldsymbol{q}^t)). \qquad \text{\# take out the common factor } \rho$$

A sufficient condition for the gradient to be $\boldsymbol{0}$ is to have

$$(\frac{2}{\rho}\boldsymbol{I} + \boldsymbol{Q}_\lambda)\boldsymbol{\gamma} + (\boldsymbol{p}^t - \boldsymbol{X}^T \boldsymbol{y} + \boldsymbol{q}^t) = \boldsymbol{0}.$$

If we solve the above equation, we obtain Equation (43):

$$\boldsymbol{\gamma} = (\frac{2}{\rho}\boldsymbol{I} + \boldsymbol{Q}_\lambda)^{-1}(\boldsymbol{X}^T \boldsymbol{y} - \boldsymbol{p}^t - \boldsymbol{q}^t).$$

**Part II**. Next, we show why $p_j^{t+1} = sign(a_j) \cdot \hat{v}_j$ together with Equation (44) gives us the solution to the second optimization problem $\boldsymbol{p}^{t+1} = \text{argmin}_{\boldsymbol{p}} G(\boldsymbol{p}) + \frac{\rho}{2}\|\boldsymbol{\theta}^{t+1} + \boldsymbol{p} - \boldsymbol{X}^T \boldsymbol{y} + \boldsymbol{q}^t\|_2^2$.

If we write out $G(\boldsymbol{p}) = \frac{1}{\lambda_2 + \lambda}\text{SumTop}_k(\{p_j^2\})$ explicitly and use $\boldsymbol{a} = \boldsymbol{X}^T\boldsymbol{y} - \boldsymbol{\theta}^{t+1} - \boldsymbol{q}^t$ as a shorthand to represent the constant, the objective function can be rewritten as:

$$\boldsymbol{p}^{t+1} = \operatorname*{argmin}_{\boldsymbol{p}} \frac{1}{\lambda_2 + \lambda}\text{SumTop}_k(\{p_j^2\}) + \frac{\rho}{2}\|\boldsymbol{p} - \boldsymbol{a}\|_2^2$$

$$= \operatorname*{argmin}_{\boldsymbol{p}} \frac{2}{\rho(\lambda_2 + \lambda)}\text{SumTop}_k(\{p_j^2\}) + \|\boldsymbol{p} - \boldsymbol{a}\|_2^2.$$

# *multiplying by $\frac{2}{\rho}$ doesn't change the optimal solution*

According to Proposition 3.1 (b) in [30], the optimal solution $\boldsymbol{p}^{t+1}$ has the property that $\text{sign}(p_j^{t+1}) = \text{sign}(a_j)$.

Inspired by this sign-preserving property, let us make a change of variable by writing $p_j = \text{sign}(a_j) \cdot v_j$. Due to this change of variable, our optimal solution $\boldsymbol{p}^{t+1}$ can be obtained by solving the following optimization problem:

$$p_j^{t+1} = \text{sign}(a_j) \cdot \hat{v}_j$$

$$\text{where } \hat{\boldsymbol{v}} = \operatorname*{argmin}_{\boldsymbol{v}} \frac{2}{\rho(\lambda_2 + \lambda)}\text{SumTop}_k(\{v_j^2\}) + \sum_{j=1}^{p}(v_j - |a_j|)^2. \qquad (45)$$

Note that in Problem (45), if we add the extra constraints $v_j \geq 0$, this will not change the optimal solution. We can show this by the argument of contradiction. Suppose there exists an optimal solution $\hat{\boldsymbol{v}}$ and an index $j$ with $\hat{v}_j < 0$. Then, if we flip the sign of $\hat{v}_j$, then the objective loss in Problem (45) is decreased. This contradicts with the assumption that $\hat{\boldsymbol{v}}$ is the optimal solution. Therefore, we are safe to add the extra constraints $v_j \geq 0$ into Problem (45) without cutting off the optimal solution.

Thus, our optimal solution $\boldsymbol{p}^{t+1}$ becomes:

$$p_j^{t+1} = \text{sign}(a_j) \cdot \hat{v}_j$$

$$\text{where } \hat{\boldsymbol{v}} = \operatorname*{argmin}_{\boldsymbol{v}} \frac{2}{\rho(\lambda_2 + \lambda)}\text{SumTop}_k(\{v_j^2\}) + \sum_{j=1}^{p}(v_j - |a_j|)^2 \text{ s.t. } v_j \geq 0 \text{ for } \forall j. \qquad (46)$$

Next, according to Proposition 3.1 (c) in [30], we have $\hat{v}_i \geq \hat{v}_l$ if $|a_i| \geq |a_l|$ in Problem (46). Thus, we add these extra constraints into Problem (46) without changing the optimal solution and $\boldsymbol{p}^{t+1}$ becomes:

$$p_j^{t+1} = \text{sign}(a_j) \cdot \hat{v}_j$$

$$\text{where } \hat{\boldsymbol{v}} = \operatorname*{argmin}_{\boldsymbol{v}} \frac{2}{\rho(\lambda_2 + \lambda)}\text{SumTop}_k(\{v_j^2\}) + \sum_{j=1}^{p}(v_j - |a_j|)^2 \qquad (47)$$

$$\text{s.t. } v_j \geq 0 \text{ for } \forall j, \text{ and } v_i \geq v_j \text{ if } |a_i| \geq |a_j|. \text{ for } \forall i, j.$$

Note that in Problem (47), because of the extra constraints $|a_i| \geq |a_j|$ for $\forall i, j$, we can rewrite $\text{SumTop}_k(\{v_j^2\}) = \sum_{j \in \mathcal{J}} v_j^2$ where $\mathcal{J}$ is the set of indices of the top $k$ terms of $\{|a_j|\}$'s. Then the optimal solution $\boldsymbol{p}^{t+1}$ becomes:

$$p_j^{t+1} = \text{sign}(a_j) \cdot \hat{v}_j$$

$$\text{where } \hat{\boldsymbol{v}} = \operatorname*{argmin}_{\boldsymbol{v}} \frac{2}{\rho(\lambda_2 + \lambda)} \sum_{j \in \mathcal{J}} v_j^2 + \sum_{j=1}^{p}(v_j - |a_j|)^2 \qquad (48)$$

$$\text{s.t. } v_j \geq 0 \text{ for } \forall j, \text{ and } v_i \geq v_j \text{ if } |a_i| \geq |a_j|. \text{ for } \forall i, j.$$

Next, we drop the constraints $v_j \geq 0$ for $\forall j$ as this does not change the optimal solution, using the same argument of contradiction we have shown before. Therefore, our optimal solution $\boldsymbol{p}^{t+1}$ can be obtained in the following way:

$$p_j^{t+1} = \text{sign}(a_j) \cdot \hat{v}_j$$

$$\text{where } \hat{\boldsymbol{v}} = \operatorname*{argmin}_{\boldsymbol{v}} \frac{2}{\rho(\lambda_2 + \lambda)} \sum_{j \in \mathcal{J}} v_j^2 + \sum_{j=1}^{p}(v_j - |a_j|)^2 \qquad (49)$$

$$\text{s.t. } v_i \geq v_j \text{ if } |a_i| \geq |a_j|. \text{ for } \forall i, j.$$

We can obtain the formulation in Problem (44) by performing some linear algebra manipulations as follows:

$$\operatorname*{argmin}_{\boldsymbol{v}} \frac{2}{\rho(\lambda_2 + \lambda)} \sum_{j \in \mathcal{J}} v_j^2 + \sum_{j=1}^{p} (v_j - |a_j|)^2$$

$$= \operatorname*{argmin}_{\boldsymbol{v}} \sum_{j \in \mathcal{J}} \left( \frac{2}{\rho(\lambda_2 + \lambda)} v_j^2 + (v_j - |a_j|)^2 \right) + \sum_{j \notin \mathcal{J}} (v_j - |a_j|)^2$$

# divide the summation into two groups based on the set $\mathcal{J}$

$$= \operatorname*{argmin}_{\boldsymbol{v}} \sum_{j \in \mathcal{J}} \left( \left( \frac{2}{\rho(\lambda_2 + \lambda)} + 1 \right) v_j^2 - 2|a_j|v_j + |a_j|^2 \right) + \sum_{j \notin \mathcal{J}} (v_j - |a_j|)^2$$

# combine the coefficients for $v_j^2$

$$= \operatorname*{argmin}_{\boldsymbol{v}} \sum_{j \in \mathcal{J}} \left( \left( \frac{2}{\rho(\lambda_2 + \lambda)} + 1 \right) \left( v_j - \frac{|a_j|}{\frac{2}{\rho(\lambda_2+\lambda)} + 1} \right)^2 - \frac{|a_j|^2}{\frac{2}{\rho(\lambda_2+\lambda)} + 1} + |a_j|^2 \right) + \sum_{j \notin \mathcal{J}} (v_j - |a_j|)^2$$

# complete the square

$$= \operatorname*{argmin}_{\boldsymbol{v}} \sum_{j \in \mathcal{J}} \left( \left( \frac{2}{\rho(\lambda_2 + \lambda)} + 1 \right) \left( v_j - \frac{|a_j|}{\frac{2}{\rho(\lambda_2+\lambda)} + 1} \right)^2 \right) + \sum_{j \notin \mathcal{J}} (v_j - |a_j|)^2$$

# get rid of the constant terms which do not affect the optimal solution

$$= \operatorname*{argmin}_{\boldsymbol{v}} \sum_{j \in \mathcal{J}} \left( w_j \left( v_j - \frac{|a_j|}{w_j} \right)^2 \right) + \sum_{j \notin \mathcal{J}} w_j \left( v_j - \frac{|a_j|}{w_j} \right)^2$$

# $w_j = \frac{2}{\rho(\lambda_2+\lambda)} + 1$ if $j \in \mathcal{J}$ and $w_j = 1$ if $j \notin \mathcal{J}$

$$= \operatorname*{argmin}_{\boldsymbol{v}} \sum_{j=1}^{p} w_j (v_j - b_j)^2.$$

# let $b_j = \frac{|a_j|}{w_j}$ and combine the two summations into a single summation

Using this new property, we can obtain the final result stated in the theorem for our optimal solution $\boldsymbol{p}^{t+1}$ to the second optimization problem:

$$p_j^{t+1} = \operatorname{sign}(a_j) \cdot \hat{v}_j$$

$$\text{where } \hat{\boldsymbol{v}} = \operatorname*{argmin}_{\boldsymbol{v}} \sum_{j=1}^{p} w_j (v_j - b_j)^2 \text{ s.t. } v_i \geq v_j \text{ if } |a_i| \geq |a_j| \text{ for } \forall i, j \qquad (50)$$

$$w_j = \begin{cases} 1 & \text{if } j \notin \mathcal{J} \\ \frac{2}{\rho(\lambda_2+\lambda)} + 1 & \text{otherwise} \end{cases}, \qquad b_j = \frac{|a_j|}{w_j}.$$

Therefore, the task of evaluating the proximal operator is essentially reduced to solving an isotonic regression problem. We would like to acknowledge that the connection between $SumTop_k(\cdot)$ and isotonic regression has also been recently discovered by [55]. However, the context and applications are totally different. While [55] discovers this relationship in the context of training neural networks, we find this connection in the context of ADMM and perspective relaxation of k-sparse ridge regression. Mathematically, our objective function is also different as our objective function contains a linear term of $\boldsymbol{p}$ in the proximal operator evaluation.

$\square$

### C.4 Proof of Theorem 3.4

**Theorem 3.4.** *Let us define a k-sparse vector pair domain to be $\Omega_k := \{(\boldsymbol{x}, \boldsymbol{y}) \in \mathbb{R}^p \times \mathbb{R}^p : \|\boldsymbol{x}\|_0 \leq k, \|\boldsymbol{y}\|_0 \leq k, \|\boldsymbol{x} - \boldsymbol{y}\|_0 \leq k\}$. Any $M_1$ satisfying $f(\boldsymbol{y}) \leq f(\boldsymbol{x}) + \nabla f(\boldsymbol{x})^T (\boldsymbol{y} - \boldsymbol{x}) + \frac{M_1}{2} \|\boldsymbol{y} - \boldsymbol{x}\|_2^2$ for all $(\boldsymbol{x}, \boldsymbol{y}) \in \Omega_1$ is called a restricted smooth parameter with support size 1, and any $m_{2k}$ satisfying*

$f(\boldsymbol{y}) \geq f(\boldsymbol{x}) + \nabla f(\boldsymbol{x})^T(\boldsymbol{y}-\boldsymbol{x}) + \frac{m_{2k}}{2}\|\boldsymbol{y}-\boldsymbol{x}\|_2^2$ *for all* $(\boldsymbol{x},\boldsymbol{y}) \in \Omega_{2k}$ *is called a restricted strongly convex parameter with support size* $2k$. *If* $\hat{\boldsymbol{\beta}}$ *is our heuristic solution by the beam-search method, and* $\boldsymbol{\beta}^*$ *is the optimal solution, then:*

$$\mathcal{L}_{ridge}(\boldsymbol{\beta}^*) \leq \mathcal{L}_{ridge}(\hat{\boldsymbol{\beta}}) \leq (1 - e^{-m_{2k}/M_1})\mathcal{L}_{ridge}(\boldsymbol{\beta}^*). \tag{51}$$

*Proof.* The left-hand-side inequality $\mathcal{L}_{\text{ridge}}(\boldsymbol{\beta}^*) \leq \mathcal{L}_{\text{ridge}}(\hat{\boldsymbol{\beta}})$ is obvious because the optimal loss should always be less than or equal to the loss found by our heuristic method. Therefore, we focus on the right-hand-side inquality $\mathcal{L}_{\text{ridge}}(\hat{\boldsymbol{\beta}}) \leq (1 - e^{-m_{2k}/M_1})\mathcal{L}_{\text{ridge}}(\boldsymbol{\beta}^*)$.

We first give an overview of our proof strategy, which is similar to the proof in [29], but our decay factor $m_{2k}/M_1$ in the exponent is much larger than theirs, which is $m_{2k}/M_{2k}$ (we can obtain the definition of $M_{2k}$ by extending the definition of $M_1$ to the support size $2k$). Let $\mathcal{L}_{\text{ridge}}(\boldsymbol{\beta}^t)$ and $\mathcal{L}_{\text{ridge}}(\boldsymbol{\beta}^{t+1})$ be the losses at the $t$-th and $(t+1)$-th iterations of our beam search algorithm. We first derive a lower bound formula on the decrease of the loss function $\mathcal{L}_{\text{ridge}}(\cdot)$ between two successive iterations, *i.e.*, $\mathcal{L}_{\text{ridge}}(\boldsymbol{\beta}^t) - \mathcal{L}_{\text{ridge}}(\boldsymbol{\beta}^{t+1})$ . We will then show that this quantity of decrease can be expressed in terms of the parameters $M_1$ and $m_{2k}$ defined in the statement of the theorem. Finally, we use the relationship between the decrease of the loss and $M_1$ and $m_{2k}$ to derive the right-hand inequality in Equation (51).

*Step 1: Decrease of Loss between Two Successive Iterations*:

For an arbitrary coordinate $j$ that is not part of the support of $\boldsymbol{\beta}^t$, or more explicitly $\beta_j^t = 0$, if we change the coefficient on the $j$-th coordinate $\boldsymbol{\beta}^t$ from 0 to $\alpha$, the loss becomes:

$\mathcal{L}_{\text{ridge}}(\boldsymbol{\beta}^t + \alpha\boldsymbol{e}_j)$      *# $\boldsymbol{e}_j$ denotes a vector with $j$-th entry to be 1 and 0 otherwise*

$=(\boldsymbol{y} - \boldsymbol{X}(\boldsymbol{\beta}^t + \alpha\boldsymbol{e}_j))^T(\boldsymbol{y} - \boldsymbol{X}(\boldsymbol{\beta}^t + \alpha\boldsymbol{e}_j)) + \lambda_2(\boldsymbol{\beta}^t + \alpha\boldsymbol{e}_j)^T(\boldsymbol{\beta}^t + \alpha\boldsymbol{e}_j)$
     *# apply the definition of ridge regression loss*

$=(\boldsymbol{y} - \boldsymbol{X}\boldsymbol{\beta}^t - \alpha\boldsymbol{X}\boldsymbol{e}_j)^T(\boldsymbol{y} - \boldsymbol{X}\boldsymbol{\beta}^t - \alpha\boldsymbol{X}\boldsymbol{e}_j) + \lambda_2(\boldsymbol{\beta}^t + \alpha\boldsymbol{e}_j)^T(\boldsymbol{\beta}^t + \alpha\boldsymbol{e}_j)$
     *# distribute into the parentheses*

$=(\boldsymbol{y} - \boldsymbol{X}\boldsymbol{\beta}^t - \alpha\boldsymbol{X}_{:j})^T(\boldsymbol{y} - \boldsymbol{X}\boldsymbol{\beta}^t - \alpha\boldsymbol{X}_{:j}) + \lambda_2(\boldsymbol{\beta}^t + \alpha\boldsymbol{e}_j)^T(\boldsymbol{\beta}^t + \alpha\boldsymbol{e}_j)$
     *# $\boldsymbol{X}_{:j}$ denotes the $j$-th column of $\boldsymbol{X}$*

$=(\boldsymbol{y} - \boldsymbol{X}\boldsymbol{\beta}^t)^T(\boldsymbol{y} - \boldsymbol{X}\boldsymbol{\beta}^t) - 2\alpha(\boldsymbol{y} - \boldsymbol{X}\boldsymbol{\beta}^t)^T\boldsymbol{X}_{:j} + \alpha^2\boldsymbol{X}_{:j}^T\boldsymbol{X}_{:j} + \lambda_2\left[(\boldsymbol{\beta}^t)^T\boldsymbol{\beta}^t + 2\alpha(\boldsymbol{\beta}^t)^T\boldsymbol{e}_j + \alpha^2\right]$
     *# expand each quadratic term into three summation terms*

$=(\boldsymbol{y} - \boldsymbol{X}\boldsymbol{\beta}^t)^T(\boldsymbol{y} - \boldsymbol{X}\boldsymbol{\beta}^t) - 2\alpha(\boldsymbol{y} - \boldsymbol{X}\boldsymbol{\beta}^t)^T\boldsymbol{X}_{:j} + \alpha^2\boldsymbol{X}_{:j}^T\boldsymbol{X}_{:j} + \lambda_2\left[(\boldsymbol{\beta}^t)^T\boldsymbol{\beta}^t + 2\alpha\beta_j^t + \alpha^2\right]$
     *# $(\boldsymbol{\beta}^t)^T\boldsymbol{e}_j = \beta_j^t$; $\boldsymbol{e}_j$ takes the $j$-th component of $\boldsymbol{\beta}^t$*

$=(\boldsymbol{y} - \boldsymbol{X}\boldsymbol{\beta}^t)^T(\boldsymbol{y} - \boldsymbol{X}\boldsymbol{\beta}^t) - 2\alpha(\boldsymbol{y} - \boldsymbol{X}\boldsymbol{\beta}^t)^T\boldsymbol{X}_{:j} + \alpha^2\boldsymbol{X}_{:j}^T\boldsymbol{X}_{:j} + \lambda_2\left[(\boldsymbol{\beta}^t)^T\boldsymbol{\beta}^t + \alpha^2\right]$
     *# $\beta_j^t = 0$ because we are optimizing on a coordinate with coefficient equal to 0*

$=(\boldsymbol{y} - \boldsymbol{X}\boldsymbol{\beta}^t)^T(\boldsymbol{y} - \boldsymbol{X}\boldsymbol{\beta}^t) + \lambda_2(\boldsymbol{\beta}^t)^T\boldsymbol{\beta}^t + (\|\boldsymbol{X}_{:j}\|_2^2 + \lambda_2)\alpha^2 - 2((\boldsymbol{y} - \boldsymbol{X}\boldsymbol{\beta}^t)^T\boldsymbol{X}_{:j})\alpha$
     *# collect relevant coefficients for $\alpha^2$ and $\alpha$*

$=\mathcal{L}_{\text{ridge}}(\boldsymbol{\beta}^t) + (\|\boldsymbol{X}_{:j}\|_2^2 + \lambda_2)\alpha^2 - 2((\boldsymbol{y} - \boldsymbol{X}\boldsymbol{\beta}^t)^T\boldsymbol{X}_{:j})\alpha$
     *# simplify the first two terms using $\mathcal{L}_{\text{ridge}}(\cdot)$*

$=\mathcal{L}_{\text{ridge}}(\boldsymbol{\beta}^t) + (\|\boldsymbol{X}_{:j}\|_2^2 + \lambda_2)\alpha^2 + \nabla_j\mathcal{L}_{\text{ridge}}(\boldsymbol{\beta}^t)\alpha$
     *# the $j$-th partial derivative is $\nabla_j\mathcal{L}_{\text{ridge}}(\boldsymbol{\beta}^t) = 2(\boldsymbol{X}\boldsymbol{\beta}^t - \boldsymbol{y})^T\boldsymbol{X}_{:j}$ according to Lemma B.7 with $\beta_j^t = 0$*

$=\mathcal{L}_{\text{ridge}}(\boldsymbol{\beta}^t) + (\|\boldsymbol{X}_{:j}\|_2^2 + \lambda_2)\left(\alpha^2 + \frac{\nabla_j\mathcal{L}_{\text{ridge}}(\boldsymbol{\beta}^t)}{\|\boldsymbol{X}_{:j}\|_2^2 + \lambda_2}\alpha\right)$
     *# extract out a coefficient*

$=\mathcal{L}_{\text{ridge}}(\boldsymbol{\beta}^t) + (\|\boldsymbol{X}_{:j}\|_2^2 + \lambda_2)\left(\alpha^2 + 2\frac{1}{2}\frac{\nabla_j\mathcal{L}_{\text{ridge}}(\boldsymbol{\beta}^t)}{\|\boldsymbol{X}_{:j}\|_2^2 + \lambda_2}\alpha + \left(\frac{1}{2}\frac{\nabla_j\mathcal{L}_{\text{ridge}}(\boldsymbol{\beta}^t)}{\|\boldsymbol{X}_{:j}\|_2^2 + \lambda_2}\right)^2 - \left(\frac{1}{2}\frac{\nabla_j\mathcal{L}_{\text{ridge}}(\boldsymbol{\beta}^t)}{\|\boldsymbol{X}_{:j}\|_2^2 + \lambda_2}\right)^2\right)$
     *# add and subtract the same term*

$=\mathcal{L}_{\text{ridge}}(\boldsymbol{\beta}^t) + (\|\boldsymbol{X}_{:j}\|_2^2 + \lambda_2)\left(\left(\alpha + \frac{1}{2}\frac{\nabla_j\mathcal{L}_{\text{ridge}}(\boldsymbol{\beta}^t)}{\|\boldsymbol{X}_{:j}\|_2^2 + \lambda_2}\right)^2 - \left(\frac{1}{2}\frac{\nabla_j\mathcal{L}_{\text{ridge}}(\boldsymbol{\beta}^t)}{\|\boldsymbol{X}_{:j}\|_2^2 + \lambda_2}\right)^2\right)$
     *# complete the square*

$$=\mathcal{L}_{\text{ridge}}(\boldsymbol{\beta}^t) - \frac{1}{4}\frac{(\nabla_j\mathcal{L}_{\text{ridge}}(\boldsymbol{\beta}^t))^2}{\|\boldsymbol{X}_{:j}\|_2^2 + \lambda_2} + (\|\boldsymbol{X}_{:j}\|_2^2 + \lambda_2)\left(\alpha + \frac{1}{2}\frac{\nabla_j\mathcal{L}_{\text{ridge}}(\boldsymbol{\beta}^t)}{\|\boldsymbol{X}_{:j}\|_2^2 + \lambda_2}\right)^2$$
# *distribute into the parentheses*

The first two terms are constant, and the third term is a quadratic function of $\alpha$.

The minimum loss we can achieve for $\mathcal{L}_{\text{ridge}}(\boldsymbol{\beta}^t + \alpha\boldsymbol{e}_j)$ by optimizing on the $j$-th coefficient is:

$$\min_{\alpha}\mathcal{L}_{\text{ridge}}(\boldsymbol{\beta}^t + \alpha\boldsymbol{e}_j) = \mathcal{L}_{\text{ridge}}(\boldsymbol{\beta}^t) - \frac{1}{4}\frac{(\nabla_j\mathcal{L}_{\text{ridge}}(\boldsymbol{\beta}^t))^2}{\|\boldsymbol{X}_{:j}\|_2^2 + \lambda_2},$$

and the maximum decrease of our ridge regression loss by optimizing a single coefficient is then:

$$\max_{\alpha}(\mathcal{L}_{\text{ridge}}(\boldsymbol{\beta}^t) - \mathcal{L}_{\text{ridge}}(\boldsymbol{\beta}^t + \alpha\boldsymbol{e}_j)) = \mathcal{L}_{\text{ridge}}(\boldsymbol{\beta}^t) - \min_{\alpha}\mathcal{L}_{\text{ridge}}(\boldsymbol{\beta}^t + \alpha\boldsymbol{e}_j) = \frac{1}{4}\frac{(\nabla_j\mathcal{L}_{\text{ridge}}(\boldsymbol{\beta}^t))^2}{\|\boldsymbol{X}_{:j}\|_2^2 + \lambda_2},$$
$$(52)$$

where $\boldsymbol{X}_{:j}$ is the $j$-th column of the design matrix $\boldsymbol{X}$.

Now, for our beam-search method, the decrease of loss between two successive iterations will be greater than the decrease of loss by optimizing on a single coordinate (because we are fine-tuning the coefficients on the newly expanded support *and* picking the best expanded support among all support candidates in the beam search), we have

$$\mathcal{L}_{\text{ridge}}(\boldsymbol{\beta}^t) - \mathcal{L}_{\text{ridge}}(\boldsymbol{\beta}^{t+1}) \geq \max_j\max_{\alpha}(\mathcal{L}_{\text{ridge}}(\boldsymbol{\beta}^t) - \mathcal{L}_{\text{ridge}}(\boldsymbol{\beta}^t + \alpha\boldsymbol{e}_j)) = \max_j\frac{1}{4}\frac{(\nabla_j\mathcal{L}_{\text{ridge}}(\boldsymbol{\beta}^t))^2}{\|\boldsymbol{X}_{:j}\|_2^2 + \lambda_2}.$$
$$(53)$$

For an arbitrary set of coordinates $\mathcal{J}'$, we have the following inequality:

$$\max_j\frac{1}{4}\frac{(\nabla_j\mathcal{L}_{\text{ridge}}(\boldsymbol{\beta}^t))^2}{\|\boldsymbol{X}_{:j}\|_2^2 + \lambda_2} \geq \frac{1}{|\mathcal{J}'|}\sum_{j'\in\mathcal{J}'}\frac{1}{4}\frac{(\nabla_{j'}\mathcal{L}_{\text{ridge}}(\boldsymbol{\beta}^t))^2}{\|\boldsymbol{X}_{:j'}\|_2^2 + \lambda_2}. \tag{54}$$

Inequality 54 holds because the largest decrease of loss among all possible coordinates is greater than or equal to the average decrease of loss among any arbitrary set of coordinates $\mathcal{J}'$.

If we choose $\mathcal{J}' = \mathcal{S}^R := \mathcal{S}^* \setminus supp(\boldsymbol{\beta}^t)$, where $\mathcal{S}^*$ is the set of coordinates for the optimal solution $\boldsymbol{\beta}^*$ and $supp(\cdot)$ denotes the support of the solution, we have

$$\mathcal{L}_{\text{ridge}}(\boldsymbol{\beta}^t) - \mathcal{L}_{\text{ridge}}(\boldsymbol{\beta}^{t+1}) \geq \max_j\frac{1}{4}\frac{(\nabla_j\mathcal{L}_{\text{ridge}}(\boldsymbol{\beta}^t))^2}{\|\boldsymbol{X}_{:j}\|_2^2 + \lambda_2}$$

$$\geq \frac{1}{|\mathcal{S}^R|}\sum_{j'\in\mathcal{S}^R}\frac{1}{4}\frac{(\nabla_{j'}\mathcal{L}_{\text{ridge}}(\boldsymbol{\beta}^t))^2}{\|\boldsymbol{X}_{:j'}\|_2^2 + \lambda_2}$$

$$= \frac{1}{4|\mathcal{S}^R|}\sum_{j'\in\mathcal{S}^R}\frac{(\nabla_{j'}\mathcal{L}_{\text{ridge}}(\boldsymbol{\beta}^t))^2}{\|\boldsymbol{X}_{:j'}\|_2^2 + \lambda_2} \qquad \text{# *pull $\frac{1}{4}$ to the front*}$$

$$\geq \frac{1}{4k}\sum_{j'\in\mathcal{S}^R}\frac{(\nabla_{j'}\mathcal{L}_{\text{ridge}}(\boldsymbol{\beta}^t))^2}{\|\boldsymbol{X}_{:j'}\|_2^2 + \lambda_2} \quad \text{# *because $\mathcal{S}^R \subset \mathcal{S}^*, |\mathcal{S}^R| \leq |\mathcal{S}^*| = k$*}$$

$$\geq \frac{1}{4k}\frac{1}{\max_{j'\in\mathcal{S}^R}(\|\boldsymbol{X}_{:j'}\|_2^2 + \lambda_2)}\sum_{j'\in\mathcal{S}^R}(\nabla_{j'}\mathcal{L}_{\text{ridge}}(\boldsymbol{\beta}^t))^2$$
# *take the maximum of the denominator and pull it to the front*

$$\Rightarrow \mathcal{L}_{\text{ridge}}(\boldsymbol{\beta}^t) - \mathcal{L}_{\text{ridge}}(\boldsymbol{\beta}^{t+1}) \geq \frac{1}{4k}\frac{1}{\max_{j'\in\mathcal{S}^R}(\|\boldsymbol{X}_{:j'}\|_2^2 + \lambda_2)}\sum_{j'\in\mathcal{S}^R}(\nabla_{j'}\mathcal{L}_{\text{ridge}}(\boldsymbol{\beta}^t))^2. \tag{55}$$

Inequality (55) is the lower bound of decrease of loss between two successive iterations of our beam-search algorithm.

In Step 2 and Step 3 below, we will provide bounds on $\max_{j' \in \mathcal{S}^R}(\|\boldsymbol{X}_{:j'}\|_2^2 + \lambda_2)$ and $\sum_{j' \in \mathcal{S}^R}(\nabla_{j'}\mathcal{L}_{\text{ridge}}(\boldsymbol{\beta}^t))^2$ in terms of $M_1$ and $m_{2k}$, respectively.

*Step 2: Upper bound on* $\max_{j' \in \mathcal{S}^R}(\|\boldsymbol{X}_{:j'}\|_2^2 + \lambda_2)$

We will show below that $\frac{M_1}{2} \geq \max_{j' \in \mathcal{S}^R}(\|\boldsymbol{X}_{:j'}\|_2^2 + \lambda_2)$.

According to the definition of the restricted smoothness of $M_1$ (See Background Concepts in Appendix B), we have that

$$\frac{M_1}{2}\|\boldsymbol{w}_2 - \boldsymbol{w}_1\|_2^2 \geq \mathcal{L}_{\text{ridge}}(\boldsymbol{w}_2) - \mathcal{L}_{\text{ridge}}(\boldsymbol{w}_1) - \nabla\mathcal{L}_{\text{ridge}}(\boldsymbol{w}_1)^T(\boldsymbol{w}_2 - \boldsymbol{w}_1) \tag{56}$$

for $\forall \boldsymbol{w}_1, \boldsymbol{w}_2 \in \mathbb{R}^p$ with $\|\boldsymbol{w}_1\|_0 \leq 1, \|\boldsymbol{w}_2\|_0 \leq 1,$ and $\|\boldsymbol{w}_2 - \boldsymbol{w}_1\|_0 \leq 1$.

The right-hand side can be rewritten as

$$\mathcal{L}_{\text{ridge}}(\boldsymbol{w}_2) - \mathcal{L}_{\text{ridge}}(\boldsymbol{w}_1) - \nabla\mathcal{L}_{\text{ridge}}(\boldsymbol{w}_1)^T(\boldsymbol{w}_2 - \boldsymbol{w}_1)$$
$$=(\|\boldsymbol{X}\boldsymbol{w}_2 - \boldsymbol{y}\|_2^2 + \lambda_2\|\boldsymbol{w}_2\|_2^2) - (\|\boldsymbol{X}\boldsymbol{w}_1 - \boldsymbol{y}\|_2^2 + \lambda_2\|\boldsymbol{w}_1\|_2^2) - 2(\boldsymbol{X}^T(\boldsymbol{X}\boldsymbol{w}_1 - \boldsymbol{y}) + \lambda_2\boldsymbol{w}_1)^T(\boldsymbol{w}_2 - \boldsymbol{w}_1)$$
$$\textcolor{green}{\text{\# write out } \mathcal{L}_{\text{ridge}}(\cdot) \text{ explicitly and substitute } \nabla\mathcal{L}_{\text{ridge}}(\boldsymbol{w}_1) =}$$
$$\textcolor{green}{2\left(\boldsymbol{X}^T(\boldsymbol{X}\boldsymbol{w}_1 - \boldsymbol{y}) + \boldsymbol{w}_1\right) \text{ according to Lemma B.7}}$$
$$=(\|(\boldsymbol{X}\boldsymbol{w}_2 - \boldsymbol{X}\boldsymbol{w}_1) + (\boldsymbol{X}\boldsymbol{w}_1 - \boldsymbol{y})\|_2^2 + \lambda_2\|(\boldsymbol{w}_2 - \boldsymbol{w}_1) + \boldsymbol{w}_1\|_2^2)$$
$$\quad - (\|\boldsymbol{X}\boldsymbol{w}_1 - \boldsymbol{y}\|_2^2 + \lambda_2\|\boldsymbol{w}_1\|_2^2) - 2(\boldsymbol{X}^T(\boldsymbol{X}\boldsymbol{w}_1 - \boldsymbol{y}) + \lambda_2\boldsymbol{w}_1)^T(\boldsymbol{w}_2 - \boldsymbol{w}_1)$$
$$\textcolor{green}{\text{\# We subtract and add identical terms inside the first parentheses}}$$
$$= (\|\boldsymbol{X}\boldsymbol{w}_2 - \boldsymbol{X}\boldsymbol{w}_1\|_2^2 + \|\boldsymbol{X}\boldsymbol{w}_1 - \boldsymbol{y}\|_2^2 + 2(\boldsymbol{X}\boldsymbol{w}_2 - \boldsymbol{X}\boldsymbol{w}_1)^T(\boldsymbol{X}\boldsymbol{w}_1 - \boldsymbol{y}))$$
$$\quad + \lambda_2\|\boldsymbol{w}_2 - \boldsymbol{w}_1\|_2^2 + \lambda_2\|\boldsymbol{w}_1\|_2^2 + 2\lambda_2(\boldsymbol{w}_2 - \boldsymbol{w}_1)^T\boldsymbol{w}_1$$
$$\quad - (\|\boldsymbol{X}\boldsymbol{w}_1 - \boldsymbol{y}\|_2^2 + \lambda_2\|\boldsymbol{w}_1\|_2^2) - 2(\boldsymbol{X}^T(\boldsymbol{X}\boldsymbol{w}_1 - \boldsymbol{y}) + \lambda_2\boldsymbol{w}_1)^T(\boldsymbol{w}_2 - \boldsymbol{w}_1)$$
$$\textcolor{green}{\text{\# expand the terms inside } \|\cdot\| \text{ for the first line into two lines}}$$
$$=(\|\boldsymbol{X}\boldsymbol{w}_2 - \boldsymbol{X}\boldsymbol{w}_1\|_2^2 + \|\boldsymbol{X}\boldsymbol{w}_1 - \boldsymbol{y}\|_2^2 + 2(\boldsymbol{X}^T(\boldsymbol{X}\boldsymbol{w}_1 - \boldsymbol{y}))^T(\boldsymbol{w}_2 - \boldsymbol{w}_1))$$
$$\quad + \lambda_2\|\boldsymbol{w}_2 - \boldsymbol{w}_1\|_2^2 + \lambda_2\|\boldsymbol{w}_1\|_2^2 + 2(\lambda_2\boldsymbol{w}_1)^T(\boldsymbol{w}_2 - \boldsymbol{w}_1)$$
$$\quad - (\|\boldsymbol{X}\boldsymbol{w}_1 - \boldsymbol{y}\|_2^2 + \lambda_2\|\boldsymbol{w}_1\|_2^2) - 2(\boldsymbol{X}^T(\boldsymbol{X}\boldsymbol{w}_1 - \boldsymbol{y}) + \lambda_2\boldsymbol{w}_1)^T(\boldsymbol{w}_2 - \boldsymbol{w}_1)$$
$$\textcolor{green}{\text{\# rewrite } (\boldsymbol{X}\boldsymbol{w}_1 - \boldsymbol{X}\boldsymbol{w}_2)^T(\boldsymbol{X}\boldsymbol{w}_1 - \boldsymbol{y}) = (\boldsymbol{X}^T(\boldsymbol{X}\boldsymbol{w}_1 - \boldsymbol{y}))^T(\boldsymbol{w}_2 - \boldsymbol{w}_1)}$$
$$=\|\boldsymbol{X}\boldsymbol{w}_2 - \boldsymbol{X}\boldsymbol{w}_1\|_2^2 + \lambda_2\|\boldsymbol{w}_2 - \boldsymbol{w}_1\|_2^2 \qquad\qquad \textcolor{green}{\text{\# cancel out terms}}$$
$$=(\boldsymbol{w}_2 - \boldsymbol{w}_1)^T(\lambda_2\mathbf{I} + \boldsymbol{X}^T\boldsymbol{X})(\boldsymbol{w}_2 - \boldsymbol{w}_1)$$

$$\Rightarrow \mathcal{L}_{\text{ridge}}(\boldsymbol{w}_2) - \mathcal{L}_{\text{ridge}}(\boldsymbol{w}_1) - \nabla\mathcal{L}_{\text{ridge}}(\boldsymbol{w}_1)^T(\boldsymbol{w}_2 - \boldsymbol{w}_1) = (\boldsymbol{w}_2 - \boldsymbol{w}_1)^T(\lambda_2\mathbf{I} + \boldsymbol{X}^T\boldsymbol{X})(\boldsymbol{w}_2 - \boldsymbol{w}_1).$$

Therefore, together with Inequality (56), we have that

$$\frac{M_1}{2}\|\boldsymbol{w}_2 - \boldsymbol{w}_1\|_2^2 \geq (\boldsymbol{w}_2 - \boldsymbol{w}_1)^T(\lambda_2\mathbf{I} + \boldsymbol{X}^T\boldsymbol{X})(\boldsymbol{w}_2 - \boldsymbol{w}_1) \tag{57}$$

for any $\boldsymbol{w}_2 - \boldsymbol{w}_1 \in \mathbb{R}^p$ and $\|\boldsymbol{w}_2 - \boldsymbol{w}_1\|_0 \leq 1$. If we let $\boldsymbol{w}_2 - \boldsymbol{w}_1 = \boldsymbol{e}_j$ where $\boldsymbol{e}_j$ is a vector with 1 on the $j$-th entry and 0 otherwise, we have

$$\frac{M_1}{2} \geq \lambda_2 + \|\boldsymbol{X}_{:j}\|_2^2$$

for $\forall j \in [1, ..., p]$. Thus, we can derive our upper bound at the beginning of Step 2:

$$\frac{M_1}{2} \geq \max_{j' \in \mathcal{S}^R}(\|\boldsymbol{X}_{:j'}\|_2^2 + \lambda_2) \tag{58}$$

*Step 3: Lower bound on* $\sum_{j' \in \mathcal{S}^R}(\nabla_{j'}\mathcal{L}_{\text{ridge}}(\boldsymbol{\beta}^t))^2$

We will show that $\sum_{j' \in \mathcal{J}'}(\nabla_{j'}\mathcal{L}_{\text{ridge}}(\boldsymbol{\beta}^t))^2 \geq 2m_{2k}(\mathcal{L}_{\text{ridge}}(\boldsymbol{\beta}^t) - \mathcal{L}_{\text{ridge}}(\boldsymbol{\beta}^*))$.

According to the definition of the restricted strong convexity parameter $m_{2k}$, we have:

$$\nabla\mathcal{L}_{\text{ridge}}(\boldsymbol{\beta}^t)^T(\boldsymbol{\beta}^* - \boldsymbol{\beta}^t) + \frac{m_{2k}}{2}\|\boldsymbol{\beta}^* - \boldsymbol{\beta}^t\|_2^2 \leq \mathcal{L}_{\text{ridge}}(\boldsymbol{\beta}^*) - \mathcal{L}_{\text{ridge}}(\boldsymbol{\beta}^t),$$

where $\boldsymbol{\beta}^t \in \mathbb{R}^p$ is our heuristic solution at the $t$-th iteration, $\boldsymbol{\beta}^*$ is the optimal $k$-sparse ridge regression solution, and $\|\boldsymbol{\beta}^t\|_0 = t \le k < 2k$, $\|\boldsymbol{\beta}^*\|_0 = k < 2k$, and $\|\boldsymbol{\beta}^* - \boldsymbol{\beta}^t\|_0 \le 2k$

By rearranging some terms, we have:

$$
\begin{aligned}
&\mathcal{L}_{\text{ridge}}(\boldsymbol{\beta}^t) - \mathcal{L}_{\text{ridge}}(\boldsymbol{\beta}^*) \\
&\le -\nabla\mathcal{L}_{\text{ridge}}(\boldsymbol{\beta}^t)^T(\boldsymbol{\beta}^* - \boldsymbol{\beta}^t) - \frac{m_{2k}}{2}\|\boldsymbol{\beta}^* - \boldsymbol{\beta}^t\|_2^2 \\
&\qquad\qquad\qquad\qquad\qquad\text{\# \textit{always holds because of restricted strong convexity}} \\
&= -\nabla\mathcal{L}_{\text{ridge}}(\boldsymbol{\beta}^t)^T(\boldsymbol{\beta}^t + \sum_{j=1}^{p}\alpha_j\boldsymbol{e}_j - \boldsymbol{\beta}^t) - \frac{m_{2k}}{2}\|\boldsymbol{\beta}^t + \sum_{j=1}^{p}\alpha_j\boldsymbol{e}_j - \boldsymbol{\beta}^t\|_2^2 \\
&\text{\# }\boldsymbol{e}_j \textit{ is a vector with 1 on the j-th entry and 0 otherwise; } \alpha_j \textit{ is a fixed number with } \alpha_j = \beta_j^* - \beta_j^t \\
&= -\nabla\mathcal{L}_{\text{ridge}}(\boldsymbol{\beta}^t)^T(\sum_{j=1}^{p}\alpha_j\boldsymbol{e}_j) - \frac{m_{2k}}{2}\|\sum_{j=1}^{p}\alpha_j\boldsymbol{e}_j\|_2^2 \qquad \text{\# } \alpha_j \textit{ is a fixed number with } \alpha_j = \beta_j^* - \beta_j^t \\
&= -\sum_{j=1}^{p}\left(\nabla_j\mathcal{L}_{\text{ridge}}(\boldsymbol{\beta}^t)\alpha_j - \frac{m_{2k}}{2}\alpha_j^2\right) \qquad\qquad\qquad \text{\# \textit{Combine into one summation}} \\
&= -\sum_{j\in\mathcal{S}^R}\left(\nabla_j\mathcal{L}_{\text{ridge}}(\boldsymbol{\beta}^t)\alpha_j - \frac{m_{2k}}{2}\alpha_j^2\right) - \sum_{j\in supp(\boldsymbol{\beta}^t)}\left(\nabla_j\mathcal{L}_{\text{ridge}}(\boldsymbol{\beta}^t)\alpha_j - \frac{m_{2k}}{2}\alpha_j^2\right) \\
&\quad - \sum_{j\notin\mathcal{S}^R\cup supp(\boldsymbol{\beta}^t)}\left(\nabla_j\mathcal{L}_{\text{ridge}}(\boldsymbol{\beta}^t)\alpha_j - \frac{m_{2k}}{2}\alpha_j^2\right) \\
&\qquad\qquad \text{\# } \textit{Recall } \mathcal{S}^R = \mathcal{S}^* \setminus supp(\boldsymbol{\beta}^t). \textit{ We divide the indices } \{1,2,...,p\} \textit{ into three groups: } \mathcal{S}^R, \\
&\qquad\qquad\quad\ supp(\boldsymbol{\beta}^t), \textit{ and the rest.} \\
&= -\sum_{j\in\mathcal{S}^R}\left(\nabla_j\mathcal{L}_{\text{ridge}}(\boldsymbol{\beta}^t)\alpha_j - \frac{m_{2k}}{2}\alpha_j^2\right) \\
&\qquad\qquad\qquad \text{\# } \nabla_j\mathcal{L}_{\text{ridge}}(\boldsymbol{\beta}^t) = 0 \textit{ for } j \in supp(\boldsymbol{\beta}^t), \textit{ and } \alpha_j = 0 \textit{ for } j \notin \mathcal{S}^R \cup supp(\boldsymbol{\beta}^t) \\
&= -\sum_{j'\in\mathcal{J}'}\left(\nabla_{j'}\mathcal{L}_{\text{ridge}}(\boldsymbol{\beta}^t)\alpha_{j'} - \frac{m_{2k}}{2}\alpha_{j'}^2\right) \qquad\qquad \text{\# \textit{Change the index notation from j to j'}} \\
&\le \max_{\boldsymbol{\nu}} -\sum_{j\in\mathcal{J}'}\left(\nabla_j\mathcal{L}_{\text{ridge}}(\boldsymbol{\beta}^t)\nu_j - \frac{m_{2k}}{2}\nu_j^2\right) \\
&\qquad\qquad \text{\# } \textit{Allow } \boldsymbol{\alpha} \textit{ to take any values with a new variable } \boldsymbol{\nu}. \textit{ The inequality holds if we take the} \\
&\qquad\qquad\quad\ \textit{maximum with respect to } \boldsymbol{\nu} \\
&= \frac{1}{2m_{2k}}\sum_{j'\in\mathcal{J}'}(\nabla_{j'}\mathcal{L}_{\text{ridge}}(\boldsymbol{\beta}^t))^2 \quad \text{\# \textit{each term can be maximized because it is a quadratic function}}
\end{aligned}
$$

This gives us the lower bound:

$$
\sum_{j'\in\mathcal{J}'}(\nabla_{j'}\mathcal{L}_{\text{ridge}}(\boldsymbol{\beta}^t))^2 \ge 2m_{2k}(\mathcal{L}_{\text{ridge}}(\boldsymbol{\beta}^t) - \mathcal{L}_{\text{ridge}}(\boldsymbol{\beta}^*)) \tag{59}
$$

*Step 4: Final Bound for the Loss of Our Heuristic Solution*

If we plug Inequalities (58) and (59) into the Inequality (55), we have:

$$
\mathcal{L}_{\text{ridge}}(\boldsymbol{\beta}^t) - \mathcal{L}_{\text{ridge}}(\boldsymbol{\beta}^{t+1}) \ge \frac{1}{4k}\frac{1}{\max_{j'\in\mathcal{S}^R}(\|\boldsymbol{X}_{:j'}\|_2^2 + \lambda_2)}\sum_{j'\in\mathcal{S}^R}(\nabla_{j'}\mathcal{L}_{\text{ridge}}(\boldsymbol{\beta}^t))^2
$$

$$
\Rightarrow \mathcal{L}_{\text{ridge}}(\boldsymbol{\beta}^t) - \mathcal{L}_{\text{ridge}}(\boldsymbol{\beta}^{t+1}) \ge \frac{m_{2k}}{kM_1}(\mathcal{L}_{\text{ridge}}(\boldsymbol{\beta}^t) - \mathcal{L}_{\text{ridge}}(\boldsymbol{\beta}^*)) \tag{60}
$$

Lastly, let's use Inequality (60) to derive the right-hand-side of Inequality (51) stated in the Theorem. Inequality (60) can be rewritten as:

$$
(\mathcal{L}_{\text{ridge}}(\boldsymbol{\beta}^t) - \mathcal{L}_{\text{ridge}}(\boldsymbol{\beta}^*)) - (\mathcal{L}_{\text{ridge}}(\boldsymbol{\beta}^{t+1}) - \mathcal{L}_{\text{ridge}}(\boldsymbol{\beta}^*)) \ge \frac{m_{2k}}{kM_1}(\mathcal{L}_{\text{ridge}}(\boldsymbol{\beta}^t) - \mathcal{L}_{\text{ridge}}(\boldsymbol{\beta}^*)),
$$

which can be rearranged to the following expression:

$$(1 - \frac{m_{2k}}{kM_1})(\mathcal{L}_{\text{ridge}}(\boldsymbol{\beta}^t) - \mathcal{L}_{\text{ridge}}(\boldsymbol{\beta}^*)) \geq (\mathcal{L}_{\text{ridge}}(\boldsymbol{\beta}^{t+1}) - \mathcal{L}_{\text{ridge}}(\boldsymbol{\beta}^*))$$

Therefore, by applying the above inequality $k$ times, we have:

$$(1 - \frac{m_{2k}}{kM_1})^k(\mathcal{L}_{\text{ridge}}(\mathbf{0}) - \mathcal{L}_{\text{ridge}}(\boldsymbol{\beta}^*)) \geq (\mathcal{L}_{\text{ridge}}(\boldsymbol{\beta}^k) - \mathcal{L}_{\text{ridge}}(\boldsymbol{\beta}^*))$$

However, $\mathcal{L}_{\text{ridge}}(\mathbf{0}) = 0$. This gives us

$$\mathcal{L}_{\text{ridge}}(\boldsymbol{\beta}^k) \leq \left(1 - (1 - \frac{m_{2k}}{kM_1})^k\right)\mathcal{L}_{\text{ridge}}(\boldsymbol{\beta}^*)$$

To get the final inequality, we start from a well-known inequality result $1 + x \leq e^x$ for any $x \in \mathbb{R}$.

Let $x = -\frac{a}{k} > -1$, then we have $1 - \frac{a}{k} \leq e^{-a/k}$. If we take both sides to the power of $k$, we have $(1 - \frac{a}{k})^k \leq e^{-a}$.

If we let $a = \frac{m_{2k}}{M_1}$ (because $\frac{m_{2k}}{M_1} < 1$ [29] we have $x = -\frac{a}{k} = -\frac{m_{2k}}{M_1 k} > -1$ for any $k \geq 1$), then we have

$$(1 - \frac{m_{2k}}{kM_1})^k \leq e^{-\frac{m_{2k}}{M_1}} \tag{61}$$

Because $\mathcal{L}_{\text{ridge}}(\boldsymbol{\beta}^*)$ is a negative number, we finally have

$$\mathcal{L}_{\text{ridge}}(\boldsymbol{\beta}^k) \leq \left(1 - (1 - \frac{m_{2k}}{kM_1})^k\right)\mathcal{L}_{\text{ridge}}(\boldsymbol{\beta}^*)$$
$$\leq (1 - e^{-\frac{m_{2k}}{M_1}})\mathcal{L}_{\text{ridge}}(\boldsymbol{\beta}^*).$$

$\square$

## C.5 Derivation of $g(a,b) = \max_c ac - \frac{c^2}{4}b$ via Fenchel Conjugate

Recall our function $g : \mathbb{R} \times \mathbb{R} \to \mathbb{R} \cup \{+\infty\}$ is defined as:

$$g(a,b) = \begin{cases} \frac{a^2}{b} & \text{if } b > 0 \\ 0 & \text{if } a = b = 0 \\ \infty & \text{otherwise} \end{cases} \tag{62}$$

The convex conjugate of $g(\cdot, \cdot)$, denoted as $g^*(\cdot, \cdot)$, is defined as [8]:

$$g^*(c,d) = \sup_{a,b} (ac + bd - g(a,b)) \tag{63}$$

Since function $g(a,b)$ is convex and lower semi-continuous, the biconjugate of $g(a,b)$ is equal to the original function, *i.e.*, $(g^*)^* = g$ [8]. Writing this explicitly, we have:

$$g(a,b) = \sup_{c,d} (ac + bd - g^*(c,d)) \tag{64}$$

For now, let us first derive an analytic formula for the convex conjugate function $g^*(\cdot, \cdot)$.

According to the definition of $g(a,b)$ in Equation (62), we have

$$h(a,b,c,d) = ac + bd - g(a,b) = \begin{cases} ac + bd - \frac{a^2}{b} & \text{if } b > 0 \\ 0 & \text{if } a = b = 0 \\ -\infty & \text{otherwise} \end{cases}.$$

For any fixed $c$ and $d$, the optimality of the first case can be achieved if the first derivatives of $ac + bd - \frac{a^2}{b}$ with respect to $a$ and $b$ are both equal to 0. This means that,

$$c - 2\frac{a}{b} = 0 \quad \text{and} \quad d + \frac{a^2}{b^2} = 0.$$

If $d = -\frac{c^2}{4}$, the optimality condition is achieved if we let $a = \frac{bc}{2}$, and then $ac + bd - \frac{a^2}{b} = \frac{bc}{2}c + b(-\frac{c^2}{4}) - \frac{c^2}{4}b = 0$.

If $d \neq -\frac{c^2}{4}$, the optimality condition can not be achieved. However, if we still let $a = \frac{bc}{2}$, we have $ac + bd - \frac{a^2}{b} = \frac{bc}{2}c + b(d) - \frac{c^2}{4}b = (\frac{c^2}{4} + d)b$. Then, we have two specific cases to discuss: 1) if $\frac{c^2}{4} + d > 0$, the supremum is $\infty$ if we let $b \to \infty$; 2) if $\frac{c^2}{4} + d < 0$, the supremum is 0 if we let $b \to 0$.

With all these cases discussed, we have that

$$g^*(c, d) = \sup_{a,b} h(a, b, c, d)$$

$$= \begin{cases} 0 & \text{if } \frac{c^2}{4} + d \leq 0 \\ \infty & \text{otherwise} \end{cases}$$

Now let us revisit Equation (64), where $g(a, b) = \sup_{c,d} ac + bd - g^*(c, d)$. Below, we discuss three cases and derive the alternative formulas for $g(a, b)$ under different scenarios: (1) $\frac{c^2}{4} + d = 0$, (2) $\frac{c^2}{4} + d \leq 0$, and (3) $\frac{c^2}{4} + d > 0$.

1. $\frac{c^2}{4} + d = 0$:

$$g_1(a, b) = \sup_{c,d, \frac{c^2}{4}+d=0} ac + bd - g^*(c, d) = \sup_c ac + b(-\frac{c^2}{4})$$

2. $\frac{c^2}{4} + d < 0$:

$$g_2(a, b) = \sup_{c,d, \frac{c^2}{4}+d<0} ac + bd - g^*(c, d) = \sup_{c,d} ac + bd$$

3. $\frac{c^2}{4} + d > 0$:

$$g_3(a, b) = \sup_{c,d, \frac{c^2}{4}+d>0} ac + bd - g^*(c, d) = -\infty$$

After discussing these three cases, we can obtain the final results by taking the maximum of the three optimal values, i.e., $g(a, b) = \max(g_1(a, b), g_2(a, b), g_3(a, b))$. The third case $\frac{c^2}{4} + d > 0$ is not interesting and can be easily eliminated. The second case $\frac{c^2}{4} + d < 0$ is also not interesting because $ac + bd = ac + b(-\frac{c^2}{4}) + b(\frac{c^2}{4} + d) \leq ac + b(-\frac{c^2}{4})$ for any $b \geq 0$, which is the domain we are interested in. This gives us $g_2(a, b) \leq g_1(a, b)$, and the equality is only achieved when $b = 0$. Thus,

$$g(a, b) = \max(g_1(a, b), g_2(a, b), g_3(a, b))$$
$$= g_1(a, b) \tag{65}$$
$$= \sup_c ac + b(-\frac{c^2}{4})$$
$$= \max_c ac - b\frac{c^2}{4}, \tag{66}$$

where in the last step we convert $\sup(\cdot)$ to $\max(\cdot)$ because the function is quadratic and the optimum can be achieved.

# D Existing MIP Formulations for $k$-sparse Ridge Regression

We want to solve the following $k$-sparse ridge regression problem:

$$\min_{\boldsymbol{\beta}, \boldsymbol{z}} \quad \boldsymbol{\beta}^T \boldsymbol{X}^T \boldsymbol{X} \boldsymbol{\beta} - 2\boldsymbol{y}^T \boldsymbol{X} \boldsymbol{\beta} + \lambda_2 \sum_{j=1}^{p} \beta_j^2 \tag{67}$$

$$\text{subject to} \quad (1 - z_j)\beta_j = 0, \quad \sum_{j=1}^{p} z_j \leq k, \quad z_j \in \{0, 1\}$$

However, we can not handle the constraint $(1 - z_j)\beta_j = 0$ directly in a mixed-integer programming (MIP) solver. Currently in the research community, there are four formulations we can write before sending the problem to a MIP solver. They are (1) SOS1, (2) big-M, (3) perspective, and (4) optimal perspective. We write each formulation explicitly below:

1. SOS1 formulation:

$$\min_{\boldsymbol{\beta}, \boldsymbol{z}} \quad \boldsymbol{\beta}^T \boldsymbol{X}^T \boldsymbol{X} \boldsymbol{\beta} - 2\boldsymbol{y}^T \boldsymbol{X} \boldsymbol{\beta} + \lambda_2 \sum_{j=1}^{p} \beta_j^2 \tag{68}$$

$$\text{subject to} \quad (1 - z_j, \beta_j) \in \text{SOS1}, \quad \sum_{j=1}^{p} z_j \leq k, \quad z_j \in \{0, 1\},$$

   where $(a, b) \in \text{SOS1}$ means that at most one variable (either $a$ or $b$ but not both) can be nonzero.

2. big-M formulation:

$$\min_{\boldsymbol{\beta}, \boldsymbol{z}} \quad \boldsymbol{\beta}^T \boldsymbol{X}^T \boldsymbol{X} \boldsymbol{\beta} - 2\boldsymbol{y}^T \boldsymbol{X} \boldsymbol{\beta} + \lambda_2 \sum_{j=1}^{p} \beta_j^2 \tag{69}$$

$$\text{subject to} \quad -Mz_j \leq \beta_j \leq Mz_j, \quad \sum_{j=1}^{p} z_j \leq k, \quad z_j \in \{0, 1\}$$

   where $M > 0$. For the big-M formulation, as we mentioned in the main paper, it is very challenging to choose the right value for $M$. If $M$ is too big, the lower bound given by the relaxed problem is too loose; if $M$ is too small, we risk cutting off the optimal solution.

3. perspective formulation:

$$\min_{\boldsymbol{\beta}, \boldsymbol{z}, \boldsymbol{s}} \quad \boldsymbol{\beta}^T \boldsymbol{X}^T \boldsymbol{X} \boldsymbol{\beta} - 2\boldsymbol{y}^T \boldsymbol{X} \boldsymbol{\beta} + \lambda_2 \sum_{j=1}^{p} s_j \tag{70}$$

$$\text{subject to} \quad \beta_j^2 \leq s_j z_j, \quad \sum_{j=1}^{p} z_j \leq k, \quad z_j \in \{0, 1\}$$

   where $s_j$'s are new variables which ensure that when $z_j = 0$, the corresponding coefficient $\beta_j = 0$. The problem is a mixed-integer quadratically-constrained quadratic problem (QCQP). The problem can be solved by a commercial solver which accepts QCQP formulations.

4. eigen-perspective formulation:

$$\min_{\boldsymbol{\beta}, \boldsymbol{z}, \boldsymbol{s}} \quad \boldsymbol{\beta}^T \left( \boldsymbol{X}^T \boldsymbol{X} - \lambda_{\min}(\boldsymbol{X}^T \boldsymbol{X}) \boldsymbol{I} \right) \boldsymbol{\beta} - 2\boldsymbol{y}^T \boldsymbol{X} \boldsymbol{\beta} + \left( \lambda_2 + \lambda_{\min}(\boldsymbol{X}^T \boldsymbol{X}) \right) \sum_{j=1}^{p} s_j \tag{71}$$

$$\text{subject to} \quad \beta_j^2 \leq s_j z_j, \quad \sum_{j=1}^{p} z_j \leq k, \quad z_j \in \{0, 1\},$$

   where $\lambda_{\min}(\cdot)$ denotes the smallest eigenvalue of a matrix. This formulation is similar to the QCQP formulation above, but we increase the $\ell_2$ coefficient by subtracting $\lambda_{\min}(\boldsymbol{X}^T \boldsymbol{X})\boldsymbol{I}$ from the diagonal of $\boldsymbol{X}^T \boldsymbol{X}$.

5. optimal perspective formulation:

$$\min_{\boldsymbol{B},\boldsymbol{\beta},\boldsymbol{z}} \quad \langle \boldsymbol{X}^T \boldsymbol{X}, \boldsymbol{B} \rangle - 2\boldsymbol{y}^T \boldsymbol{X} \boldsymbol{\beta} \tag{72}$$

$$\text{subject to} \quad \boldsymbol{B} - \boldsymbol{\beta}\boldsymbol{\beta}^T \succeq 0, \quad \beta_j^2 \leq B_{jj} z_j, \quad \sum_{j=1}^{p} z_j \leq k, \quad z_j \in \{0,1\},$$

where $\boldsymbol{A} \succeq 0$ means that $\boldsymbol{A}$ is positive semidefinite. Such formulation is called a mixed-integer semidefinite programming (MI-SDP) problem and is proposed in [62, 28, 37]. Currently, there is no commercial solver that can solve MI-SDP problems: Gurobi does not support SDP; MOSEK can solve SDP but does not support MI-SDP. As shown by [28], SDP is not scalable to high dimensions. We will confirm this observation in our experiments by solving the relaxed convex SDP problem (relaxing $\{0,1\}$ to $[0,1]$) via MOSEK, the state-of-the-art conic solver. After we get the solution from the relaxed problem, we find the indices corresponding to the top $k$ largest $z_j$'s and retrieve a feasible solution on these features.

# E    Algorithmic Charts

## E.1    Overall Branch-and-bound Framework

We first provide the overall branch-and-bound framework. Then we go into details on how to calculate the lower bound, how to obtain a near-optimal solution and upper bound (via beam search), and how to do branching. A visualization of the branch-and-bound tree can be found in Figure 5.

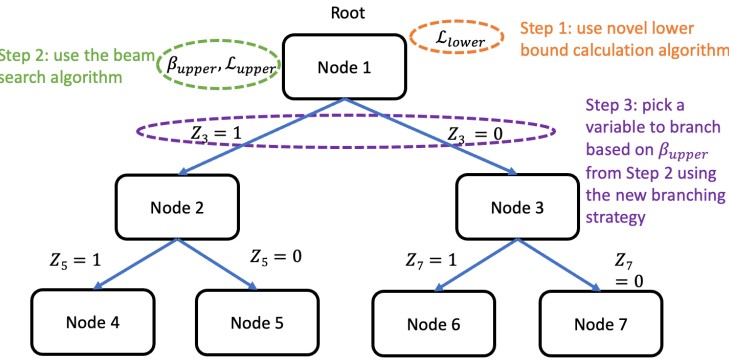

Figure 5: Branch-and-bound diagram.

**Algorithm 1** OKRidge($\mathcal{D}$,$k$,$\lambda_2$, $\epsilon$, $T_{\text{iter}}$, $T_{\text{max}}$) $\to$ $\boldsymbol{\beta}$

---

**Input:** dataset $\mathcal{D}$, sparsity constraint $k$, $\ell_2$ coefficient $\lambda_2$, optimality gap $\epsilon$, iteration limit $T_{\text{iter}}$ for subgradient ascent, and time limit $T_{\text{max}}$.

**Output:** a sparse continuous coefficient vector $\boldsymbol{\beta} \in \mathbb{R}^p$ with $\|\boldsymbol{\beta}\|_0 \leq k$.

1: $\epsilon_{\text{best}}, \mathcal{L}_{\text{lower}}^{\text{best}}, \boldsymbol{\beta}_{\text{upper}}^{\text{best}}, \mathcal{L}_{\text{upper}}^{\text{best}} = \infty, -\infty, \mathbf{0}, 0$ ▷*Initialize best optimality gap to $\infty$, best lower bound to $-\infty$, best solution to $\mathbf{0}$ and its loss to 0.*

2: $\mathcal{Q} \leftarrow$ initialize an empty queue (FIFO) structure ▷*FIFO corresponds to a breadth-first search.*

3: $\mathcal{N}_{\text{select}} = \emptyset, \mathcal{N}_{\text{avoid}} = \emptyset$ ▷*Coordinates in $\mathcal{N}_{select}$ can be selected during heuristic search and lower bound calculation for a node; Coordinates in $\mathcal{N}_{avoid}$ must be avoided.*

4: RootNode = CreateNode($\mathcal{D}, k, \lambda_2, \mathcal{N}_{\text{select}}, \mathcal{N}_{\text{avoid}}$) ▷*Create the first node, which is the root node in the BnB tree.*

5: $\mathcal{Q} = \mathcal{Q}$.push(RootNode) ▷*Put the root node in the queue.*

6: $l_{\text{unsolved}} = 1$ ▷*Smallest depth where at least one node is unsolved in the BnB tree*

7: **WHILE** $\mathcal{Q} \neq \emptyset$ and $T_{\text{elapsed}} < T_{\text{max}}$ **do** ▷*Keep searching until the time limit is reached or there is no node in the queue.*

8:     Node = $\mathcal{Q}$.getNode()

9:     $\mathcal{L}_{\text{lower}}$ = Node.lowerSolveFast() ▷*Get the lower bound for this node through the fast method*

10:     **if** $\mathcal{L}_{\text{lower}} < \mathcal{L}_{\text{best}}^{\text{upper}}$ **then** $\mathcal{L}_{\text{lower}} =$ Node.lowerSolveTight() ▷*Get a tighter lower bound for this node through the subgradient ascent*

11:     **if** $\mathcal{L}_{\text{lower}} \geq \mathcal{L}_{\text{best}}^{\text{upper}}$ **then continue** ▷*Prune Node if its lower bound is higher than or equal to the current best upper bound.*

12:     **if** All nodes in depth $l_{\text{unsolved}}$ of the BnB tree are solved **then**

13:         $\mathcal{L}_{\text{lower}}^{\text{best}} = \max_q(\{\mathcal{L}^q, \text{for } q \in \{1, 2, ..., l_{\text{unsolved}}\})$ ▷*Tighten the best lower bound; $\mathcal{L}^q$ is the smallest lower bound of all nodes in depth $q$*

14:         $l_{\text{unsolved}} = l_{\text{unsolved}} + 1$ ▷*Raise up the smallest unsolved depth by 1*

15:         $\epsilon_{\text{best}} = (\mathcal{L}_{\text{upper}}^{\text{best}} - \mathcal{L}_{\text{lower}}^{\text{best}})/|\mathcal{L}_{\text{upper}}^{\text{best}}|$

16:         **if** $\epsilon_{\text{best}} < \epsilon$ **then return** $\boldsymbol{\beta}_{\text{best}}$ ▷*Optimality gap tolerance is reached, return the current best solution*

17:     $\boldsymbol{\beta}_{\text{upper}}, \mathcal{L}_{\text{upper}} =$ Node.upperSolve() ▷*Get feasible sparse solution and its loss from beam search, which is an upper bound for our BnB tree.*

18:     **if** $\mathcal{L}_{\text{upper}} < \mathcal{L}_{\text{upper}}^{\text{best}}$ **then**

19:         $\mathcal{L}_{\text{upper}}^{\text{best}}, \boldsymbol{\beta}_{\text{upper}}^{\text{best}} = \mathcal{L}_{\text{upper}}, \boldsymbol{\beta}_{\text{upper}}$ ▷*Update our best solution and best upper bound*

20:         $\epsilon_{\text{best}} = (\mathcal{L}_{\text{upper}}^{\text{best}} - \mathcal{L}_{\text{lower}}^{\text{best}})/|\mathcal{L}_{\text{upper}}^{\text{best}}|$

21:         **if** $\epsilon_{\text{best}} < \epsilon$ **then return** $\boldsymbol{\beta}_{\text{best}}$ ▷*Optimality gap tolerance is reached, return the current best solution*

22:     $j =$ Node.getBranchCoord($\boldsymbol{\beta}_{\text{upper}}$) ▷*Get which feature coordinate to branch*

23:     ChildNode1 = CreateNode($\mathcal{D}, k, \lambda_2, (\text{Node} \to \mathcal{N}_{\text{select}}) \cup \{j\}, \text{Node} \to \mathcal{N}_{\text{avoid}}$) ▷*New child node where feature $j$ must be selected*

24:     ChildNode2 = CreateNode($\mathcal{D}, k, \lambda_2, \text{Node} \to \mathcal{N}_{\text{select}}, (\text{Node} \to \mathcal{N}_{\text{avoid}}) \cup \{j\}$) ▷*New child node where feature $j$ must be avoided*

25:     $\mathcal{Q}$.push(ChildNode1, ChildNode2) ▷*Put the two newly generated nodes into our queue.*

26: **return** $\boldsymbol{\beta}_{\text{best}}$ ▷*Return the current best solution when time limit is reached or there is no node in the queue*

---

## E.2 Fast Lower Bound Calculation

---

**Algorithm 2** Node.lowerSolveFast() $\rightarrow \mathcal{L}_{\text{lower}}$

---

**Attributes in the Node class:** dataset $\mathcal{D} = \{(\boldsymbol{x}_i, y_i)\}_{i=1}^n$, sparsity constraint $k$, $\ell_2$ coefficient $\lambda_2$, set of features that must be selected $\mathcal{N}_{\text{select}}$, set of features that must be avoided $\mathcal{N}_{\text{avoid}}$.
**Output:** a lower bound for the node $\mathcal{L}_{\text{lower}}$.

1: $\boldsymbol{\gamma} = \operatorname{argmin}_{\boldsymbol{\beta}, \beta_j=0 \text{ if } j \in \mathcal{N}_{\text{avoid}}} \boldsymbol{\beta}^T \boldsymbol{X}^T \boldsymbol{X} \boldsymbol{\beta} - 2\boldsymbol{y}^T \boldsymbol{X} \boldsymbol{\beta} + \lambda_2 \|\boldsymbol{\beta}\|_2^2$        ▷*Solve the ridge regression problem. We avoid using coordinates belonging to $\mathcal{N}_{avoid}$ by fixing the coefficients on those coordinates to be $0$ .*

2: $\mathcal{L}_{\text{ridge}} = \boldsymbol{\gamma}^T \boldsymbol{X}^T \boldsymbol{X} \boldsymbol{\gamma} - 2\boldsymbol{y}^T \boldsymbol{X} \boldsymbol{\gamma} + \lambda_2 \|\boldsymbol{\gamma}\|_2^2$        ▷*Loss for the ridge regression.*

3: $\mathcal{L}_{\text{extra}} = (\lambda_2 + \lambda_{\min}(\boldsymbol{X}^T \boldsymbol{X})) \cdot \text{SumBottom}_{p-|\mathcal{N}_{\text{avoid}}|-k}(\{\beta_j^2, \text{ for } \forall j \in [1, ..., p] \setminus (\mathcal{N}_{\text{avoid}} \cup \mathcal{N}_{\text{select}})\})$        ▷*First of all, we are restricted to the coordinates not belonging to $\mathcal{N}_{avoid}$. Second, coordinates in the set $\mathcal{N}_{select}$ must be selected. If we work through the strong convexity argument in Section 3.1 in the main paper, we would get this formula.*

4: $\mathcal{L}_{\text{lower}} = \mathcal{L}_{\text{ridge}} + \mathcal{L}_{\text{extra}}$

5: **return** $\mathcal{L}_{\text{lower}}$

---

### E.3 Refine the Lower Bound through the ADMM method

Next, we show how to get a tighter lower bound through the ADMM method.

---

**Algorithm 3** Node.lowerSolveTightADMM() $\rightarrow \mathcal{L}_{\text{lower}}$

---

**Attributes in the Node class:** dataset $\mathcal{D} = \{(\boldsymbol{x}_i, y_i)\}_{i=1}^n$, sparsity constraint $k$, $\ell_2$ coefficient $\lambda_2$, set of features that must be selected $\mathcal{N}_{\text{select}}$, and iteration limit $T_{\text{iter}}$ for ADMM (For simplicity of presentation, we assume $\mathcal{N}_{\text{avoid}}$ is empty. If $\mathcal{N}_{\text{avoid}} \neq \emptyset$, we can do a selection and relabeling step on $\boldsymbol{X}, \boldsymbol{y}, \boldsymbol{Q}_\lambda, \mathcal{N}_{\text{select}}$ to focus solely on coordinates not belonging to $\mathcal{N}_{\text{avoid}}$).
**Output:** a lower bound for the node $\mathcal{L}_{\text{lower}}$.

1: $\boldsymbol{\gamma}^1 = \text{argmin}_{\boldsymbol{\beta}} \, \boldsymbol{\beta}^T \boldsymbol{X}^T \boldsymbol{X} \boldsymbol{\beta} - 2\boldsymbol{y}^T \boldsymbol{X} \boldsymbol{\beta} + \lambda_2 \|\boldsymbol{\beta}\|_2^2$ ▷*Initialize $\boldsymbol{\gamma}$ by solving the ridge regression problem. This will be used as a warm start for the ADMM method below. To avoid redundant computation, we can use the value for $\boldsymbol{\gamma}$ from Node.lowerSolveFast().*

2: $\boldsymbol{p}^1 = \boldsymbol{X}^T \boldsymbol{y} - \boldsymbol{Q}_\lambda \boldsymbol{\gamma}$ ▷*Initialize $\boldsymbol{p}$ as stated in Problem 18.*

3: $\boldsymbol{q}^1 = \boldsymbol{0}$ ▷*Initialize the scaled dual variable $\boldsymbol{q}$ in the ADMM.*

4: $\rho = 2/\sqrt{\lambda_{\max}(\boldsymbol{Q}_\lambda)\lambda_{\min>0}(\boldsymbol{Q}_\lambda)}$ ▷*Calculate the step size used in the ADMM*

5: **for** t = 2, 3, ...$T_{\text{iter}}$. **do**

6: $\quad \boldsymbol{\gamma}^t = (\frac{2}{\rho}\boldsymbol{I} + \boldsymbol{Q}_\lambda)^{-1}(\boldsymbol{X}^T\boldsymbol{y} - \boldsymbol{p}^{t-1} - \boldsymbol{q}^{t-1})$ ▷*Update $\boldsymbol{\gamma}$ in the ADMM.*

7: $\quad \boldsymbol{\theta}^t = 2\boldsymbol{Q}_\lambda \boldsymbol{\gamma}^t + \boldsymbol{p}^{t-1} - \boldsymbol{X}^T\boldsymbol{y}$ ▷*Update $\boldsymbol{\theta}$ in the ADMM.*

8: $\quad \boldsymbol{a} = \boldsymbol{X}^T\boldsymbol{y} - \boldsymbol{\theta}^t - \boldsymbol{q}^{t-1}$ ▷*This line and the next 3 lines are used to calculate variables used for setting up the isotonic regression problem.*

9: $\quad$ Let $\mathcal{J}$ be the indices of the top $k - |\mathcal{N}_{\text{select}}|$ terms of $|a_j|$ with the constraint that any element in $\mathcal{J}$ does not belong to $\mathcal{N}_{\text{select}}$.

10: $\quad$ Let $w_j = 1$ if $j \notin \mathcal{J}$ and $j \notin \mathcal{N}_{\text{select}}$ and let $w_j = \frac{2}{\rho(\lambda_2+\lambda)+1}$ otherwise.

11: $\quad$ Let $b_j = \frac{|a_j|}{w_j}$.

12: $\quad$ Solve the problem $\hat{\boldsymbol{v}} = \text{argmin}_{\boldsymbol{v}} \sum_j w_j(v_j - b_j)^2$ such that for $\forall i, l \notin \mathcal{N}_{\text{select}}, v_i \geq v_l$ if $|a_i| \geq |a_l|$. ▷*We can decompose this optimization problem into two independent problems with coordinates belonging to $\mathcal{N}_{select}$ (can be solved individually) and coordinates not belonging to $\mathcal{N}_{select}$ (can be solved via standard isotonic regression).*

13: $\quad \boldsymbol{p}^t = sign(\boldsymbol{a}) \odot \hat{\boldsymbol{v}}$ ▷*Update $\boldsymbol{p}$ in the ADMM. The symbol $\odot$ denotes the element-wise (Hadamard) product.*

14: $\quad \boldsymbol{q}^t = \boldsymbol{q}^{t-1} + \boldsymbol{\theta}^t + \boldsymbol{p}^t - \boldsymbol{X}^T\boldsymbol{y}$ ▷*Update $\boldsymbol{q}$ in the ADMM.*

15: **end for**

16: $\mathcal{L}_{\text{lower}} = h(\boldsymbol{\gamma}^{T_{\text{iter}}})$ ▷*Calculate the lower bound.*

17: **return** $\mathcal{L}_{\text{lower}}$

---

### E.4 (Optional) An alternative to the Lower Bound Calculation via Subgradient Ascent and the CMF method

In addition to the ADMM method, we also provide an alternative algorithm showing how to get a tighter lower bound through the subgradient ascent algorithm using the CMF method. Note that this subgradient method has been superseded by the ADMM method, which converges faster. We provide this method in the Appendix for the sake of completeness and for future readers to consult.

---

**Algorithm 4** Node.lowerSolveTightSubgradient() $\rightarrow \mathcal{L}_{\text{lower}}$

---

**Attributes in the Node class:** dataset $\mathcal{D} = \{(\boldsymbol{x}_i, y_i)\}_{i=1}^n$, sparsity constraint $k$, $\ell_2$ coefficient $\lambda_2$, set of features that must be selected $\mathcal{N}_{\text{select}}$, set of features that must be avoided $\mathcal{N}_{\text{avoid}}$, and iteration limit $T_{\text{iter}}$ for subgradient ascent
**Output:** a lower bound for the node $\mathcal{L}_{\text{lower}}$.

1: $\boldsymbol{\gamma} = \text{argmin}_{\boldsymbol{\beta}, \beta_j = 0 \text{ if } j \in \mathcal{N}_{\text{avoid}}} \boldsymbol{\beta}^T \boldsymbol{X}^T \boldsymbol{X} \boldsymbol{\beta} - 2\boldsymbol{y}^T \boldsymbol{X} \boldsymbol{\beta} + \lambda_2 \|\boldsymbol{\beta}\|_2^2$     ▷*Solve the ridge regression problem and avoid using coordinates in $\mathcal{N}_{avoid}$. This will be used as a warm start for the subgradient ascent procedure below. To avoid redundant computation, we can use the value for $\boldsymbol{\gamma}$ from Node.lowerSolveFast()*
2: $\boldsymbol{v} = \boldsymbol{\gamma}$     ▷*Initialization for $\boldsymbol{v}^t$ at iteration step $t = 1$*
3: $h^* = \mathcal{L}_{\text{ridge}}(\boldsymbol{\beta}_{upper}^{root})$  ▷*get an estimate of $max_{\boldsymbol{\gamma}} h(\boldsymbol{\gamma})$; if $\boldsymbol{\beta}_{upper}^{root}$ has not been solved at the root node, solve $\boldsymbol{\beta}_{upper}^{root}$ by calling the upperSolve() function at the root node.*
4: $\mathcal{L}_{\text{lower}} = h(\boldsymbol{\gamma})$
5: **for** t = 2, 3, ...$T_{\text{iter}}$ **do**
6:     $\boldsymbol{\gamma}, \boldsymbol{v} = \text{Node.subgradAscentCMF}(\boldsymbol{\gamma}, \boldsymbol{v})$   ▷*Perform one step of subgradient ascent using the CMF method*
7:     $\mathcal{L}_{\text{lower}} = \max(\mathcal{L}_{\text{lower}}, h(\boldsymbol{\gamma}))$   ▷*Loss decreases stochastically during the subgradient ascent procedure. We keep the largest value of $h(\boldsymbol{\gamma})$*
8: **end for**
9: **return** $\mathcal{L}_{\text{lower}}$

---

In the above algorithm, we call a new sub algorithm called subGradAscentCMF. We provide the details on how to do this below. We try to maximize $h(\boldsymbol{\gamma})$ through iterative subgradient ascent. A more sophisticated algorithm (CMF method [23]) can be applied to adaptively choose the step size, which has been shown to achieve faster convergence in practice. The computational procedure for the subgradient ascent algorithm using the CMF method [18, 23] is as follows. Given $\boldsymbol{\gamma}^{t-1}$ at iteration $t - 1$, $\boldsymbol{\gamma}^t$ can be computed as:

$$\boldsymbol{w}^t = \frac{\partial \mathcal{L}_{\text{ridge}-\lambda}^{\text{saddle}}(\boldsymbol{\gamma}, Z(\boldsymbol{\gamma}^{t-1}))}{\partial \boldsymbol{\gamma}} \bigg|_{\boldsymbol{\gamma} = \boldsymbol{\gamma}^{t-1}} \qquad \text{\# get the subgradient}$$

$$s_t = \max(0, -d_t (\boldsymbol{v}^{t-1})^T \boldsymbol{w}^t / \|\boldsymbol{v}^{t-1}\|_2^2)$$
$$\text{\# } s_t \text{ will be used as a smoothing factor between } \boldsymbol{v}^{t-1} \text{ and } \boldsymbol{w}^t$$

$$\boldsymbol{v}^t = \boldsymbol{w}^t + s_t \boldsymbol{v}^{t-1} \qquad \text{\# combine } \boldsymbol{v}^{t-1} \text{ and } \boldsymbol{w}^t \text{ using the smoothing factor}$$

$$\alpha_t = (h^* - h(\boldsymbol{\gamma}^{t-1})) / \|\boldsymbol{v}^t\|_2^2$$
$$\text{\# calculate the step size; } h^* \text{ is an estimate of } \max_{\boldsymbol{\gamma}} h(\boldsymbol{\gamma}); \text{ here we let } h^* = \mathcal{L}_{\text{ridge}}(\boldsymbol{\beta}_{upper}^{root})$$

$$\boldsymbol{\gamma}^t = \boldsymbol{\gamma}^{t-1} + \alpha_t \boldsymbol{v}^t \qquad \text{\# perform subgradient ascent}$$

where at the beginning of the iteration ($t = 1$), $\boldsymbol{v}^1 = \boldsymbol{0}$ and we choose $d_t = 1.5$ as a constant for all iterations as recommended by the original paper. Below, we provide the detailed algorithm to perform subgradient ascent using the CMF method.

**Algorithm 5** Node.subgradAscentCMF($\boldsymbol{\gamma}, \boldsymbol{v}, h^*) \rightarrow \boldsymbol{\gamma}, \boldsymbol{v}$

---

**Attributes in the Node class:** dataset $\mathcal{D} = \{(\boldsymbol{x}_i, y_i)\}_{i=1}^{n}$, sparsity constraint $k$, $\ell_2$ coefficient $\lambda_2$, set of features that must be selected $\mathcal{N}_{\text{select}}$, set of features that must be avoided $\mathcal{N}_{\text{avoid}}$

**Input:** Current variable $\boldsymbol{\gamma}$ for the lower bound function $h(\cdot)$ for Equation 14, current ascent vector $\boldsymbol{v}$, and $h^*$ is an estimate of $\max_{\boldsymbol{\gamma}} h(\boldsymbol{\gamma})$.

**Output:** updated variable $\boldsymbol{\gamma}$ and updated ascent vector $\boldsymbol{v}$.

1: $\hat{\boldsymbol{z}} = \operatorname{argmin}_{\boldsymbol{z}} \mathcal{L}_{\text{ridge}-\lambda}^{\text{saddle}}(\boldsymbol{\gamma}, \boldsymbol{z})$     ▷*get the minimizer for $\boldsymbol{z}$ at the current value for $\boldsymbol{\gamma}$; optimization is performed under the constraint $\sum_{j=1}^{p} z_j \leq k, z_j \in [0,1], z_j = 1$ for $j \in \mathcal{N}_{\text{select}}$ and $z_j = 0$ for $j \in \mathcal{N}_{\text{select}}$*

2: $\boldsymbol{w} = \left(\partial \mathcal{L}_{\text{ridge}-\lambda}^{\text{saddle}}(\boldsymbol{\gamma}, \hat{\boldsymbol{z}})\right) / (\partial \boldsymbol{\gamma})$     ▷*get the subgradient*

3: $\boldsymbol{w}_j = 0$ for $j \in \mathcal{N}_{\text{avoid}}$   ▷*We don't perform any ascent steps on the coordinates belong to $\mathcal{N}_{\text{avoid}}$.*

4: $s = \max(0, -1.5\boldsymbol{v}^T \boldsymbol{w}/\|\boldsymbol{v}\|_2^2)$     ▷*Calculate the smoothing factor*

5: $\boldsymbol{v} = \boldsymbol{w} + s\boldsymbol{v}$     ▷*update the ascent vector by combining the current subgradient and previous ascent vector*

6: $\alpha = (h^* - h(\boldsymbol{\gamma})) / \|\boldsymbol{v}\|_2^2$     ▷*calculate the step size for the ascent step*

7: $\boldsymbol{\gamma} = \boldsymbol{\gamma} + \alpha\boldsymbol{v}$     ▷*Perform one step of ascent along the direction of the ascent vector*

8: **return** $\boldsymbol{\gamma}, \boldsymbol{v}$

---

### E.5 Beam Search Algorithm

We provide a visualization of the beam search algorithm in Figure 6

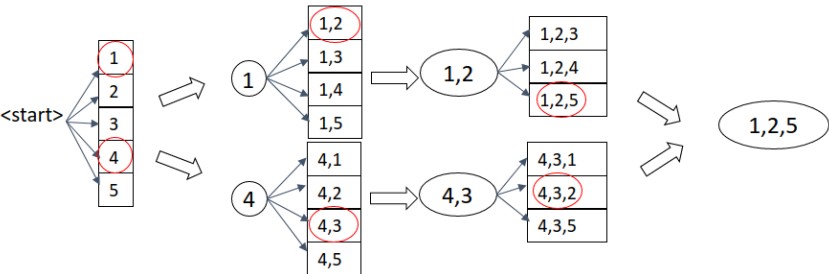

Figure 6: We add into our support one feature at a time by picking the feature which would result in the largest decrease in the objective function. After that, we finetune all coefficients on the support. We keep the top $B$ solutions during each stage of support expansion.

---

**Algorithm 6** Node.upperSolve($B = 50$) $\rightarrow$ $\boldsymbol{\beta}_{\text{upper}}, \mathcal{L}_{\text{upper}}$

---

**Attributes in the Node class:** dataset $\mathcal{D} = \{(\boldsymbol{x}_i, y_i)\}_{i=1}^n$, sparsity constraint $k$, $\ell_2$ coefficient $\lambda_2$, a set of feature coordinates that must be selected $\mathcal{N}_{\text{select}}$, and a set of feature coordinates that must be avoided $\mathcal{N}_{\text{avoid}}$.

**Input:** beam search size $B$, which is set to be 50 as the default value.

**Output:** a near-optimal solution $\boldsymbol{\beta}_{\text{upper}}$ with $\|\beta_{\text{upper}}\|_0 = k$ and the corresponding loss $\mathcal{L}_{\text{upper}}$.

1: $\boldsymbol{\beta}' = \text{argmin}_{\boldsymbol{\beta}, \beta_j = 0 \text{ for } j \notin \mathcal{N}_{\text{select}}} \boldsymbol{\beta}^T \boldsymbol{X}^T \boldsymbol{X} \boldsymbol{\beta} - 2\boldsymbol{y}^T \boldsymbol{X}\boldsymbol{\beta} + \lambda_2 \|\boldsymbol{\beta}\|_2^2$   ▷*Initialize the coefficients by fitting on the must-be-selected coordinates.*

2: $\mathcal{W} = \{\boldsymbol{\beta}'\}$        ▷*collection of solutions at each iteration (initially we only have one solution)*

3: $\mathcal{F} = \emptyset$                      ▷*Initialize the collection of found supports as an empty set*

4: **for** $t = |\mathcal{N}_{\text{select}}| + 1, ..., k$ **do**

5:    $\mathcal{W}_{\text{tmp}} \leftarrow \emptyset$

6:    **for** $\boldsymbol{\beta}' \in \mathcal{W}$ **do**                        ▷*Each of these has support $t - 1$*

7:      $(\mathcal{W}', \mathcal{F}) \leftarrow \text{ExpandSuppBy1}(\mathcal{D}, \boldsymbol{\beta}', \mathcal{F}, B, \mathcal{N}_{\text{avoid}})$.    ▷*Returns $\leq B$ models with supp. $t$*

8:      $\mathcal{W}_{\text{tmp}} \leftarrow \mathcal{W}_{\text{tmp}} \cup \mathcal{W}'$

9:    **end for**

10:    Reset $\mathcal{W}$ to be the $B$ solutions in $\mathcal{W}_{\text{tmp}}$ with the smallest ridge regression loss values.

11: **end for**

12: Pick $\boldsymbol{\beta}_{\text{upper}}$ from $\mathcal{W}$ with the smallest ridge regression loss.

13: $\mathcal{L}_{\text{upper}} = \boldsymbol{\beta}_{\text{upper}}^T \boldsymbol{X}^T \boldsymbol{X} \boldsymbol{\beta}_{\text{upper}} - 2\boldsymbol{y}^T \boldsymbol{X} \boldsymbol{\beta}_{\text{upper}} + \lambda_2 \|\boldsymbol{\beta}_{\text{upper}}\|_2^2$

14: **return** $\boldsymbol{\beta}_{\text{upper}}, \mathcal{L}_{\text{upper}}$.

---

---

**Algorithm 7** ExpandSuppBy1($\mathcal{D}, \boldsymbol{\beta}', \mathcal{F}, B, \mathcal{N}_{\text{avoid}}$) $\rightarrow$ $\mathcal{W}, \mathcal{F}$

---

**Input:** dataset $\mathcal{D} = \{(\boldsymbol{x}_i, y_i)\}_{i=1}^n$, current solution $\boldsymbol{\beta}'$, set of found supports $\mathcal{F}$, beam search size $B$, and a set of feature coordinates that must be avoided $\mathcal{N}_{\text{avoid}}$.

**Output:** a collection of solutions $\mathcal{W} = \{\boldsymbol{\beta}^t\}$ with $\|\boldsymbol{\beta}^t\|_0 = \|\boldsymbol{\beta}'\|_0 + 1$ for $\forall t$. All of these solutions include the support of $\boldsymbol{\beta}'$ plus one more feature. None of the solutions have the same support as any element of $\mathcal{F}$, meaning we do not discover the same support set multiple times. We also output the updated $\mathcal{F}$.

1: Let $\mathcal{S}^c \leftarrow \{j \mid \beta_j' = 0 \text{ and } j \notin \mathcal{N}_{\text{avoid}}\}$                      ▷*Non-support of the given solution*

2: Pick up to $B$ coords ($j$'s) in $\mathcal{S}^c$ with largest decreases in $\Delta_j \mathcal{L}_{\text{ridge}}(\cdot)$, call this set $\mathcal{J}'$.   ▷*We will use these supports, which include the support of $\boldsymbol{w}$ plus one more.*

3: $\mathcal{W} \leftarrow \emptyset$

4: **for** $j \in \mathcal{J}'$ **do**                                   ▷*Optimize on the top $B$ coordinates*

5:    **if** $(\text{supp}(\boldsymbol{w}) \cup \{j\}) \in \mathcal{F}$ **then**

6:      **continue**                                ▷*We've already seen this support, so skip.*

7:    **end if**

8:    $\mathcal{F} \leftarrow \mathcal{F} \cup \{supp(\boldsymbol{w}) \cup \{j\}\}$.                              ▷*Add new support to $\mathcal{F}$.*

9:    $\boldsymbol{u}^* \in \text{argmin}_{\boldsymbol{u}} \mathcal{L}_{\text{ridge}}(\mathcal{D}, \boldsymbol{u})$ with $\text{supp}(\boldsymbol{u}) = \text{supp}(\boldsymbol{w}') \cup \{j\}$ .   ▷*Update coefficients on the newly expanded support*

10:    $\mathcal{W} \leftarrow \mathcal{W} \cup \{\boldsymbol{u}^*\}$                              ▷*cache the solution to the central pool*

11: **end for**

12: **return** $\mathcal{W}$ and $\mathcal{F}$.

---

## E.6 Branching

---

**Algorithm 8** Node.getBranchCoord($\boldsymbol{\beta}$) $\rightarrow$ $j^*$

---

**Attributes in the Node class:** dataset $\mathcal{D} = \{(\boldsymbol{x}_i, y_i)\}_{i=1}^n$, sparsity constraint $k$, $\ell_2$ coefficient $\lambda_2$, a set of feature coordinates that must be selected $\mathcal{N}_{\text{select}}$, and a set of feature coordinates that must be avoided $\mathcal{N}_{\text{avoid}}$.
**Output:** a single coordinate $j$ on which the node should branch in the BnB tree.

1: $j^* = \text{argmax}_{j \notin \mathcal{N}_{\text{select}}} -2\boldsymbol{\beta}^T(\boldsymbol{X}^T\boldsymbol{X} + \lambda_2\boldsymbol{I})\beta_j\boldsymbol{e}_j + ((\boldsymbol{X}^T\boldsymbol{X})_{jj} + \lambda_2)\beta_j^2 + 2\boldsymbol{y}^T\boldsymbol{X}\beta_j\boldsymbol{e}_j$ ▷*select the coordinate whose coefficient, if we set to 0, would result in the greatest increase in our ridge regression loss*
2: **return** $j^*$

---

## E.7 A Note on the Connection with the Interlacing Eigenvalue Property

During the branch-and-bound procedure, finding the smallest eigenvalue of any principle submatrix of $\boldsymbol{X}^T\boldsymbol{X}$ is computationally expensive if we do it for every node. Fortunately, we can get a lower bound for the smallest eigenvalue based on the following Interlacing Eigenvalue Property [36]:

**Theorem E.1.** *Suppose $\boldsymbol{A} \in \mathbb{R}^{n \times n}$ is symmetric. Let $\boldsymbol{B} \in \mathbb{R}^{m \times m}$ with $m < n$ be a principal submatrix (obtained by deleting both $i$-th row and $i$-th column for some values of $i$). Suppose $\boldsymbol{A}$ has eigenvalues $\gamma_1 \leq ... \leq \gamma_n$ and $\boldsymbol{B}$ has eigenvalues $\alpha_1 \leq ... \leq \alpha_m$. Then*

$$\gamma_k \leq \alpha_k \leq \gamma_{k+n-m} \quad for\ k = 1, ..., m$$

*And if $m = n - 1$,*

$$\gamma_1 \leq \alpha_1 \leq \gamma_2 \leq \alpha_2 \leq ... \leq \alpha_{n-1} \leq \gamma_n$$

Using this Theorem, we can let $\lambda = \lambda_{\min}(\boldsymbol{X}^T\boldsymbol{X})$ for every node in the branch-and-bound tree in our algorithm. This saves us a lot of computational time because we only have to calculate the smallest eigenvalue once in our root node. However, if the minimum eigenvalue were to be recalculated at each node efficiently, the lower bounds would be even tighter.

## F More Experimental Setup Details

### F.1 Computing Platform

All experiments were run on the 10x TensorEX TS2-673917-DPN Intel Xeon Gold 6226 Processor, 2.7Ghz. We set the memory limit to be 100GB.

### F.2 Baselines, Licenses, and Links

**Commerical Solvers** For the MIP formulations listed in Appendix D, we used Gurobi and MOSEK. We implemented the SOS1, bigM, perspective, and eigen-perspective formulations in Gurobi. The Gurobi version is 10.0, which can be installed through conda (`https://anaconda.org/gurobi/gurobi`). We used the Academic Site License. We implemented the perspective formulations and relaxed convex optimal perspective formulations in MOSEK. The MOSEK version is 10.0, which can be installed through conda (`https://anaconda.org/MOSEK/mosek`). We used the Personal Academic License.

**PySINDy, STLSQ, SSR, E-STLSQ** We used the PySINDy library (`https://github.com/dynamicslab/pysindy`) to perform the differential equation experiments. The license is MIT. The heuristic methods STLSQ, SSR, and E-STLSQ are implemented in PySINDy.

**MIOSR** For the differential equation experiments, the MIOSR code is available on GitHub(`https://github.com/wesg52/sindy_mio_paper`). The license for this code repository is MIT.

**SubsetSelectionCIO** The SubsetSelectionCIO code is available on GitHub (`https://github.com/jeanpauphilet/SubsetSelectionCIO.jl`) with commit version 8b03d1fba9262b150d6c0f230c6bf9e8ee57f92e. The license for this code repository is MIT "Expat" License (`https://github.com/jeanpauphilet/SubsetSelectionCIO.jl/blob/master/LICENSE.md`). Due to package compatibility, we use Gurobi 9.0 for SubsetSelectionCIO. See the SusbetSelectionCIO package version specification at this link: `https://github.com/jeanpauphilet/SubsetSelectionCIO.jl/blob/master/Project.toml`.

**l0bnb** The l0bnb code is available on GitHub (`https://github.com/alisaab/l0bnb`) with commit version 54375f8baeb64baf751e3d0effc86f8b05a386ce. The license for this code repository is MIT.

**Sparse Simplex** The Sparse Simplex code is available at the author's website (`https://sites.google.com/site/amirbeck314/software?authuser=0`). We ran the greedy sparse simplex algorithm. The license for this code repository is GNU General Public License 2.0.

**Block Decomposition** The block decomposition code is available at the author's website (`https://yuangzh.github.io`). The author didn't specify a license for this code repository. However, we didn't modify the code and only used it for benchmarking purposes, which is allowed by the software community. Besides the block decompostion algorithm, the code repository also contains reimplementations for other algorithms, such as GP, SSP, OMP, proximal gradient l0c, QPM, ROMP. We used these reimplementations in the differential equation experiments as well. Please see [60] for details of these baselines.

### F.3 Differential Equation Experiments

We compare the certifiable method MIOSR along with other baselines on solving dynamical system problems. In particular, for each dynamical system, we perform 20 trials with random initial conditions for 12 different trajectory lengths. Each dynamical system has observed positions every 0.002 seconds, so a larger trajectory length corresponds to a larger sample size $n$. Derivatives were approximated using a smoothing finite difference with a window size of 9 points. The derivatives were calculated automatically by the PySINDY library. Each trajectory has 0.2% noise added to it to demonstrate robustness to noise. For the SINDy approach, true dynamics exist in the finite-dimensional candidate functions/features. We investigate which method recovers the correct dynamics more effectively.

### F.3.1 Model Selection

Model selection was performed via cross-validation with training data encompassing the first 2/3rds of a trajectory and the final third used for validation. Predicted derivatives were compared to the validation set, with the best model being the one that minimizes $AIC_c$ for sparse regression in the setting of dynamical systems [45].

### F.3.2 Hyperparameter Choices

With respect to each trajectory, each algorithm was trained with various combinations of hyperparameter choices. For both MIOSR and our method OKRidge, the ridge regression hyperparameter choices are $\lambda_2 \in \{10^{-5}, 10^{-3}, 10^{-2}, 0.05, 0.2\}$, and the sparsity level hyperparameter choices are $k \in \{1, 2, 3, 4, 5\}$. As in the experiments of MIOSR [9], we set the time limit for each optimization to be 30 seconds for both MIOSR and our method OKRidge. For SSR and STLSQ, the ridge regression hyperparameter choices are $\lambda_2 \in \{10^{-5}, 10^{-3}, 10^{-2}, 0.05, 0.2\}$. SSR and STLSQ may automatically pick a sparsity level, but they require a thresholding hyperparameter. We choose 50 thresholding hyperparameters with values between $10^{-3}$ and $10^{1.5}$, depending on the system being fit. Together there are 250 different hyperparameter combinations we need to run on each trajectory length for for SSR and STLSQ. When fitting E-STLSQ, cross-validation is not performed as the chosen hyperparameters for STLSQ are used instead. For SSR, MIOSR, and OKRidge, we cross validate on each dimension of the data and combine the strongest results into a final model. For STLSQ and E-STLSQ, no post-processing is done and the same hyperparameters are used for each dimension.

### F.3.3 Evaluation Metrics

Three metrics are considered to evaluate each algorithm: true positivity rate, $L_2$ coefficient error, and root-mean-squared error (RMSE). The true positivity rate is defined as the ratio of the number of correctly chosen features divided by the sum of the number of true features and incorrectly chosen features. This metric equals 1 if and only if the correct form of the dynamical system is recovered. $L_2$ coefficient error is defined as the sum of squared errors of the predicted and the true coefficients. Finally, for each model fit, we produce 10 simulations with different initial conditions of the true model, and calculate the root-mean-squared error (RMSE) of the predicted derivatives and the true derivatives. Besides these three metrics, we also report the running times, comparing OKRidge with other optimal ridge regression methods.

# G  More Experimental Results on Synthetic Benchmarks

## G.1  Results Summarization

$\ell_2$ **Perturbation Study**   We first compare with SOS1, Big-M (Big-M50 means that the big-M value is set to 50), Perspective, and Eigen-Perspective formulations with different $\ell_2$ regularizations, with and without using beam search as the warm start. We have three levels of $\ell_2$ regularizations: $0.001, 0.1, 10.0$. The results are shown in Tables 1-60 in Appendix G.2. When the method has "-" in the upper bound or gap column, it means that method does not produce any nontrivial solution within the time limit.

Extensive results have shown that OKRidge outperforms all exisiting MIP formulations solved by the state-of-the-art MIP solver Gurobi. With the beam search solutions (usually they are already the optimal solutions) as the warm starts, Gurobi are faster than without using warm starts. However, OKRidge still outperforms all existing MIP formulations. If Gurobi can solve the problem within the time limit, OKRidge is usually usually orders of magnitude faster than all baselines. When Gurobi cannot solve the problem within the time limit, OKRidge can either solve the problem to optimality or achieve a much smaller optimality gap than all baselines, especially in the presence of high feature correlation and/or high feature dimension.

**Big-M Perturbation Study**   We also conduct perturbation study on the big-M values. We pick three values: 50, 20, and 5. In practice, it is difficult to select the right value because setting the big-M value too small will accidentally cut off the optimal solutions. Nonetheless, we still pick a small big-M value (5; the optimal coefficients are all 1's in the synthetic benchmarks) to do a thorough comparison. The results are shown in Tables 66-120 in Appendix G.3. When the method has "-" in the upper bound or gap column, it means that method does not produce any nontrivial solution within the time limit.

The results show that although choosing a small big-M value can often lead to smaller running time, OKRidge is still orders of magnitude faster than the big-M formulation, and achieves the smallest optimality gap when all algorithms reach the time limit. Additionally, we find that having a small big-M value will sometimes lead to increased running time on solving the root relaxation. Sometimes, when the root relaxation is not finished within the 1h time limit, Gurobi will provide a loose lower bound, resulting in large optimality gaps. Aside from accidentally cutting off the optimal solution, this is another reason against choosing a very small big-M value.

**Comparison with MOSEK, SubsetSelectionCIO, and l0bnb**   For the purpose of thorough comparison, we also solve the problem with other MIP solvers/formulations. The results are shown in Tables 121-Table 135 in Appendix G.4. When the method has "-" in the upper bound or gap column, it means that method does not produce any nontrivial solution within the time limit. For the MOSEK solver, when the upper bound and gap columns have "-" values, and the time is exactly 3600.00, this means that MOSEK runs out of memory (100GB).

First, we try to solve the problem using the commercial conic solver called MOSEK. We use MOSEK to solve the perspective formulation, with and without using beam search solutions as warm starts. The results indicate that Gurobi is in general much faster and more memory-efficient than MOSEK in solving the perspective formulation. We also use MOSEK to solve the relaxed convex optimal perspective formulation (relaxing the binary constraint $z_j \in \{0, 1\}$ to the interval constraint $z_j \in [0, 1]$). we only solve the relaxed convex SDP problem because MOSEK does not support MI-SDP. The results show that MOSEK has limited capacity in solving the perspective formulation. MOSEK also runs out of memory for large-scale problems ($n = 100000$, $p = 3000$ or $p = 5000$). Besides, the experimental results show that convex SDP problem is not scalable to high dimensions. For the high-dimensional cases, OKRidge solving the MIP problem is usually much faster than MOSEK solving a convex SDP problem.

Second, we uses the SubsetSelectionCIO method, which solves the original sparse regression problem using branch-and-cut (lazy-constraint callbacks) in Gurobi. However, as the results indicate, SubsetSelectionCIO produces wrong optimal solutions when $\ell_2$ regularization is small and the number of samples is large ($n = 100000$). When SubsetSelectionCIO does not have errors, the running times or optimality gaps are much worse than those of OKRidge.

Lastly, we solve the sparse ridge regression problem using l0bnb. In contrast to other methods, l0bnb controlls the sparsity level implicitly through the $\ell_0$ regularization, instead of the $\ell_0$ constraint. l0bnb needs to try several different $\ell_0$ regularizations to find the desired sparsity level. For each $\ell_0$ regularization coefficient, we set the time limit to be 180 seconds, and we stop the l0bnb algorithm after a time limit of 7200 seconds (this is longer than 3600 seconds because we want to give enough time for l0bnb to try different $\ell_0$ coefficients). Moreover, l0bnb imposes a box constraint (similar to the big-M method) on all coefficients. The box constraint tightens the lower bound and potentially accelerates the pruning process, but it could also cut off the optimal solution if the "M" value is not chosen properly or large enough. To our best knowledge, there is no reliable way to calculate the "M" value. The l0bnb package internally estimates the "M" value in a heuristic way. Despite the box constraint giving l0bnb an extra advantage, we still compare this baseline with OKRidge. As the results indicate, l0bnb often takes a long time to find the desired sparsity level by trying different $\ell_0$ regularizations. In some cases, l0bnb finds a solution with support size different than the required sparsity level (demonstrating the difficulty of finding a good $\ell_0$ coefficient). In other cases, l0bnb still has an optimality gap after reaching the time limit while OKRidge can finish certification within seconds.

**Discussion on the Case $p \gg n$**   For the application of discovering differential equations, the datasets usually have $n \geq p$. Therefore, the case $p \gg n$ is not the main focus of this paper. Nonetheless, we provide additional experiments to give a complete view of our algorithm.

We set the number of features $p = 3000$ and the number of samples $n = 100, 200, 500$. The feature correlation is set to be $\rho = 0.1$ since this is already a very challenging case. The results are shown in Tables 151-153 in Appendix G.5. The results show that OKRidge is competitive with other MIP methods. l0bnb is the only method that can solve the problem when $n = 500$ within the time limit. However, this algorithm has difficulty in finding the desired sparsity level in the case of $n = 100$ and $n = 200$ even with a large time limit of 7200 seconds. In some cases, MOSEK gives the best optimality gap. However, when $n \geq p$ as we have shown in the previous subsections, MOSEK is not scalable to high dimensions. Additionally, the results in differential equations show that MOSEK and l0bnb are not competitive in discovering differential equations (datasets have $n \geq p$), which is the main focus of this paper.

In the future, we plan to extend OKRidge to develop a more scalable algorithm for the case $p \gg n$.

**Fast Solve vs. ADMM**   In the main paper, we proposed two ways to calculate the saddle point problem. The first approach, fast solve, is based on an analytical formula. The second approach is based on the ADMM method. Intuitively, since the fast solve method is only maximizing a lower bound of the saddle point objective and ADMM is maximizing the saddle point objective directly, we expect that ADMM would give much tighter lower bounds than the fast solve approach does. Here, we give empirical evidence to support our claim. The results can be seen in Figures 7 and 8.

## G.2 Comparison with SOS1, Big-M, Perspective, and Eigen-Perspective Formulations with different $\ell_2$ Regularizations

### G.2.1 Synthetic 1 Benchmark with $\lambda_2 = 0.001$ with No Warmstart

| # of features | method | upper bound | time(s) | gap(%) |
|---|---|---|---|---|
| 100 | SOS1 | $-2.10878 \cdot 10^6$ | 1.68 | 0.00 |
| 100 | big-M50 | $-2.10874 \cdot 10^6$ | 0.39 | 0.00 |
| 100 | persp. | $-2.10878 \cdot 10^6$ | 0.08 | 0.00 |
| 100 | eig. persp. | $-2.10878 \cdot 10^6$ | **0.01** | **0.00** |
| 100 | ours | $-2.10878 \cdot 10^6$ | 0.29 | 0.00 |
| 500 | SOS1 | $-2.12833 \cdot 10^6$ | 8.04 | 0.00 |
| 500 | big-M50 | $-2.12828 \cdot 10^6$ | 36.80 | 0.00 |
| 500 | persp. | $-2.12833 \cdot 10^6$ | 20.29 | 0.00 |
| 500 | eig. persp. | $-2.12832 \cdot 10^6$ | 0.69 | 0.00 |
| 500 | ours | $-2.12833 \cdot 10^6$ | **0.37** | **0.00** |
| 1000 | SOS1 | $-2.11005 \cdot 10^6$ | 82.53 | 0.00 |
| 1000 | big-M50 | $-2.11005 \cdot 10^6$ | 1911.41 | 0.00 |
| 1000 | persp. | $-2.11005 \cdot 10^6$ | 1143.11 | 0.00 |
| 1000 | eig. persp. | $-2.11005 \cdot 10^6$ | 4.11 | 0.00 |
| 1000 | ours | $-2.11005 \cdot 10^6$ | **0.58** | **0.00** |
| 3000 | SOS1 | $-2.10230 \cdot 10^6$ | 3600.15 | 0.59 |
| 3000 | big-M50 | $-7.83869 \cdot 10^5$ | 3601.20 | 169.78 |
| 3000 | persp. | $-2.10231 \cdot 10^6$ | 2288.93 | 0.00 |
| 3000 | eig. persp. | $-2.10230 \cdot 10^6$ | 102.18 | 0.00 |
| 3000 | ours | $-2.10230 \cdot 10^6$ | **1.73** | **0.00** |
| 5000 | SOS1 | $-1.98939 \cdot 10^6$ | 3600.13 | 6.43 |
| 5000 | big-M50 | $-1.87965 \cdot 10^6$ | 3603.25 | 12.65 |
| 5000 | persp. | $-2.09637 \cdot 10^6$ | 3600.62 | 1.00 |
| 5000 | eig. persp. | - | 3880.65 | - |
| 5000 | ours | $-2.09635 \cdot 10^6$ | **4.22** | **0.00** |

Table 1: Comparison of time and optimality gap on the synthetic datasets (n = 100000, k = 10, $\rho = 0.1$, $\lambda_2 = 0.001$).

| # of features | method | upper bound | time(s) | gap(%) |
|:---:|:---:|:---:|:---:|:---:|
| 100 | SOS1 | $-5.28726 \cdot 10^6$ | 3.37 | 0.00 |
| 100 | big-M50 | $-5.28725 \cdot 10^6$ | 0.42 | 0.00 |
| 100 | persp. | $-5.28726 \cdot 10^6$ | 0.07 | 0.00 |
| 100 | eig. persp. | $-5.28726 \cdot 10^6$ | **0.02** | **0.00** |
| 100 | ours | $-5.28726 \cdot 10^6$ | 0.29 | 0.00 |
| 500 | SOS1 | $-5.31852 \cdot 10^6$ | 9.97 | 0.00 |
| 500 | big-M50 | $-5.31819 \cdot 10^6$ | 89.04 | 0.00 |
| 500 | persp. | $-5.31852 \cdot 10^6$ | 8.44 | 0.00 |
| 500 | eig. persp. | $-5.31852 \cdot 10^6$ | 0.99 | 0.00 |
| 500 | ours | $-5.31852 \cdot 10^6$ | **0.41** | **0.00** |
| 1000 | SOS1 | $-5.26974 \cdot 10^6$ | 88.42 | 0.00 |
| 1000 | big-M50 | $-5.26974 \cdot 10^6$ | 1528.98 | 0.00 |
| 1000 | persp. | $-5.26974 \cdot 10^6$ | 54.96 | 0.00 |
| 1000 | eig. persp. | $-5.26974 \cdot 10^6$ | 300.72 | 0.00 |
| 1000 | ours | $-5.26974 \cdot 10^6$ | **0.52** | **0.00** |
| 3000 | SOS1 | $-5.16793 \cdot 10^6$ | 3600.38 | 2.71 |
| 3000 | big-M50 | $-3.72978 \cdot 10^6$ | 3602.55 | 42.31 |
| 3000 | persp. | $-5.27668 \cdot 10^6$ | 3600.33 | 0.59 |
| 3000 | eig. persp. | $-5.27668 \cdot 10^6$ | 100.58 | 0.00 |
| 3000 | ours | $-5.27668 \cdot 10^6$ | **1.75** | **0.00** |
| 5000 | SOS1 | $-5.16088 \cdot 10^6$ | 3600.14 | 3.16 |
| 5000 | big-M50 | $-4.16126 \cdot 10^6$ | 3602.53 | 27.95 |
| 5000 | persp. | $-5.27126 \cdot 10^6$ | 3600.72 | 1.00 |
| 5000 | eig. persp. | - | 3890.12 | - |
| 5000 | ours | $-5.27126 \cdot 10^6$ | **4.32** | **0.00** |

Table 2: Comparison of time and optimality gap on the synthetic datasets (n = 100000, k = 10, $\rho = 0.3$, $\lambda_2 = 0.001$).

| # of features | method | upper bound | time(s) | gap(%) |
|---|---|---|---|---|
| 100 | SOS1 | $-1.10084 \cdot 10^7$ | 2.97 | 0.00 |
| 100 | big-M50 | $-1.10084 \cdot 10^7$ | 0.41 | 0.00 |
| 100 | persp. | $-1.10084 \cdot 10^7$ | **0.09** | **0.00** |
| 100 | eig. persp. | $-1.10084 \cdot 10^7$ | 0.11 | 0.00 |
| 100 | ours | $-1.10084 \cdot 10^7$ | 0.29 | 0.00 |
| 500 | SOS1 | $-1.10518 \cdot 10^7$ | 9.65 | 0.00 |
| 500 | big-M50 | $-1.10515 \cdot 10^7$ | 89.59 | 0.00 |
| 500 | persp. | $-1.10518 \cdot 10^7$ | 3600.01 | 0.08 |
| 500 | eig. persp. | $-1.10518 \cdot 10^7$ | 25.13 | 0.00 |
| 500 | ours | $-1.10518 \cdot 10^7$ | **0.40** | **0.00** |
| 1000 | SOS1 | $-1.09508 \cdot 10^7$ | 92.16 | 0.00 |
| 1000 | big-M50 | $-1.09506 \cdot 10^7$ | 1757.09 | 0.00 |
| 1000 | persp. | $-1.09508 \cdot 10^7$ | 3600.04 | 0.20 |
| 1000 | eig. persp. | $-1.09508 \cdot 10^7$ | 226.82 | 0.00 |
| 1000 | ours | $-1.09508 \cdot 10^7$ | **0.59** | **0.00** |
| 3000 | SOS1 | $-1.10012 \cdot 10^7$ | 3600.10 | 0.52 |
| 3000 | big-M50 | - | 3615.51 | - |
| 3000 | persp. | $-1.10012 \cdot 10^7$ | 3600.91 | 0.59 |
| 3000 | eig. persp. | $-1.10012 \cdot 10^7$ | 100.85 | 0.00 |
| 3000 | ours | $-1.10012 \cdot 10^7$ | **1.74** | **0.00** |
| 5000 | SOS1 | $-1.08838 \cdot 10^7$ | 3600.13 | 2.05 |
| 5000 | big-M50 | - | 3612.62 | - |
| 5000 | persp. | $-1.09966 \cdot 10^7$ | 3600.17 | 1.00 |
| 5000 | eig. persp. | - | 3732.98 | - |
| 5000 | ours | $-1.09966 \cdot 10^7$ | **4.00** | **0.00** |

Table 3: Comparison of time and optimality gap on the synthetic datasets (n = 100000, k = 10, $\rho = 0.5$, $\lambda_2 = 0.001$).

| # of features | method | upper bound | time(s) | gap(%) |
|---|---|---|---|---|
| 100 | SOS1 | $-2.43575 \cdot 10^7$ | 2.83 | 0.00 |
| 100 | big-M50 | $-2.43573 \cdot 10^7$ | 0.46 | 0.00 |
| 100 | persp. | $-2.43575 \cdot 10^7$ | 0.21 | 0.00 |
| 100 | eig. persp. | $-2.43575 \cdot 10^7$ | **0.02** | **0.00** |
| 100 | ours | $-2.43575 \cdot 10^7$ | 0.32 | 0.00 |
| 500 | SOS1 | $-2.44185 \cdot 10^7$ | 10.25 | 0.00 |
| 500 | big-M50 | $-2.44178 \cdot 10^7$ | 92.53 | 0.00 |
| 500 | persp. | $-2.44185 \cdot 10^7$ | 114.80 | 0.00 |
| 500 | eig. persp. | $-2.44185 \cdot 10^7$ | 26.49 | 0.00 |
| 500 | ours | $-2.44185 \cdot 10^7$ | **0.46** | **0.00** |
| 1000 | SOS1 | $-2.41989 \cdot 10^7$ | 88.38 | 0.00 |
| 1000 | big-M50 | $-2.41987 \cdot 10^7$ | 1361.05 | 0.00 |
| 1000 | persp. | $-2.41989 \cdot 10^7$ | 3600.08 | 0.20 |
| 1000 | eig. persp. | $-2.41989 \cdot 10^7$ | 232.96 | 0.00 |
| 1000 | ours | $-2.41989 \cdot 10^7$ | **0.52** | **0.00** |
| 3000 | SOS1 | $-2.43713 \cdot 10^7$ | 3600.47 | 0.59 |
| 3000 | big-M50 | - | 3640.12 | - |
| 3000 | persp. | $-2.43713 \cdot 10^7$ | 3600.70 | 0.59 |
| 3000 | eig. persp. | - | 3606.15 | - |
| 3000 | ours | $-2.43713 \cdot 10^7$ | **1.77** | **0.00** |
| 5000 | SOS1 | $-2.42551 \cdot 10^7$ | 3600.13 | 1.48 |
| 5000 | big-M50 | $-2.43694 \cdot 10^7$ | 3643.28 | 1.00 |
| 5000 | persp. | $-2.43694 \cdot 10^7$ | 3604.07 | 1.00 |
| 5000 | eig. persp. | - | 3711.01 | - |
| 5000 | ours | $-2.43694 \cdot 10^7$ | **4.15** | **0.00** |

Table 4: Comparison of time and optimality gap on the synthetic datasets (n = 100000, k = 10, $\rho = 0.7$, $\lambda_2 = 0.001$).

| # of features | method | upper bound | time(s) | gap(%) |
|---|---|---|---|---|
| 100 | SOS1 | $-9.11026 \cdot 10^7$ | **0.05** | **0.00** |
| 100 | big-M50 | $-9.11026 \cdot 10^7$ | 0.13 | 0.00 |
| 100 | persp. | $-9.11026 \cdot 10^7$ | 0.27 | 0.00 |
| 100 | eig. persp. | $-9.11019 \cdot 10^7$ | 1.63 | 0.00 |
| 100 | ours | $-9.11026 \cdot 10^7$ | 0.30 | 0.00 |
| 500 | SOS1 | $-9.12047 \cdot 10^7$ | 21.43 | 0.00 |
| 500 | big-M50 | $-9.12047 \cdot 10^7$ | 94.03 | 0.00 |
| 500 | persp. | $-9.12047 \cdot 10^7$ | 176.04 | 0.00 |
| 500 | eig. persp. | $-9.12047 \cdot 10^7$ | 85.92 | 0.00 |
| 500 | ours | $-9.12047 \cdot 10^7$ | **0.38** | **0.00** |
| 1000 | SOS1 | $-9.04075 \cdot 10^7$ | 1085.91 | 0.00 |
| 1000 | big-M50 | $-9.04075 \cdot 10^7$ | 3600.01 | 0.07 |
| 1000 | persp. | $-9.04075 \cdot 10^7$ | 3600.01 | 0.20 |
| 1000 | eig. persp. | $-9.04075 \cdot 10^7$ | 341.28 | 0.00 |
| 1000 | ours | $-9.04075 \cdot 10^7$ | **0.52** | **0.00** |
| 3000 | SOS1 | $-9.09247 \cdot 10^7$ | 3600.05 | 0.98 |
| 3000 | big-M50 | - | 3605.36 | - |
| 3000 | persp. | $-9.12761 \cdot 10^7$ | 3600.05 | 0.59 |
| 3000 | eig. persp. | - | 3600.41 | - |
| 3000 | ours | $-9.12761 \cdot 10^7$ | **20.74** | **0.00** |
| 5000 | SOS1 | $-9.00628 \cdot 10^7$ | 3600.16 | 2.38 |
| 5000 | big-M50 | - | 3607.68 | - |
| 5000 | persp. | $-9.12274 \cdot 10^7$ | 3600.13 | 1.07 |
| 5000 | eig. persp. | - | 3810.32 | - |
| 5000 | ours | $-9.12916 \cdot 10^7$ | **58.29** | **0.00** |

Table 5: Comparison of time and optimality gap on the synthetic datasets (n = 100000, k = 10, $\rho = 0.9$, $\lambda_2 = 0.001$).

### G.2.2 Synthetic 1 Benchmark with $\lambda_2 = 0.001$ with Warmstart

| # of features | method | upper bound | time(s) | gap(%) |
|---|---|---|---|---|
| 100 | SOS1 + warm start | $-2.10878 \cdot 10^6$ | 0.05 | 0.00 |
| 100 | big-M50 + warm start | $-2.10878 \cdot 10^6$ | 0.08 | 0.00 |
| 100 | persp. + warm start | $-2.10878 \cdot 10^6$ | 0.08 | 0.00 |
| 100 | eig. persp. + warm start | $-2.10878 \cdot 10^6$ | **0.01** | **0.00** |
| 100 | ours | $-2.10878 \cdot 10^6$ | 0.29 | 0.00 |
| 500 | SOS1 + warm start | $-2.12833 \cdot 10^6$ | 6.31 | 0.00 |
| 500 | big-M50 + warm start | $-2.12833 \cdot 10^6$ | 37.62 | 0.00 |
| 500 | persp. + warm start | $-2.12833 \cdot 10^6$ | 19.60 | 0.00 |
| 500 | eig. persp. + warm start | $-2.12833 \cdot 10^6$ | 0.61 | 0.00 |
| 500 | ours | $-2.12833 \cdot 10^6$ | **0.37** | **0.00** |
| 1000 | SOS1 + warm start | $-2.11005 \cdot 10^6$ | 77.82 | 0.00 |
| 1000 | big-M50 + warm start | $-2.11005 \cdot 10^6$ | 28.51 | 0.00 |
| 1000 | persp. + warm start | $-2.11005 \cdot 10^6$ | 1148.21 | 0.00 |
| 1000 | eig. persp. + warm start | $-2.11005 \cdot 10^6$ | 4.08 | 0.00 |
| 1000 | ours | $-2.11005 \cdot 10^6$ | **0.58** | **0.00** |
| 3000 | SOS1 + warm start | $-2.10230 \cdot 10^6$ | 2100.58 | 0.00 |
| 3000 | big-M50 + warm start | $-2.10230 \cdot 10^6$ | 407.92 | 0.00 |
| 3000 | persp. + warm start | $-2.10231 \cdot 10^6$ | 2275.23 | 0.00 |
| 3000 | eig. persp. + warm start | $-2.10230 \cdot 10^6$ | 102.19 | 0.00 |
| 3000 | ours | $-2.10230 \cdot 10^6$ | **1.73** | **0.00** |
| 5000 | SOS1 + warm start | $-2.09635 \cdot 10^6$ | 3601.23 | 0.95 |
| 5000 | big-M50 + warm start | $-2.09635 \cdot 10^6$ | 1380.38 | 0.00 |
| 5000 | persp. + warm start | $-2.09637 \cdot 10^6$ | 3600.68 | 1.00 |
| 5000 | eig. persp. + warm start | $-2.09635 \cdot 10^6$ | 483.38 | 0.00 |
| 5000 | ours | $-2.09635 \cdot 10^6$ | **4.22** | **0.00** |

Table 6: Comparison of time and optimality gap on the synthetic datasets (n = 100000, k = 10, $\rho = 0.1$, $\lambda_2 = 0.001$). All baselines use our beam search solution as a warm start.

| # of features | method | upper bound | time(s) | gap(%) |
|---|---|---|---|---|
| 100 | SOS1 + warm start | $-5.28726 \cdot 10^6$ | 0.05 | 0.00 |
| 100 | big-M50 + warm start | $-5.28726 \cdot 10^6$ | 0.26 | 0.00 |
| 100 | persp. + warm start | $-5.28726 \cdot 10^6$ | 0.07 | 0.00 |
| 100 | eig. persp. + warm start | $-5.28726 \cdot 10^6$ | **0.02** | **0.00** |
| 100 | ours | $-5.28726 \cdot 10^6$ | 0.29 | 0.00 |
| 500 | SOS1 + warm start | $-5.31852 \cdot 10^6$ | 9.61 | 0.00 |
| 500 | big-M50 + warm start | $-5.31852 \cdot 10^6$ | 3.00 | 0.00 |
| 500 | persp. + warm start | $-5.31852 \cdot 10^6$ | 8.06 | 0.00 |
| 500 | eig. persp. + warm start | $-5.31852 \cdot 10^6$ | 0.63 | 0.00 |
| 500 | ours | $-5.31852 \cdot 10^6$ | **0.41** | **0.00** |
| 1000 | SOS1 + warm start | $-5.26974 \cdot 10^6$ | 88.36 | 0.00 |
| 1000 | big-M50 + warm start | $-5.26974 \cdot 10^6$ | 13.79 | 0.00 |
| 1000 | persp. + warm start | $-5.26974 \cdot 10^6$ | 54.26 | 0.00 |
| 1000 | eig. persp. + warm start | $-5.26974 \cdot 10^6$ | 4.18 | 0.00 |
| 1000 | ours | $-5.26974 \cdot 10^6$ | **0.52** | **0.00** |
| 3000 | SOS1 + warm start | $-5.27668 \cdot 10^6$ | 2740.42 | 0.00 |
| 3000 | big-M50 + warm start | $-5.27668 \cdot 10^6$ | 313.22 | 0.00 |
| 3000 | persp. + warm start | $-5.27668 \cdot 10^6$ | 144.09 | 0.00 |
| 3000 | eig. persp. + warm start | $-5.27668 \cdot 10^6$ | 101.95 | 0.00 |
| 3000 | ours | $-5.27668 \cdot 10^6$ | **1.75** | **0.00** |
| 5000 | SOS1 + warm start | $-5.27126 \cdot 10^6$ | 3600.51 | 1.00 |
| 5000 | big-M50 + warm start | $-5.27126 \cdot 10^6$ | 1705.37 | 0.00 |
| 5000 | persp. + warm start | $-5.27126 \cdot 10^6$ | 543.39 | 0.00 |
| 5000 | eig. persp. + warm start | $-5.27126 \cdot 10^6$ | 485.74 | 0.00 |
| 5000 | ours | $-5.27126 \cdot 10^6$ | **4.32** | **0.00** |

Table 7: Comparison of time and optimality gap on the synthetic datasets (n = 100000, k = 10, $\rho = 0.3$, $\lambda_2 = 0.001$). All baselines use our beam search solution as a warm start.

| # of features | method | upper bound | time(s) | gap(%) |
|---|---|---|---|---|
| 100 | SOS1 + warm start | $-1.10084 \cdot 10^7$ | 0.05 | 0.00 |
| 100 | big-M50 + warm start | $-1.10084 \cdot 10^7$ | 0.06 | 0.00 |
| 100 | persp. + warm start | $-1.10084 \cdot 10^7$ | 0.09 | 0.00 |
| 100 | eig. persp. + warm start | $-1.10084 \cdot 10^7$ | **0.02** | **0.00** |
| 100 | ours | $-1.10084 \cdot 10^7$ | 0.29 | 0.00 |
| 500 | SOS1 + warm start | $-1.10518 \cdot 10^7$ | 7.68 | 0.00 |
| 500 | big-M50 + warm start | $-1.10518 \cdot 10^7$ | 11.59 | 0.00 |
| 500 | persp. + warm start | $-1.10518 \cdot 10^7$ | 3600.02 | 0.08 |
| 500 | eig. persp. + warm start | $-1.10518 \cdot 10^7$ | 0.63 | 0.00 |
| 500 | ours | $-1.10518 \cdot 10^7$ | **0.40** | **0.00** |
| 1000 | SOS1 + warm start | $-1.09508 \cdot 10^7$ | 90.16 | 0.00 |
| 1000 | big-M50 + warm start | $-1.09508 \cdot 10^7$ | 21.16 | 0.00 |
| 1000 | persp. + warm start | $-1.09508 \cdot 10^7$ | 9.85 | 0.00 |
| 1000 | eig. persp. + warm start | $-1.09508 \cdot 10^7$ | 4.04 | 0.00 |
| 1000 | ours | $-1.09508 \cdot 10^7$ | **0.59** | **0.00** |
| 3000 | SOS1 + warm start | $-1.10012 \cdot 10^7$ | 3044.33 | 0.00 |
| 3000 | big-M50 + warm start | $-1.10012 \cdot 10^7$ | 243.42 | 0.00 |
| 3000 | persp. + warm start | $-1.10012 \cdot 10^7$ | 141.68 | 0.00 |
| 3000 | eig. persp. + warm start | $-1.10012 \cdot 10^7$ | 100.50 | 0.00 |
| 3000 | ours | $-1.10012 \cdot 10^7$ | **1.74** | **0.00** |
| 5000 | SOS1 + warm start | $-1.09966 \cdot 10^7$ | 3603.69 | 0.99 |
| 5000 | big-M50 + warm start | $-1.09966 \cdot 10^7$ | 3600.95 | 1.00 |
| 5000 | persp. + warm start | $-1.09966 \cdot 10^7$ | 3603.33 | 1.00 |
| 5000 | eig. persp. + warm start | $-1.09966 \cdot 10^7$ | 445.26 | 0.00 |
| 5000 | ours | $-1.09966 \cdot 10^7$ | **4.00** | **0.00** |

Table 8: Comparison of time and optimality gap on the synthetic datasets (n = 100000, k = 10, $\rho = 0.5$, $\lambda_2 = 0.001$). All baselines use our beam search solution as a warm start.

| # of features | method | upper bound | time(s) | gap(%) |
|:---:|:---:|:---:|:---:|:---:|
| 100 | SOS1 + warm start | $-2.43575 \cdot 10^7$ | 0.05 | 0.00 |
| 100 | big-M50 + warm start | $-2.43575 \cdot 10^7$ | 0.05 | 0.00 |
| 100 | persp. + warm start | $-2.43575 \cdot 10^7$ | 0.19 | 0.00 |
| 100 | eig. persp. + warm start | $-2.43575 \cdot 10^7$ | **0.02** | **0.00** |
| 100 | ours | $-2.43575 \cdot 10^7$ | 0.32 | 0.00 |
| 500 | SOS1 + warm start | $-2.44185 \cdot 10^7$ | 8.94 | 0.00 |
| 500 | big-M50 + warm start | $-2.44185 \cdot 10^7$ | 4.02 | 0.00 |
| 500 | persp. + warm start | $-2.44185 \cdot 10^7$ | 271.52 | 0.00 |
| 500 | eig. persp. + warm start | $-2.44185 \cdot 10^7$ | 0.60 | 0.00 |
| 500 | ours | $-2.44185 \cdot 10^7$ | **0.46** | **0.00** |
| 1000 | SOS1 + warm start | $-2.41989 \cdot 10^7$ | 83.96 | 0.00 |
| 1000 | big-M50 + warm start | $-2.41989 \cdot 10^7$ | 11.63 | 0.00 |
| 1000 | persp. + warm start | $-2.41989 \cdot 10^7$ | 3600.13 | 0.20 |
| 1000 | eig. persp. + warm start | $-2.41989 \cdot 10^7$ | 4.02 | 0.00 |
| 1000 | ours | $-2.41989 \cdot 10^7$ | **0.52** | **0.00** |
| 3000 | SOS1 + warm start | $-2.43713 \cdot 10^7$ | 3601.04 | 0.59 |
| 3000 | big-M50 + warm start | $-2.43713 \cdot 10^7$ | 3608.31 | 0.59 |
| 3000 | persp. + warm start | $-2.43713 \cdot 10^7$ | 3600.34 | 0.59 |
| 3000 | eig. persp. + warm start | $-2.43713 \cdot 10^7$ | 110.94 | 0.00 |
| 3000 | ours | $-2.43713 \cdot 10^7$ | **1.77** | **0.00** |
| 5000 | SOS1 + warm start | $-2.43694 \cdot 10^7$ | 3600.18 | 1.00 |
| 5000 | big-M50 + warm start | $-2.43694 \cdot 10^7$ | 3670.94 | 1.00 |
| 5000 | persp. + warm start | $-2.43694 \cdot 10^7$ | 3600.18 | 1.00 |
| 5000 | eig. persp. + warm start | $-2.43694 \cdot 10^7$ | 442.81 | 0.00 |
| 5000 | ours | $-2.43694 \cdot 10^7$ | **4.15** | **0.00** |

Table 9: Comparison of time and optimality gap on the synthetic datasets (n = 100000, k = 10, $\rho = 0.7$, $\lambda_2 = 0.001$). All baselines use our beam search solution as a warm start.

| # of features | method | upper bound | time(s) | gap(%) |
|---|---|---|---|---|
| 100 | SOS1 + warm start | $-9.11026 \cdot 10^7$ | 0.04 | 0.00 |
| 100 | big-M50 + warm start | $-9.11026 \cdot 10^7$ | 0.03 | 0.00 |
| 100 | persp. + warm start | $-9.11026 \cdot 10^7$ | 0.26 | 0.00 |
| 100 | eig. persp. + warm start | $-9.11026 \cdot 10^7$ | **0.02** | **0.00** |
| 100 | ours | $-9.11026 \cdot 10^7$ | 0.30 | 0.00 |
| 500 | SOS1 + warm start | $-9.12047 \cdot 10^7$ | 48.24 | 0.00 |
| 500 | big-M50 + warm start | $-9.12047 \cdot 10^7$ | 34.86 | 0.00 |
| 500 | persp. + warm start | $-9.12047 \cdot 10^7$ | 3600.02 | 0.09 |
| 500 | eig. persp. + warm start | $-9.12047 \cdot 10^7$ | 0.59 | 0.00 |
| 500 | ours | $-9.12047 \cdot 10^7$ | **0.38** | **0.00** |
| 1000 | SOS1 + warm start | $-9.04075 \cdot 10^7$ | 1090.45 | 0.00 |
| 1000 | big-M50 + warm start | $-9.04075 \cdot 10^7$ | 3600.01 | 0.08 |
| 1000 | persp. + warm start | $-9.04075 \cdot 10^7$ | 3600.08 | 0.20 |
| 1000 | eig. persp. + warm start | $-9.04075 \cdot 10^7$ | 4.15 | 0.00 |
| 1000 | ours | $-9.04075 \cdot 10^7$ | **0.52** | **0.00** |
| 3000 | SOS1 + warm start | $-9.12761 \cdot 10^7$ | 3600.07 | 0.59 |
| 3000 | big-M50 + warm start | $-9.12761 \cdot 10^7$ | 3612.88 | 0.59 |
| 3000 | persp. + warm start | $-9.12761 \cdot 10^7$ | 3600.07 | 0.59 |
| 3000 | eig. persp. + warm start | $-9.12761 \cdot 10^7$ | 100.26 | 0.00 |
| 3000 | ours | $-9.12761 \cdot 10^7$ | **20.74** | **0.00** |
| 5000 | SOS1 + warm start | $-9.12916 \cdot 10^7$ | 3601.57 | 1.00 |
| 5000 | big-M50 + warm start | $-9.12916 \cdot 10^7$ | 3622.52 | 1.00 |
| 5000 | persp. + warm start | $-9.12916 \cdot 10^7$ | 3602.87 | 1.00 |
| 5000 | eig. persp. + warm start | $-9.12916 \cdot 10^7$ | 439.82 | 0.00 |
| 5000 | ours | $-9.12916 \cdot 10^7$ | **58.29** | **0.00** |

Table 10: Comparison of time and optimality gap on the synthetic datasets (n = 100000, k = 10, $\rho = 0.9$, $\lambda_2 = 0.001$). All baselines use our beam search solution as a warm start.

### G.2.3 Synthetic 1 Benchmark with $\lambda_2 = 0.1$ with No Warmstart

| # of features | method | upper bound | time(s) | gap(%) |
|---|---|---|---|---|
| 100 | SOS1 | $-2.10878 \cdot 10^6$ | 3.04 | 0.00 |
| 100 | big-M50 | $-2.10878 \cdot 10^6$ | 0.86 | 0.00 |
| 100 | persp. | $-2.10878 \cdot 10^6$ | 0.07 | 0.00 |
| 100 | eig. persp. | $-2.10878 \cdot 10^6$ | **0.02** | **0.00** |
| 100 | ours | $-2.10878 \cdot 10^6$ | 0.36 | 0.00 |
| 500 | SOS1 | $-2.12833 \cdot 10^6$ | 8.13 | 0.00 |
| 500 | big-M50 | $-2.12833 \cdot 10^6$ | 37.96 | 0.00 |
| 500 | persp. | $-2.12833 \cdot 10^6$ | 3.87 | 0.00 |
| 500 | eig. persp. | $-2.12833 \cdot 10^6$ | 0.63 | 0.00 |
| 500 | ours | $-2.12833 \cdot 10^6$ | **0.37** | **0.00** |
| 1000 | SOS1 | $-2.11005 \cdot 10^6$ | 81.40 | 0.00 |
| 1000 | big-M50 | $-2.11005 \cdot 10^6$ | 1007.27 | 0.00 |
| 1000 | persp. | $-2.11005 \cdot 10^6$ | 1195.64 | 0.00 |
| 1000 | eig. persp. | $-2.11005 \cdot 10^6$ | 4.07 | 0.00 |
| 1000 | ours | $-2.11005 \cdot 10^6$ | **0.57** | **0.00** |
| 3000 | SOS1 | $-2.10230 \cdot 10^6$ | 3601.25 | 0.59 |
| 3000 | big-M50 | $-7.99504 \cdot 10^5$ | 3608.36 | 164.51 |
| 3000 | persp. | $-2.10230 \cdot 10^6$ | 2055.12 | 0.00 |
| 3000 | eig. persp. | $-2.10230 \cdot 10^6$ | 103.29 | 0.00 |
| 3000 | ours | $-2.10230 \cdot 10^6$ | **1.79** | **0.00** |
| 5000 | SOS1 | $-1.98939 \cdot 10^6$ | 3600.15 | 6.43 |
| 5000 | big-M50 | $-9.69753 \cdot 10^5$ | 3619.50 | 118.35 |
| 5000 | persp. | $-2.09635 \cdot 10^6$ | 3600.72 | 1.00 |
| 5000 | eig. persp. | $-2.09635 \cdot 10^6$ | 584.03 | 0.00 |
| 5000 | ours | $-2.09635 \cdot 10^6$ | **4.20** | **0.00** |

Table 11: Comparison of time and optimality gap on the synthetic datasets (n = 100000, k = 10, $\rho = 0.1$, $\lambda_2 = 0.1$).

| # of features | method | upper bound | time(s) | gap(%) |
|---|---|---|---|---|
| 100 | SOS1 | $-5.28726 \cdot 10^6$ | 3.37 | 0.00 |
| 100 | big-M50 | $-4.60269 \cdot 10^6$ | 1.83 | 0.00 |
| 100 | persp. | $-5.28726 \cdot 10^6$ | **0.07** | **0.00** |
| 100 | eig. persp. | $-5.28726 \cdot 10^6$ | 0.11 | 0.00 |
| 100 | ours | $-5.28726 \cdot 10^6$ | 0.27 | 0.00 |
| 500 | SOS1 | $-5.31852 \cdot 10^6$ | 10.00 | 0.00 |
| 500 | big-M50 | $-5.31827 \cdot 10^6$ | 85.52 | 0.00 |
| 500 | persp. | $-5.31852 \cdot 10^6$ | 8.41 | 0.00 |
| 500 | eig. persp. | $-5.31852 \cdot 10^6$ | 1.07 | 0.00 |
| 500 | ours | $-5.31852 \cdot 10^6$ | **0.41** | **0.00** |
| 1000 | SOS1 | $-5.26974 \cdot 10^6$ | 88.50 | 0.00 |
| 1000 | big-M50 | $-5.26974 \cdot 10^6$ | 1576.75 | 0.00 |
| 1000 | persp. | $-5.26974 \cdot 10^6$ | 91.53 | 0.00 |
| 1000 | eig. persp. | $-5.26974 \cdot 10^6$ | 4.11 | 0.00 |
| 1000 | ours | $-5.26974 \cdot 10^6$ | **0.52** | **0.00** |
| 3000 | SOS1 | $-5.16793 \cdot 10^6$ | 3600.25 | 2.71 |
| 3000 | big-M50 | $-3.73049 \cdot 10^6$ | 3603.09 | 42.28 |
| 3000 | persp. | $-5.27668 \cdot 10^6$ | 3600.09 | 0.59 |
| 3000 | eig. persp. | $-5.27668 \cdot 10^6$ | 103.60 | 0.00 |
| 3000 | ours | $-5.27668 \cdot 10^6$ | **1.84** | **0.00** |
| 5000 | SOS1 | $-5.16087 \cdot 10^6$ | 3600.13 | 3.16 |
| 5000 | big-M50 | $-3.98069 \cdot 10^6$ | 3606.79 | 33.75 |
| 5000 | persp. | $-5.27126 \cdot 10^6$ | 3601.72 | 1.00 |
| 5000 | eig. persp. | $-5.27126 \cdot 10^6$ | 611.45 | 0.00 |
| 5000 | ours | $-5.27126 \cdot 10^6$ | **4.42** | **0.00** |

Table 12: Comparison of time and optimality gap on the synthetic datasets (n = 100000, k = 10, $\rho = 0.3$, $\lambda_2 = 0.1$).

| # of features | method | upper bound | time(s) | gap(%) |
|---|---|---|---|---|
| 100 | SOS1 | $-1.10084 \cdot 10^7$ | 2.97 | 0.00 |
| 100 | big-M50 | $-1.10084 \cdot 10^7$ | 0.44 | 0.00 |
| 100 | persp. | $-1.10084 \cdot 10^7$ | 0.96 | 0.00 |
| 100 | eig. persp. | $-1.10084 \cdot 10^7$ | **0.13** | **0.00** |
| 100 | ours | $-1.10084 \cdot 10^7$ | 0.32 | 0.00 |
| 500 | SOS1 | $-1.10518 \cdot 10^7$ | 9.66 | 0.00 |
| 500 | big-M50 | $-1.10515 \cdot 10^7$ | 86.52 | 0.00 |
| 500 | persp. | $-1.10518 \cdot 10^7$ | 3600.03 | 0.08 |
| 500 | eig. persp. | $-1.10518 \cdot 10^7$ | 0.62 | 0.00 |
| 500 | ours | $-1.10518 \cdot 10^7$ | **0.40** | **0.00** |
| 1000 | SOS1 | $-1.09508 \cdot 10^7$ | 87.74 | 0.00 |
| 1000 | big-M50 | $-1.09506 \cdot 10^7$ | 1539.25 | 0.00 |
| 1000 | persp. | $-1.09508 \cdot 10^7$ | 3600.12 | 0.19 |
| 1000 | eig. persp. | $-1.09508 \cdot 10^7$ | 335.72 | 0.00 |
| 1000 | ours | $-1.09508 \cdot 10^7$ | **0.53** | **0.00** |
| 3000 | SOS1 | $-1.10012 \cdot 10^7$ | 3600.36 | 0.51 |
| 3000 | big-M50 | - | 3608.09 | - |
| 3000 | persp. | $-1.10012 \cdot 10^7$ | 3600.08 | 0.59 |
| 3000 | eig. persp. | $-1.10012 \cdot 10^7$ | 103.03 | 0.00 |
| 3000 | ours | $-1.10012 \cdot 10^7$ | **1.67** | **0.00** |
| 5000 | SOS1 | $-1.08838 \cdot 10^7$ | 3602.31 | 2.05 |
| 5000 | big-M50 | - | 3649.14 | - |
| 5000 | persp. | $-1.09966 \cdot 10^7$ | 3601.44 | 1.00 |
| 5000 | eig. persp. | - | 3758.62 | - |
| 5000 | ours | $-1.09966 \cdot 10^7$ | **4.13** | **0.00** |

Table 13: Comparison of time and optimality gap on the synthetic datasets (n = 100000, k = 10, $\rho = 0.5$, $\lambda_2 = 0.1$).

| # of features | method | upper bound | time(s) | gap(%) |
|---|---|---|---|---|
| 100 | SOS1 | $-2.43575 \cdot 10^7$ | **0.05** | **0.00** |
| 100 | big-M50 | $-2.43573 \cdot 10^7$ | 0.41 | 0.00 |
| 100 | persp. | $-2.43575 \cdot 10^7$ | 0.43 | 0.00 |
| 100 | eig. persp. | $-2.43575 \cdot 10^7$ | 0.13 | 0.00 |
| 100 | ours | $-2.43575 \cdot 10^7$ | 0.34 | 0.00 |
| 500 | SOS1 | $-2.44185 \cdot 10^7$ | 9.10 | 0.00 |
| 500 | big-M50 | $-2.44167 \cdot 10^7$ | 96.59 | 0.00 |
| 500 | persp. | $-2.44185 \cdot 10^7$ | 148.21 | 0.00 |
| 500 | eig. persp. | $-2.44185 \cdot 10^7$ | 23.81 | 0.00 |
| 500 | ours | $-2.44185 \cdot 10^7$ | **0.44** | **0.00** |
| 1000 | SOS1 | $-2.41989 \cdot 10^7$ | 89.99 | 0.00 |
| 1000 | big-M50 | $-2.41986 \cdot 10^7$ | 1608.58 | 0.00 |
| 1000 | persp. | $-2.41989 \cdot 10^7$ | 3600.10 | 0.19 |
| 1000 | eig. persp. | $-2.41989 \cdot 10^7$ | 4.42 | 0.00 |
| 1000 | ours | $-2.41989 \cdot 10^7$ | **0.53** | **0.00** |
| 3000 | SOS1 | $-2.42629 \cdot 10^7$ | 3601.14 | 1.04 |
| 3000 | big-M50 | - | 3605.20 | - |
| 3000 | persp. | $-2.43713 \cdot 10^7$ | 3601.16 | 0.59 |
| 3000 | eig. persp. | - | 3658.27 | - |
| 3000 | ours | $-2.43713 \cdot 10^7$ | **1.75** | **0.00** |
| 5000 | SOS1 | $-2.42551 \cdot 10^7$ | 3600.14 | 1.48 |
| 5000 | big-M50 | $-2.43694 \cdot 10^7$ | 3650.52 | 1.00 |
| 5000 | persp. | $-2.43694 \cdot 10^7$ | 3606.01 | 1.00 |
| 5000 | eig. persp. | - | 3750.44 | - |
| 5000 | ours | $-2.43694 \cdot 10^7$ | **4.22** | **0.00** |

Table 14: Comparison of time and optimality gap on the synthetic datasets (n = 100000, k = 10, $\rho = 0.7$, $\lambda_2 = 0.1$).

| # of features | method | upper bound | time(s) | gap(%) |
|---|---|---|---|---|
| 100 | SOS1 | $-9.11026 \cdot 10^7$ | **0.05** | **0.00** |
| 100 | big-M50 | $-9.11026 \cdot 10^7$ | 0.18 | 0.00 |
| 100 | persp. | $-9.11026 \cdot 10^7$ | 0.30 | 0.00 |
| 100 | eig. persp. | $-9.11026 \cdot 10^7$ | 0.13 | 0.00 |
| 100 | ours | $-9.11026 \cdot 10^7$ | 0.31 | 0.00 |
| 500 | SOS1 | $-9.12047 \cdot 10^7$ | 22.15 | 0.00 |
| 500 | big-M50 | $-9.12047 \cdot 10^7$ | 398.02 | 0.00 |
| 500 | persp. | $-9.12047 \cdot 10^7$ | 3600.01 | 0.08 |
| 500 | eig. persp. | $-9.12046 \cdot 10^7$ | 84.16 | 0.00 |
| 500 | ours | $-9.12047 \cdot 10^7$ | **0.38** | **0.00** |
| 1000 | SOS1 | $-9.04075 \cdot 10^7$ | 1090.30 | 0.00 |
| 1000 | big-M50 | $-9.04075 \cdot 10^7$ | 3600.05 | 0.01 |
| 1000 | persp. | $-9.04075 \cdot 10^7$ | 3600.08 | 0.20 |
| 1000 | eig. persp. | $-9.04074 \cdot 10^7$ | 881.89 | 0.00 |
| 1000 | ours | $-9.04075 \cdot 10^7$ | **0.52** | **0.00** |
| 3000 | SOS1 | $-9.11671 \cdot 10^7$ | 3600.55 | 0.71 |
| 3000 | big-M50 | - | 3619.63 | - |
| 3000 | persp. | $-9.12761 \cdot 10^7$ | 3600.05 | 0.59 |
| 3000 | eig. persp. | - | 3600.21 | - |
| 3000 | ours | $-9.12761 \cdot 10^7$ | **20.88** | **0.00** |
| 5000 | SOS1 | $-9.02495 \cdot 10^7$ | 3602.25 | 2.17 |
| 5000 | big-M50 | - | 3620.37 | - |
| 5000 | persp. | $-9.12900 \cdot 10^7$ | 3600.18 | 1.00 |
| 5000 | eig. persp. | - | 4028.26 | - |
| 5000 | ours | $-9.12916 \cdot 10^7$ | **58.07** | **0.00** |

Table 15: Comparison of time and optimality gap on the synthetic datasets (n = 100000, k = 10, $\rho = 0.9$, $\lambda_2 = 0.1$).

### G.2.4 Synthetic 1 Benchmark with $\lambda_2 = 0.1$ with Warmstart

| # of features | method | upper bound | time(s) | gap(%) |
|---|---|---|---|---|
| 100 | SOS1 + warm start | $-2.10878 \cdot 10^6$ | 0.05 | 0.00 |
| 100 | big-M50 + warm start | $-2.10878 \cdot 10^6$ | 0.08 | 0.00 |
| 100 | persp. + warm start | $-2.10878 \cdot 10^6$ | 0.07 | 0.00 |
| 100 | eig. persp. + warm start | $-2.10878 \cdot 10^6$ | **0.02** | **0.00** |
| 100 | ours | $-2.10878 \cdot 10^6$ | 0.36 | 0.00 |
| 500 | SOS1 + warm start | $-2.12833 \cdot 10^6$ | 5.04 | 0.00 |
| 500 | big-M50 + warm start | $-2.12833 \cdot 10^6$ | 3.78 | 0.00 |
| 500 | persp. + warm start | $-2.12833 \cdot 10^6$ | 3.84 | 0.00 |
| 500 | eig. persp. + warm start | $-2.12833 \cdot 10^6$ | 0.60 | 0.00 |
| 500 | ours | $-2.12833 \cdot 10^6$ | **0.37** | **0.00** |
| 1000 | SOS1 + warm start | $-2.11005 \cdot 10^6$ | 74.29 | 0.00 |
| 1000 | big-M50 + warm start | $-2.11005 \cdot 10^6$ | 26.94 | 0.00 |
| 1000 | persp. + warm start | $-2.11005 \cdot 10^6$ | 1175.58 | 0.00 |
| 1000 | eig. persp. + warm start | $-2.11005 \cdot 10^6$ | 4.18 | 0.00 |
| 1000 | ours | $-2.11005 \cdot 10^6$ | **0.57** | **0.00** |
| 3000 | SOS1 + warm start | $-2.10230 \cdot 10^6$ | 2144.33 | 0.00 |
| 3000 | big-M50 + warm start | $-2.10230 \cdot 10^6$ | 193.85 | 0.00 |
| 3000 | persp. + warm start | $-2.10230 \cdot 10^6$ | 2108.24 | 0.00 |
| 3000 | eig. persp. + warm start | $-2.10230 \cdot 10^6$ | 110.85 | 0.00 |
| 3000 | ours | $-2.10230 \cdot 10^6$ | **1.79** | **0.00** |
| 5000 | SOS1 + warm start | $-2.09635 \cdot 10^6$ | 3600.66 | 0.95 |
| 5000 | big-M50 + warm start | $-2.09635 \cdot 10^6$ | 1366.14 | 0.00 |
| 5000 | persp. + warm start | $-2.09635 \cdot 10^6$ | 3601.28 | 1.00 |
| 5000 | eig. persp. + warm start | $-2.09635 \cdot 10^6$ | 460.24 | 0.00 |
| 5000 | ours | $-2.09635 \cdot 10^6$ | **4.20** | **0.00** |

Table 16: Comparison of time and optimality gap on the synthetic datasets (n = 100000, k = 10, $\rho = 0.1$, $\lambda_2 = 0.1$). All baselines use our beam search solution as a warm start.

| # of features | method | upper bound | time(s) | gap(%) |
|---|---|---|---|---|
| 100 | SOS1 + warm start | $-5.28726 \cdot 10^6$ | 0.05 | 0.00 |
| 100 | big-M50 + warm start | $-5.28726 \cdot 10^6$ | 0.07 | 0.00 |
| 100 | persp. + warm start | $-5.28726 \cdot 10^6$ | 0.07 | 0.00 |
| 100 | eig. persp. + warm start | $-5.28726 \cdot 10^6$ | **0.02** | **0.00** |
| 100 | ours | $-5.28726 \cdot 10^6$ | 0.27 | 0.00 |
| 500 | SOS1 + warm start | $-5.31852 \cdot 10^6$ | 9.77 | 0.00 |
| 500 | big-M50 + warm start | $-5.31852 \cdot 10^6$ | 39.88 | 0.00 |
| 500 | persp. + warm start | $-5.31852 \cdot 10^6$ | 8.26 | 0.00 |
| 500 | eig. persp. + warm start | $-5.31852 \cdot 10^6$ | 0.63 | 0.00 |
| 500 | ours | $-5.31852 \cdot 10^6$ | **0.41** | **0.00** |
| 1000 | SOS1 + warm start | $-5.26974 \cdot 10^6$ | 89.01 | 0.00 |
| 1000 | big-M50 + warm start | $-5.26974 \cdot 10^6$ | 14.33 | 0.00 |
| 1000 | persp. + warm start | $-5.26974 \cdot 10^6$ | 91.61 | 0.00 |
| 1000 | eig. persp. + warm start | $-5.26974 \cdot 10^6$ | 4.09 | 0.00 |
| 1000 | ours | $-5.26974 \cdot 10^6$ | **0.52** | **0.00** |
| 3000 | SOS1 + warm start | $-5.27668 \cdot 10^6$ | 2743.09 | 0.00 |
| 3000 | big-M50 + warm start | $-5.27668 \cdot 10^6$ | 317.41 | 0.00 |
| 3000 | persp. + warm start | $-5.27668 \cdot 10^6$ | 139.45 | 0.00 |
| 3000 | eig. persp. + warm start | $-5.27668 \cdot 10^6$ | 111.91 | 0.00 |
| 3000 | ours | $-5.27668 \cdot 10^6$ | **1.84** | **0.00** |
| 5000 | SOS1 + warm start | $-5.27126 \cdot 10^6$ | 3600.80 | 1.00 |
| 5000 | big-M50 + warm start | $-5.27126 \cdot 10^6$ | 1658.42 | 0.00 |
| 5000 | persp. + warm start | $-5.27126 \cdot 10^6$ | 546.50 | 0.00 |
| 5000 | eig. persp. + warm start | $-5.27126 \cdot 10^6$ | 454.67 | 0.00 |
| 5000 | ours | $-5.27126 \cdot 10^6$ | **4.42** | **0.00** |

Table 17: Comparison of time and optimality gap on the synthetic datasets (n = 100000, k = 10, $\rho = 0.3$, $\lambda_2 = 0.1$). All baselines use our beam search solution as a warm start.

| # of features | method | upper bound | time(s) | gap(%) |
|---|---|---|---|---|
| 100 | SOS1 + warm start | $-1.10084 \cdot 10^7$ | 0.05 | 0.00 |
| 100 | big-M50 + warm start | $-1.10084 \cdot 10^7$ | 0.06 | 0.00 |
| 100 | persp. + warm start | $-1.10084 \cdot 10^7$ | 0.90 | 0.00 |
| 100 | eig. persp. + warm start | $-1.10084 \cdot 10^7$ | **0.02** | **0.00** |
| 100 | ours | $-1.10084 \cdot 10^7$ | 0.32 | 0.00 |
| 500 | SOS1 + warm start | $-1.10518 \cdot 10^7$ | 9.26 | 0.00 |
| 500 | big-M50 + warm start | $-1.10518 \cdot 10^7$ | 2.28 | 0.00 |
| 500 | persp. + warm start | $-1.10518 \cdot 10^7$ | 3600.02 | 0.08 |
| 500 | eig. persp. + warm start | $-1.10518 \cdot 10^7$ | 0.64 | 0.00 |
| 500 | ours | $-1.10518 \cdot 10^7$ | **0.40** | **0.00** |
| 1000 | SOS1 + warm start | $-1.09508 \cdot 10^7$ | 89.44 | 0.00 |
| 1000 | big-M50 + warm start | $-1.09508 \cdot 10^7$ | 14.11 | 0.00 |
| 1000 | persp. + warm start | $-1.09508 \cdot 10^7$ | 10.09 | 0.00 |
| 1000 | eig. persp. + warm start | $-1.09508 \cdot 10^7$ | 4.05 | 0.00 |
| 1000 | ours | $-1.09508 \cdot 10^7$ | **0.53** | **0.00** |
| 3000 | SOS1 + warm start | $-1.10012 \cdot 10^7$ | 3046.03 | 0.00 |
| 3000 | big-M50 + warm start | $-1.10012 \cdot 10^7$ | 222.56 | 0.00 |
| 3000 | persp. + warm start | $-1.10012 \cdot 10^7$ | 141.64 | 0.00 |
| 3000 | eig. persp. + warm start | $-1.10012 \cdot 10^7$ | 110.14 | 0.00 |
| 3000 | ours | $-1.10012 \cdot 10^7$ | **1.67** | **0.00** |
| 5000 | SOS1 + warm start | $-1.09966 \cdot 10^7$ | 3600.73 | 0.99 |
| 5000 | big-M50 + warm start | $-1.09966 \cdot 10^7$ | 3606.43 | 1.00 |
| 5000 | persp. + warm start | $-1.09966 \cdot 10^7$ | 3600.38 | 1.00 |
| 5000 | eig. persp. + warm start | $-1.09966 \cdot 10^7$ | 455.87 | 0.00 |
| 5000 | ours | $-1.09966 \cdot 10^7$ | **4.13** | **0.00** |

Table 18: Comparison of time and optimality gap on the synthetic datasets (n = 100000, k = 10, $\rho = 0.5$, $\lambda_2 = 0.1$). All baselines use our beam search solution as a warm start.

| # of features | method | upper bound | time(s) | gap(%) |
|---|---|---|---|---|
| 100 | SOS1 + warm start | $-2.43575 \cdot 10^7$ | 0.05 | 0.00 |
| 100 | big-M50 + warm start | $-2.43575 \cdot 10^7$ | 0.06 | 0.00 |
| 100 | persp. + warm start | $-2.43575 \cdot 10^7$ | 0.43 | 0.00 |
| 100 | eig. persp. + warm start | $-2.43575 \cdot 10^7$ | **0.02** | **0.00** |
| 100 | ours | $-2.43575 \cdot 10^7$ | 0.34 | 0.00 |
| 500 | SOS1 + warm start | $-2.44185 \cdot 10^7$ | 9.20 | 0.00 |
| 500 | big-M50 + warm start | $-2.44185 \cdot 10^7$ | 4.00 | 0.00 |
| 500 | persp. + warm start | $-2.44185 \cdot 10^7$ | 146.41 | 0.00 |
| 500 | eig. persp. + warm start | $-2.44185 \cdot 10^7$ | 0.62 | 0.00 |
| 500 | ours | $-2.44185 \cdot 10^7$ | **0.44** | **0.00** |
| 1000 | SOS1 + warm start | $-2.41989 \cdot 10^7$ | 88.02 | 0.00 |
| 1000 | big-M50 + warm start | $-2.41989 \cdot 10^7$ | 12.62 | 0.00 |
| 1000 | persp. + warm start | $-2.41989 \cdot 10^7$ | 1117.62 | 0.00 |
| 1000 | eig. persp. + warm start | $-2.41989 \cdot 10^7$ | 4.05 | 0.00 |
| 1000 | ours | $-2.41989 \cdot 10^7$ | **0.53** | **0.00** |
| 3000 | SOS1 + warm start | $-2.43713 \cdot 10^7$ | 3600.80 | 0.59 |
| 3000 | big-M50 + warm start | $-2.43713 \cdot 10^7$ | 3600.43 | 0.59 |
| 3000 | persp. + warm start | $-2.43713 \cdot 10^7$ | 3600.68 | 0.59 |
| 3000 | eig. persp. + warm start | $-2.43713 \cdot 10^7$ | 101.46 | 0.00 |
| 3000 | ours | $-2.43713 \cdot 10^7$ | **1.75** | **0.00** |
| 5000 | SOS1 + warm start | $-2.43694 \cdot 10^7$ | 3600.86 | 1.00 |
| 5000 | big-M50 + warm start | $-2.43694 \cdot 10^7$ | 3616.71 | 1.00 |
| 5000 | persp. + warm start | $-2.43694 \cdot 10^7$ | 3603.18 | 1.00 |
| 5000 | eig. persp. + warm start | $-2.43694 \cdot 10^7$ | 455.57 | 0.00 |
| 5000 | ours | $-2.43694 \cdot 10^7$ | **4.22** | **0.00** |

Table 19: Comparison of time and optimality gap on the synthetic datasets (n = 100000, k = 10, $\rho = 0.7$, $\lambda_2 = 0.1$). All baselines use our beam search solution as a warm start.

| # of features | method | upper bound | time(s) | gap(%) |
|---|---|---|---|---|
| 100 | SOS1 + warm start | $-9.11026 \cdot 10^7$ | 0.04 | 0.00 |
| 100 | big-M50 + warm start | $-9.11026 \cdot 10^7$ | 0.04 | 0.00 |
| 100 | persp. + warm start | $-9.11026 \cdot 10^7$ | 0.29 | 0.00 |
| 100 | eig. persp. + warm start | $-9.11026 \cdot 10^7$ | **0.02** | **0.00** |
| 100 | ours | $-9.11026 \cdot 10^7$ | 0.31 | 0.00 |
| 500 | SOS1 + warm start | $-9.12047 \cdot 10^7$ | 52.72 | 0.00 |
| 500 | big-M50 + warm start | $-9.12047 \cdot 10^7$ | 14.34 | 0.00 |
| 500 | persp. + warm start | $-9.12047 \cdot 10^7$ | 166.73 | 0.00 |
| 500 | eig. persp. + warm start | $-9.12047 \cdot 10^7$ | 0.60 | 0.00 |
| 500 | ours | $-9.12047 \cdot 10^7$ | **0.38** | **0.00** |
| 1000 | SOS1 + warm start | $-9.04075 \cdot 10^7$ | 1068.59 | 0.00 |
| 1000 | big-M50 + warm start | $-9.04075 \cdot 10^7$ | 3600.16 | 0.01 |
| 1000 | persp. + warm start | $-9.04075 \cdot 10^7$ | 3600.01 | 0.20 |
| 1000 | eig. persp. + warm start | $-9.04075 \cdot 10^7$ | 4.17 | 0.00 |
| 1000 | ours | $-9.04075 \cdot 10^7$ | **0.52** | **0.00** |
| 3000 | SOS1 + warm start | $-9.12761 \cdot 10^7$ | 3600.39 | 0.59 |
| 3000 | big-M50 + warm start | $-9.12761 \cdot 10^7$ | 3611.02 | 0.59 |
| 3000 | persp. + warm start | $-9.12761 \cdot 10^7$ | 3600.23 | 0.59 |
| 3000 | eig. persp. + warm start | $-9.12761 \cdot 10^7$ | 101.65 | 0.00 |
| 3000 | ours | $-9.12761 \cdot 10^7$ | **20.88** | **0.00** |
| 5000 | SOS1 + warm start | $-9.12916 \cdot 10^7$ | 3603.57 | 1.00 |
| 5000 | big-M50 + warm start | $-9.12916 \cdot 10^7$ | 3606.64 | 1.00 |
| 5000 | persp. + warm start | $-9.12916 \cdot 10^7$ | 3601.56 | 1.00 |
| 5000 | eig. persp. + warm start | $-9.12916 \cdot 10^7$ | 454.63 | 0.00 |
| 5000 | ours | $-9.12916 \cdot 10^7$ | **58.07** | **0.00** |

Table 20: Comparison of time and optimality gap on the synthetic datasets (n = 100000, k = 10, $\rho = 0.9$, $\lambda_2 = 0.1$). All baselines use our beam search solution as a warm start.

## G.2.5 Synthetic 1 Benchmark with $\lambda_2 = 10.0$ with No Warmstart

| # of features | method | upper bound | time(s) | gap(%) |
|---|---|---|---|---|
| 100 | SOS1 | $-2.10868 \cdot 10^6$ | 3.37 | 0.00 |
| 100 | big-M50 | $-2.10868 \cdot 10^6$ | 0.28 | 0.00 |
| 100 | persp. | $-2.10868 \cdot 10^6$ | 2.06 | 0.00 |
| 100 | eig. persp. | $-2.10868 \cdot 10^6$ | **0.02** | **0.00** |
| 100 | ours | $-2.10868 \cdot 10^6$ | 0.30 | 0.00 |
| 500 | SOS1 | $-2.12823 \cdot 10^6$ | 6.58 | 0.00 |
| 500 | big-M50 | $-2.12819 \cdot 10^6$ | 42.85 | 0.00 |
| 500 | persp. | $-2.12823 \cdot 10^6$ | 143.68 | 0.00 |
| 500 | eig. persp. | $-2.12823 \cdot 10^6$ | 0.65 | 0.00 |
| 500 | ours | $-2.12823 \cdot 10^6$ | **0.38** | **0.00** |
| 1000 | SOS1 | $-2.10995 \cdot 10^6$ | 70.57 | 0.00 |
| 1000 | big-M50 | $-2.10995 \cdot 10^6$ | 1072.15 | 0.00 |
| 1000 | persp. | $-2.10995 \cdot 10^6$ | 452.04 | 0.00 |
| 1000 | eig. persp. | $-2.10995 \cdot 10^6$ | 4.22 | 0.00 |
| 1000 | ours | $-2.10995 \cdot 10^6$ | **0.57** | **0.00** |
| 3000 | SOS1 | $-1.87788 \cdot 10^6$ | 3600.81 | 12.60 |
| 3000 | big-M50 | $-7.92671 \cdot 10^5$ | 3605.01 | 166.77 |
| 3000 | persp. | $-2.10220 \cdot 10^6$ | 3600.04 | 0.55 |
| 3000 | eig. persp. | $-2.10220 \cdot 10^6$ | 102.51 | 0.00 |
| 3000 | ours | $-2.10220 \cdot 10^6$ | **1.68** | **0.00** |
| 5000 | SOS1 | $-1.98929 \cdot 10^6$ | 3600.16 | 6.43 |
| 5000 | big-M50 | $-7.89596 \cdot 10^5$ | 3617.52 | 168.16 |
| 5000 | persp. | $-2.09584 \cdot 10^6$ | 3601.36 | 1.00 |
| 5000 | eig. persp. | $-2.09625 \cdot 10^6$ | 475.37 | 0.00 |
| 5000 | ours | $-2.09625 \cdot 10^6$ | **5.09** | **0.00** |

Table 21: Comparison of time and optimality gap on the synthetic datasets (n = 100000, k = 10, $\rho = 0.1$, $\lambda_2 = 10.0$).

| # of features | method | upper bound | time(s) | gap(%) |
|---|---|---|---|---|
| 100 | SOS1 | $-5.28716 \cdot 10^6$ | 2.98 | 0.00 |
| 100 | big-M50 | $-5.28713 \cdot 10^6$ | 0.40 | 0.00 |
| 100 | persp. | $-5.28716 \cdot 10^6$ | 0.77 | 0.00 |
| 100 | eig. persp. | $-5.28716 \cdot 10^6$ | **0.10** | **0.00** |
| 100 | ours | $-5.28716 \cdot 10^6$ | 0.30 | 0.00 |
| 500 | SOS1 | $-5.31842 \cdot 10^6$ | 10.06 | 0.00 |
| 500 | big-M50 | $-5.31819 \cdot 10^6$ | 89.34 | 0.00 |
| 500 | persp. | $-5.31841 \cdot 10^6$ | 8.73 | 0.00 |
| 500 | eig. persp. | $-5.31842 \cdot 10^6$ | 0.60 | 0.00 |
| 500 | ours | $-5.31842 \cdot 10^6$ | **0.41** | **0.00** |
| 1000 | SOS1 | $-5.26964 \cdot 10^6$ | 87.65 | 0.00 |
| 1000 | big-M50 | $-5.26964 \cdot 10^6$ | 1869.66 | 0.00 |
| 1000 | persp. | $-5.26964 \cdot 10^6$ | 97.81 | 0.00 |
| 1000 | eig. persp. | $-5.26964 \cdot 10^6$ | 226.32 | 0.00 |
| 1000 | ours | $-5.26964 \cdot 10^6$ | **0.51** | **0.00** |
| 3000 | SOS1 | $-5.27658 \cdot 10^6$ | 3600.46 | 0.55 |
| 3000 | big-M50 | $-3.96610 \cdot 10^6$ | 3600.32 | 33.83 |
| 3000 | persp. | $-5.27658 \cdot 10^6$ | 3600.06 | 0.58 |
| 3000 | eig. persp. | - | 3615.07 | - |
| 3000 | ours | $-5.27658 \cdot 10^6$ | **1.76** | **0.00** |
| 5000 | SOS1 | $-5.16077 \cdot 10^6$ | 3603.93 | 3.16 |
| 5000 | big-M50 | $-3.96757 \cdot 10^6$ | 3607.78 | 34.19 |
| 5000 | persp. | $-5.27116 \cdot 10^6$ | 3600.43 | 1.00 |
| 5000 | eig. persp. | $-5.27116 \cdot 10^6$ | 614.83 | 0.00 |
| 5000 | ours | $-5.27116 \cdot 10^6$ | **4.32** | **0.00** |

Table 22: Comparison of time and optimality gap on the synthetic datasets (n = 100000, k = 10, $\rho = 0.3$, $\lambda_2 = 10.0$).

| # of features | method | upper bound | time(s) | gap(%) |
|---|---|---|---|---|
| 100 | SOS1 | $-1.10083 \cdot 10^7$ | 2.83 | 0.00 |
| 100 | big-M50 | $-1.10083 \cdot 10^7$ | 0.44 | 0.00 |
| 100 | persp. | $-1.10083 \cdot 10^7$ | 0.46 | 0.00 |
| 100 | eig. persp. | $-1.10083 \cdot 10^7$ | **0.13** | **0.00** |
| 100 | ours | $-1.10083 \cdot 10^7$ | 0.32 | 0.00 |
| 500 | SOS1 | $-1.10517 \cdot 10^7$ | 11.73 | 0.00 |
| 500 | big-M50 | $-1.10515 \cdot 10^7$ | 91.37 | 0.00 |
| 500 | persp. | $-1.10517 \cdot 10^7$ | 3600.01 | 0.08 |
| 500 | eig. persp. | $-1.10517 \cdot 10^7$ | 0.63 | 0.00 |
| 500 | ours | $-1.10517 \cdot 10^7$ | **0.41** | **0.00** |
| 1000 | SOS1 | $-1.09507 \cdot 10^7$ | 90.27 | 0.00 |
| 1000 | big-M50 | $-1.09505 \cdot 10^7$ | 1534.37 | 0.00 |
| 1000 | persp. | $-1.09507 \cdot 10^7$ | 3600.04 | 0.19 |
| 1000 | eig. persp. | $-1.09507 \cdot 10^7$ | 3.92 | 0.00 |
| 1000 | ours | $-1.09507 \cdot 10^7$ | **0.55** | **0.00** |
| 3000 | SOS1 | $-1.10011 \cdot 10^7$ | 3600.92 | 0.51 |
| 3000 | big-M50 | - | 3603.26 | - |
| 3000 | persp. | $-1.10011 \cdot 10^7$ | 3601.24 | 0.58 |
| 3000 | eig. persp. | - | 3646.60 | - |
| 3000 | ours | $-1.10011 \cdot 10^7$ | **1.73** | **0.00** |
| 5000 | SOS1 | $-1.08825 \cdot 10^7$ | 3600.12 | 2.06 |
| 5000 | big-M50 | $-9.62941 \cdot 10^6$ | 3608.37 | 15.34 |
| 5000 | persp. | $-1.09965 \cdot 10^7$ | 3602.42 | 1.00 |
| 5000 | eig. persp. | - | 3600.21 | - |
| 5000 | ours | $-1.09965 \cdot 10^7$ | **4.24** | **0.00** |

Table 23: Comparison of time and optimality gap on the synthetic datasets (n = 100000, k = 10, $\rho = 0.5$, $\lambda_2 = 10.0$).

| # of features | method | upper bound | time(s) | gap(%) |
|---|---|---|---|---|
| 100 | SOS1 | $-2.43574 \cdot 10^7$ | **0.05** | **0.00** |
| 100 | big-M50 | $-2.43574 \cdot 10^7$ | 0.42 | 0.00 |
| 100 | persp. | $-2.43574 \cdot 10^7$ | 0.55 | 0.00 |
| 100 | eig. persp. | $-2.43574 \cdot 10^7$ | 0.12 | 0.00 |
| 100 | ours | $-2.43574 \cdot 10^7$ | 0.30 | 0.00 |
| 500 | SOS1 | $-2.44184 \cdot 10^7$ | 10.39 | 0.00 |
| 500 | big-M50 | $-2.44178 \cdot 10^7$ | 85.12 | 0.00 |
| 500 | persp. | $-2.44184 \cdot 10^7$ | 248.24 | 0.00 |
| 500 | eig. persp. | $-2.44184 \cdot 10^7$ | 0.61 | 0.00 |
| 500 | ours | $-2.44184 \cdot 10^7$ | **0.44** | **0.00** |
| 1000 | SOS1 | $-2.41988 \cdot 10^7$ | 91.02 | 0.00 |
| 1000 | big-M50 | $-2.41986 \cdot 10^7$ | 1415.19 | 0.00 |
| 1000 | persp. | $-2.41988 \cdot 10^7$ | 3600.04 | 0.16 |
| 1000 | eig. persp. | $-2.41988 \cdot 10^7$ | 9.28 | 0.00 |
| 1000 | ours | $-2.41988 \cdot 10^7$ | **0.52** | **0.00** |
| 3000 | SOS1 | $-2.43712 \cdot 10^7$ | 3601.13 | 0.59 |
| 3000 | big-M50 | - | 3615.34 | - |
| 3000 | persp. | $-2.43712 \cdot 10^7$ | 3600.26 | 0.58 |
| 3000 | eig. persp. | - | 3669.39 | - |
| 3000 | ours | $-2.43712 \cdot 10^7$ | **1.81** | **0.00** |
| 5000 | SOS1 | $-2.42550 \cdot 10^7$ | 3600.13 | 1.48 |
| 5000 | big-M50 | $-2.43693 \cdot 10^7$ | 3607.06 | 1.00 |
| 5000 | persp. | $-2.43693 \cdot 10^7$ | 3601.61 | 1.00 |
| 5000 | eig. persp. | $-2.43693 \cdot 10^7$ | 619.11 | 0.00 |
| 5000 | ours | $-2.43693 \cdot 10^7$ | **4.08** | **0.00** |

Table 24: Comparison of time and optimality gap on the synthetic datasets (n = 100000, k = 10, $\rho = 0.7$, $\lambda_2 = 10.0$).

| # of features | method | upper bound | time(s) | gap(%) |
|---|---|---|---|---|
| 100 | SOS1 | $-9.11025 \cdot 10^7$ | **0.05** | **0.00** |
| 100 | big-M50 | $-9.11025 \cdot 10^7$ | 0.17 | 0.00 |
| 100 | persp. | $-9.11025 \cdot 10^7$ | 0.29 | 0.00 |
| 100 | eig. persp. | $-9.11018 \cdot 10^7$ | 11.35 | 0.00 |
| 100 | ours | $-9.11025 \cdot 10^7$ | 0.32 | 0.00 |
| 500 | SOS1 | $-9.12046 \cdot 10^7$ | 13.62 | 0.00 |
| 500 | big-M50 | $-9.12046 \cdot 10^7$ | 117.86 | 0.00 |
| 500 | persp. | $-9.12046 \cdot 10^7$ | 997.83 | 0.00 |
| 500 | eig. persp. | $-9.12046 \cdot 10^7$ | 84.29 | 0.00 |
| 500 | ours | $-9.12046 \cdot 10^7$ | **0.38** | **0.00** |
| 1000 | SOS1 | $-9.04074 \cdot 10^7$ | 1201.62 | 0.00 |
| 1000 | big-M50 | $-9.04074 \cdot 10^7$ | 3600.04 | 0.01 |
| 1000 | persp. | $-9.04074 \cdot 10^7$ | 3600.03 | 0.20 |
| 1000 | eig. persp. | $-9.04074 \cdot 10^7$ | 343.13 | 0.00 |
| 1000 | ours | $-9.04074 \cdot 10^7$ | **0.52** | **0.00** |
| 3000 | SOS1 | $-9.12760 \cdot 10^7$ | 3601.23 | 0.59 |
| 3000 | big-M50 | - | 3611.90 | - |
| 3000 | persp. | $-9.12755 \cdot 10^7$ | 3601.40 | 0.59 |
| 3000 | eig. persp. | - | 3625.51 | - |
| 3000 | ours | $-9.12760 \cdot 10^7$ | **20.84** | **0.00** |
| 5000 | SOS1 | $-8.98304 \cdot 10^7$ | 3600.14 | 2.65 |
| 5000 | big-M50 | - | 3615.45 | - |
| 5000 | persp. | $-9.12241 \cdot 10^7$ | 3600.13 | 1.07 |
| 5000 | eig. persp. | - | 3819.54 | - |
| 5000 | ours | $-9.12915 \cdot 10^7$ | **57.41** | **0.00** |

Table 25: Comparison of time and optimality gap on the synthetic datasets (n = 100000, k = 10, $\rho = 0.9$, $\lambda_2 = 10.0$).

### G.2.6 Synthetic 1 Benchmark with $\lambda_2 = 10.0$ with Warmstart

| # of features | method | upper bound | time(s) | gap(%) |
|---|---|---|---|---|
| 100 | SOS1 + warm start | $-2.10868 \cdot 10^6$ | 0.05 | 0.00 |
| 100 | big-M50 + warm start | $-2.10868 \cdot 10^6$ | 0.25 | 0.00 |
| 100 | persp. + warm start | $-2.10868 \cdot 10^6$ | 2.05 | 0.00 |
| 100 | eig. persp. + warm start | $-2.10868 \cdot 10^6$ | **0.01** | **0.00** |
| 100 | ours | $-2.10868 \cdot 10^6$ | 0.30 | 0.00 |
| 500 | SOS1 + warm start | $-2.12823 \cdot 10^6$ | 6.54 | 0.00 |
| 500 | big-M50 + warm start | $-2.12823 \cdot 10^6$ | 3.76 | 0.00 |
| 500 | persp. + warm start | $-2.12823 \cdot 10^6$ | 144.06 | 0.00 |
| 500 | eig. persp. + warm start | $-2.12823 \cdot 10^6$ | 0.61 | 0.00 |
| 500 | ours | $-2.12823 \cdot 10^6$ | **0.38** | **0.00** |
| 1000 | SOS1 + warm start | $-2.10995 \cdot 10^6$ | 67.85 | 0.00 |
| 1000 | big-M50 + warm start | $-2.10995 \cdot 10^6$ | 23.00 | 0.00 |
| 1000 | persp. + warm start | $-2.10995 \cdot 10^6$ | 436.41 | 0.00 |
| 1000 | eig. persp. + warm start | $-2.10995 \cdot 10^6$ | 4.18 | 0.00 |
| 1000 | ours | $-2.10995 \cdot 10^6$ | **0.57** | **0.00** |
| 3000 | SOS1 + warm start | $-2.10220 \cdot 10^6$ | 2171.96 | 0.00 |
| 3000 | big-M50 + warm start | $-2.10220 \cdot 10^6$ | 241.10 | 0.00 |
| 3000 | persp. + warm start | $-2.10220 \cdot 10^6$ | 3571.06 | 0.00 |
| 3000 | eig. persp. + warm start | $-2.10220 \cdot 10^6$ | 99.06 | 0.00 |
| 3000 | ours | $-2.10220 \cdot 10^6$ | **1.68** | **0.00** |
| 5000 | SOS1 + warm start | $-2.09625 \cdot 10^6$ | 3602.96 | 0.95 |
| 5000 | big-M50 + warm start | $-2.09625 \cdot 10^6$ | 1309.75 | 0.00 |
| 5000 | persp. + warm start | $-2.09625 \cdot 10^6$ | 3601.45 | 0.98 |
| 5000 | eig. persp. + warm start | $-2.09625 \cdot 10^6$ | 442.61 | 0.00 |
| 5000 | ours | $-2.09625 \cdot 10^6$ | **5.09** | **0.00** |

Table 26: Comparison of time and optimality gap on the synthetic datasets (n = 100000, k = 10, $\rho = 0.1$, $\lambda_2 = 10.0$). All baselines use our beam search solution as a warm start.

| # of features | method | upper bound | time(s) | gap(%) |
|---|---|---|---|---|
| 100 | SOS1 + warm start | $-5.28716 \cdot 10^6$ | 0.05 | 0.00 |
| 100 | big-M50 + warm start | $-5.28716 \cdot 10^6$ | 0.25 | 0.00 |
| 100 | persp. + warm start | $-5.28716 \cdot 10^6$ | 0.89 | 0.00 |
| 100 | eig. persp. + warm start | $-5.28716 \cdot 10^6$ | **0.02** | **0.00** |
| 100 | ours | $-5.28716 \cdot 10^6$ | 0.30 | 0.00 |
| 500 | SOS1 + warm start | $-5.31842 \cdot 10^6$ | 7.34 | 0.00 |
| 500 | big-M50 + warm start | $-5.31842 \cdot 10^6$ | 2.93 | 0.00 |
| 500 | persp. + warm start | $-5.31842 \cdot 10^6$ | 8.30 | 0.00 |
| 500 | eig. persp. + warm start | $-5.31842 \cdot 10^6$ | 0.62 | 0.00 |
| 500 | ours | $-5.31842 \cdot 10^6$ | **0.41** | **0.00** |
| 1000 | SOS1 + warm start | $-5.26964 \cdot 10^6$ | 84.96 | 0.00 |
| 1000 | big-M50 + warm start | $-5.26964 \cdot 10^6$ | 15.10 | 0.00 |
| 1000 | persp. + warm start | $-5.26964 \cdot 10^6$ | 97.38 | 0.00 |
| 1000 | eig. persp. + warm start | $-5.26964 \cdot 10^6$ | 4.10 | 0.00 |
| 1000 | ours | $-5.26964 \cdot 10^6$ | **0.51** | **0.00** |
| 3000 | SOS1 + warm start | $-5.27658 \cdot 10^6$ | 2809.30 | 0.00 |
| 3000 | big-M50 + warm start | $-5.27658 \cdot 10^6$ | 315.30 | 0.00 |
| 3000 | persp. + warm start | $-5.27658 \cdot 10^6$ | 152.21 | 0.00 |
| 3000 | eig. persp. + warm start | $-5.27658 \cdot 10^6$ | 100.20 | 0.00 |
| 3000 | ours | $-5.27658 \cdot 10^6$ | **1.76** | **0.00** |
| 5000 | SOS1 + warm start | $-5.27116 \cdot 10^6$ | 3601.86 | 0.99 |
| 5000 | big-M50 + warm start | $-5.27116 \cdot 10^6$ | 1100.87 | 0.00 |
| 5000 | persp. + warm start | $-5.27116 \cdot 10^6$ | 633.11 | 0.00 |
| 5000 | eig. persp. + warm start | $-5.27116 \cdot 10^6$ | 460.01 | 0.00 |
| 5000 | ours | $-5.27116 \cdot 10^6$ | **4.32** | **0.00** |

Table 27: Comparison of time and optimality gap on the synthetic datasets (n = 100000, k = 10, $\rho = 0.3$, $\lambda_2 = 10.0$). All baselines use our beam search solution as a warm start.

| # of features | method | upper bound | time(s) | gap(%) |
|---|---|---|---|---|
| 100 | SOS1 + warm start | $-1.10083 \cdot 10^7$ | 0.05 | 0.00 |
| 100 | big-M50 + warm start | $-1.10083 \cdot 10^7$ | 0.06 | 0.00 |
| 100 | persp. + warm start | $-1.10083 \cdot 10^7$ | 0.46 | 0.00 |
| 100 | eig. persp. + warm start | $-1.10083 \cdot 10^7$ | **0.02** | **0.00** |
| 100 | ours | $-1.10083 \cdot 10^7$ | 0.32 | 0.00 |
| 500 | SOS1 + warm start | $-1.10517 \cdot 10^7$ | 10.26 | 0.00 |
| 500 | big-M50 + warm start | $-1.10517 \cdot 10^7$ | 8.51 | 0.00 |
| 500 | persp. + warm start | $-1.10517 \cdot 10^7$ | 3600.01 | 0.08 |
| 500 | eig. persp. + warm start | $-1.10517 \cdot 10^7$ | 0.64 | 0.00 |
| 500 | ours | $-1.10517 \cdot 10^7$ | **0.41** | **0.00** |
| 1000 | SOS1 + warm start | $-1.09507 \cdot 10^7$ | 88.05 | 0.00 |
| 1000 | big-M50 + warm start | $-1.09507 \cdot 10^7$ | 14.03 | 0.00 |
| 1000 | persp. + warm start | $-1.09507 \cdot 10^7$ | 10.25 | 0.00 |
| 1000 | eig. persp. + warm start | $-1.09507 \cdot 10^7$ | 4.13 | 0.00 |
| 1000 | ours | $-1.09507 \cdot 10^7$ | **0.55** | **0.00** |
| 3000 | SOS1 + warm start | $-1.10011 \cdot 10^7$ | 3008.66 | 0.00 |
| 3000 | big-M50 + warm start | $-1.10011 \cdot 10^7$ | 201.32 | 0.00 |
| 3000 | persp. + warm start | $-1.10011 \cdot 10^7$ | 153.29 | 0.00 |
| 3000 | eig. persp. + warm start | $-1.10011 \cdot 10^7$ | 102.13 | 0.00 |
| 3000 | ours | $-1.10011 \cdot 10^7$ | **1.73** | **0.00** |
| 5000 | SOS1 + warm start | $-1.09965 \cdot 10^7$ | 3601.18 | 0.99 |
| 5000 | big-M50 + warm start | $-1.09965 \cdot 10^7$ | 3601.20 | 1.00 |
| 5000 | persp. + warm start | $-1.09965 \cdot 10^7$ | 3602.12 | 1.00 |
| 5000 | eig. persp. + warm start | $-1.09965 \cdot 10^7$ | 499.65 | 0.00 |
| 5000 | ours | $-1.09965 \cdot 10^7$ | **4.24** | **0.00** |

Table 28: Comparison of time and optimality gap on the synthetic datasets (n = 100000, k = 10, $\rho = 0.5$, $\lambda_2 = 10.0$). All baselines use our beam search solution as a warm start.

| # of features | method | upper bound | time(s) | gap(%) |
|---|---|---|---|---|
| 100 | SOS1 + warm start | $-2.43574 \cdot 10^7$ | 0.05 | 0.00 |
| 100 | big-M50 + warm start | $-2.43574 \cdot 10^7$ | 0.16 | 0.00 |
| 100 | persp. + warm start | $-2.43574 \cdot 10^7$ | 0.51 | 0.00 |
| 100 | eig. persp. + warm start | $-2.43574 \cdot 10^7$ | **0.02** | **0.00** |
| 100 | ours | $-2.43574 \cdot 10^7$ | 0.30 | 0.00 |
| 500 | SOS1 + warm start | $-2.44184 \cdot 10^7$ | 8.95 | 0.00 |
| 500 | big-M50 + warm start | $-2.44184 \cdot 10^7$ | 2.11 | 0.00 |
| 500 | persp. + warm start | $-2.44184 \cdot 10^7$ | 215.86 | 0.00 |
| 500 | eig. persp. + warm start | $-2.44184 \cdot 10^7$ | 0.61 | 0.00 |
| 500 | ours | $-2.44184 \cdot 10^7$ | **0.44** | **0.00** |
| 1000 | SOS1 + warm start | $-2.41988 \cdot 10^7$ | 86.91 | 0.00 |
| 1000 | big-M50 + warm start | $-2.41988 \cdot 10^7$ | 12.39 | 0.00 |
| 1000 | persp. + warm start | $-2.41988 \cdot 10^7$ | 3600.03 | 0.08 |
| 1000 | eig. persp. + warm start | $-2.41988 \cdot 10^7$ | 4.14 | 0.00 |
| 1000 | ours | $-2.41988 \cdot 10^7$ | **0.52** | **0.00** |
| 3000 | SOS1 + warm start | $-2.43712 \cdot 10^7$ | 3601.60 | 0.59 |
| 3000 | big-M50 + warm start | $-2.43712 \cdot 10^7$ | 3600.43 | 0.59 |
| 3000 | persp. + warm start | $-2.43712 \cdot 10^7$ | 3601.09 | 0.58 |
| 3000 | eig. persp. + warm start | $-2.43712 \cdot 10^7$ | 113.06 | 0.00 |
| 3000 | ours | $-2.43712 \cdot 10^7$ | **1.81** | **0.00** |
| 5000 | SOS1 + warm start | $-2.43693 \cdot 10^7$ | 3601.29 | 1.00 |
| 5000 | big-M50 + warm start | $-2.43693 \cdot 10^7$ | 3624.24 | 1.00 |
| 5000 | persp. + warm start | $-2.43693 \cdot 10^7$ | 3600.19 | 1.00 |
| 5000 | eig. persp. + warm start | $-2.43693 \cdot 10^7$ | 457.54 | 0.00 |
| 5000 | ours | $-2.43693 \cdot 10^7$ | **4.08** | **0.00** |

Table 29: Comparison of time and optimality gap on the synthetic datasets (n = 100000, k = 10, $\rho = 0.7$, $\lambda_2 = 10.0$). All baselines use our beam search solution as a warm start.

| # of features | method | upper bound | time(s) | gap(%) |
|---|---|---|---|---|
| 100 | SOS1 + warm start | $-9.11025 \cdot 10^7$ | 0.04 | 0.00 |
| 100 | big-M50 + warm start | $-9.11025 \cdot 10^7$ | 0.04 | 0.00 |
| 100 | persp. + warm start | $-9.11025 \cdot 10^7$ | 0.29 | 0.00 |
| 100 | eig. persp. + warm start | $-9.11025 \cdot 10^7$ | **0.03** | **0.00** |
| 100 | ours | $-9.11025 \cdot 10^7$ | 0.32 | 0.00 |
| 500 | SOS1 + warm start | $-9.12046 \cdot 10^7$ | 54.77 | 0.00 |
| 500 | big-M50 + warm start | $-9.12046 \cdot 10^7$ | 36.63 | 0.00 |
| 500 | persp. + warm start | $-9.12046 \cdot 10^7$ | 306.17 | 0.00 |
| 500 | eig. persp. + warm start | $-9.12046 \cdot 10^7$ | 0.59 | 0.00 |
| 500 | ours | $-9.12046 \cdot 10^7$ | **0.38** | **0.00** |
| 1000 | SOS1 + warm start | $-9.04074 \cdot 10^7$ | 1178.82 | 0.00 |
| 1000 | big-M50 + warm start | $-9.04074 \cdot 10^7$ | 3600.19 | 0.06 |
| 1000 | persp. + warm start | $-9.04074 \cdot 10^7$ | 3600.04 | 0.19 |
| 1000 | eig. persp. + warm start | $-9.04074 \cdot 10^7$ | 4.15 | 0.00 |
| 1000 | ours | $-9.04074 \cdot 10^7$ | **0.52** | **0.00** |
| 3000 | SOS1 + warm start | $-9.12760 \cdot 10^7$ | 3600.52 | 0.59 |
| 3000 | big-M50 + warm start | $-9.12760 \cdot 10^7$ | 3606.60 | 0.59 |
| 3000 | persp. + warm start | $-9.12760 \cdot 10^7$ | 3600.05 | 0.59 |
| 3000 | eig. persp. + warm start | $-9.12760 \cdot 10^7$ | 103.22 | 0.00 |
| 3000 | ours | $-9.12760 \cdot 10^7$ | **20.84** | **0.00** |
| 5000 | SOS1 + warm start | $-9.12915 \cdot 10^7$ | 3602.40 | 1.00 |
| 5000 | big-M50 + warm start | $-9.12915 \cdot 10^7$ | 3606.47 | 1.00 |
| 5000 | persp. + warm start | $-9.12915 \cdot 10^7$ | 3600.16 | 1.00 |
| 5000 | eig. persp. + warm start | $-9.12915 \cdot 10^7$ | 490.05 | 0.00 |
| 5000 | ours | $-9.12915 \cdot 10^7$ | **57.41** | **0.00** |

Table 30: Comparison of time and optimality gap on the synthetic datasets (n = 100000, k = 10, $\rho = 0.9$, $\lambda_2 = 10.0$). All baselines use our beam search solution as a warm start.

### G.2.7 Synthetic 2 Benchmark with $\lambda_2 = 0.001$ with No Warmstart

| # of samples | method | upper bound | time(s) | gap(%) |
|---|---|---|---|---|
| 3000 | SOS1 | $-5.42178 \cdot 10^4$ | 3600.10 | 39.82 |
| 3000 | big-M50 | $-6.34585 \cdot 10^4$ | **3600.07** | **19.16** |
| 3000 | persp. | $-4.05327 \cdot 10^4$ | 3600.06 | 87.02 |
| 3000 | eig. persp. | $-5.42702 \cdot 10^4$ | 3600.07 | 39.69 |
| 3000 | ours | $-6.34585 \cdot 10^4$ | 3600.49 | 19.45 |
| 4000 | SOS1 | $-8.64294 \cdot 10^4$ | 3601.19 | 15.37 |
| 4000 | big-M50 | $-8.64294 \cdot 10^4$ | 3600.07 | 15.32 |
| 4000 | persp. | $-8.29421 \cdot 10^4$ | 3600.06 | 20.22 |
| 4000 | eig. persp. | $-8.29301 \cdot 10^4$ | 3600.08 | 6.80 |
| 4000 | ours | $-8.64294 \cdot 10^4$ | **20.25** | **0.00** |
| 5000 | SOS1 | $-9.85324 \cdot 10^4$ | 3600.10 | 24.46 |
| 5000 | big-M50 | $-1.09576 \cdot 10^5$ | 3600.81 | 10.25 |
| 5000 | persp. | $-1.09576 \cdot 10^5$ | 3600.07 | 11.91 |
| 5000 | eig. persp. | $-1.09576 \cdot 10^5$ | 3600.07 | 0.05 |
| 5000 | ours | $-1.09576 \cdot 10^5$ | **20.51** | **0.00** |
| 6000 | SOS1 | $-1.15043 \cdot 10^5$ | 3600.09 | 21.96 |
| 6000 | big-M50 | $-1.27643 \cdot 10^5$ | 3600.08 | 9.93 |
| 6000 | persp. | $-1.27643 \cdot 10^5$ | 3600.14 | 9.93 |
| 6000 | eig. persp. | $-1.27643 \cdot 10^5$ | 3600.06 | 12.64 |
| 6000 | ours | $-1.27643 \cdot 10^5$ | **1.70** | **0.00** |
| 7000 | SOS1 | $-1.50614 \cdot 10^5$ | 3600.86 | 8.32 |
| 7000 | big-M50 | $-1.50614 \cdot 10^5$ | 3600.09 | 8.29 |
| 7000 | persp. | $-1.50614 \cdot 10^5$ | 3600.21 | 8.33 |
| 7000 | eig. persp. | $-1.50614 \cdot 10^5$ | 106.73 | 0.00 |
| 7000 | ours | $-1.50614 \cdot 10^5$ | **1.67** | **0.00** |

Table 31: Comparison of time and optimality gap on the synthetic datasets (p = 3000, k = 10, $\rho = 0.1$, $\lambda_2 = 0.001$).

| # of samples | method | upper bound | time(s) | gap(%) |
|---|---|---|---|---|
| 3000 | SOS1 | $-1.50509 \cdot 10^5$ | 3601.34 | 26.68 |
| 3000 | big-M50 | $-1.59509 \cdot 10^5$ | **3600.08** | **19.07** |
| 3000 | persp. | $-1.48091 \cdot 10^5$ | 3600.99 | 28.74 |
| 3000 | eig. persp. | $-1.35700 \cdot 10^5$ | 3600.07 | 40.51 |
| 3000 | ours | $-1.59509 \cdot 10^5$ | 3600.67 | 19.53 |
| 4000 | SOS1 | $-2.09237 \cdot 10^5$ | 3600.08 | 19.87 |
| 4000 | big-M50 | $-2.17574 \cdot 10^5$ | 3600.07 | 14.78 |
| 4000 | persp. | $-2.17574 \cdot 10^5$ | 3600.06 | 14.91 |
| 4000 | eig. persp. | $-2.02936 \cdot 10^5$ | 3600.06 | 23.67 |
| 4000 | ours | $-2.17574 \cdot 10^5$ | **100.13** | **0.00** |
| 5000 | SOS1 | $-2.71986 \cdot 10^5$ | 3600.29 | 12.00 |
| 5000 | big-M50 | $-2.71986 \cdot 10^5$ | 3600.07 | 11.22 |
| 5000 | persp. | $-2.71986 \cdot 10^5$ | 3600.40 | 11.98 |
| 5000 | eig. persp. | $-2.71986 \cdot 10^5$ | 2152.71 | 0.00 |
| 5000 | ours | $-2.71986 \cdot 10^5$ | **20.80** | **0.00** |
| 6000 | SOS1 | $-3.16761 \cdot 10^5$ | 3600.70 | 9.96 |
| 6000 | big-M50 | $-3.16761 \cdot 10^5$ | 3600.09 | 9.96 |
| 6000 | persp. | $-3.16761 \cdot 10^5$ | 3600.06 | 9.96 |
| 6000 | eig. persp. | $-3.16761 \cdot 10^5$ | 1968.93 | 0.00 |
| 6000 | ours | $-3.16761 \cdot 10^5$ | **20.70** | **0.00** |
| 7000 | SOS1 | $-3.72610 \cdot 10^5$ | 3601.25 | 8.33 |
| 7000 | big-M50 | $-3.72610 \cdot 10^5$ | 3601.19 | 8.33 |
| 7000 | persp. | $-3.72610 \cdot 10^5$ | 3600.14 | 8.33 |
| 7000 | eig. persp. | $-3.72610 \cdot 10^5$ | 2906.29 | 0.00 |
| 7000 | ours | $-3.72610 \cdot 10^5$ | **1.98** | **0.00** |

Table 32: Comparison of time and optimality gap on the synthetic datasets (p = 3000, k = 10, $\rho = 0.3$, $\lambda_2 = 0.001$).

| # of samples | method | upper bound | time(s) | gap(%) |
|---|---|---|---|---|
| 3000 | SOS1 | $-3.24748 \cdot 10^5$ | 3600.06 | 22.83 |
| 3000 | big-M50 | $-3.33604 \cdot 10^5$ | **3600.08** | **18.79** |
| 3000 | persp. | $-3.03254 \cdot 10^5$ | 3600.06 | 31.54 |
| 3000 | eig. persp. | $-3.04485 \cdot 10^5$ | 3600.80 | 31.01 |
| 3000 | ours | $-3.33604 \cdot 10^5$ | 3600.58 | 19.57 |
| 4000 | SOS1 | $-4.49670 \cdot 10^5$ | 3600.55 | 16.16 |
| 4000 | big-M50 | $-4.53261 \cdot 10^5$ | 3600.07 | 14.98 |
| 4000 | persp. | $-4.50269 \cdot 10^5$ | 3600.06 | 15.99 |
| 4000 | eig. persp. | $-4.30194 \cdot 10^5$ | 3600.06 | 21.59 |
| 4000 | ours | $-4.53261 \cdot 10^5$ | **3600.25** | **0.63** |
| 5000 | SOS1 | $-5.57342 \cdot 10^5$ | 3600.12 | 13.18 |
| 5000 | big-M50 | $-5.63007 \cdot 10^5$ | 3601.19 | 11.68 |
| 5000 | persp. | $-5.63007 \cdot 10^5$ | 3600.57 | 11.75 |
| 5000 | eig. persp. | $-5.63007 \cdot 10^5$ | 3600.05 | 0.29 |
| 5000 | ours | $-5.63007 \cdot 10^5$ | **36.97** | **0.00** |
| 6000 | SOS1 | $-6.51710 \cdot 10^5$ | 3600.08 | 11.01 |
| 6000 | big-M50 | $-6.57799 \cdot 10^5$ | 3600.34 | 9.98 |
| 6000 | persp. | $-6.57799 \cdot 10^5$ | 3600.15 | 9.97 |
| 6000 | eig. persp. | $-6.57799 \cdot 10^5$ | 1685.67 | 0.00 |
| 6000 | ours | $-6.57799 \cdot 10^5$ | **20.75** | **0.00** |
| 7000 | SOS1 | $-7.46135 \cdot 10^5$ | 3600.71 | 12.13 |
| 7000 | big-M50 | $-7.72267 \cdot 10^5$ | 3600.11 | 8.33 |
| 7000 | persp. | $-7.72267 \cdot 10^5$ | 3600.50 | 8.32 |
| 7000 | eig. persp. | $-7.72267 \cdot 10^5$ | 3600.27 | 10.37 |
| 7000 | ours | $-7.72267 \cdot 10^5$ | **21.02** | **0.00** |

Table 33: Comparison of time and optimality gap on the synthetic datasets (p = 3000, k = 10, $\rho = 0.5$, $\lambda_2 = 0.001$).

| # of samples | method | upper bound | time(s) | gap(%) |
|---|---|---|---|---|
| 3000 | SOS1 | $-7.34855 \cdot 10^5$ | 3600.26 | 20.63 |
| 3000 | big-M50 | $-7.41187 \cdot 10^5$ | **3600.05** | **18.15** |
| 3000 | persp. | $-6.86643 \cdot 10^5$ | 3600.74 | 29.10 |
| 3000 | eig. persp. | $-7.02765 \cdot 10^5$ | 3600.06 | 26.14 |
| 3000 | ours | $-7.41187 \cdot 10^5$ | 3601.14 | 19.60 |
| 4000 | SOS1 | $-1.00137 \cdot 10^6$ | 3601.49 | 15.39 |
| 4000 | big-M50 | $-1.00103 \cdot 10^6$ | 3600.07 | 15.19 |
| 4000 | persp. | $-9.96124 \cdot 10^5$ | 3600.10 | 15.98 |
| 4000 | eig. persp. | $-9.90959 \cdot 10^5$ | 3600.06 | 17.00 |
| 4000 | ours | $-1.00287 \cdot 10^6$ | **3600.37** | **1.88** |
| 5000 | SOS1 | $-1.23058 \cdot 10^6$ | 3600.25 | 12.96 |
| 5000 | big-M50 | $-1.24027 \cdot 10^6$ | 3601.10 | 11.88 |
| 5000 | persp. | $-1.23565 \cdot 10^6$ | 3601.00 | 12.49 |
| 5000 | eig. persp. | $-1.22403 \cdot 10^6$ | 3600.05 | 2.31 |
| 5000 | ours | $-1.24027 \cdot 10^6$ | **466.42** | **0.00** |
| 6000 | SOS1 | $-1.44944 \cdot 10^6$ | 3600.09 | 10.36 |
| 6000 | big-M50 | $-1.45417 \cdot 10^6$ | 3600.11 | 9.78 |
| 6000 | persp. | $-1.45417 \cdot 10^6$ | 3600.55 | 9.82 |
| 6000 | eig. persp. | $-1.45417 \cdot 10^6$ | 3430.19 | 0.00 |
| 6000 | ours | $-1.45417 \cdot 10^6$ | **57.67** | **0.00** |
| 7000 | SOS1 | $-1.68757 \cdot 10^6$ | 3600.20 | 9.45 |
| 7000 | big-M50 | $-1.70483 \cdot 10^6$ | 3601.55 | 8.34 |
| 7000 | persp. | $-1.70483 \cdot 10^6$ | 3600.96 | 8.13 |
| 7000 | eig. persp. | $-1.70483 \cdot 10^6$ | 3432.42 | 0.00 |
| 7000 | ours | $-1.70483 \cdot 10^6$ | **21.42** | **0.00** |

Table 34: Comparison of time and optimality gap on the synthetic datasets (p = 3000, k = 10, $\rho = 0.7$, $\lambda_2 = 0.001$).

| # of samples | method | upper bound | time(s) | gap(%) |
|---|---|---|---|---|
| 3000 | SOS1 | $-2.77903 \cdot 10^6$ | 3600.07 | 19.87 |
| 3000 | big-M50 | $-2.78521 \cdot 10^6$ | **3600.15** | **15.90** |
| 3000 | persp. | $-2.76075 \cdot 10^6$ | 3600.05 | 20.66 |
| 3000 | eig. persp. | $-2.74468 \cdot 10^6$ | 3600.81 | 21.37 |
| 3000 | ours | $-2.78521 \cdot 10^6$ | 3600.01 | 19.60 |
| 4000 | SOS1 | $-3.75203 \cdot 10^6$ | 3600.12 | 15.10 |
| 4000 | big-M50 | $-3.75202 \cdot 10^6$ | 3600.06 | 13.33 |
| 4000 | persp. | $-3.72754 \cdot 10^6$ | 3600.07 | 15.88 |
| 4000 | eig. persp. | $-3.72921 \cdot 10^6$ | 3600.51 | 16.56 |
| 4000 | ours | $-3.75203 \cdot 10^6$ | **3600.42** | **3.50** |
| 5000 | SOS1 | $-4.61204 \cdot 10^6$ | 3600.15 | 12.27 |
| 5000 | big-M50 | $-4.61877 \cdot 10^6$ | 3600.19 | 11.37 |
| 5000 | persp. | $-4.60317 \cdot 10^6$ | 3600.36 | 12.48 |
| 5000 | eig. persp. | $-4.60063 \cdot 10^6$ | 3600.26 | 13.32 |
| 5000 | ours | $-4.61877 \cdot 10^6$ | **3600.57** | **0.93** |
| 6000 | SOS1 | $-5.41709 \cdot 10^6$ | 3600.27 | 10.45 |
| 6000 | big-M50 | $-5.43826 \cdot 10^6$ | 3600.06 | 9.77 |
| 6000 | persp. | $-5.41956 \cdot 10^6$ | 3600.06 | 10.39 |
| 6000 | eig. persp. | $-5.40204 \cdot 10^6$ | 3600.06 | 11.80 |
| 6000 | ours | $-5.43826 \cdot 10^6$ | **3600.65** | **0.33** |
| 7000 | SOS1 | $-6.35797 \cdot 10^6$ | 3600.07 | 8.51 |
| 7000 | big-M50 | $-6.36767 \cdot 10^6$ | 3600.38 | 8.22 |
| 7000 | persp. | $-6.35116 \cdot 10^6$ | 3600.24 | 8.62 |
| 7000 | eig. persp. | $-6.36156 \cdot 10^6$ | 3610.76 | 0.53 |
| 7000 | ours | $-6.36767 \cdot 10^6$ | **3600.78** | **0.11** |

Table 35: Comparison of time and optimality gap on the synthetic datasets (p = 3000, k = 10, $\rho = 0.9$, $\lambda_2 = 0.001$).

### G.2.8 Synthetic 2 Benchmark with $\lambda_2 = 0.001$ with Warmstart

| # of samples | method | upper bound | time(s) | gap(%) |
|---|---|---|---|---|
| 3000 | SOS1 + warm start | $-6.34585 \cdot 10^4$ | 3600.96 | 19.46 |
| 3000 | big-M50 + warm start | $-6.34585 \cdot 10^4$ | **3601.20** | **19.16** |
| 3000 | persp. + warm start | $-6.34585 \cdot 10^4$ | 3600.10 | 19.45 |
| 3000 | eig. persp. + warm start | $-6.34585 \cdot 10^4$ | 3600.13 | 19.46 |
| 3000 | ours | $-6.34585 \cdot 10^4$ | 3600.49 | 19.45 |
| 4000 | SOS1 + warm start | $-8.64294 \cdot 10^4$ | 3600.56 | 15.37 |
| 4000 | big-M50 + warm start | $-8.64294 \cdot 10^4$ | 3600.87 | 15.32 |
| 4000 | persp. + warm start | $-8.64294 \cdot 10^4$ | 3600.07 | 14.33 |
| 4000 | eig. persp. + warm start | $-8.64294 \cdot 10^4$ | 3600.12 | 1.64 |
| 4000 | ours | $-8.64294 \cdot 10^4$ | **20.25** | **0.00** |
| 5000 | SOS1 + warm start | $-1.09576 \cdot 10^5$ | 3600.38 | 11.91 |
| 5000 | big-M50 + warm start | $-1.09576 \cdot 10^5$ | 3601.86 | 10.25 |
| 5000 | persp. + warm start | $-1.09576 \cdot 10^5$ | 3600.07 | 11.91 |
| 5000 | eig. persp. + warm start | $-1.09576 \cdot 10^5$ | 3389.68 | 0.00 |
| 5000 | ours | $-1.09576 \cdot 10^5$ | **20.51** | **0.00** |
| 6000 | SOS1 + warm start | $-1.27643 \cdot 10^5$ | 3600.46 | 9.92 |
| 6000 | big-M50 + warm start | $-1.27643 \cdot 10^5$ | 3601.14 | 9.93 |
| 6000 | persp. + warm start | $-1.27643 \cdot 10^5$ | 3600.05 | 9.93 |
| 6000 | eig. persp. + warm start | $-1.27643 \cdot 10^5$ | 3600.07 | 12.65 |
| 6000 | ours | $-1.27643 \cdot 10^5$ | **1.70** | **0.00** |
| 7000 | SOS1 + warm start | $-1.50614 \cdot 10^5$ | 3600.07 | 5.84 |
| 7000 | big-M50 + warm start | $-1.50614 \cdot 10^5$ | 3600.13 | 8.29 |
| 7000 | persp. + warm start | $-1.50614 \cdot 10^5$ | 3600.17 | 8.33 |
| 7000 | eig. persp. + warm start | $-1.50614 \cdot 10^5$ | 101.37 | 0.00 |
| 7000 | ours | $-1.50614 \cdot 10^5$ | **1.67** | **0.00** |

Table 36: Comparison of time and optimality gap on the synthetic datasets (p = 3000, k = 10, $\rho = 0.1$, $\lambda_2 = 0.001$). All baselines use our beam search solution as a warm start.

| # of samples | method | upper bound | time(s) | gap(%) |
|---|---|---|---|---|
| 3000 | SOS1 + warm start | $-1.59509 \cdot 10^5$ | 3600.74 | 19.53 |
| 3000 | big-M50 + warm start | $-1.59509 \cdot 10^5$ | **3600.60** | **19.07** |
| 3000 | persp. + warm start | $-1.59509 \cdot 10^5$ | 3601.09 | 19.53 |
| 3000 | eig. persp. + warm start | $-1.59509 \cdot 10^5$ | 3601.11 | 19.54 |
| 3000 | ours | $-1.59509 \cdot 10^5$ | 3600.67 | 19.53 |
| 4000 | SOS1 + warm start | $-2.17574 \cdot 10^5$ | 3600.07 | 15.28 |
| 4000 | big-M50 + warm start | $-2.17574 \cdot 10^5$ | 3600.10 | 14.78 |
| 4000 | persp. + warm start | $-2.17574 \cdot 10^5$ | 3601.12 | 14.91 |
| 4000 | eig. persp. + warm start | $-2.17574 \cdot 10^5$ | 3600.11 | 15.34 |
| 4000 | ours | $-2.17574 \cdot 10^5$ | **100.13** | **0.00** |
| 5000 | SOS1 + warm start | $-2.71986 \cdot 10^5$ | 3600.44 | 12.00 |
| 5000 | big-M50 + warm start | $-2.71986 \cdot 10^5$ | 3600.06 | 11.22 |
| 5000 | persp. + warm start | $-2.71986 \cdot 10^5$ | 3600.14 | 11.35 |
| 5000 | eig. persp. + warm start | $-2.71986 \cdot 10^5$ | 2118.55 | 0.00 |
| 5000 | ours | $-2.71986 \cdot 10^5$ | **20.80** | **0.00** |
| 6000 | SOS1 + warm start | $-3.16761 \cdot 10^5$ | 3600.73 | 9.96 |
| 6000 | big-M50 + warm start | $-3.16761 \cdot 10^5$ | 3600.07 | 9.96 |
| 6000 | persp. + warm start | $-3.16761 \cdot 10^5$ | 3601.28 | 9.96 |
| 6000 | eig. persp. + warm start | $-3.16761 \cdot 10^5$ | 3600.50 | 11.79 |
| 6000 | ours | $-3.16761 \cdot 10^5$ | **20.70** | **0.00** |
| 7000 | SOS1 + warm start | $-3.72610 \cdot 10^5$ | 3600.61 | 8.33 |
| 7000 | big-M50 + warm start | $-3.72610 \cdot 10^5$ | 3600.08 | 8.33 |
| 7000 | persp. + warm start | $-3.72610 \cdot 10^5$ | 3600.08 | 8.33 |
| 7000 | eig. persp. + warm start | $-3.72610 \cdot 10^5$ | 3600.81 | 9.45 |
| 7000 | ours | $-3.72610 \cdot 10^5$ | **1.98** | **0.00** |

Table 37: Comparison of time and optimality gap on the synthetic datasets ($p = 3000$, $k = 10$, $\rho = 0.3$, $\lambda_2 = 0.001$). All baselines use our beam search solution as a warm start.

| # of samples | method | upper bound | time(s) | gap(%) |
|---|---|---|---|---|
| 3000 | SOS1 + warm start | $-3.33604 \cdot 10^5$ | 3600.69 | 19.57 |
| 3000 | big-M50 + warm start | $-3.33604 \cdot 10^5$ | **3600.17** | **18.79** |
| 3000 | persp. + warm start | $-3.33604 \cdot 10^5$ | 3600.06 | 19.57 |
| 3000 | eig. persp. + warm start | $-3.33604 \cdot 10^5$ | 3600.06 | 19.58 |
| 3000 | ours | $-3.33604 \cdot 10^5$ | 3600.58 | 19.57 |
| 4000 | SOS1 + warm start | $-4.53261 \cdot 10^5$ | 3600.07 | 15.24 |
| 4000 | big-M50 + warm start | $-4.53261 \cdot 10^5$ | 3600.17 | 14.98 |
| 4000 | persp. + warm start | $-4.53261 \cdot 10^5$ | 3600.08 | 15.23 |
| 4000 | eig. persp. + warm start | $-4.53261 \cdot 10^5$ | 3600.09 | 15.41 |
| 4000 | ours | $-4.53261 \cdot 10^5$ | **3600.25** | **0.63** |
| 5000 | SOS1 + warm start | $-5.63007 \cdot 10^5$ | 3600.25 | 12.04 |
| 5000 | big-M50 + warm start | $-5.63007 \cdot 10^5$ | 3600.29 | 11.70 |
| 5000 | persp. + warm start | $-5.63007 \cdot 10^5$ | 3600.76 | 11.75 |
| 5000 | eig. persp. + warm start | $-5.63007 \cdot 10^5$ | 3600.39 | 0.37 |
| 5000 | ours | $-5.63007 \cdot 10^5$ | **36.97** | **0.00** |
| 6000 | SOS1 + warm start | $-6.57799 \cdot 10^5$ | 3600.29 | 9.98 |
| 6000 | big-M50 + warm start | $-6.57799 \cdot 10^5$ | 3600.94 | 9.98 |
| 6000 | persp. + warm start | $-6.57799 \cdot 10^5$ | 3601.41 | 9.57 |
| 6000 | eig. persp. + warm start | $-6.57799 \cdot 10^5$ | 1420.54 | 0.00 |
| 6000 | ours | $-6.57799 \cdot 10^5$ | **20.75** | **0.00** |
| 7000 | SOS1 + warm start | $-7.72267 \cdot 10^5$ | 3600.06 | 8.33 |
| 7000 | big-M50 + warm start | $-7.72267 \cdot 10^5$ | 3600.55 | 8.33 |
| 7000 | persp. + warm start | $-7.72267 \cdot 10^5$ | 3600.39 | 7.86 |
| 7000 | eig. persp. + warm start | $-7.72267 \cdot 10^5$ | 3600.81 | 10.37 |
| 7000 | ours | $-7.72267 \cdot 10^5$ | **21.02** | **0.00** |

Table 38: Comparison of time and optimality gap on the synthetic datasets (p = 3000, k = 10, $\rho = 0.5$, $\lambda_2 = 0.001$). All baselines use our beam search solution as a warm start.

| # of samples | method | upper bound | time(s) | gap(%) |
|---|---|---|---|---|
| 3000 | SOS1 + warm start | $-7.41187 \cdot 10^5$ | 3600.08 | 19.60 |
| 3000 | big-M50 + warm start | $-7.41187 \cdot 10^5$ | **3600.76** | **18.15** |
| 3000 | persp. + warm start | $-7.41187 \cdot 10^5$ | 3600.32 | 19.60 |
| 3000 | eig. persp. + warm start | $-7.41187 \cdot 10^5$ | 3600.07 | 19.60 |
| 3000 | ours | $-7.41187 \cdot 10^5$ | 3601.14 | 19.60 |
| 4000 | SOS1 + warm start | $-1.00287 \cdot 10^6$ | 3600.07 | 15.22 |
| 4000 | big-M50 + warm start | $-1.00287 \cdot 10^6$ | 3600.07 | 14.98 |
| 4000 | persp. + warm start | $-1.00287 \cdot 10^6$ | 3600.06 | 15.20 |
| 4000 | eig. persp. + warm start | $-1.00287 \cdot 10^6$ | 3600.07 | 15.61 |
| 4000 | ours | $-1.00287 \cdot 10^6$ | **3600.37** | **1.88** |
| 5000 | SOS1 + warm start | $-1.24027 \cdot 10^6$ | 3600.06 | 12.08 |
| 5000 | big-M50 + warm start | $-1.24027 \cdot 10^6$ | 3600.05 | 11.88 |
| 5000 | persp. + warm start | $-1.24027 \cdot 10^6$ | 3600.07 | 12.07 |
| 5000 | eig. persp. + warm start | $-1.24027 \cdot 10^6$ | 3600.20 | 0.97 |
| 5000 | ours | $-1.24027 \cdot 10^6$ | **466.42** | **0.00** |
| 6000 | SOS1 + warm start | $-1.45417 \cdot 10^6$ | 3601.20 | 10.00 |
| 6000 | big-M50 + warm start | $-1.45417 \cdot 10^6$ | 3600.45 | 9.78 |
| 6000 | persp. + warm start | $-1.45417 \cdot 10^6$ | 3600.04 | 9.82 |
| 6000 | eig. persp. + warm start | $-1.45417 \cdot 10^6$ | 2727.13 | 0.00 |
| 6000 | ours | $-1.45417 \cdot 10^6$ | **57.67** | **0.00** |
| 7000 | SOS1 + warm start | $-1.70483 \cdot 10^6$ | 3600.69 | 8.34 |
| 7000 | big-M50 + warm start | $-1.70483 \cdot 10^6$ | 3601.38 | 8.34 |
| 7000 | persp. + warm start | $-1.70483 \cdot 10^6$ | 3601.09 | 8.13 |
| 7000 | eig. persp. + warm start | $-1.70483 \cdot 10^6$ | 1517.88 | 0.00 |
| 7000 | ours | $-1.70483 \cdot 10^6$ | **21.42** | **0.00** |

Table 39: Comparison of time and optimality gap on the synthetic datasets (p = 3000, k = 10, $\rho = 0.7$, $\lambda_2 = 0.001$). All baselines use our beam search solution as a warm start.

| # of samples | method | upper bound | time(s) | gap(%) |
|---|---|---|---|---|
| 3000 | SOS1 + warm start | $-2.78521 \cdot 10^6$ | 3600.34 | 19.60 |
| 3000 | big-M50 + warm start | $-2.78521 \cdot 10^6$ | **3600.74** | **15.90** |
| 3000 | persp. + warm start | $-2.78521 \cdot 10^6$ | 3600.06 | 19.60 |
| 3000 | eig. persp. + warm start | $-2.78521 \cdot 10^6$ | 3600.44 | 19.61 |
| 3000 | ours | $-2.78521 \cdot 10^6$ | 3600.01 | 19.60 |
| 4000 | SOS1 + warm start | $-3.75203 \cdot 10^6$ | 3600.06 | 15.10 |
| 4000 | big-M50 + warm start | $-3.75203 \cdot 10^6$ | 3600.07 | 13.33 |
| 4000 | persp. + warm start | $-3.75203 \cdot 10^6$ | 3600.06 | 15.12 |
| 4000 | eig. persp. + warm start | $-3.75203 \cdot 10^6$ | 3600.29 | 15.85 |
| 4000 | ours | $-3.75203 \cdot 10^6$ | **3600.42** | **3.50** |
| 5000 | SOS1 + warm start | $-4.61877 \cdot 10^6$ | 3600.07 | 12.11 |
| 5000 | big-M50 + warm start | $-4.61877 \cdot 10^6$ | 3601.36 | 11.37 |
| 5000 | persp. + warm start | $-4.61877 \cdot 10^6$ | 3600.06 | 12.10 |
| 5000 | eig. persp. + warm start | $-4.61877 \cdot 10^6$ | 3604.20 | 12.88 |
| 5000 | ours | $-4.61877 \cdot 10^6$ | **3600.57** | **0.93** |
| 6000 | SOS1 + warm start | $-5.43826 \cdot 10^6$ | 3600.07 | 10.02 |
| 6000 | big-M50 + warm start | $-5.43826 \cdot 10^6$ | 3600.22 | 9.77 |
| 6000 | persp. + warm start | $-5.43826 \cdot 10^6$ | 3601.43 | 10.01 |
| 6000 | eig. persp. + warm start | $-5.43826 \cdot 10^6$ | 3600.95 | 0.85 |
| 6000 | ours | $-5.43826 \cdot 10^6$ | **3600.65** | **0.33** |
| 7000 | SOS1 + warm start | $-6.36767 \cdot 10^6$ | 3600.06 | 8.35 |
| 7000 | big-M50 + warm start | $-6.36767 \cdot 10^6$ | 3600.12 | 8.22 |
| 7000 | persp. + warm start | $-6.36767 \cdot 10^6$ | 3600.06 | 8.34 |
| 7000 | eig. persp. + warm start | $-6.36767 \cdot 10^6$ | 3600.28 | 0.35 |
| 7000 | ours | $-6.36767 \cdot 10^6$ | **3600.78** | **0.11** |

Table 40: Comparison of time and optimality gap on the synthetic datasets (p = 3000, k = 10, $\rho = 0.9$, $\lambda_2 = 0.001$). All baselines use our beam search solution as a warm start.

### G.2.9 Synthetic 2 Benchmark with $\lambda_2 = 0.1$ with No Warmstart

| # of samples | method | upper bound | time(s) | gap(%) |
|---|---|---|---|---|
| 3000 | SOS1 | $-6.34575 \cdot 10^4$ | 3600.08 | 19.36 |
| 3000 | big-M50 | $-6.34575 \cdot 10^4$ | **3600.07** | **19.14** |
| 3000 | persp. | $-4.81335 \cdot 10^4$ | 3600.14 | 57.24 |
| 3000 | eig. persp. | $-4.80471 \cdot 10^4$ | 3600.06 | 57.76 |
| 3000 | ours | $-6.34575 \cdot 10^4$ | 3600.33 | 19.31 |
| 4000 | SOS1 | $-8.64284 \cdot 10^4$ | 3600.20 | 15.37 |
| 4000 | big-M50 | $-8.64284 \cdot 10^4$ | 3600.23 | 15.33 |
| 4000 | persp. | $-8.29378 \cdot 10^4$ | 3600.27 | 20.19 |
| 4000 | eig. persp. | $-8.64284 \cdot 10^4$ | 3600.49 | 1.61 |
| 4000 | ours | $-8.64284 \cdot 10^4$ | **20.81** | **0.00** |
| 5000 | SOS1 | $-1.09575 \cdot 10^5$ | 3600.96 | 11.91 |
| 5000 | big-M50 | $-1.09575 \cdot 10^5$ | 3600.08 | 11.60 |
| 5000 | persp. | $-1.09575 \cdot 10^5$ | 3600.07 | 11.90 |
| 5000 | eig. persp. | $-1.09575 \cdot 10^5$ | 3600.18 | 0.23 |
| 5000 | ours | $-1.09575 \cdot 10^5$ | **20.65** | **0.00** |
| 6000 | SOS1 | $-1.27642 \cdot 10^5$ | 3600.09 | 9.92 |
| 6000 | big-M50 | $-1.27642 \cdot 10^5$ | 3601.02 | 9.92 |
| 6000 | persp. | $-1.27642 \cdot 10^5$ | 3600.93 | 9.92 |
| 6000 | eig. persp. | $-1.27642 \cdot 10^5$ | 3600.74 | 12.65 |
| 6000 | ours | $-1.27642 \cdot 10^5$ | **1.70** | **0.00** |
| 7000 | SOS1 | $-1.50613 \cdot 10^5$ | 3600.12 | 8.32 |
| 7000 | big-M50 | $-1.50613 \cdot 10^5$ | 3600.45 | 8.32 |
| 7000 | persp. | $-1.50613 \cdot 10^5$ | 3600.20 | 8.29 |
| 7000 | eig. persp. | $-1.50613 \cdot 10^5$ | 113.48 | 0.00 |
| 7000 | ours | $-1.50613 \cdot 10^5$ | **1.67** | **0.00** |

Table 41: Comparison of time and optimality gap on the synthetic datasets (p = 3000, k = 10, $\rho = 0.1$, $\lambda_2 = 0.1$).

| # of samples | method | upper bound | time(s) | gap(%) |
|---|---|---|---|---|
| 3000 | SOS1 | $-1.52666 \cdot 10^5$ | 3600.76 | 24.80 |
| 3000 | big-M50 | $-1.59508 \cdot 10^5$ | **3601.50** | **19.07** |
| 3000 | persp. | $-1.28190 \cdot 10^5$ | 3600.48 | 48.57 |
| 3000 | eig. persp. | $-1.34180 \cdot 10^5$ | 3600.42 | 42.09 |
| 3000 | ours | $-1.59508 \cdot 10^5$ | 3600.27 | 19.40 |
| 4000 | SOS1 | $-2.04336 \cdot 10^5$ | 3600.51 | 22.74 |
| 4000 | big-M50 | $-2.17573 \cdot 10^5$ | 3600.23 | 14.78 |
| 4000 | persp. | $-2.17573 \cdot 10^5$ | 3600.79 | 14.88 |
| 4000 | eig. persp. | $-2.09228 \cdot 10^5$ | 3600.20 | 19.95 |
| 4000 | ours | $-2.17573 \cdot 10^5$ | **99.53** | **0.00** |
| 5000 | SOS1 | $-2.66411 \cdot 10^5$ | 3600.59 | 14.34 |
| 5000 | big-M50 | $-2.71985 \cdot 10^5$ | 3600.09 | 11.22 |
| 5000 | persp. | $-2.71985 \cdot 10^5$ | 3600.21 | 11.96 |
| 5000 | eig. persp. | $-2.71985 \cdot 10^5$ | 2019.93 | 0.00 |
| 5000 | ours | $-2.71985 \cdot 10^5$ | **21.01** | **0.00** |
| 6000 | SOS1 | $-3.16760 \cdot 10^5$ | 3600.27 | 9.96 |
| 6000 | big-M50 | $-3.16760 \cdot 10^5$ | 3600.07 | 9.96 |
| 6000 | persp. | $-3.16760 \cdot 10^5$ | 3600.07 | 9.96 |
| 6000 | eig. persp. | $-3.16760 \cdot 10^5$ | 1196.41 | 0.00 |
| 6000 | ours | $-3.16760 \cdot 10^5$ | **20.91** | **0.00** |
| 7000 | SOS1 | $-3.55314 \cdot 10^5$ | 3600.32 | 13.60 |
| 7000 | big-M50 | $-3.72609 \cdot 10^5$ | 3600.80 | 8.33 |
| 7000 | persp. | $-3.72609 \cdot 10^5$ | 3600.06 | 8.31 |
| 7000 | eig. persp. | $-3.72609 \cdot 10^5$ | 3600.07 | 9.46 |
| 7000 | ours | $-3.72609 \cdot 10^5$ | **1.73** | **0.00** |

Table 42: Comparison of time and optimality gap on the synthetic datasets (p = 3000, k = 10, $\rho = 0.3$, $\lambda_2 = 0.1$).

| # of samples | method | upper bound | time(s) | gap(%) |
|---|---|---|---|---|
| 3000 | SOS1 | $-3.23979 \cdot 10^5$ | 3601.13 | 23.04 |
| 3000 | big-M50 | $-3.33603 \cdot 10^5$ | **3601.02** | **18.78** |
| 3000 | persp. | $-2.94042 \cdot 10^5$ | 3600.41 | 35.54 |
| 3000 | eig. persp. | $-3.07151 \cdot 10^5$ | 3600.07 | 29.86 |
| 3000 | ours | $-3.33603 \cdot 10^5$ | 3600.68 | 19.44 |
| 4000 | SOS1 | $-4.35290 \cdot 10^5$ | 3600.08 | 20.00 |
| 4000 | big-M50 | $-4.53260 \cdot 10^5$ | 3600.12 | 14.95 |
| 4000 | persp. | $-4.50184 \cdot 10^5$ | 3600.07 | 15.99 |
| 4000 | eig. persp. | $-4.41599 \cdot 10^5$ | 3600.06 | 18.45 |
| 4000 | ours | $-4.53260 \cdot 10^5$ | **3600.41** | **0.62** |
| 5000 | SOS1 | $-5.63006 \cdot 10^5$ | 3600.09 | 12.04 |
| 5000 | big-M50 | $-5.63006 \cdot 10^5$ | 3600.07 | 11.69 |
| 5000 | persp. | $-5.63006 \cdot 10^5$ | 3600.16 | 11.74 |
| 5000 | eig. persp. | $-5.63006 \cdot 10^5$ | 3600.43 | 0.46 |
| 5000 | ours | $-5.63006 \cdot 10^5$ | **37.26** | **0.00** |
| 6000 | SOS1 | $-6.57798 \cdot 10^5$ | 3600.98 | 9.98 |
| 6000 | big-M50 | $-6.57798 \cdot 10^5$ | 3600.53 | 9.98 |
| 6000 | persp. | $-6.57798 \cdot 10^5$ | 3600.13 | 9.96 |
| 6000 | eig. persp. | $-6.57798 \cdot 10^5$ | 1737.91 | 0.00 |
| 6000 | ours | $-6.57798 \cdot 10^5$ | **20.64** | **0.00** |
| 7000 | SOS1 | $-7.46134 \cdot 10^5$ | 3600.11 | 12.13 |
| 7000 | big-M50 | $-7.72266 \cdot 10^5$ | 3600.96 | 8.33 |
| 7000 | persp. | $-7.72266 \cdot 10^5$ | 3600.23 | 8.31 |
| 7000 | eig. persp. | $-7.72266 \cdot 10^5$ | 3601.08 | 10.36 |
| 7000 | ours | $-7.72266 \cdot 10^5$ | **20.91** | **0.00** |

Table 43: Comparison of time and optimality gap on the synthetic datasets (p = 3000, k = 10, $\rho = 0.5$, $\lambda_2 = 0.1$).

| # of samples | method | upper bound | time(s) | gap(%) |
|---|---|---|---|---|
| 3000 | SOS1 | $-7.36308 \cdot 10^5$ | 3601.07 | 20.30 |
| 3000 | big-M50 | $-7.41186 \cdot 10^5$ | **3600.75** | **18.15** |
| 3000 | persp. | $-7.11307 \cdot 10^5$ | 3600.38 | 24.54 |
| 3000 | eig. persp. | $-7.14103 \cdot 10^5$ | 3600.06 | 24.13 |
| 3000 | ours | $-7.41186 \cdot 10^5$ | 3600.59 | 19.46 |
| 4000 | SOS1 | $-1.00107 \cdot 10^6$ | 3600.08 | 15.42 |
| 4000 | big-M50 | $-1.00287 \cdot 10^6$ | 3600.07 | 14.98 |
| 4000 | persp. | $-9.98563 \cdot 10^5$ | 3600.06 | 15.67 |
| 4000 | eig. persp. | $-9.88486 \cdot 10^5$ | 3600.06 | 17.29 |
| 4000 | ours | $-1.00287 \cdot 10^6$ | **3600.08** | **1.88** |
| 5000 | SOS1 | $-1.23516 \cdot 10^6$ | 3600.08 | 12.54 |
| 5000 | big-M50 | $-1.24027 \cdot 10^6$ | 3600.07 | 11.88 |
| 5000 | persp. | $-1.23565 \cdot 10^6$ | 3600.08 | 12.47 |
| 5000 | eig. persp. | $-1.22304 \cdot 10^6$ | 3600.46 | 14.19 |
| 5000 | ours | $-1.24027 \cdot 10^6$ | **469.43** | **0.00** |
| 6000 | SOS1 | $-1.45416 \cdot 10^6$ | 3600.07 | 10.00 |
| 6000 | big-M50 | $-1.45416 \cdot 10^6$ | 3601.17 | 9.78 |
| 6000 | persp. | $-1.45416 \cdot 10^6$ | 3601.30 | 9.82 |
| 6000 | eig. persp. | $-1.45416 \cdot 10^6$ | 3600.05 | 0.03 |
| 6000 | ours | $-1.45416 \cdot 10^6$ | **59.16** | **0.00** |
| 7000 | SOS1 | $-1.69945 \cdot 10^6$ | 3600.08 | 8.68 |
| 7000 | big-M50 | $-1.70483 \cdot 10^6$ | 3602.06 | 8.34 |
| 7000 | persp. | $-1.70483 \cdot 10^6$ | 3600.92 | 8.13 |
| 7000 | eig. persp. | $-1.70483 \cdot 10^6$ | 1537.97 | 0.00 |
| 7000 | ours | $-1.70483 \cdot 10^6$ | **21.12** | **0.00** |

Table 44: Comparison of time and optimality gap on the synthetic datasets ($p = 3000$, $k = 10$, $\rho = 0.7$, $\lambda_2 = 0.1$).

| # of samples | method | upper bound | time(s) | gap(%) |
|---|---|---|---|---|
| 3000 | SOS1 | $-2.77826 \cdot 10^6$ | 3600.41 | 19.82 |
| 3000 | big-M50 | $-2.78468 \cdot 10^6$ | **3600.31** | **15.93** |
| 3000 | persp. | $-2.76199 \cdot 10^6$ | 3600.06 | 20.50 |
| 3000 | eig. persp. | $-2.76520 \cdot 10^6$ | 3600.06 | 20.47 |
| 3000 | ours | $-2.78521 \cdot 10^6$ | 3600.23 | 19.46 |
| 4000 | SOS1 | $-3.74434 \cdot 10^6$ | 3600.35 | 15.33 |
| 4000 | big-M50 | $-3.75203 \cdot 10^6$ | 3600.08 | 13.35 |
| 4000 | persp. | $-3.73180 \cdot 10^6$ | 3600.06 | 15.73 |
| 4000 | eig. persp. | $-3.74150 \cdot 10^6$ | 3600.69 | 16.17 |
| 4000 | ours | $-3.75203 \cdot 10^6$ | **3601.10** | **3.50** |
| 5000 | SOS1 | $-4.60905 \cdot 10^6$ | 3600.07 | 12.34 |
| 5000 | big-M50 | $-4.61877 \cdot 10^6$ | 3600.08 | 11.40 |
| 5000 | persp. | $-4.54788 \cdot 10^6$ | 3600.79 | 13.84 |
| 5000 | eig. persp. | $-4.59650 \cdot 10^6$ | 3600.06 | 13.42 |
| 5000 | ours | $-4.61877 \cdot 10^6$ | **3600.40** | **0.93** |
| 6000 | SOS1 | $-5.40939 \cdot 10^6$ | 3600.07 | 10.60 |
| 6000 | big-M50 | $-5.43826 \cdot 10^6$ | 3600.23 | 9.79 |
| 6000 | persp. | $-5.42336 \cdot 10^6$ | 3600.65 | 10.30 |
| 6000 | eig. persp. | $-5.43107 \cdot 10^6$ | 3600.99 | 11.20 |
| 6000 | ours | $-5.43826 \cdot 10^6$ | **3600.26** | **0.33** |
| 7000 | SOS1 | $-6.33762 \cdot 10^6$ | 3600.07 | 8.86 |
| 7000 | big-M50 | $-6.36767 \cdot 10^6$ | 3600.07 | 8.22 |
| 7000 | persp. | $-6.26888 \cdot 10^6$ | 3600.06 | 10.04 |
| 7000 | eig. persp. | $-6.34367 \cdot 10^6$ | 3600.05 | 10.10 |
| 7000 | ours | $-6.36767 \cdot 10^6$ | **3600.58** | **0.11** |

Table 45: Comparison of time and optimality gap on the synthetic datasets ($p = 3000$, $k = 10$, $\rho = 0.9$, $\lambda_2 = 0.1$).

### G.2.10  Synthetic 2 Benchmark with $\lambda_2 = 0.1$ with Warmstart

| # of samples | method | upper bound | time(s) | gap(%) |
|---|---|---|---|---|
| 3000 | SOS1 + warm start | $-6.34575 \cdot 10^4$ | 3600.07 | 19.36 |
| 3000 | big-M50 + warm start | $-6.34575 \cdot 10^4$ | **3600.12** | **19.14** |
| 3000 | persp. + warm start | $-6.34575 \cdot 10^4$ | 3600.14 | 19.27 |
| 3000 | eig. persp. + warm start | $-6.34575 \cdot 10^4$ | 3600.13 | 19.45 |
| 3000 | ours | $-6.34575 \cdot 10^4$ | 3600.33 | 19.31 |
| 4000 | SOS1 + warm start | $-8.64284 \cdot 10^4$ | 3600.36 | 15.36 |
| 4000 | big-M50 + warm start | $-8.64284 \cdot 10^4$ | 3600.05 | 15.33 |
| 4000 | persp. + warm start | $-8.64284 \cdot 10^4$ | 3600.52 | 14.30 |
| 4000 | eig. persp. + warm start | $-8.64284 \cdot 10^4$ | 3600.17 | 1.64 |
| 4000 | ours | $-8.64284 \cdot 10^4$ | **20.81** | **0.00** |
| 5000 | SOS1 + warm start | $-1.09575 \cdot 10^5$ | 3600.99 | 11.91 |
| 5000 | big-M50 + warm start | $-1.09575 \cdot 10^5$ | 3601.88 | 11.60 |
| 5000 | persp. + warm start | $-1.09575 \cdot 10^5$ | 3600.14 | 11.90 |
| 5000 | eig. persp. + warm start | $-1.09575 \cdot 10^5$ | 2971.48 | 0.00 |
| 5000 | ours | $-1.09575 \cdot 10^5$ | **20.65** | **0.00** |
| 6000 | SOS1 + warm start | $-1.27642 \cdot 10^5$ | 3600.07 | 9.92 |
| 6000 | big-M50 + warm start | $-1.27642 \cdot 10^5$ | 3600.62 | 9.93 |
| 6000 | persp. + warm start | $-1.27642 \cdot 10^5$ | 3600.06 | 9.92 |
| 6000 | eig. persp. + warm start | $-1.27642 \cdot 10^5$ | 3601.19 | 12.65 |
| 6000 | ours | $-1.27642 \cdot 10^5$ | **1.70** | **0.00** |
| 7000 | SOS1 + warm start | $-1.50613 \cdot 10^5$ | 3600.13 | 5.84 |
| 7000 | big-M50 + warm start | $-1.50613 \cdot 10^5$ | 3600.07 | 8.32 |
| 7000 | persp. + warm start | $-1.50613 \cdot 10^5$ | 3600.19 | 8.29 |
| 7000 | eig. persp. + warm start | $-1.50613 \cdot 10^5$ | 103.79 | 0.00 |
| 7000 | ours | $-1.50613 \cdot 10^5$ | **1.67** | **0.00** |

Table 46: Comparison of time and optimality gap on the synthetic datasets (p = 3000, k = 10, $\rho = 0.1$, $\lambda_2 = 0.1$). All baselines use our beam search solution as a warm start.

| # of samples | method | upper bound | time(s) | gap(%) |
|---|---|---|---|---|
| 3000 | SOS1 + warm start | $-1.59508 \cdot 10^5$ | 3600.06 | 19.45 |
| 3000 | big-M50 + warm start | $-1.59508 \cdot 10^5$ | **3600.22** | **19.07** |
| 3000 | persp. + warm start | $-1.59508 \cdot 10^5$ | 3600.06 | 19.40 |
| 3000 | eig. persp. + warm start | $-1.59508 \cdot 10^5$ | 3600.14 | 19.52 |
| 3000 | ours | $-1.59508 \cdot 10^5$ | 3600.27 | 19.40 |
| 4000 | SOS1 + warm start | $-2.17573 \cdot 10^5$ | 3600.40 | 15.28 |
| 4000 | big-M50 + warm start | $-2.17573 \cdot 10^5$ | 3600.10 | 14.78 |
| 4000 | persp. + warm start | $-2.17573 \cdot 10^5$ | 3600.06 | 14.88 |
| 4000 | eig. persp. + warm start | $-2.17573 \cdot 10^5$ | 3600.10 | 15.34 |
| 4000 | ours | $-2.17573 \cdot 10^5$ | **99.53** | **0.00** |
| 5000 | SOS1 + warm start | $-2.71985 \cdot 10^5$ | 3601.08 | 12.00 |
| 5000 | big-M50 + warm start | $-2.71985 \cdot 10^5$ | 3600.07 | 11.22 |
| 5000 | persp. + warm start | $-2.71985 \cdot 10^5$ | 3600.06 | 11.34 |
| 5000 | eig. persp. + warm start | $-2.71985 \cdot 10^5$ | 1642.59 | 0.00 |
| 5000 | ours | $-2.71985 \cdot 10^5$ | **21.01** | **0.00** |
| 6000 | SOS1 + warm start | $-3.16760 \cdot 10^5$ | 3600.83 | 9.96 |
| 6000 | big-M50 + warm start | $-3.16760 \cdot 10^5$ | 3600.07 | 9.96 |
| 6000 | persp. + warm start | $-3.16760 \cdot 10^5$ | 3600.05 | 9.96 |
| 6000 | eig. persp. + warm start | $-3.16760 \cdot 10^5$ | 3600.06 | 11.77 |
| 6000 | ours | $-3.16760 \cdot 10^5$ | **20.91** | **0.00** |
| 7000 | SOS1 + warm start | $-3.72609 \cdot 10^5$ | 3600.67 | 8.33 |
| 7000 | big-M50 + warm start | $-3.72609 \cdot 10^5$ | 3601.58 | 8.33 |
| 7000 | persp. + warm start | $-3.72609 \cdot 10^5$ | 3601.01 | 8.31 |
| 7000 | eig. persp. + warm start | $-3.72609 \cdot 10^5$ | 3600.55 | 9.45 |
| 7000 | ours | $-3.72609 \cdot 10^5$ | **1.73** | **0.00** |

Table 47: Comparison of time and optimality gap on the synthetic datasets (p = 3000, k = 10, $\rho = 0.3$, $\lambda_2 = 0.1$). All baselines use our beam search solution as a warm start.

| # of samples | method | upper bound | time(s) | gap(%) |
|---|---|---|---|---|
| 3000 | SOS1 + warm start | $-3.33603 \cdot 10^5$ | 3600.06 | 19.49 |
| 3000 | big-M50 + warm start | $-3.33603 \cdot 10^5$ | **3601.53** | **18.78** |
| 3000 | persp. + warm start | $-3.33603 \cdot 10^5$ | 3600.07 | 19.47 |
| 3000 | eig. persp. + warm start | $-3.33603 \cdot 10^5$ | 3600.06 | 19.56 |
| 3000 | ours | $-3.33603 \cdot 10^5$ | 3600.68 | 19.44 |
| 4000 | SOS1 + warm start | $-4.53260 \cdot 10^5$ | 3600.06 | 15.24 |
| 4000 | big-M50 + warm start | $-4.53260 \cdot 10^5$ | 3600.07 | 15.04 |
| 4000 | persp. + warm start | $-4.53260 \cdot 10^5$ | 3600.07 | 15.20 |
| 4000 | eig. persp. + warm start | $-4.53260 \cdot 10^5$ | 3601.61 | 15.40 |
| 4000 | ours | $-4.53260 \cdot 10^5$ | **3600.41** | **0.62** |
| 5000 | SOS1 + warm start | $-5.63006 \cdot 10^5$ | 3600.47 | 12.04 |
| 5000 | big-M50 + warm start | $-5.63006 \cdot 10^5$ | 3601.70 | 11.69 |
| 5000 | persp. + warm start | $-5.63006 \cdot 10^5$ | 3600.06 | 11.74 |
| 5000 | eig. persp. + warm start | $-5.63006 \cdot 10^5$ | 3600.50 | 0.47 |
| 5000 | ours | $-5.63006 \cdot 10^5$ | **37.26** | **0.00** |
| 6000 | SOS1 + warm start | $-6.57798 \cdot 10^5$ | 3601.18 | 9.98 |
| 6000 | big-M50 + warm start | $-6.57798 \cdot 10^5$ | 3600.18 | 9.98 |
| 6000 | persp. + warm start | $-6.57798 \cdot 10^5$ | 3600.06 | 9.56 |
| 6000 | eig. persp. + warm start | $-6.57798 \cdot 10^5$ | 1376.14 | 0.00 |
| 6000 | ours | $-6.57798 \cdot 10^5$ | **20.64** | **0.00** |
| 7000 | SOS1 + warm start | $-7.72266 \cdot 10^5$ | 3600.10 | 8.33 |
| 7000 | big-M50 + warm start | $-7.72266 \cdot 10^5$ | 3600.63 | 8.33 |
| 7000 | persp. + warm start | $-7.72266 \cdot 10^5$ | 3600.34 | 7.86 |
| 7000 | eig. persp. + warm start | $-7.72266 \cdot 10^5$ | 3600.44 | 10.37 |
| 7000 | ours | $-7.72266 \cdot 10^5$ | **20.91** | **0.00** |

Table 48: Comparison of time and optimality gap on the synthetic datasets (p = 3000, k = 10, $\rho = 0.5$, $\lambda_2 = 0.1$). All baselines use our beam search solution as a warm start.

| # of samples | method | upper bound | time(s) | gap(%) |
|---|---|---|---|---|
| 3000 | SOS1 + warm start | $-7.41186 \cdot 10^5$ | 3600.06 | 19.51 |
| 3000 | big-M50 + warm start | $-7.41186 \cdot 10^5$ | **3601.56** | **18.15** |
| 3000 | persp. + warm start | $-7.41186 \cdot 10^5$ | 3600.87 | 19.52 |
| 3000 | eig. persp. + warm start | $-7.41186 \cdot 10^5$ | 3600.05 | 19.59 |
| 3000 | ours | $-7.41186 \cdot 10^5$ | 3600.59 | 19.46 |
| 4000 | SOS1 + warm start | $-1.00287 \cdot 10^6$ | 3600.07 | 15.21 |
| 4000 | big-M50 + warm start | $-1.00287 \cdot 10^6$ | 3600.09 | 14.98 |
| 4000 | persp. + warm start | $-1.00287 \cdot 10^6$ | 3600.07 | 15.18 |
| 4000 | eig. persp. + warm start | $-1.00287 \cdot 10^6$ | 3600.06 | 15.61 |
| 4000 | ours | $-1.00287 \cdot 10^6$ | **3600.08** | **1.88** |
| 5000 | SOS1 + warm start | $-1.24027 \cdot 10^6$ | 3601.15 | 12.08 |
| 5000 | big-M50 + warm start | $-1.24027 \cdot 10^6$ | 3600.40 | 11.88 |
| 5000 | persp. + warm start | $-1.24027 \cdot 10^6$ | 3601.42 | 12.05 |
| 5000 | eig. persp. + warm start | $-1.24027 \cdot 10^6$ | 3600.19 | 0.97 |
| 5000 | ours | $-1.24027 \cdot 10^6$ | **469.43** | **0.00** |
| 6000 | SOS1 + warm start | $-1.45416 \cdot 10^6$ | 3600.09 | 10.00 |
| 6000 | big-M50 + warm start | $-1.45416 \cdot 10^6$ | 3600.05 | 9.78 |
| 6000 | persp. + warm start | $-1.45416 \cdot 10^6$ | 3600.10 | 9.82 |
| 6000 | eig. persp. + warm start | $-1.45416 \cdot 10^6$ | 3600.16 | 0.22 |
| 6000 | ours | $-1.45416 \cdot 10^6$ | **59.16** | **0.00** |
| 7000 | SOS1 + warm start | $-1.70483 \cdot 10^6$ | 3600.74 | 8.34 |
| 7000 | big-M50 + warm start | $-1.70483 \cdot 10^6$ | 3600.07 | 8.34 |
| 7000 | persp. + warm start | $-1.70483 \cdot 10^6$ | 3600.06 | 8.13 |
| 7000 | eig. persp. + warm start | $-1.70483 \cdot 10^6$ | 1417.14 | 0.00 |
| 7000 | ours | $-1.70483 \cdot 10^6$ | **21.12** | **0.00** |

Table 49: Comparison of time and optimality gap on the synthetic datasets (p = 3000, k = 10, $\rho = 0.7$, $\lambda_2 = 0.1$). All baselines use our beam search solution as a warm start.

| # of samples | method | upper bound | time(s) | gap(%) |
|---|---|---|---|---|
| 3000 | SOS1 + warm start | $-2.78521 \cdot 10^6$ | 3600.07 | 19.52 |
| 3000 | big-M50 + warm start | $-2.78521 \cdot 10^6$ | **3600.07** | **15.91** |
| 3000 | persp. + warm start | $-2.78521 \cdot 10^6$ | 3600.06 | 19.50 |
| 3000 | eig. persp. + warm start | $-2.78521 \cdot 10^6$ | 3600.06 | 19.60 |
| 3000 | ours | $-2.78521 \cdot 10^6$ | 3600.23 | 19.46 |
| 4000 | SOS1 + warm start | $-3.75203 \cdot 10^6$ | 3600.07 | 15.10 |
| 4000 | big-M50 + warm start | $-3.75203 \cdot 10^6$ | 3600.27 | 13.35 |
| 4000 | persp. + warm start | $-3.75203 \cdot 10^6$ | 3600.51 | 15.10 |
| 4000 | eig. persp. + warm start | $-3.75203 \cdot 10^6$ | 3600.07 | 15.85 |
| 4000 | ours | $-3.75203 \cdot 10^6$ | **3601.10** | **3.50** |
| 5000 | SOS1 + warm start | $-4.61877 \cdot 10^6$ | 3600.07 | 12.11 |
| 5000 | big-M50 + warm start | $-4.61877 \cdot 10^6$ | 3600.12 | 11.40 |
| 5000 | persp. + warm start | $-4.61877 \cdot 10^6$ | 3600.06 | 12.09 |
| 5000 | eig. persp. + warm start | $-4.61877 \cdot 10^6$ | 3600.35 | 12.88 |
| 5000 | ours | $-4.61877 \cdot 10^6$ | **3600.40** | **0.93** |
| 6000 | SOS1 + warm start | $-5.43826 \cdot 10^6$ | 3600.37 | 10.02 |
| 6000 | big-M50 + warm start | $-5.43826 \cdot 10^6$ | 3600.07 | 9.79 |
| 6000 | persp. + warm start | $-5.43826 \cdot 10^6$ | 3600.07 | 10.00 |
| 6000 | eig. persp. + warm start | $-5.43826 \cdot 10^6$ | 3600.08 | 11.05 |
| 6000 | ours | $-5.43826 \cdot 10^6$ | **3600.26** | **0.33** |
| 7000 | SOS1 + warm start | $-6.36767 \cdot 10^6$ | 3600.50 | 8.35 |
| 7000 | big-M50 + warm start | $-6.36767 \cdot 10^6$ | 3600.07 | 8.22 |
| 7000 | persp. + warm start | $-6.36767 \cdot 10^6$ | 3600.06 | 8.33 |
| 7000 | eig. persp. + warm start | $-6.36767 \cdot 10^6$ | 3600.05 | 0.38 |
| 7000 | ours | $-6.36767 \cdot 10^6$ | **3600.58** | **0.11** |

Table 50: Comparison of time and optimality gap on the synthetic datasets (p = 3000, k = 10, $\rho = 0.9$, $\lambda_2 = 0.1$). All baselines use our beam search solution as a warm start.

### G.2.11 Synthetic 2 Benchmark with $\lambda_2 = 10.0$ with No Warmstart

| # of samples | method | upper bound | time(s) | gap(%) |
|---|---|---|---|---|
| 3000 | SOS1 | $-6.33576 \cdot 10^4$ | 3600.08 | 18.27 |
| 3000 | big-M50 | $-6.33576 \cdot 10^4$ | 3600.31 | 18.47 |
| 3000 | persp. | $-6.33576 \cdot 10^4$ | 3600.20 | 17.53 |
| 3000 | eig. persp. | $-4.46130 \cdot 10^4$ | 3601.33 | 69.85 |
| 3000 | ours | $-6.33576 \cdot 10^4$ | **3601.41** | **4.54** |
| 4000 | SOS1 | $-8.63287 \cdot 10^4$ | 3600.28 | 15.23 |
| 4000 | big-M50 | $-8.63287 \cdot 10^4$ | 3600.85 | 15.21 |
| 4000 | persp. | $-8.63287 \cdot 10^4$ | 3600.15 | 12.69 |
| 4000 | eig. persp. | $-8.58997 \cdot 10^4$ | 3600.10 | 1.74 |
| 4000 | ours | $-8.63287 \cdot 10^4$ | **20.33** | **0.00** |
| 5000 | SOS1 | $-1.09474 \cdot 10^5$ | 3600.08 | 11.86 |
| 5000 | big-M50 | $-1.09474 \cdot 10^5$ | 3600.28 | 11.85 |
| 5000 | persp. | $-1.09474 \cdot 10^5$ | 3600.80 | 10.44 |
| 5000 | eig. persp. | $-1.09474 \cdot 10^5$ | 3600.06 | 13.55 |
| 5000 | ours | $-1.09474 \cdot 10^5$ | **20.71** | **0.00** |
| 6000 | SOS1 | $-1.27542 \cdot 10^5$ | 3600.19 | 9.90 |
| 6000 | big-M50 | $-1.27542 \cdot 10^5$ | 3600.37 | 9.90 |
| 6000 | persp. | $-1.27542 \cdot 10^5$ | 3600.14 | 9.03 |
| 6000 | eig. persp. | $-1.27542 \cdot 10^5$ | 2609.62 | 0.00 |
| 6000 | ours | $-1.27542 \cdot 10^5$ | **1.69** | **0.00** |
| 7000 | SOS1 | $-1.42695 \cdot 10^5$ | 3600.12 | 14.24 |
| 7000 | big-M50 | $-1.50512 \cdot 10^5$ | 3600.56 | 8.31 |
| 7000 | persp. | $-1.50512 \cdot 10^5$ | 3600.15 | 7.73 |
| 7000 | eig. persp. | $-1.50512 \cdot 10^5$ | 107.90 | 0.00 |
| 7000 | ours | $-1.50512 \cdot 10^5$ | **1.70** | **0.00** |

Table 51: Comparison of time and optimality gap on the synthetic datasets (p = 3000, k = 10, $\rho = 0.1$, $\lambda_2 = 10.0$).

| # of samples | method | upper bound | time(s) | gap(%) |
|---|---|---|---|---|
| 3000 | SOS1 | $-1.52806 \cdot 10^5$ | 3601.20 | 23.64 |
| 3000 | big-M50 | $-1.59409 \cdot 10^5$ | 3600.72 | 18.52 |
| 3000 | persp. | $-1.59409 \cdot 10^5$ | 3600.14 | 17.84 |
| 3000 | eig. persp. | $-1.25897 \cdot 10^5$ | 3600.88 | 51.38 |
| 3000 | ours | $-1.59409 \cdot 10^5$ | **3600.62** | **4.68** |
| 4000 | SOS1 | $-2.17472 \cdot 10^5$ | 3600.70 | 15.13 |
| 4000 | big-M50 | $-2.17472 \cdot 10^5$ | 3600.07 | 14.64 |
| 4000 | persp. | $-2.17472 \cdot 10^5$ | 3600.05 | 12.93 |
| 4000 | eig. persp. | $-2.03767 \cdot 10^5$ | 3600.20 | 23.07 |
| 4000 | ours | $-2.17472 \cdot 10^5$ | **86.90** | **0.00** |
| 5000 | SOS1 | $-2.71885 \cdot 10^5$ | 3600.32 | 11.94 |
| 5000 | big-M50 | $-2.71885 \cdot 10^5$ | 3603.74 | 11.16 |
| 5000 | persp. | $-2.71885 \cdot 10^5$ | 3600.05 | 10.84 |
| 5000 | eig. persp. | $-2.71885 \cdot 10^5$ | 1969.80 | 0.00 |
| 5000 | ours | $-2.71885 \cdot 10^5$ | **20.75** | **0.00** |
| 6000 | SOS1 | $-3.10076 \cdot 10^5$ | 3600.08 | 12.27 |
| 6000 | big-M50 | $-3.16660 \cdot 10^5$ | 3600.07 | 9.93 |
| 6000 | persp. | $-3.16660 \cdot 10^5$ | 3600.06 | 9.24 |
| 6000 | eig. persp. | $-3.16660 \cdot 10^5$ | 2020.73 | 0.00 |
| 6000 | ours | $-3.16660 \cdot 10^5$ | **21.05** | **0.00** |
| 7000 | SOS1 | $-3.72509 \cdot 10^5$ | 3600.17 | 8.31 |
| 7000 | big-M50 | $-3.72509 \cdot 10^5$ | 3601.17 | 8.31 |
| 7000 | persp. | $-3.72509 \cdot 10^5$ | 3600.13 | 7.78 |
| 7000 | eig. persp. | $-3.72509 \cdot 10^5$ | 2139.72 | 0.00 |
| 7000 | ours | $-3.72509 \cdot 10^5$ | **1.71** | **0.00** |

Table 52: Comparison of time and optimality gap on the synthetic datasets (p = 3000, k = 10, $\rho = 0.3$, $\lambda_2 = 10.0$).

| # of samples | method | upper bound | time(s) | gap(%) |
|---|---|---|---|---|
| 3000 | SOS1 | $-3.31006 \cdot 10^5$ | 3600.07 | 19.45 |
| 3000 | big-M50 | $-3.33503 \cdot 10^5$ | 3600.07 | 18.43 |
| 3000 | persp. | $-3.30991 \cdot 10^5$ | 3600.07 | 18.99 |
| 3000 | eig. persp. | $-2.97162 \cdot 10^5$ | 3600.08 | 34.18 |
| 3000 | ours | $-3.33503 \cdot 10^5$ | **3600.25** | **6.21** |
| 4000 | SOS1 | $-4.36975 \cdot 10^5$ | 3600.35 | 19.35 |
| 4000 | big-M50 | $-4.53159 \cdot 10^5$ | 3600.16 | 14.84 |
| 4000 | persp. | $-4.50480 \cdot 10^5$ | 3600.38 | 12.96 |
| 4000 | eig. persp. | $-4.38770 \cdot 10^5$ | 3600.11 | 19.13 |
| 4000 | ours | $-4.53159 \cdot 10^5$ | **3600.95** | **0.35** |
| 5000 | SOS1 | $-5.37846 \cdot 10^5$ | 3600.98 | 17.20 |
| 5000 | big-M50 | $-5.62906 \cdot 10^5$ | 3600.16 | 11.59 |
| 5000 | persp. | $-5.62906 \cdot 10^5$ | 3600.07 | 10.92 |
| 5000 | eig. persp. | $-5.62906 \cdot 10^5$ | 3600.19 | 12.71 |
| 5000 | ours | $-5.62906 \cdot 10^5$ | **31.47** | **0.00** |
| 6000 | SOS1 | $-6.51605 \cdot 10^5$ | 3600.45 | 10.98 |
| 6000 | big-M50 | $-6.57698 \cdot 10^5$ | 3600.16 | 9.95 |
| 6000 | persp. | $-6.57698 \cdot 10^5$ | 3601.04 | 9.39 |
| 6000 | eig. persp. | $-6.57698 \cdot 10^5$ | 2550.81 | 0.00 |
| 6000 | ours | $-6.57698 \cdot 10^5$ | **21.24** | **0.00** |
| 7000 | SOS1 | $-7.65939 \cdot 10^5$ | 3600.07 | 9.20 |
| 7000 | big-M50 | $-7.72166 \cdot 10^5$ | 3600.12 | 8.31 |
| 7000 | persp. | $-7.72166 \cdot 10^5$ | 3600.13 | 7.90 |
| 7000 | eig. persp. | $-7.72166 \cdot 10^5$ | 3600.06 | 10.13 |
| 7000 | ours | $-7.72166 \cdot 10^5$ | **20.37** | **0.00** |

Table 53: Comparison of time and optimality gap on the synthetic datasets (p = 3000, k = 10, $\rho = 0.5$, $\lambda_2 = 10.0$).

| # of samples | method | upper bound | time(s) | gap(%) |
|---|---|---|---|---|
| 3000 | SOS1 | $-7.35981 \cdot 10^5$ | 3600.07 | 19.40 |
| 3000 | big-M50 | $-7.41086 \cdot 10^5$ | 3600.65 | 17.88 |
| 3000 | persp. | $-7.25990 \cdot 10^5$ | 3600.38 | 20.81 |
| 3000 | eig. persp. | $-7.18339 \cdot 10^5$ | 3600.06 | 23.35 |
| 3000 | ours | $-7.41086 \cdot 10^5$ | **3600.48** | **8.32** |
| 4000 | SOS1 | $-9.99525 \cdot 10^5$ | 3600.07 | 15.44 |
| 4000 | big-M50 | $-1.00277 \cdot 10^6$ | 3600.09 | 14.85 |
| 4000 | persp. | $-9.91636 \cdot 10^5$ | 3601.08 | 14.29 |
| 4000 | eig. persp. | $-9.92065 \cdot 10^5$ | 3600.10 | 16.78 |
| 4000 | ours | $-1.00277 \cdot 10^6$ | **3600.62** | **1.66** |
| 5000 | SOS1 | $-1.23548 \cdot 10^6$ | 3600.08 | 12.44 |
| 5000 | big-M50 | $-1.24017 \cdot 10^6$ | 3600.25 | 11.82 |
| 5000 | persp. | $-1.23555 \cdot 10^6$ | 3600.42 | 11.18 |
| 5000 | eig. persp. | $-1.21712 \cdot 10^6$ | 3600.05 | 14.73 |
| 5000 | ours | $-1.24017 \cdot 10^6$ | **386.72** | **0.00** |
| 6000 | SOS1 | $-1.42629 \cdot 10^6$ | 3600.81 | 12.11 |
| 6000 | big-M50 | $-1.45406 \cdot 10^6$ | 3600.07 | 9.75 |
| 6000 | persp. | $-1.45406 \cdot 10^6$ | 3600.07 | 9.44 |
| 6000 | eig. persp. | $-1.45406 \cdot 10^6$ | 2941.57 | 0.00 |
| 6000 | ours | $-1.45406 \cdot 10^6$ | **54.13** | **0.00** |
| 7000 | SOS1 | $-1.69139 \cdot 10^6$ | 3600.49 | 9.17 |
| 7000 | big-M50 | $-1.70473 \cdot 10^6$ | 3600.81 | 8.32 |
| 7000 | persp. | $-1.70473 \cdot 10^6$ | 3600.06 | 7.85 |
| 7000 | eig. persp. | $-1.70473 \cdot 10^6$ | 1585.19 | 0.00 |
| 7000 | ours | $-1.70473 \cdot 10^6$ | **21.30** | **0.00** |

Table 54: Comparison of time and optimality gap on the synthetic datasets (p = 3000, k = 10, $\rho = 0.7$, $\lambda_2 = 10.0$).

| # of samples | method | upper bound | time(s) | gap(%) |
|---|---|---|---|---|
| 3000 | SOS1 | $-2.78026 \cdot 10^6$ | 3600.08 | 18.78 |
| 3000 | big-M50 | $-2.78507 \cdot 10^6$ | 3600.17 | 15.95 |
| 3000 | persp. | $-2.75178 \cdot 10^6$ | 3600.06 | 20.10 |
| 3000 | eig. persp. | $-2.73809 \cdot 10^6$ | 3600.11 | 21.60 |
| 3000 | ours | $-2.78507 \cdot 10^6$ | **3600.78** | **9.79** |
| 4000 | SOS1 | $-3.74696 \cdot 10^6$ | 3600.08 | 15.12 |
| 4000 | big-M50 | $-3.75192 \cdot 10^6$ | 3600.06 | 13.19 |
| 4000 | persp. | $-3.73412 \cdot 10^6$ | 3600.07 | 14.42 |
| 4000 | eig. persp. | $-3.73458 \cdot 10^6$ | 3600.81 | 16.31 |
| 4000 | ours | $-3.75192 \cdot 10^6$ | **3600.32** | **3.23** |
| 5000 | SOS1 | $-4.60115 \cdot 10^6$ | 3600.29 | 12.47 |
| 5000 | big-M50 | $-4.61867 \cdot 10^6$ | 3600.17 | 11.32 |
| 5000 | persp. | $-4.60427 \cdot 10^6$ | 3600.07 | 11.73 |
| 5000 | eig. persp. | $-4.58981 \cdot 10^6$ | 3600.07 | 13.56 |
| 5000 | ours | $-4.61867 \cdot 10^6$ | **3600.30** | **0.89** |
| 6000 | SOS1 | $-5.42828 \cdot 10^6$ | 3600.08 | 10.18 |
| 6000 | big-M50 | $-5.43816 \cdot 10^6$ | 3600.12 | 9.75 |
| 6000 | persp. | $-5.41942 \cdot 10^6$ | 3600.07 | 9.90 |
| 6000 | eig. persp. | $-5.41142 \cdot 10^6$ | 3600.42 | 11.59 |
| 6000 | ours | $-5.43816 \cdot 10^6$ | **3600.22** | **0.32** |
| 7000 | SOS1 | $-6.33201 \cdot 10^6$ | 3600.05 | 8.93 |
| 7000 | big-M50 | $-6.36757 \cdot 10^6$ | 3600.32 | 8.20 |
| 7000 | persp. | $-6.35270 \cdot 10^6$ | 3600.06 | 8.23 |
| 7000 | eig. persp. | $-6.36342 \cdot 10^6$ | 3600.05 | 0.46 |
| 7000 | ours | $-6.36757 \cdot 10^6$ | **3600.06** | **0.10** |

Table 55: Comparison of time and optimality gap on the synthetic datasets (p = 3000, k = 10, $\rho = 0.9$, $\lambda_2 = 10.0$).

### G.2.12 Synthetic 2 Benchmark with $\lambda_2 = 10.0$ with Warmstart

| # of samples | method | upper bound | time(s) | gap(%) |
|---|---|---|---|---|
| 3000 | SOS1 + warm start | $-6.33576 \cdot 10^4$ | 3601.40 | 18.27 |
| 3000 | big-M50 + warm start | $-6.33576 \cdot 10^4$ | 3600.12 | 18.47 |
| 3000 | persp. + warm start | $-6.33576 \cdot 10^4$ | 3601.92 | 17.53 |
| 3000 | eig. persp. + warm start | $-6.33576 \cdot 10^4$ | 3600.22 | 19.61 |
| 3000 | ours | $-6.33576 \cdot 10^4$ | **3601.41** | **4.54** |
| 4000 | SOS1 + warm start | $-8.63287 \cdot 10^4$ | 3600.08 | 15.23 |
| 4000 | big-M50 + warm start | $-8.63287 \cdot 10^4$ | 3600.29 | 15.21 |
| 4000 | persp. + warm start | $-8.63287 \cdot 10^4$ | 3600.30 | 12.69 |
| 4000 | eig. persp. + warm start | $-8.63287 \cdot 10^4$ | 3600.38 | 1.32 |
| 4000 | ours | $-8.63287 \cdot 10^4$ | **20.33** | **0.00** |
| 5000 | SOS1 + warm start | $-1.09474 \cdot 10^5$ | 3600.06 | 11.86 |
| 5000 | big-M50 + warm start | $-1.09474 \cdot 10^5$ | 3600.15 | 11.85 |
| 5000 | persp. + warm start | $-1.09474 \cdot 10^5$ | 3601.20 | 10.44 |
| 5000 | eig. persp. + warm start | $-1.09474 \cdot 10^5$ | 3601.16 | 13.51 |
| 5000 | ours | $-1.09474 \cdot 10^5$ | **20.71** | **0.00** |
| 6000 | SOS1 + warm start | $-1.27542 \cdot 10^5$ | 3600.07 | 9.90 |
| 6000 | big-M50 + warm start | $-1.27542 \cdot 10^5$ | 3601.39 | 9.89 |
| 6000 | persp. + warm start | $-1.27542 \cdot 10^5$ | 3600.21 | 9.03 |
| 6000 | eig. persp. + warm start | $-1.27542 \cdot 10^5$ | 3601.11 | 12.70 |
| 6000 | ours | $-1.27542 \cdot 10^5$ | **1.69** | **0.00** |
| 7000 | SOS1 + warm start | $-1.50512 \cdot 10^5$ | 3600.28 | 5.83 |
| 7000 | big-M50 + warm start | $-1.50512 \cdot 10^5$ | 3601.02 | 8.31 |
| 7000 | persp. + warm start | $-1.50512 \cdot 10^5$ | 3600.16 | 7.73 |
| 7000 | eig. persp. + warm start | $-1.50512 \cdot 10^5$ | 104.66 | 0.00 |
| 7000 | ours | $-1.50512 \cdot 10^5$ | **1.70** | **0.00** |

Table 56: Comparison of time and optimality gap on the synthetic datasets (p = 3000, k = 10, $\rho = 0.1$, $\lambda_2 = 10.0$). All baselines use our beam search solution as a warm start.

| # of samples | method | upper bound | time(s) | gap(%) |
|---|---|---|---|---|
| 3000 | SOS1 + warm start | $-1.59409 \cdot 10^5$ | 3601.55 | 18.52 |
| 3000 | big-M50 + warm start | $-1.59409 \cdot 10^5$ | 3600.67 | 18.52 |
| 3000 | persp. + warm start | $-1.59409 \cdot 10^5$ | 3600.26 | 17.84 |
| 3000 | eig. persp. + warm start | $-1.59409 \cdot 10^5$ | 3600.07 | 19.55 |
| 3000 | ours | $-1.59409 \cdot 10^5$ | **3600.62** | **4.68** |
| 4000 | SOS1 + warm start | $-2.17472 \cdot 10^5$ | 3600.06 | 15.13 |
| 4000 | big-M50 + warm start | $-2.17472 \cdot 10^5$ | 3600.08 | 14.64 |
| 4000 | persp. + warm start | $-2.17472 \cdot 10^5$ | 3600.78 | 12.93 |
| 4000 | eig. persp. + warm start | $-2.17472 \cdot 10^5$ | 3600.35 | 2.79 |
| 4000 | ours | $-2.17472 \cdot 10^5$ | **86.90** | **0.00** |
| 5000 | SOS1 + warm start | $-2.71885 \cdot 10^5$ | 3601.07 | 11.94 |
| 5000 | big-M50 + warm start | $-2.71885 \cdot 10^5$ | 3600.07 | 11.16 |
| 5000 | persp. + warm start | $-2.71885 \cdot 10^5$ | 3600.22 | 10.25 |
| 5000 | eig. persp. + warm start | $-2.71885 \cdot 10^5$ | 3109.70 | 0.00 |
| 5000 | ours | $-2.71885 \cdot 10^5$ | **20.75** | **0.00** |
| 6000 | SOS1 + warm start | $-3.16660 \cdot 10^5$ | 3600.92 | 9.93 |
| 6000 | big-M50 + warm start | $-3.16660 \cdot 10^5$ | 3600.06 | 9.93 |
| 6000 | persp. + warm start | $-3.16660 \cdot 10^5$ | 3600.04 | 9.24 |
| 6000 | eig. persp. + warm start | $-3.16660 \cdot 10^5$ | 3601.01 | 11.80 |
| 6000 | ours | $-3.16660 \cdot 10^5$ | **21.05** | **0.00** |
| 7000 | SOS1 + warm start | $-3.72509 \cdot 10^5$ | 3600.29 | 8.31 |
| 7000 | big-M50 + warm start | $-3.72509 \cdot 10^5$ | 3601.61 | 8.31 |
| 7000 | persp. + warm start | $-3.72509 \cdot 10^5$ | 3600.12 | 7.78 |
| 7000 | eig. persp. + warm start | $-3.72509 \cdot 10^5$ | 3600.07 | 9.46 |
| 7000 | ours | $-3.72509 \cdot 10^5$ | **1.71** | **0.00** |

Table 57: Comparison of time and optimality gap on the synthetic datasets (p = 3000, k = 10, $\rho = 0.3$, $\lambda_2 = 10.0$). All baselines use our beam search solution as a warm start.

| # of samples | method | upper bound | time(s) | gap(%) |
|---|---|---|---|---|
| 3000 | SOS1 + warm start | $-3.33503 \cdot 10^5$ | 3600.95 | 18.55 |
| 3000 | big-M50 + warm start | $-3.33503 \cdot 10^5$ | 3600.08 | 18.43 |
| 3000 | persp. + warm start | $-3.33503 \cdot 10^5$ | 3600.40 | 18.09 |
| 3000 | eig. persp. + warm start | $-3.33503 \cdot 10^5$ | 3600.06 | 19.55 |
| 3000 | ours | $-3.33503 \cdot 10^5$ | **3600.25** | **6.21** |
| 4000 | SOS1 + warm start | $-4.53159 \cdot 10^5$ | 3600.06 | 15.09 |
| 4000 | big-M50 + warm start | $-4.53159 \cdot 10^5$ | 3600.06 | 14.84 |
| 4000 | persp. + warm start | $-4.53159 \cdot 10^5$ | 3673.11 | 12.29 |
| 4000 | eig. persp. + warm start | $-4.53159 \cdot 10^5$ | 3600.09 | 15.34 |
| 4000 | ours | $-4.53159 \cdot 10^5$ | **3600.95** | **0.35** |
| 5000 | SOS1 + warm start | $-5.62906 \cdot 10^5$ | 3600.07 | 11.98 |
| 5000 | big-M50 + warm start | $-5.62906 \cdot 10^5$ | 3600.08 | 11.59 |
| 5000 | persp. + warm start | $-5.62906 \cdot 10^5$ | 3600.06 | 10.92 |
| 5000 | eig. persp. + warm start | $-5.62906 \cdot 10^5$ | 3600.31 | 0.24 |
| 5000 | ours | $-5.62906 \cdot 10^5$ | **31.47** | **0.00** |
| 6000 | SOS1 + warm start | $-6.57698 \cdot 10^5$ | 3600.88 | 9.95 |
| 6000 | big-M50 + warm start | $-6.57698 \cdot 10^5$ | 3600.14 | 9.95 |
| 6000 | persp. + warm start | $-6.57698 \cdot 10^5$ | 3600.06 | 9.02 |
| 6000 | eig. persp. + warm start | $-6.57698 \cdot 10^5$ | 1703.35 | 0.00 |
| 6000 | ours | $-6.57698 \cdot 10^5$ | **21.24** | **0.00** |
| 7000 | SOS1 + warm start | $-7.72166 \cdot 10^5$ | 3600.06 | 8.32 |
| 7000 | big-M50 + warm start | $-7.72166 \cdot 10^5$ | 3601.52 | 8.31 |
| 7000 | persp. + warm start | $-7.72166 \cdot 10^5$ | 3600.16 | 7.90 |
| 7000 | eig. persp. + warm start | $-7.72166 \cdot 10^5$ | 3600.41 | 10.39 |
| 7000 | ours | $-7.72166 \cdot 10^5$ | **20.37** | **0.00** |

Table 58: Comparison of time and optimality gap on the synthetic datasets ($p = 3000$, $k = 10$, $\rho = 0.5$, $\lambda_2 = 10.0$). All baselines use our beam search solution as a warm start.

| # of samples | method | upper bound | time(s) | gap(%) |
|---|---|---|---|---|
| 3000 | SOS1 + warm start | $-7.41086 \cdot 10^5$ | 3600.63 | 18.58 |
| 3000 | big-M50 + warm start | $-7.41086 \cdot 10^5$ | 3601.32 | 17.88 |
| 3000 | persp. + warm start | $-7.41086 \cdot 10^5$ | 3600.59 | 18.35 |
| 3000 | eig. persp. + warm start | $-7.41086 \cdot 10^5$ | 3600.49 | 19.55 |
| 3000 | ours | $-7.41086 \cdot 10^5$ | **3600.48** | **8.32** |
| 4000 | SOS1 + warm start | $-1.00277 \cdot 10^6$ | 3600.31 | 15.06 |
| 4000 | big-M50 + warm start | $-1.00277 \cdot 10^6$ | 3600.44 | 14.85 |
| 4000 | persp. + warm start | $-1.00277 \cdot 10^6$ | 3600.06 | 13.03 |
| 4000 | eig. persp. + warm start | $-1.00277 \cdot 10^6$ | 3600.21 | 15.54 |
| 4000 | ours | $-1.00277 \cdot 10^6$ | **3600.62** | **1.66** |
| 5000 | SOS1 + warm start | $-1.24017 \cdot 10^6$ | 3600.99 | 12.01 |
| 5000 | big-M50 + warm start | $-1.24017 \cdot 10^6$ | 3600.27 | 11.82 |
| 5000 | persp. + warm start | $-1.24017 \cdot 10^6$ | 3600.75 | 10.76 |
| 5000 | eig. persp. + warm start | $-1.24017 \cdot 10^6$ | 3600.07 | 12.59 |
| 5000 | ours | $-1.24017 \cdot 10^6$ | **386.72** | **0.00** |
| 6000 | SOS1 + warm start | $-1.45406 \cdot 10^6$ | 3600.07 | 9.97 |
| 6000 | big-M50 + warm start | $-1.45406 \cdot 10^6$ | 3600.26 | 9.75 |
| 6000 | persp. + warm start | $-1.45406 \cdot 10^6$ | 3600.31 | 9.44 |
| 6000 | eig. persp. + warm start | $-1.45406 \cdot 10^6$ | 2855.45 | 0.00 |
| 6000 | ours | $-1.45406 \cdot 10^6$ | **54.13** | **0.00** |
| 7000 | SOS1 + warm start | $-1.70473 \cdot 10^6$ | 3601.18 | 8.32 |
| 7000 | big-M50 + warm start | $-1.70473 \cdot 10^6$ | 3600.20 | 8.32 |
| 7000 | persp. + warm start | $-1.70473 \cdot 10^6$ | 3600.18 | 7.85 |
| 7000 | eig. persp. + warm start | $-1.70473 \cdot 10^6$ | 1264.32 | 0.00 |
| 7000 | ours | $-1.70473 \cdot 10^6$ | **21.30** | **0.00** |

Table 59: Comparison of time and optimality gap on the synthetic datasets (p = 3000, k = 10, $\rho = 0.7$, $\lambda_2 = 10.0$). All baselines use our beam search solution as a warm start.

| # of samples | method | upper bound | time(s) | gap(%) |
|---|---|---|---|---|
| 3000 | SOS1 + warm start | $-2.78507 \cdot 10^6$ | 3601.20 | 18.57 |
| 3000 | big-M50 + warm start | $-2.78507 \cdot 10^6$ | 3600.07 | 15.95 |
| 3000 | persp. + warm start | $-2.78507 \cdot 10^6$ | 3600.06 | 18.67 |
| 3000 | eig. persp. + warm start | $-2.78507 \cdot 10^6$ | 3600.36 | 19.55 |
| 3000 | ours | $-2.78507 \cdot 10^6$ | **3600.78** | **9.79** |
| 4000 | SOS1 + warm start | $-3.75192 \cdot 10^6$ | 3600.30 | 14.97 |
| 4000 | big-M50 + warm start | $-3.75192 \cdot 10^6$ | 3600.06 | 13.19 |
| 4000 | persp. + warm start | $-3.75192 \cdot 10^6$ | 3600.05 | 13.88 |
| 4000 | eig. persp. + warm start | $-3.75192 \cdot 10^6$ | 3600.06 | 15.77 |
| 4000 | ours | $-3.75192 \cdot 10^6$ | **3600.32** | **3.23** |
| 5000 | SOS1 + warm start | $-4.61867 \cdot 10^6$ | 3601.14 | 12.04 |
| 5000 | big-M50 + warm start | $-4.61867 \cdot 10^6$ | 3600.06 | 11.32 |
| 5000 | persp. + warm start | $-4.61867 \cdot 10^6$ | 3600.06 | 11.39 |
| 5000 | eig. persp. + warm start | $-4.61867 \cdot 10^6$ | 3600.07 | 12.85 |
| 5000 | ours | $-4.61867 \cdot 10^6$ | **3600.30** | **0.89** |
| 6000 | SOS1 + warm start | $-5.43816 \cdot 10^6$ | 3600.06 | 9.98 |
| 6000 | big-M50 + warm start | $-5.43816 \cdot 10^6$ | 3600.39 | 9.75 |
| 6000 | persp. + warm start | $-5.43816 \cdot 10^6$ | 3600.10 | 9.52 |
| 6000 | eig. persp. + warm start | $-5.43816 \cdot 10^6$ | 3600.13 | 11.05 |
| 6000 | ours | $-5.43816 \cdot 10^6$ | **3600.22** | **0.32** |
| 7000 | SOS1 + warm start | $-6.36757 \cdot 10^6$ | 3600.54 | 8.33 |
| 7000 | big-M50 + warm start | $-6.36757 \cdot 10^6$ | 3600.10 | 8.20 |
| 7000 | persp. + warm start | $-6.36757 \cdot 10^6$ | 3600.13 | 7.98 |
| 7000 | eig. persp. + warm start | $-6.36757 \cdot 10^6$ | 3600.25 | 0.34 |
| 7000 | ours | $-6.36757 \cdot 10^6$ | **3600.06** | **0.10** |

Table 60: Comparison of time and optimality gap on the synthetic datasets (p = 3000, k = 10, $\rho = 0.9$, $\lambda_2 = 10.0$). All baselines use our beam search solution as a warm start.

## G.3 Comparison with the Big-M Formulation with different $M$ Values

### G.3.1 Synthetic 1 Benchmark with $\lambda_2 = 0.001$ with No Warmstart

| # of features | method | upper bound | time(s) | gap(%) |
|---|---|---|---|---|
| 100 | big-M50 | $-2.10874 \cdot 10^6$ | 0.39 | 0.00 |
| 100 | big-M20 | $-2.10878 \cdot 10^6$ | 0.44 | 0.00 |
| 100 | big-M5 | $-2.10878 \cdot 10^6$ | 0.32 | 0.00 |
| 100 | ours | $-2.10878 \cdot 10^6$ | **0.29** | **0.00** |
| 500 | big-M50 | $-2.12828 \cdot 10^6$ | 36.80 | 0.00 |
| 500 | big-M20 | $-2.12833 \cdot 10^6$ | 81.36 | 0.00 |
| 500 | big-M5 | $-2.12833 \cdot 10^6$ | 52.95 | 0.00 |
| 500 | ours | $-2.12833 \cdot 10^6$ | **0.37** | **0.00** |
| 1000 | big-M50 | $-2.11005 \cdot 10^6$ | 1911.41 | 0.00 |
| 1000 | big-M20 | $-2.11005 \cdot 10^6$ | 1260.21 | 0.00 |
| 1000 | big-M5 | $-2.11005 \cdot 10^6$ | 771.63 | 0.00 |
| 1000 | ours | $-2.11005 \cdot 10^6$ | **0.58** | **0.00** |
| 3000 | big-M50 | $-7.83869 \cdot 10^5$ | 3601.20 | 169.78 |
| 3000 | big-M20 | $-1.14989 \cdot 10^6$ | 3603.59 | 83.91 |
| 3000 | big-M5 | $-2.10230 \cdot 10^6$ | 1998.40 | 0.00 |
| 3000 | ours | $-2.10230 \cdot 10^6$ | **1.73** | **0.00** |
| 5000 | big-M50 | $-1.87965 \cdot 10^6$ | 3603.25 | 12.65 |
| 5000 | big-M20 | $-9.85913 \cdot 10^5$ | 3602.87 | 114.78 |
| 5000 | big-M5 | - | 3618.45 | - |
| 5000 | ours | $-2.09635 \cdot 10^6$ | **4.22** | **0.00** |

Table 61: Comparison of time and optimality gap on the synthetic datasets (n = 100000, k = 10, $\rho = 0.1$, $\lambda_2 = 0.001$).

| # of features | method | upper bound | time(s) | gap(%) |
|---|---|---|---|---|
| 100 | big-M50 | $-5.28725 \cdot 10^6$ | 0.42 | 0.00 |
| 100 | big-M20 | $-5.28726 \cdot 10^6$ | 0.45 | 0.00 |
| 100 | big-M5 | $-5.28726 \cdot 10^6$ | 0.36 | 0.00 |
| 100 | ours | $-5.28726 \cdot 10^6$ | **0.29** | **0.00** |
| 500 | big-M50 | $-5.31819 \cdot 10^6$ | 89.04 | 0.00 |
| 500 | big-M20 | $-5.31845 \cdot 10^6$ | 75.86 | 0.00 |
| 500 | big-M5 | $-5.31852 \cdot 10^6$ | 51.98 | 0.00 |
| 500 | ours | $-5.31852 \cdot 10^6$ | **0.41** | **0.00** |
| 1000 | big-M50 | $-5.26974 \cdot 10^6$ | 1528.98 | 0.00 |
| 1000 | big-M20 | $-5.26973 \cdot 10^6$ | 1180.33 | 0.00 |
| 1000 | big-M5 | $-5.26974 \cdot 10^6$ | 794.14 | 0.00 |
| 1000 | ours | $-5.26974 \cdot 10^6$ | **0.52** | **0.00** |
| 3000 | big-M50 | $-3.72978 \cdot 10^6$ | 3602.55 | 42.31 |
| 3000 | big-M20 | $-3.97937 \cdot 10^6$ | 3604.47 | 33.38 |
| 3000 | big-M5 | - | 3660.43 | - |
| 3000 | ours | $-5.27668 \cdot 10^6$ | **1.75** | **0.00** |
| 5000 | big-M50 | $-4.16126 \cdot 10^6$ | 3602.53 | 27.95 |
| 5000 | big-M20 | $-4.20081 \cdot 10^6$ | 3601.38 | 26.74 |
| 5000 | big-M5 | $-5.27126 \cdot 10^6$ | 3600.14 | 1.00 |
| 5000 | ours | $-5.27126 \cdot 10^6$ | **4.32** | **0.00** |

Table 62: Comparison of time and optimality gap on the synthetic datasets (n = 100000, k = 10, $\rho = 0.3$, $\lambda_2 = 0.001$).

| # of features | method | upper bound | time(s) | gap(%) |
|---|---|---|---|---|
| 100 | big-M50 | $-1.10084 \cdot 10^7$ | 0.41 | 0.00 |
| 100 | big-M20 | $-1.10084 \cdot 10^7$ | 0.37 | 0.00 |
| 100 | big-M5 | $-1.10084 \cdot 10^7$ | 0.33 | 0.00 |
| 100 | ours | $-1.10084 \cdot 10^7$ | **0.29** | **0.00** |
| 500 | big-M50 | $-1.10515 \cdot 10^7$ | 89.59 | 0.00 |
| 500 | big-M20 | $-1.10518 \cdot 10^7$ | 79.56 | 0.00 |
| 500 | big-M5 | $-1.10518 \cdot 10^7$ | 54.59 | 0.00 |
| 500 | ours | $-1.10518 \cdot 10^7$ | **0.40** | **0.00** |
| 1000 | big-M50 | $-1.09506 \cdot 10^7$ | 1757.09 | 0.00 |
| 1000 | big-M20 | $-1.09508 \cdot 10^7$ | 1133.80 | 0.00 |
| 1000 | big-M5 | $-1.09508 \cdot 10^7$ | 744.15 | 0.00 |
| 1000 | ours | $-1.09508 \cdot 10^7$ | **0.59** | **0.00** |
| 3000 | big-M50 | - | 3615.51 | - |
| 3000 | big-M20 | - | 3635.15 | - |
| 3000 | big-M5 | - | 3648.05 | - |
| 3000 | ours | $-1.10012 \cdot 10^7$ | **1.74** | **0.00** |
| 5000 | big-M50 | - | 3612.62 | - |
| 5000 | big-M20 | $-1.01062 \cdot 10^7$ | 3612.19 | 9.90 |
| 5000 | big-M5 | - | 3605.20 | - |
| 5000 | ours | $-1.09966 \cdot 10^7$ | **4.00** | **0.00** |

Table 63: Comparison of time and optimality gap on the synthetic datasets (n = 100000, k = 10, $\rho = 0.5$, $\lambda_2 = 0.001$).

| # of features | method | upper bound | time(s) | gap(%) |
|---|---|---|---|---|
| 100 | big-M50 | $-2.43573 \cdot 10^7$ | 0.46 | 0.00 |
| 100 | big-M20 | $-2.43575 \cdot 10^7$ | 0.38 | 0.00 |
| 100 | big-M5 | $-2.43575 \cdot 10^7$ | **0.31** | **0.00** |
| 100 | ours | $-2.43575 \cdot 10^7$ | 0.32 | 0.00 |
| 500 | big-M50 | $-2.44178 \cdot 10^7$ | 92.53 | 0.00 |
| 500 | big-M20 | $-2.44185 \cdot 10^7$ | 75.08 | 0.00 |
| 500 | big-M5 | $-2.44185 \cdot 10^7$ | 52.27 | 0.00 |
| 500 | ours | $-2.44185 \cdot 10^7$ | **0.46** | **0.00** |
| 1000 | big-M50 | $-2.41987 \cdot 10^7$ | 1361.05 | 0.00 |
| 1000 | big-M20 | $-2.41988 \cdot 10^7$ | 974.43 | 0.00 |
| 1000 | big-M5 | $-2.41989 \cdot 10^7$ | 681.46 | 0.00 |
| 1000 | ours | $-2.41989 \cdot 10^7$ | **0.52** | **0.00** |
| 3000 | big-M50 | - | 3640.12 | - |
| 3000 | big-M20 | - | 3637.70 | - |
| 3000 | big-M5 | $-2.43713 \cdot 10^7$ | 3600.23 | 0.56 |
| 3000 | ours | $-2.43713 \cdot 10^7$ | **1.77** | **0.00** |
| 5000 | big-M50 | $-2.43694 \cdot 10^7$ | 3643.28 | 1.00 |
| 5000 | big-M20 | $-2.43694 \cdot 10^7$ | 3602.42 | 1.00 |
| 5000 | big-M5 | - | 3631.91 | - |
| 5000 | ours | $-2.43694 \cdot 10^7$ | **4.15** | **0.00** |

Table 64: Comparison of time and optimality gap on the synthetic datasets (n = 100000, k = 10, $\rho = 0.7$, $\lambda_2 = 0.001$).

| # of features | method | upper bound | time(s) | gap(%) |
|---|---|---|---|---|
| 100 | big-M50 | $-9.11026 \cdot 10^7$ | 0.13 | 0.00 |
| 100 | big-M20 | $-9.11026 \cdot 10^7$ | 0.13 | 0.00 |
| 100 | big-M5 | $-9.11026 \cdot 10^7$ | **0.10** | **0.00** |
| 100 | ours | $-9.11026 \cdot 10^7$ | 0.30 | 0.00 |
| 500 | big-M50 | $-9.12047 \cdot 10^7$ | 94.03 | 0.00 |
| 500 | big-M20 | $-9.12047 \cdot 10^7$ | 90.67 | 0.00 |
| 500 | big-M5 | $-9.12047 \cdot 10^7$ | 185.31 | 0.00 |
| 500 | ours | $-9.12047 \cdot 10^7$ | **0.38** | **0.00** |
| 1000 | big-M50 | $-9.04075 \cdot 10^7$ | 3600.01 | 0.07 |
| 1000 | big-M20 | $-9.04075 \cdot 10^7$ | 3600.50 | 0.20 |
| 1000 | big-M5 | $-9.04075 \cdot 10^7$ | 2081.96 | 0.00 |
| 1000 | ours | $-9.04075 \cdot 10^7$ | **0.52** | **0.00** |
| 3000 | big-M50 | - | 3605.36 | - |
| 3000 | big-M20 | - | 3623.06 | - |
| 3000 | big-M5 | - | 3611.34 | - |
| 3000 | ours | $-9.12761 \cdot 10^7$ | **20.74** | **0.00** |
| 5000 | big-M50 | - | 3607.68 | - |
| 5000 | big-M20 | - | 3616.13 | - |
| 5000 | big-M5 | - | 3627.90 | - |
| 5000 | ours | $-9.12916 \cdot 10^7$ | **58.29** | **0.00** |

Table 65: Comparison of time and optimality gap on the synthetic datasets (n = 100000, k = 10, $\rho = 0.9$, $\lambda_2 = 0.001$).

**G.3.2  Synthetic 1 Benchmark with $\lambda_2 = 0.001$ with Warmstart**

| # of features | method | upper bound | time(s) | gap(%) |
|:---:|:---:|:---:|:---:|:---:|
| 100 | big-M50 + warm start | $-2.10878 \cdot 10^6$ | 0.08 | 0.00 |
| 100 | big-M20 + warm start | $-2.10878 \cdot 10^6$ | 0.07 | 0.00 |
| 100 | big-M5 + warm start | $-2.10878 \cdot 10^6$ | **0.05** | **0.00** |
| 100 | ours | $-2.10878 \cdot 10^6$ | 0.29 | 0.00 |
| 500 | big-M50 + warm start | $-2.12833 \cdot 10^6$ | 37.62 | 0.00 |
| 500 | big-M20 + warm start | $-2.12833 \cdot 10^6$ | 2.82 | 0.00 |
| 500 | big-M5 + warm start | $-2.12833 \cdot 10^6$ | 3.00 | 0.00 |
| 500 | ours | $-2.12833 \cdot 10^6$ | **0.37** | **0.00** |
| 1000 | big-M50 + warm start | $-2.11005 \cdot 10^6$ | 28.51 | 0.00 |
| 1000 | big-M20 + warm start | $-2.11005 \cdot 10^6$ | 17.68 | 0.00 |
| 1000 | big-M5 + warm start | $-2.11005 \cdot 10^6$ | 25.46 | 0.00 |
| 1000 | ours | $-2.11005 \cdot 10^6$ | **0.58** | **0.00** |
| 3000 | big-M50 + warm start | $-2.10230 \cdot 10^6$ | 407.92 | 0.00 |
| 3000 | big-M20 + warm start | $-2.10230 \cdot 10^6$ | 255.31 | 0.00 |
| 3000 | big-M5 + warm start | $-2.10230 \cdot 10^6$ | 400.99 | 0.00 |
| 3000 | ours | $-2.10230 \cdot 10^6$ | **1.73** | **0.00** |
| 5000 | big-M50 + warm start | $-2.09635 \cdot 10^6$ | 1380.38 | 0.00 |
| 5000 | big-M20 + warm start | $-2.09635 \cdot 10^6$ | 1107.68 | 0.00 |
| 5000 | big-M5 + warm start | $-2.09635 \cdot 10^6$ | 1673.77 | 0.00 |
| 5000 | ours | $-2.09635 \cdot 10^6$ | **4.22** | **0.00** |

Table 66: Comparison of time and optimality gap on the synthetic datasets (n = 100000, k = 10, $\rho = 0.1$, $\lambda_2 = 0.001$). All baselines use our beam search solution as a warm start.

| # of features | method | upper bound | time(s) | gap(%) |
|---|---|---|---|---|
| 100 | big-M50 + warm start | $-5.28726 \cdot 10^6$ | 0.26 | 0.00 |
| 100 | big-M20 + warm start | $-5.28726 \cdot 10^6$ | 0.17 | 0.00 |
| 100 | big-M5 + warm start | $-5.28726 \cdot 10^6$ | **0.05** | **0.00** |
| 100 | ours | $-5.28726 \cdot 10^6$ | 0.29 | 0.00 |
| 500 | big-M50 + warm start | $-5.31852 \cdot 10^6$ | 3.00 | 0.00 |
| 500 | big-M20 + warm start | $-5.31852 \cdot 10^6$ | 12.76 | 0.00 |
| 500 | big-M5 + warm start | $-5.31852 \cdot 10^6$ | 1.94 | 0.00 |
| 500 | ours | $-5.31852 \cdot 10^6$ | **0.41** | **0.00** |
| 1000 | big-M50 + warm start | $-5.26974 \cdot 10^6$ | 13.79 | 0.00 |
| 1000 | big-M20 + warm start | $-5.26974 \cdot 10^6$ | 18.04 | 0.00 |
| 1000 | big-M5 + warm start | $-5.26974 \cdot 10^6$ | 12.19 | 0.00 |
| 1000 | ours | $-5.26974 \cdot 10^6$ | **0.52** | **0.00** |
| 3000 | big-M50 + warm start | $-5.27668 \cdot 10^6$ | 313.22 | 0.00 |
| 3000 | big-M20 + warm start | $-5.27668 \cdot 10^6$ | 461.86 | 0.00 |
| 3000 | big-M5 + warm start | $-5.27668 \cdot 10^6$ | 245.59 | 0.00 |
| 3000 | ours | $-5.27668 \cdot 10^6$ | **1.75** | **0.00** |
| 5000 | big-M50 + warm start | $-5.27126 \cdot 10^6$ | 1705.37 | 0.00 |
| 5000 | big-M20 + warm start | $-5.27126 \cdot 10^6$ | 1666.20 | 0.00 |
| 5000 | big-M5 + warm start | $-5.27126 \cdot 10^6$ | 3601.58 | 1.00 |
| 5000 | ours | $-5.27126 \cdot 10^6$ | **4.32** | **0.00** |

Table 67: Comparison of time and optimality gap on the synthetic datasets (n = 100000, k = 10, $\rho = 0.3$, $\lambda_2 = 0.001$). All baselines use our beam search solution as a warm start.

| # of features | method | upper bound | time(s) | gap(%) |
|---|---|---|---|---|
| 100 | big-M50 + warm start | $-1.10084 \cdot 10^7$ | 0.06 | 0.00 |
| 100 | big-M20 + warm start | $-1.10084 \cdot 10^7$ | 0.06 | 0.00 |
| 100 | big-M5 + warm start | $-1.10084 \cdot 10^7$ | **0.04** | **0.00** |
| 100 | ours | $-1.10084 \cdot 10^7$ | 0.29 | 0.00 |
| 500 | big-M50 + warm start | $-1.10518 \cdot 10^7$ | 11.59 | 0.00 |
| 500 | big-M20 + warm start | $-1.10518 \cdot 10^7$ | 6.55 | 0.00 |
| 500 | big-M5 + warm start | $-1.10518 \cdot 10^7$ | 1.97 | 0.00 |
| 500 | ours | $-1.10518 \cdot 10^7$ | **0.40** | **0.00** |
| 1000 | big-M50 + warm start | $-1.09508 \cdot 10^7$ | 21.16 | 0.00 |
| 1000 | big-M20 + warm start | $-1.09508 \cdot 10^7$ | 14.95 | 0.00 |
| 1000 | big-M5 + warm start | $-1.09508 \cdot 10^7$ | 12.99 | 0.00 |
| 1000 | ours | $-1.09508 \cdot 10^7$ | **0.59** | **0.00** |
| 3000 | big-M50 + warm start | $-1.10012 \cdot 10^7$ | 243.42 | 0.00 |
| 3000 | big-M20 + warm start | $-1.10012 \cdot 10^7$ | 301.69 | 0.00 |
| 3000 | big-M5 + warm start | $-1.10012 \cdot 10^7$ | 242.64 | 0.00 |
| 3000 | ours | $-1.10012 \cdot 10^7$ | **1.74** | **0.00** |
| 5000 | big-M50 + warm start | $-1.09966 \cdot 10^7$ | 3600.95 | 1.00 |
| 5000 | big-M20 + warm start | $-1.09966 \cdot 10^7$ | 3600.28 | 1.00 |
| 5000 | big-M5 + warm start | $-1.09966 \cdot 10^7$ | 3602.10 | 570161330.64 |
| 5000 | ours | $-1.09966 \cdot 10^7$ | **4.00** | **0.00** |

Table 68: Comparison of time and optimality gap on the synthetic datasets (n = 100000, k = 10, $\rho = 0.5$, $\lambda_2 = 0.001$). All baselines use our beam search solution as a warm start. The method big-M5 has some large optimality gaps because Gurobi couldn't finish solving the root relaxation within 1h.

| # of features | method | upper bound | time(s) | gap(%) |
|---|---|---|---|---|
| 100 | big-M50 + warm start | $-2.43575 \cdot 10^7$ | 0.05 | 0.00 |
| 100 | big-M20 + warm start | $-2.43575 \cdot 10^7$ | 0.26 | 0.00 |
| 100 | big-M5 + warm start | $-2.43575 \cdot 10^7$ | **0.05** | **0.00** |
| 100 | ours | $-2.43575 \cdot 10^7$ | 0.32 | 0.00 |
| 500 | big-M50 + warm start | $-2.44185 \cdot 10^7$ | 4.02 | 0.00 |
| 500 | big-M20 + warm start | $-2.44185 \cdot 10^7$ | 2.15 | 0.00 |
| 500 | big-M5 + warm start | $-2.44185 \cdot 10^7$ | 20.44 | 0.00 |
| 500 | ours | $-2.44185 \cdot 10^7$ | **0.46** | **0.00** |
| 1000 | big-M50 + warm start | $-2.41989 \cdot 10^7$ | 11.63 | 0.00 |
| 1000 | big-M20 + warm start | $-2.41989 \cdot 10^7$ | 13.11 | 0.00 |
| 1000 | big-M5 + warm start | $-2.41989 \cdot 10^7$ | 10.49 | 0.00 |
| 1000 | ours | $-2.41989 \cdot 10^7$ | **0.52** | **0.00** |
| 3000 | big-M50 + warm start | $-2.43713 \cdot 10^7$ | 3608.31 | 0.59 |
| 3000 | big-M20 + warm start | $-2.43713 \cdot 10^7$ | 3600.14 | 0.59 |
| 3000 | big-M5 + warm start | $-2.43713 \cdot 10^7$ | 3600.95 | 0.56 |
| 3000 | ours | $-2.43713 \cdot 10^7$ | **1.77** | **0.00** |
| 5000 | big-M50 + warm start | $-2.43694 \cdot 10^7$ | 3670.94 | 1.00 |
| 5000 | big-M20 + warm start | $-2.43694 \cdot 10^7$ | 3604.91 | 1.00 |
| 5000 | big-M5 + warm start | $-2.43694 \cdot 10^7$ | 3646.34 | 600317552.81 |
| 5000 | ours | $-2.43694 \cdot 10^7$ | **4.15** | **0.00** |

Table 69: Comparison of time and optimality gap on the synthetic datasets (n = 100000, k = 10, $\rho = 0.7$, $\lambda_2 = 0.001$). All baselines use our beam search solution as a warm start. The method big-M5 has some large optimality gaps because Gurobi couldn't finish solving the root relaxation within 1h.

| # of features | method | upper bound | time(s) | gap(%) |
|---|---|---|---|---|
| 100 | big-M50 + warm start | $-9.11026 \cdot 10^7$ | 0.03 | 0.00 |
| 100 | big-M20 + warm start | $-9.11026 \cdot 10^7$ | **0.03** | **0.00** |
| 100 | big-M5 + warm start | $-9.11026 \cdot 10^7$ | 0.04 | 0.00 |
| 100 | ours | $-9.11026 \cdot 10^7$ | 0.30 | 0.00 |
| 500 | big-M50 + warm start | $-9.12047 \cdot 10^7$ | 34.86 | 0.00 |
| 500 | big-M20 + warm start | $-9.12047 \cdot 10^7$ | 28.75 | 0.00 |
| 500 | big-M5 + warm start | $-9.12047 \cdot 10^7$ | 18.67 | 0.00 |
| 500 | ours | $-9.12047 \cdot 10^7$ | **0.38** | **0.00** |
| 1000 | big-M50 + warm start | $-9.04075 \cdot 10^7$ | 3600.01 | 0.08 |
| 1000 | big-M20 + warm start | $-9.04075 \cdot 10^7$ | 3600.04 | 0.20 |
| 1000 | big-M5 + warm start | $-9.04075 \cdot 10^7$ | 1662.95 | 0.00 |
| 1000 | ours | $-9.04075 \cdot 10^7$ | **0.52** | **0.00** |
| 3000 | big-M50 + warm start | $-9.12761 \cdot 10^7$ | 3612.88 | 0.59 |
| 3000 | big-M20 + warm start | $-9.12761 \cdot 10^7$ | 3600.12 | 0.59 |
| 3000 | big-M5 + warm start | $-9.12761 \cdot 10^7$ | 3613.07 | 222997912.50 |
| 3000 | ours | $-9.12761 \cdot 10^7$ | **20.74** | **0.00** |
| 5000 | big-M50 + warm start | $-9.12916 \cdot 10^7$ | 3622.52 | 1.00 |
| 5000 | big-M20 + warm start | $-9.12916 \cdot 10^7$ | 3606.87 | 1.00 |
| 5000 | big-M5 + warm start | $-9.12916 \cdot 10^7$ | 3600.86 | 618133887.09 |
| 5000 | ours | $-9.12916 \cdot 10^7$ | **58.29** | **0.00** |

Table 70: Comparison of time and optimality gap on the synthetic datasets (n = 100000, k = 10, $\rho = 0.9$, $\lambda_2 = 0.001$). All baselines use our beam search solution as a warm start. The method big-M5 has some large optimality gaps because Gurobi couldn't finish solving the root relaxation within 1h.

### G.3.3 Synthetic 1 Benchmark with $\lambda_2 = 0.1$ with No Warmstart

| # of features | method | upper bound | time(s) | gap(%) |
|---|---|---|---|---|
| 100 | big-M50 | $-2.10878 \cdot 10^6$ | 0.86 | 0.00 |
| 100 | big-M20 | $-2.10878 \cdot 10^6$ | 0.41 | 0.00 |
| 100 | big-M5 | $-2.10878 \cdot 10^6$ | **0.32** | **0.00** |
| 100 | ours | $-2.10878 \cdot 10^6$ | 0.36 | 0.00 |
| 500 | big-M50 | $-2.12833 \cdot 10^6$ | 37.96 | 0.00 |
| 500 | big-M20 | $-2.12833 \cdot 10^6$ | 27.57 | 0.00 |
| 500 | big-M5 | $-2.12833 \cdot 10^6$ | 55.79 | 0.00 |
| 500 | ours | $-2.12833 \cdot 10^6$ | **0.37** | **0.00** |
| 1000 | big-M50 | $-2.11005 \cdot 10^6$ | 1007.27 | 0.00 |
| 1000 | big-M20 | $-2.11005 \cdot 10^6$ | 373.44 | 0.00 |
| 1000 | big-M5 | $-2.11005 \cdot 10^6$ | 74.44 | 0.00 |
| 1000 | ours | $-2.11005 \cdot 10^6$ | **0.57** | **0.00** |
| 3000 | big-M50 | $-7.99504 \cdot 10^5$ | 3608.36 | 164.51 |
| 3000 | big-M20 | $-1.15295 \cdot 10^6$ | 3603.16 | 83.42 |
| 3000 | big-M5 | $-2.10230 \cdot 10^6$ | 1289.57 | 0.00 |
| 3000 | ours | $-2.10230 \cdot 10^6$ | **1.79** | **0.00** |
| 5000 | big-M50 | $-9.69753 \cdot 10^5$ | 3619.50 | 118.35 |
| 5000 | big-M20 | $-9.85912 \cdot 10^5$ | 3611.26 | 114.78 |
| 5000 | big-M5 | - | 3610.20 | - |
| 5000 | ours | $-2.09635 \cdot 10^6$ | **4.20** | **0.00** |

Table 71: Comparison of time and optimality gap on the synthetic datasets (n = 100000, k = 10, $\rho = 0.1$, $\lambda_2 = 0.1$).

| # of features | method | upper bound | time(s) | gap(%) |
|---|---|---|---|---|
| 100 | big-M50 | $-4.60269 \cdot 10^6$ | 1.83 | 0.00 |
| 100 | big-M20 | $-5.28726 \cdot 10^6$ | 0.40 | 0.00 |
| 100 | big-M5 | $-5.28726 \cdot 10^6$ | 0.34 | 0.00 |
| 100 | ours | $-5.28726 \cdot 10^6$ | **0.27** | **0.00** |
| 500 | big-M50 | $-5.31827 \cdot 10^6$ | 85.52 | 0.00 |
| 500 | big-M20 | $-5.31844 \cdot 10^6$ | 72.87 | 0.00 |
| 500 | big-M5 | $-5.31852 \cdot 10^6$ | 52.81 | 0.00 |
| 500 | ours | $-5.31852 \cdot 10^6$ | **0.41** | **0.00** |
| 1000 | big-M50 | $-5.26974 \cdot 10^6$ | 1576.75 | 0.00 |
| 1000 | big-M20 | $-5.26974 \cdot 10^6$ | 1112.97 | 0.00 |
| 1000 | big-M5 | $-5.26974 \cdot 10^6$ | 791.17 | 0.00 |
| 1000 | ours | $-5.26974 \cdot 10^6$ | **0.52** | **0.00** |
| 3000 | big-M50 | $-3.73049 \cdot 10^6$ | 3603.09 | 42.28 |
| 3000 | big-M20 | $-3.98562 \cdot 10^6$ | 3602.15 | 33.17 |
| 3000 | big-M5 | - | 3634.94 | - |
| 3000 | ours | $-5.27668 \cdot 10^6$ | **1.84** | **0.00** |
| 5000 | big-M50 | $-3.98069 \cdot 10^6$ | 3606.79 | 33.75 |
| 5000 | big-M20 | $-4.58351 \cdot 10^6$ | 3615.50 | 16.16 |
| 5000 | big-M5 | $-5.27126 \cdot 10^6$ | 3600.14 | 1.00 |
| 5000 | ours | $-5.27126 \cdot 10^6$ | **4.42** | **0.00** |

Table 72: Comparison of time and optimality gap on the synthetic datasets (n = 100000, k = 10, $\rho = 0.3$, $\lambda_2 = 0.1$).

| # of features | method | upper bound | time(s) | gap(%) |
|---|---|---|---|---|
| 100 | big-M50 | $-1.10084 \cdot 10^7$ | 0.44 | 0.00 |
| 100 | big-M20 | $-1.10084 \cdot 10^7$ | 0.37 | 0.00 |
| 100 | big-M5 | $-1.10084 \cdot 10^7$ | 0.33 | 0.00 |
| 100 | ours | $-1.10084 \cdot 10^7$ | **0.32** | **0.00** |
| 500 | big-M50 | $-1.10515 \cdot 10^7$ | 86.52 | 0.00 |
| 500 | big-M20 | $-1.10518 \cdot 10^7$ | 77.61 | 0.00 |
| 500 | big-M5 | $-1.10518 \cdot 10^7$ | 51.24 | 0.00 |
| 500 | ours | $-1.10518 \cdot 10^7$ | **0.40** | **0.00** |
| 1000 | big-M50 | $-1.09506 \cdot 10^7$ | 1539.25 | 0.00 |
| 1000 | big-M20 | $-1.09508 \cdot 10^7$ | 1118.41 | 0.00 |
| 1000 | big-M5 | $-1.09508 \cdot 10^7$ | 893.44 | 0.00 |
| 1000 | ours | $-1.09508 \cdot 10^7$ | **0.53** | **0.00** |
| 3000 | big-M50 | - | 3608.09 | - |
| 3000 | big-M20 | - | 3614.30 | - |
| 3000 | big-M5 | - | 3610.50 | - |
| 3000 | ours | $-1.10012 \cdot 10^7$ | **1.67** | **0.00** |
| 5000 | big-M50 | - | 3649.14 | - |
| 5000 | big-M20 | $-1.01130 \cdot 10^7$ | 3600.71 | 9.83 |
| 5000 | big-M5 | - | 3628.44 | - |
| 5000 | ours | $-1.09966 \cdot 10^7$ | **4.13** | **0.00** |

Table 73: Comparison of time and optimality gap on the synthetic datasets (n = 100000, k = 10, $\rho = 0.5$, $\lambda_2 = 0.1$).

| # of features | method | upper bound | time(s) | gap(%) |
|---|---|---|---|---|
| 100 | big-M50 | $-2.43573 \cdot 10^7$ | 0.41 | 0.00 |
| 100 | big-M20 | $-2.43575 \cdot 10^7$ | 0.39 | 0.00 |
| 100 | big-M5 | $-2.43575 \cdot 10^7$ | **0.30** | **0.00** |
| 100 | ours | $-2.43575 \cdot 10^7$ | 0.34 | 0.00 |
| 500 | big-M50 | $-2.44167 \cdot 10^7$ | 96.59 | 0.00 |
| 500 | big-M20 | $-2.44185 \cdot 10^7$ | 72.41 | 0.00 |
| 500 | big-M5 | $-2.44185 \cdot 10^7$ | 54.87 | 0.00 |
| 500 | ours | $-2.44185 \cdot 10^7$ | **0.44** | **0.00** |
| 1000 | big-M50 | $-2.41986 \cdot 10^7$ | 1608.58 | 0.00 |
| 1000 | big-M20 | $-2.41988 \cdot 10^7$ | 984.16 | 0.00 |
| 1000 | big-M5 | $-2.41989 \cdot 10^7$ | 639.70 | 0.00 |
| 1000 | ours | $-2.41989 \cdot 10^7$ | **0.53** | **0.00** |
| 3000 | big-M50 | - | 3605.20 | - |
| 3000 | big-M20 | - | 3617.57 | - |
| 3000 | big-M5 | $-2.43713 \cdot 10^7$ | 3600.08 | 0.56 |
| 3000 | ours | $-2.43713 \cdot 10^7$ | **1.75** | **0.00** |
| 5000 | big-M50 | $-2.43694 \cdot 10^7$ | 3650.52 | 1.00 |
| 5000 | big-M20 | $-2.43694 \cdot 10^7$ | 3624.42 | 1.00 |
| 5000 | big-M5 | - | 3650.91 | - |
| 5000 | ours | $-2.43694 \cdot 10^7$ | **4.22** | **0.00** |

Table 74: Comparison of time and optimality gap on the synthetic datasets (n = 100000, k = 10, $\rho = 0.7$, $\lambda_2 = 0.1$).

| # of features | method | upper bound | time(s) | gap(%) |
|---|---|---|---|---|
| 100 | big-M50 | $-9.11026 \cdot 10^7$ | 0.18 | 0.00 |
| 100 | big-M20 | $-9.11026 \cdot 10^7$ | **0.11** | **0.00** |
| 100 | big-M5 | $-9.11026 \cdot 10^7$ | 0.12 | 0.00 |
| 100 | ours | $-9.11026 \cdot 10^7$ | 0.31 | 0.00 |
| 500 | big-M50 | $-9.12047 \cdot 10^7$ | 398.02 | 0.00 |
| 500 | big-M20 | $-9.12047 \cdot 10^7$ | 113.34 | 0.00 |
| 500 | big-M5 | $-9.12047 \cdot 10^7$ | 187.63 | 0.00 |
| 500 | ours | $-9.12047 \cdot 10^7$ | **0.38** | **0.00** |
| 1000 | big-M50 | $-9.04075 \cdot 10^7$ | 3600.05 | 0.01 |
| 1000 | big-M20 | $-9.04075 \cdot 10^7$ | 3600.24 | 0.20 |
| 1000 | big-M5 | $-9.04075 \cdot 10^7$ | 2116.59 | 0.00 |
| 1000 | ours | $-9.04075 \cdot 10^7$ | **0.52** | **0.00** |
| 3000 | big-M50 | - | 3619.63 | - |
| 3000 | big-M20 | - | 3613.39 | - |
| 3000 | big-M5 | - | 3609.80 | - |
| 3000 | ours | $-9.12761 \cdot 10^7$ | **20.88** | **0.00** |
| 5000 | big-M50 | - | 3620.37 | - |
| 5000 | big-M20 | - | 3603.08 | - |
| 5000 | big-M5 | - | 3623.71 | - |
| 5000 | ours | $-9.12916 \cdot 10^7$ | **58.07** | **0.00** |

Table 75: Comparison of time and optimality gap on the synthetic datasets (n = 100000, k = 10, $\rho = 0.9$, $\lambda_2 = 0.1$).

### G.3.4 Synthetic 1 Benchmark with $\lambda_2 = 0.1$ with Warmstart

| # of features | method | upper bound | time(s) | gap(%) |
|---|---|---|---|---|
| 100 | big-M50 + warm start | $-2.10878 \cdot 10^6$ | 0.08 | 0.00 |
| 100 | big-M20 + warm start | $-2.10878 \cdot 10^6$ | 0.17 | 0.00 |
| 100 | big-M5 + warm start | $-2.10878 \cdot 10^6$ | **0.05** | **0.00** |
| 100 | ours | $-2.10878 \cdot 10^6$ | 0.36 | 0.00 |
| 500 | big-M50 + warm start | $-2.12833 \cdot 10^6$ | 3.78 | 0.00 |
| 500 | big-M20 + warm start | $-2.12833 \cdot 10^6$ | 2.75 | 0.00 |
| 500 | big-M5 + warm start | $-2.12833 \cdot 10^6$ | 3.10 | 0.00 |
| 500 | ours | $-2.12833 \cdot 10^6$ | **0.37** | **0.00** |
| 1000 | big-M50 + warm start | $-2.11005 \cdot 10^6$ | 26.94 | 0.00 |
| 1000 | big-M20 + warm start | $-2.11005 \cdot 10^6$ | 18.08 | 0.00 |
| 1000 | big-M5 + warm start | $-2.11005 \cdot 10^6$ | 16.21 | 0.00 |
| 1000 | ours | $-2.11005 \cdot 10^6$ | **0.57** | **0.00** |
| 3000 | big-M50 + warm start | $-2.10230 \cdot 10^6$ | 193.85 | 0.00 |
| 3000 | big-M20 + warm start | $-2.10230 \cdot 10^6$ | 255.55 | 0.00 |
| 3000 | big-M5 + warm start | $-2.10230 \cdot 10^6$ | 283.37 | 0.00 |
| 3000 | ours | $-2.10230 \cdot 10^6$ | **1.79** | **0.00** |
| 5000 | big-M50 + warm start | $-2.09635 \cdot 10^6$ | 1366.14 | 0.00 |
| 5000 | big-M20 + warm start | $-2.09635 \cdot 10^6$ | 1114.05 | 0.00 |
| 5000 | big-M5 + warm start | $-2.09635 \cdot 10^6$ | 1660.23 | 0.00 |
| 5000 | ours | $-2.09635 \cdot 10^6$ | **4.20** | **0.00** |

Table 76: Comparison of time and optimality gap on the synthetic datasets (n = 100000, k = 10, $\rho = 0.1$, $\lambda_2 = 0.1$). All baselines use our beam search solution as a warm start.

| # of features | method | upper bound | time(s) | gap(%) |
|---|---|---|---|---|
| 100 | big-M50 + warm start | $-5.28726 \cdot 10^6$ | 0.07 | 0.00 |
| 100 | big-M20 + warm start | $-5.28726 \cdot 10^6$ | 0.07 | 0.00 |
| 100 | big-M5 + warm start | $-5.28726 \cdot 10^6$ | **0.05** | **0.00** |
| 100 | ours | $-5.28726 \cdot 10^6$ | 0.27 | 0.00 |
| 500 | big-M50 + warm start | $-5.31852 \cdot 10^6$ | 39.88 | 0.00 |
| 500 | big-M20 + warm start | $-5.31852 \cdot 10^6$ | 3.02 | 0.00 |
| 500 | big-M5 + warm start | $-5.31852 \cdot 10^6$ | 1.86 | 0.00 |
| 500 | ours | $-5.31852 \cdot 10^6$ | **0.41** | **0.00** |
| 1000 | big-M50 + warm start | $-5.26974 \cdot 10^6$ | 14.33 | 0.00 |
| 1000 | big-M20 + warm start | $-5.26974 \cdot 10^6$ | 14.92 | 0.00 |
| 1000 | big-M5 + warm start | $-5.26974 \cdot 10^6$ | 13.35 | 0.00 |
| 1000 | ours | $-5.26974 \cdot 10^6$ | **0.52** | **0.00** |
| 3000 | big-M50 + warm start | $-5.27668 \cdot 10^6$ | 317.41 | 0.00 |
| 3000 | big-M20 + warm start | $-5.27668 \cdot 10^6$ | 470.35 | 0.00 |
| 3000 | big-M5 + warm start | $-5.27668 \cdot 10^6$ | 249.02 | 0.00 |
| 3000 | ours | $-5.27668 \cdot 10^6$ | **1.84** | **0.00** |
| 5000 | big-M50 + warm start | $-5.27126 \cdot 10^6$ | 1658.42 | 0.00 |
| 5000 | big-M20 + warm start | $-5.27126 \cdot 10^6$ | 3607.98 | 1.00 |
| 5000 | big-M5 + warm start | $-5.27126 \cdot 10^6$ | 3600.14 | 1.00 |
| 5000 | ours | $-5.27126 \cdot 10^6$ | **4.42** | **0.00** |

Table 77: Comparison of time and optimality gap on the synthetic datasets (n = 100000, k = 10, $\rho = 0.3$, $\lambda_2 = 0.1$). All baselines use our beam search solution as a warm start.

| # of features | method | upper bound | time(s) | gap(%) |
|---|---|---|---|---|
| 100 | big-M50 + warm start | $-1.10084 \cdot 10^7$ | 0.06 | 0.00 |
| 100 | big-M20 + warm start | $-1.10084 \cdot 10^7$ | 0.06 | 0.00 |
| 100 | big-M5 + warm start | $-1.10084 \cdot 10^7$ | **0.04** | **0.00** |
| 100 | ours | $-1.10084 \cdot 10^7$ | 0.32 | 0.00 |
| 500 | big-M50 + warm start | $-1.10518 \cdot 10^7$ | 2.28 | 0.00 |
| 500 | big-M20 + warm start | $-1.10518 \cdot 10^7$ | 6.61 | 0.00 |
| 500 | big-M5 + warm start | $-1.10518 \cdot 10^7$ | 13.48 | 0.00 |
| 500 | ours | $-1.10518 \cdot 10^7$ | **0.40** | **0.00** |
| 1000 | big-M50 + warm start | $-1.09508 \cdot 10^7$ | 14.11 | 0.00 |
| 1000 | big-M20 + warm start | $-1.09508 \cdot 10^7$ | 13.83 | 0.00 |
| 1000 | big-M5 + warm start | $-1.09508 \cdot 10^7$ | 12.46 | 0.00 |
| 1000 | ours | $-1.09508 \cdot 10^7$ | **0.53** | **0.00** |
| 3000 | big-M50 + warm start | $-1.10012 \cdot 10^7$ | 222.56 | 0.00 |
| 3000 | big-M20 + warm start | $-1.10012 \cdot 10^7$ | 290.65 | 0.00 |
| 3000 | big-M5 + warm start | $-1.10012 \cdot 10^7$ | 251.83 | 0.00 |
| 3000 | ours | $-1.10012 \cdot 10^7$ | **1.67** | **0.00** |
| 5000 | big-M50 + warm start | $-1.09966 \cdot 10^7$ | 3606.43 | 1.00 |
| 5000 | big-M20 + warm start | $-1.09966 \cdot 10^7$ | 3611.47 | 1.00 |
| 5000 | big-M5 + warm start | $-1.09966 \cdot 10^7$ | 3631.72 | 570161381.99 |
| 5000 | ours | $-1.09966 \cdot 10^7$ | **4.13** | **0.00** |

Table 78: Comparison of time and optimality gap on the synthetic datasets (n = 100000, k = 10, $\rho = 0.5$, $\lambda_2 = 0.1$). All baselines use our beam search solution as a warm start. The method big-M5 has some large optimality gaps because Gurobi couldn't finish solving the root relaxation within 1h.

| # of features | method | upper bound | time(s) | gap(%) |
|---|---|---|---|---|
| 100 | big-M50 + warm start | $-2.43575 \cdot 10^7$ | 0.06 | 0.00 |
| 100 | big-M20 + warm start | $-2.43575 \cdot 10^7$ | 0.24 | 0.00 |
| 100 | big-M5 + warm start | $-2.43575 \cdot 10^7$ | **0.04** | **0.00** |
| 100 | ours | $-2.43575 \cdot 10^7$ | 0.34 | 0.00 |
| 500 | big-M50 + warm start | $-2.44185 \cdot 10^7$ | 4.00 | 0.00 |
| 500 | big-M20 + warm start | $-2.44185 \cdot 10^7$ | 8.60 | 0.00 |
| 500 | big-M5 + warm start | $-2.44185 \cdot 10^7$ | 19.75 | 0.00 |
| 500 | ours | $-2.44185 \cdot 10^7$ | **0.44** | **0.00** |
| 1000 | big-M50 + warm start | $-2.41989 \cdot 10^7$ | 12.62 | 0.00 |
| 1000 | big-M20 + warm start | $-2.41989 \cdot 10^7$ | 12.12 | 0.00 |
| 1000 | big-M5 + warm start | $-2.41989 \cdot 10^7$ | 9.30 | 0.00 |
| 1000 | ours | $-2.41989 \cdot 10^7$ | **0.53** | **0.00** |
| 3000 | big-M50 + warm start | $-2.43713 \cdot 10^7$ | 3600.43 | 0.59 |
| 3000 | big-M20 + warm start | $-2.43713 \cdot 10^7$ | 3600.05 | 0.59 |
| 3000 | big-M5 + warm start | $-2.43713 \cdot 10^7$ | 3600.15 | 0.56 |
| 3000 | ours | $-2.43713 \cdot 10^7$ | **1.75** | **0.00** |
| 5000 | big-M50 + warm start | $-2.43694 \cdot 10^7$ | 3616.71 | 1.00 |
| 5000 | big-M20 + warm start | $-2.43694 \cdot 10^7$ | 3615.82 | 1.00 |
| 5000 | big-M5 + warm start | $-2.43694 \cdot 10^7$ | 3626.45 | 600317577.18 |
| 5000 | ours | $-2.43694 \cdot 10^7$ | **4.22** | **0.00** |

Table 79: Comparison of time and optimality gap on the synthetic datasets (n = 100000, k = 10, $\rho = 0.7$, $\lambda_2 = 0.1$). All baselines use our beam search solution as a warm start. The method big-M5 has some large optimality gaps because Gurobi couldn't finish solving the root relaxation within 1h.

| # of features | method | upper bound | time(s) | gap(%) |
|---|---|---|---|---|
| 100 | big-M50 + warm start | $-9.11026 \cdot 10^7$ | 0.04 | 0.00 |
| 100 | big-M20 + warm start | $-9.11026 \cdot 10^7$ | **0.03** | **0.00** |
| 100 | big-M5 + warm start | $-9.11026 \cdot 10^7$ | 0.03 | 0.00 |
| 100 | ours | $-9.11026 \cdot 10^7$ | 0.31 | 0.00 |
| 500 | big-M50 + warm start | $-9.12047 \cdot 10^7$ | 14.34 | 0.00 |
| 500 | big-M20 + warm start | $-9.12047 \cdot 10^7$ | 28.63 | 0.00 |
| 500 | big-M5 + warm start | $-9.12047 \cdot 10^7$ | 18.60 | 0.00 |
| 500 | ours | $-9.12047 \cdot 10^7$ | **0.38** | **0.00** |
| 1000 | big-M50 + warm start | $-9.04075 \cdot 10^7$ | 3600.16 | 0.01 |
| 1000 | big-M20 + warm start | $-9.04075 \cdot 10^7$ | 3600.11 | 0.20 |
| 1000 | big-M5 + warm start | $-9.04075 \cdot 10^7$ | 1687.23 | 0.00 |
| 1000 | ours | $-9.04075 \cdot 10^7$ | **0.52** | **0.00** |
| 3000 | big-M50 + warm start | $-9.12761 \cdot 10^7$ | 3611.02 | 0.59 |
| 3000 | big-M20 + warm start | $-9.12761 \cdot 10^7$ | 3601.67 | 0.59 |
| 3000 | big-M5 + warm start | $-9.12761 \cdot 10^7$ | 3602.09 | 222997914.92 |
| 3000 | ours | $-9.12761 \cdot 10^7$ | **20.88** | **0.00** |
| 5000 | big-M50 + warm start | $-9.12916 \cdot 10^7$ | 3606.64 | 1.00 |
| 5000 | big-M20 + warm start | $-9.12916 \cdot 10^7$ | 3610.82 | 1.00 |
| 5000 | big-M5 + warm start | $-9.12916 \cdot 10^7$ | 3632.24 | 618133893.79 |
| 5000 | ours | $-9.12916 \cdot 10^7$ | **58.07** | **0.00** |

Table 80: Comparison of time and optimality gap on the synthetic datasets (n = 100000, k = 10, $\rho = 0.9$, $\lambda_2 = 0.1$). All baselines use our beam search solution as a warm start. The method big-M5 has some large optimality gaps because Gurobi couldn't finish solving the root relaxation within 1h.

### G.3.5 Synthetic 1 Benchmark with $\lambda_2 = 10.0$ with No Warmstart

| # of features | method | upper bound | time(s) | gap(%) |
|---|---|---|---|---|
| 100 | big-M50 | $-2.10868 \cdot 10^6$ | **0.28** | **0.00** |
| 100 | big-M20 | $-2.10868 \cdot 10^6$ | 0.44 | 0.00 |
| 100 | big-M5 | $-2.10868 \cdot 10^6$ | 0.32 | 0.00 |
| 100 | ours | $-2.10868 \cdot 10^6$ | 0.30 | 0.00 |
| 500 | big-M50 | $-2.12819 \cdot 10^6$ | 42.85 | 0.00 |
| 500 | big-M20 | $-2.12823 \cdot 10^6$ | 28.73 | 0.00 |
| 500 | big-M5 | $-2.12823 \cdot 10^6$ | 57.10 | 0.00 |
| 500 | ours | $-2.12823 \cdot 10^6$ | **0.38** | **0.00** |
| 1000 | big-M50 | $-2.10995 \cdot 10^6$ | 1072.15 | 0.00 |
| 1000 | big-M20 | $-2.10995 \cdot 10^6$ | 451.79 | 0.00 |
| 1000 | big-M5 | $-2.10995 \cdot 10^6$ | 904.01 | 0.00 |
| 1000 | ours | $-2.10995 \cdot 10^6$ | **0.57** | **0.00** |
| 3000 | big-M50 | $-7.92671 \cdot 10^5$ | 3605.01 | 166.77 |
| 3000 | big-M20 | $-9.76712 \cdot 10^5$ | 3602.25 | 116.51 |
| 3000 | big-M5 | - | 3623.24 | - |
| 3000 | ours | $-2.10220 \cdot 10^6$ | **1.68** | **0.00** |
| 5000 | big-M50 | $-7.89596 \cdot 10^5$ | 3617.52 | 168.16 |
| 5000 | big-M20 | $-9.77684 \cdot 10^5$ | 3612.02 | 116.57 |
| 5000 | big-M5 | $-1.87959 \cdot 10^6$ | 3600.23 | 12.65 |
| 5000 | ours | $-2.09625 \cdot 10^6$ | **5.09** | **0.00** |

Table 81: Comparison of time and optimality gap on the synthetic datasets (n = 100000, k = 10, $\rho = 0.1$, $\lambda_2 = 10.0$).

| # of features | method | upper bound | time(s) | gap(%) |
|---|---|---|---|---|
| 100 | big-M50 | $-5.28713 \cdot 10^6$ | 0.40 | 0.00 |
| 100 | big-M20 | $-5.28716 \cdot 10^6$ | 0.39 | 0.00 |
| 100 | big-M5 | $-5.28716 \cdot 10^6$ | 0.35 | 0.00 |
| 100 | ours | $-5.28716 \cdot 10^6$ | **0.30** | **0.00** |
| 500 | big-M50 | $-5.31819 \cdot 10^6$ | 89.34 | 0.00 |
| 500 | big-M20 | $-5.31835 \cdot 10^6$ | 73.89 | 0.00 |
| 500 | big-M5 | $-5.31842 \cdot 10^6$ | 52.55 | 0.00 |
| 500 | ours | $-5.31842 \cdot 10^6$ | **0.41** | **0.00** |
| 1000 | big-M50 | $-5.26964 \cdot 10^6$ | 1869.66 | 0.00 |
| 1000 | big-M20 | $-5.26964 \cdot 10^6$ | 1241.04 | 0.00 |
| 1000 | big-M5 | $-5.26964 \cdot 10^6$ | 711.84 | 0.00 |
| 1000 | ours | $-5.26964 \cdot 10^6$ | **0.51** | **0.00** |
| 3000 | big-M50 | $-3.96610 \cdot 10^6$ | 3600.32 | 33.83 |
| 3000 | big-M20 | $-3.98698 \cdot 10^6$ | 3601.87 | 33.13 |
| 3000 | big-M5 | - | 3604.15 | - |
| 3000 | ours | $-5.27658 \cdot 10^6$ | **1.76** | **0.00** |
| 5000 | big-M50 | $-3.96757 \cdot 10^6$ | 3607.78 | 34.19 |
| 5000 | big-M20 | $-4.37277 \cdot 10^6$ | 3610.74 | 21.76 |
| 5000 | big-M5 | $-5.27116 \cdot 10^6$ | 3600.14 | 1.00 |
| 5000 | ours | $-5.27116 \cdot 10^6$ | **4.32** | **0.00** |

Table 82: Comparison of time and optimality gap on the synthetic datasets (n = 100000, k = 10, $\rho = 0.3$, $\lambda_2 = 10.0$).

| # of features | method | upper bound | time(s) | gap(%) |
|---|---|---|---|---|
| 100 | big-M50 | $-1.10083 \cdot 10^7$ | 0.44 | 0.00 |
| 100 | big-M20 | $-1.10083 \cdot 10^7$ | 0.37 | 0.00 |
| 100 | big-M5 | $-1.10083 \cdot 10^7$ | 0.33 | 0.00 |
| 100 | ours | $-1.10083 \cdot 10^7$ | **0.32** | **0.00** |
| 500 | big-M50 | $-1.10515 \cdot 10^7$ | 91.37 | 0.00 |
| 500 | big-M20 | $-1.10517 \cdot 10^7$ | 74.71 | 0.00 |
| 500 | big-M5 | $-1.10517 \cdot 10^7$ | 99.37 | 0.00 |
| 500 | ours | $-1.10517 \cdot 10^7$ | **0.41** | **0.00** |
| 1000 | big-M50 | $-1.09505 \cdot 10^7$ | 1534.37 | 0.00 |
| 1000 | big-M20 | $-1.09507 \cdot 10^7$ | 1102.78 | 0.00 |
| 1000 | big-M5 | $-1.09507 \cdot 10^7$ | 814.92 | 0.00 |
| 1000 | ours | $-1.09507 \cdot 10^7$ | **0.55** | **0.00** |
| 3000 | big-M50 | - | 3603.26 | - |
| 3000 | big-M20 | - | 3606.10 | - |
| 3000 | big-M5 | - | 3606.02 | - |
| 3000 | ours | $-1.10011 \cdot 10^7$ | **1.73** | **0.00** |
| 5000 | big-M50 | $-9.62941 \cdot 10^6$ | 3608.37 | 15.34 |
| 5000 | big-M20 | $-1.00932 \cdot 10^7$ | 3602.48 | 10.04 |
| 5000 | big-M5 | - | 3633.80 | - |
| 5000 | ours | $-1.09965 \cdot 10^7$ | **4.24** | **0.00** |

Table 83: Comparison of time and optimality gap on the synthetic datasets (n = 100000, k = 10, $\rho = 0.5$, $\lambda_2 = 10.0$).

| # of features | method | upper bound | time(s) | gap(%) |
|---|---|---|---|---|
| 100 | big-M50 | $-2.43574 \cdot 10^7$ | 0.42 | 0.00 |
| 100 | big-M20 | $-2.43574 \cdot 10^7$ | 0.36 | 0.00 |
| 100 | big-M5 | $-2.43574 \cdot 10^7$ | 0.37 | 0.00 |
| 100 | ours | $-2.43574 \cdot 10^7$ | **0.30** | **0.00** |
| 500 | big-M50 | $-2.44178 \cdot 10^7$ | 85.12 | 0.00 |
| 500 | big-M20 | $-2.44184 \cdot 10^7$ | 70.81 | 0.00 |
| 500 | big-M5 | $-2.44184 \cdot 10^7$ | 53.85 | 0.00 |
| 500 | ours | $-2.44184 \cdot 10^7$ | **0.44** | **0.00** |
| 1000 | big-M50 | $-2.41986 \cdot 10^7$ | 1415.19 | 0.00 |
| 1000 | big-M20 | $-2.41987 \cdot 10^7$ | 878.61 | 0.00 |
| 1000 | big-M5 | $-2.41988 \cdot 10^7$ | 613.98 | 0.00 |
| 1000 | ours | $-2.41988 \cdot 10^7$ | **0.52** | **0.00** |
| 3000 | big-M50 | - | 3615.34 | - |
| 3000 | big-M20 | - | 3632.97 | - |
| 3000 | big-M5 | $-2.43712 \cdot 10^7$ | 3600.25 | 0.56 |
| 3000 | ours | $-2.43712 \cdot 10^7$ | **1.81** | **0.00** |
| 5000 | big-M50 | $-2.43693 \cdot 10^7$ | 3607.06 | 1.00 |
| 5000 | big-M20 | $-2.43693 \cdot 10^7$ | 3603.18 | 1.00 |
| 5000 | big-M5 | - | 3626.02 | - |
| 5000 | ours | $-2.43693 \cdot 10^7$ | **4.08** | **0.00** |

Table 84: Comparison of time and optimality gap on the synthetic datasets (n = 100000, k = 10, $\rho = 0.7$, $\lambda_2 = 10.0$).

| # of features | method | upper bound | time(s) | gap(%) |
|---|---|---|---|---|
| 100 | big-M50 | $-9.11025 \cdot 10^7$ | 0.17 | 0.00 |
| 100 | big-M20 | $-9.11025 \cdot 10^7$ | 0.12 | 0.00 |
| 100 | big-M5 | $-9.11025 \cdot 10^7$ | **0.10** | **0.00** |
| 100 | ours | $-9.11025 \cdot 10^7$ | 0.32 | 0.00 |
| 500 | big-M50 | $-9.12046 \cdot 10^7$ | 117.86 | 0.00 |
| 500 | big-M20 | $-9.12046 \cdot 10^7$ | 128.18 | 0.00 |
| 500 | big-M5 | $-9.12046 \cdot 10^7$ | 184.18 | 0.00 |
| 500 | ours | $-9.12046 \cdot 10^7$ | **0.38** | **0.00** |
| 1000 | big-M50 | $-9.04074 \cdot 10^7$ | 3600.04 | 0.01 |
| 1000 | big-M20 | $-9.04074 \cdot 10^7$ | 3600.17 | 0.20 |
| 1000 | big-M5 | $-9.04074 \cdot 10^7$ | 2000.72 | 0.00 |
| 1000 | ours | $-9.04074 \cdot 10^7$ | **0.52** | **0.00** |
| 3000 | big-M50 | - | 3611.90 | - |
| 3000 | big-M20 | - | 3625.57 | - |
| 3000 | big-M5 | - | 3607.35 | - |
| 3000 | ours | $-9.12760 \cdot 10^7$ | **20.84** | **0.00** |
| 5000 | big-M50 | - | 3615.45 | - |
| 5000 | big-M20 | - | 3613.78 | - |
| 5000 | big-M5 | - | 3613.28 | - |
| 5000 | ours | $-9.12915 \cdot 10^7$ | **57.41** | **0.00** |

Table 85: Comparison of time and optimality gap on the synthetic datasets (n = 100000, k = 10, $\rho = 0.9$, $\lambda_2 = 10.0$).

### G.3.6 Synthetic 1 Benchmark with $\lambda_2 = 10.0$ with Warmstart

| # of features | method | upper bound | time(s) | gap(%) |
|---|---|---|---|---|
| 100 | big-M50 + warm start | $-2.10868 \cdot 10^6$ | 0.25 | 0.00 |
| 100 | big-M20 + warm start | $-2.10868 \cdot 10^6$ | 0.17 | 0.00 |
| 100 | big-M5 + warm start | $-2.10868 \cdot 10^6$ | **0.05** | **0.00** |
| 100 | ours | $-2.10868 \cdot 10^6$ | 0.30 | 0.00 |
| 500 | big-M50 + warm start | $-2.12823 \cdot 10^6$ | 3.76 | 0.00 |
| 500 | big-M20 + warm start | $-2.12823 \cdot 10^6$ | 2.68 | 0.00 |
| 500 | big-M5 + warm start | $-2.12823 \cdot 10^6$ | 2.17 | 0.00 |
| 500 | ours | $-2.12823 \cdot 10^6$ | **0.38** | **0.00** |
| 1000 | big-M50 + warm start | $-2.10995 \cdot 10^6$ | 23.00 | 0.00 |
| 1000 | big-M20 + warm start | $-2.10995 \cdot 10^6$ | 18.30 | 0.00 |
| 1000 | big-M5 + warm start | $-2.10995 \cdot 10^6$ | 26.24 | 0.00 |
| 1000 | ours | $-2.10995 \cdot 10^6$ | **0.57** | **0.00** |
| 3000 | big-M50 + warm start | $-2.10220 \cdot 10^6$ | 241.10 | 0.00 |
| 3000 | big-M20 + warm start | $-2.10220 \cdot 10^6$ | 302.29 | 0.00 |
| 3000 | big-M5 + warm start | $-2.10220 \cdot 10^6$ | 273.21 | 0.00 |
| 3000 | ours | $-2.10220 \cdot 10^6$ | **1.68** | **0.00** |
| 5000 | big-M50 + warm start | $-2.09625 \cdot 10^6$ | 1309.75 | 0.00 |
| 5000 | big-M20 + warm start | $-2.09625 \cdot 10^6$ | 1142.75 | 0.00 |
| 5000 | big-M5 + warm start | $-2.09625 \cdot 10^6$ | 1636.89 | 0.00 |
| 5000 | ours | $-2.09625 \cdot 10^6$ | **5.09** | **0.00** |

Table 86: Comparison of time and optimality gap on the synthetic datasets (n = 100000, k = 10, $\rho = 0.1$, $\lambda_2 = 10.0$). All baselines use our beam search solution as a warm start.

| # of features | method | upper bound | time(s) | gap(%) |
|---|---|---|---|---|
| 100 | big-M50 + warm start | $-5.28716 \cdot 10^6$ | 0.25 | 0.00 |
| 100 | big-M20 + warm start | $-5.28716 \cdot 10^6$ | **0.05** | **0.00** |
| 100 | big-M5 + warm start | $-5.28716 \cdot 10^6$ | 0.05 | 0.00 |
| 100 | ours | $-5.28716 \cdot 10^6$ | 0.30 | 0.00 |
| 500 | big-M50 + warm start | $-5.31842 \cdot 10^6$ | 2.93 | 0.00 |
| 500 | big-M20 + warm start | $-5.31842 \cdot 10^6$ | 2.99 | 0.00 |
| 500 | big-M5 + warm start | $-5.31842 \cdot 10^6$ | 1.86 | 0.00 |
| 500 | ours | $-5.31842 \cdot 10^6$ | **0.41** | **0.00** |
| 1000 | big-M50 + warm start | $-5.26964 \cdot 10^6$ | 15.10 | 0.00 |
| 1000 | big-M20 + warm start | $-5.26964 \cdot 10^6$ | 14.76 | 0.00 |
| 1000 | big-M5 + warm start | $-5.26964 \cdot 10^6$ | 11.79 | 0.00 |
| 1000 | ours | $-5.26964 \cdot 10^6$ | **0.51** | **0.00** |
| 3000 | big-M50 + warm start | $-5.27658 \cdot 10^6$ | 315.30 | 0.00 |
| 3000 | big-M20 + warm start | $-5.27658 \cdot 10^6$ | 390.22 | 0.00 |
| 3000 | big-M5 + warm start | $-5.27658 \cdot 10^6$ | 249.80 | 0.00 |
| 3000 | ours | $-5.27658 \cdot 10^6$ | **1.76** | **0.00** |
| 5000 | big-M50 + warm start | $-5.27116 \cdot 10^6$ | 1100.87 | 0.00 |
| 5000 | big-M20 + warm start | $-5.27116 \cdot 10^6$ | 1014.36 | 0.00 |
| 5000 | big-M5 + warm start | $-5.27116 \cdot 10^6$ | 3627.13 | 1.00 |
| 5000 | ours | $-5.27116 \cdot 10^6$ | **4.32** | **0.00** |

Table 87: Comparison of time and optimality gap on the synthetic datasets (n = 100000, k = 10, $\rho = 0.3$, $\lambda_2 = 10.0$). All baselines use our beam search solution as a warm start.

| # of features | method | upper bound | time(s) | gap(%) |
|---|---|---|---|---|
| 100 | big-M50 + warm start | $-1.10083 \cdot 10^7$ | 0.06 | 0.00 |
| 100 | big-M20 + warm start | $-1.10083 \cdot 10^7$ | 0.05 | 0.00 |
| 100 | big-M5 + warm start | $-1.10083 \cdot 10^7$ | **0.05** | **0.00** |
| 100 | ours | $-1.10083 \cdot 10^7$ | 0.32 | 0.00 |
| 500 | big-M50 + warm start | $-1.10517 \cdot 10^7$ | 8.51 | 0.00 |
| 500 | big-M20 + warm start | $-1.10517 \cdot 10^7$ | 6.01 | 0.00 |
| 500 | big-M5 + warm start | $-1.10517 \cdot 10^7$ | 13.97 | 0.00 |
| 500 | ours | $-1.10517 \cdot 10^7$ | **0.41** | **0.00** |
| 1000 | big-M50 + warm start | $-1.09507 \cdot 10^7$ | 14.03 | 0.00 |
| 1000 | big-M20 + warm start | $-1.09507 \cdot 10^7$ | 15.08 | 0.00 |
| 1000 | big-M5 + warm start | $-1.09507 \cdot 10^7$ | 11.81 | 0.00 |
| 1000 | ours | $-1.09507 \cdot 10^7$ | **0.55** | **0.00** |
| 3000 | big-M50 + warm start | $-1.10011 \cdot 10^7$ | 201.32 | 0.00 |
| 3000 | big-M20 + warm start | $-1.10011 \cdot 10^7$ | 292.08 | 0.00 |
| 3000 | big-M5 + warm start | $-1.10011 \cdot 10^7$ | 241.63 | 0.00 |
| 3000 | ours | $-1.10011 \cdot 10^7$ | **1.73** | **0.00** |
| 5000 | big-M50 + warm start | $-1.09965 \cdot 10^7$ | 3601.20 | 1.00 |
| 5000 | big-M20 + warm start | $-1.09965 \cdot 10^7$ | 3623.71 | 1.00 |
| 5000 | big-M5 + warm start | $-1.09965 \cdot 10^7$ | 3629.56 | 570166517.91 |
| 5000 | ours | $-1.09965 \cdot 10^7$ | **4.24** | **0.00** |

Table 88: Comparison of time and optimality gap on the synthetic datasets (n = 100000, k = 10, $\rho = 0.5$, $\lambda_2 = 10.0$). All baselines use our beam search solution as a warm start. The method big-M5 has some large optimality gaps because Gurobi couldn't finish solving the root relaxation within 1h.

| # of features | method | upper bound | time(s) | gap(%) |
|---|---|---|---|---|
| 100 | big-M50 + warm start | $-2.43574 \cdot 10^7$ | 0.16 | 0.00 |
| 100 | big-M20 + warm start | $-2.43574 \cdot 10^7$ | 0.17 | 0.00 |
| 100 | big-M5 + warm start | $-2.43574 \cdot 10^7$ | **0.04** | **0.00** |
| 100 | ours | $-2.43574 \cdot 10^7$ | 0.30 | 0.00 |
| 500 | big-M50 + warm start | $-2.44184 \cdot 10^7$ | 2.11 | 0.00 |
| 500 | big-M20 + warm start | $-2.44184 \cdot 10^7$ | 11.80 | 0.00 |
| 500 | big-M5 + warm start | $-2.44184 \cdot 10^7$ | 1.69 | 0.00 |
| 500 | ours | $-2.44184 \cdot 10^7$ | **0.44** | **0.00** |
| 1000 | big-M50 + warm start | $-2.41988 \cdot 10^7$ | 12.39 | 0.00 |
| 1000 | big-M20 + warm start | $-2.41988 \cdot 10^7$ | 13.36 | 0.00 |
| 1000 | big-M5 + warm start | $-2.41988 \cdot 10^7$ | 10.59 | 0.00 |
| 1000 | ours | $-2.41988 \cdot 10^7$ | **0.52** | **0.00** |
| 3000 | big-M50 + warm start | $-2.43712 \cdot 10^7$ | 3600.43 | 0.59 |
| 3000 | big-M20 + warm start | $-2.43712 \cdot 10^7$ | 3600.08 | 0.59 |
| 3000 | big-M5 + warm start | $-2.43712 \cdot 10^7$ | 3600.72 | 0.56 |
| 3000 | ours | $-2.43712 \cdot 10^7$ | **1.81** | **0.00** |
| 5000 | big-M50 + warm start | $-2.43693 \cdot 10^7$ | 3624.24 | 1.00 |
| 5000 | big-M20 + warm start | $-2.43693 \cdot 10^7$ | 3603.69 | 1.00 |
| 5000 | big-M5 + warm start | $-2.43693 \cdot 10^7$ | 3646.36 | 600320013.98 |
| 5000 | ours | $-2.43693 \cdot 10^7$ | **4.08** | **0.00** |

Table 89: Comparison of time and optimality gap on the synthetic datasets (n = 100000, k = 10, $\rho = 0.7$, $\lambda_2 = 10.0$). All baselines use our beam search solution as a warm start. The method big-M5 has some large optimality gaps because Gurobi couldn't finish solving the root relaxation within 1h.

| # of features | method | upper bound | time(s) | gap(%) |
|---|---|---|---|---|
| 100 | big-M50 + warm start | $-9.11025 \cdot 10^7$ | 0.04 | 0.00 |
| 100 | big-M20 + warm start | $-9.11025 \cdot 10^7$ | **0.04** | **0.00** |
| 100 | big-M5 + warm start | $-9.11025 \cdot 10^7$ | 0.04 | 0.00 |
| 100 | ours | $-9.11025 \cdot 10^7$ | 0.32 | 0.00 |
| 500 | big-M50 + warm start | $-9.12046 \cdot 10^7$ | 36.63 | 0.00 |
| 500 | big-M20 + warm start | $-9.12046 \cdot 10^7$ | 28.85 | 0.00 |
| 500 | big-M5 + warm start | $-9.12046 \cdot 10^7$ | 18.35 | 0.00 |
| 500 | ours | $-9.12046 \cdot 10^7$ | **0.38** | **0.00** |
| 1000 | big-M50 + warm start | $-9.04074 \cdot 10^7$ | 3600.19 | 0.06 |
| 1000 | big-M20 + warm start | $-9.04074 \cdot 10^7$ | 3600.33 | 0.20 |
| 1000 | big-M5 + warm start | $-9.04074 \cdot 10^7$ | 1545.53 | 0.00 |
| 1000 | ours | $-9.04074 \cdot 10^7$ | **0.52** | **0.00** |
| 3000 | big-M50 + warm start | $-9.12760 \cdot 10^7$ | 3606.60 | 0.59 |
| 3000 | big-M20 + warm start | $-9.12760 \cdot 10^7$ | 3603.80 | 0.59 |
| 3000 | big-M5 + warm start | $-9.12760 \cdot 10^7$ | 3605.53 | 222998156.52 |
| 3000 | ours | $-9.12760 \cdot 10^7$ | **20.84** | **0.00** |
| 5000 | big-M50 + warm start | $-9.12915 \cdot 10^7$ | 3606.47 | 1.00 |
| 5000 | big-M20 + warm start | $-9.12915 \cdot 10^7$ | 3602.91 | 1.00 |
| 5000 | big-M5 + warm start | $-9.12915 \cdot 10^7$ | 3634.46 | 618134564.44 |
| 5000 | ours | $-9.12915 \cdot 10^7$ | **57.41** | **0.00** |

Table 90: Comparison of time and optimality gap on the synthetic datasets (n = 100000, k = 10, $\rho = 0.9$, $\lambda_2 = 10.0$). All baselines use our beam search solution as a warm start. The method big-M5 has some large optimality gaps because Gurobi couldn't finish solving the root relaxation within 1h.

### G.3.7 Synthetic 2 Benchmark with $\lambda_2 = 0.001$ with No Warmstart

| # of samples | method | upper bound | time(s) | gap(%) |
|---|---|---|---|---|
| 3000 | big-M50 | $-6.34585 \cdot 10^4$ | 3600.07 | 19.16 |
| 3000 | big-M20 | $-6.34585 \cdot 10^4$ | 3600.07 | 18.11 |
| 3000 | big-M5 | $-6.34585 \cdot 10^4$ | **2252.12** | **0.00** |
| 3000 | ours | $-6.34585 \cdot 10^4$ | 3600.49 | 19.45 |
| 4000 | big-M50 | $-8.64294 \cdot 10^4$ | 3600.07 | 15.32 |
| 4000 | big-M20 | $-8.64294 \cdot 10^4$ | 3600.09 | 14.37 |
| 4000 | big-M5 | $-8.64294 \cdot 10^4$ | 1536.42 | 0.00 |
| 4000 | ours | $-8.64294 \cdot 10^4$ | **20.25** | **0.00** |
| 5000 | big-M50 | $-1.09576 \cdot 10^5$ | 3600.81 | 10.25 |
| 5000 | big-M20 | $-1.09576 \cdot 10^5$ | 3600.09 | 11.72 |
| 5000 | big-M5 | $-1.09576 \cdot 10^5$ | 1154.40 | 0.00 |
| 5000 | ours | $-1.09576 \cdot 10^5$ | **20.51** | **0.00** |
| 6000 | big-M50 | $-1.27643 \cdot 10^5$ | 3600.08 | 9.93 |
| 6000 | big-M20 | $-1.27643 \cdot 10^5$ | 3600.59 | 9.92 |
| 6000 | big-M5 | $-1.27643 \cdot 10^5$ | 1018.24 | 0.00 |
| 6000 | ours | $-1.27643 \cdot 10^5$ | **1.70** | **0.00** |
| 7000 | big-M50 | $-1.50614 \cdot 10^5$ | 3600.09 | 8.29 |
| 7000 | big-M20 | $-1.50614 \cdot 10^5$ | 3600.07 | 5.79 |
| 7000 | big-M5 | $-1.50614 \cdot 10^5$ | 851.91 | 0.00 |
| 7000 | ours | $-1.50614 \cdot 10^5$ | **1.67** | **0.00** |

Table 91: Comparison of time and optimality gap on the synthetic datasets (p = 3000, k = 10, $\rho = 0.1$, $\lambda_2 = 0.001$).

| # of samples | method | upper bound | time(s) | gap(%) |
|---|---|---|---|---|
| 3000 | big-M50 | $-1.59509 \cdot 10^5$ | 3600.08 | 19.07 |
| 3000 | big-M20 | $-1.59509 \cdot 10^5$ | 3600.67 | 17.35 |
| 3000 | big-M5 | $-1.59509 \cdot 10^5$ | **3600.06** | **6.40** |
| 3000 | ours | $-1.59509 \cdot 10^5$ | 3600.67 | 19.53 |
| 4000 | big-M50 | $-2.17574 \cdot 10^5$ | 3600.07 | 14.78 |
| 4000 | big-M20 | $-2.17574 \cdot 10^5$ | 3600.14 | 14.34 |
| 4000 | big-M5 | $-2.17574 \cdot 10^5$ | 3600.40 | 4.63 |
| 4000 | ours | $-2.17574 \cdot 10^5$ | **100.13** | **0.00** |
| 5000 | big-M50 | $-2.71986 \cdot 10^5$ | 3600.07 | 11.22 |
| 5000 | big-M20 | $-2.71986 \cdot 10^5$ | 3600.07 | 11.12 |
| 5000 | big-M5 | $-2.71986 \cdot 10^5$ | 2568.73 | 0.00 |
| 5000 | ours | $-2.71986 \cdot 10^5$ | **20.80** | **0.00** |
| 6000 | big-M50 | $-3.16761 \cdot 10^5$ | 3600.09 | 9.96 |
| 6000 | big-M20 | $-3.16761 \cdot 10^5$ | 3600.06 | 9.95 |
| 6000 | big-M5 | $-3.16761 \cdot 10^5$ | 1834.08 | 0.00 |
| 6000 | ours | $-3.16761 \cdot 10^5$ | **20.70** | **0.00** |
| 7000 | big-M50 | $-3.72610 \cdot 10^5$ | 3601.19 | 8.33 |
| 7000 | big-M20 | $-3.72610 \cdot 10^5$ | 3600.40 | 8.13 |
| 7000 | big-M5 | $-3.72610 \cdot 10^5$ | 1682.96 | 0.00 |
| 7000 | ours | $-3.72610 \cdot 10^5$ | **1.98** | **0.00** |

Table 92: Comparison of time and optimality gap on the synthetic datasets (p = 3000, k = 10, $\rho = 0.3$, $\lambda_2 = 0.001$).

| # of samples | method | upper bound | time(s) | gap(%) |
|---|---|---|---|---|
| 3000 | big-M50 | $-3.33604 \cdot 10^5$ | 3600.08 | 18.79 |
| 3000 | big-M20 | $-3.31111 \cdot 10^5$ | 3600.31 | 16.99 |
| 3000 | big-M5 | $-3.33604 \cdot 10^5$ | **3600.23** | **7.53** |
| 3000 | ours | $-3.33604 \cdot 10^5$ | 3600.58 | 19.57 |
| 4000 | big-M50 | $-4.53261 \cdot 10^5$ | 3600.07 | 14.98 |
| 4000 | big-M20 | $-4.53261 \cdot 10^5$ | 3600.48 | 13.50 |
| 4000 | big-M5 | $-4.53261 \cdot 10^5$ | 3600.17 | 6.23 |
| 4000 | ours | $-4.53261 \cdot 10^5$ | **3600.25** | **0.63** |
| 5000 | big-M50 | $-5.63007 \cdot 10^5$ | 3601.19 | 11.68 |
| 5000 | big-M20 | $-5.63007 \cdot 10^5$ | 3600.05 | 11.25 |
| 5000 | big-M5 | $-5.63007 \cdot 10^5$ | 3600.07 | 3.85 |
| 5000 | ours | $-5.63007 \cdot 10^5$ | **36.97** | **0.00** |
| 6000 | big-M50 | $-6.57799 \cdot 10^5$ | 3600.34 | 9.98 |
| 6000 | big-M20 | $-6.57799 \cdot 10^5$ | 3600.43 | 9.36 |
| 6000 | big-M5 | $-6.57799 \cdot 10^5$ | 3600.05 | 2.87 |
| 6000 | ours | $-6.57799 \cdot 10^5$ | **20.75** | **0.00** |
| 7000 | big-M50 | $-7.72267 \cdot 10^5$ | 3600.11 | 8.33 |
| 7000 | big-M20 | $-7.72267 \cdot 10^5$ | 3600.96 | 7.85 |
| 7000 | big-M5 | $-7.72267 \cdot 10^5$ | 3600.06 | 2.40 |
| 7000 | ours | $-7.72267 \cdot 10^5$ | **21.02** | **0.00** |

Table 93: Comparison of time and optimality gap on the synthetic datasets (p = 3000, k = 10, $\rho = 0.5$, $\lambda_2 = 0.001$).

| # of samples | method | upper bound | time(s) | gap(%) |
|---|---|---|---|---|
| 3000 | big-M50 | $-7.41187 \cdot 10^5$ | 3600.05 | 18.15 |
| 3000 | big-M20 | $-7.41187 \cdot 10^5$ | 3600.20 | 14.15 |
| 3000 | big-M5 | $-7.41187 \cdot 10^5$ | **3600.22** | **5.94** |
| 3000 | ours | $-7.41187 \cdot 10^5$ | 3601.14 | 19.60 |
| 4000 | big-M50 | $-1.00103 \cdot 10^6$ | 3600.07 | 15.19 |
| 4000 | big-M20 | $-1.00287 \cdot 10^6$ | 3600.21 | 11.81 |
| 4000 | big-M5 | $-1.00287 \cdot 10^6$ | 3600.11 | 5.06 |
| 4000 | ours | $-1.00287 \cdot 10^6$ | **3600.37** | **1.88** |
| 5000 | big-M50 | $-1.24027 \cdot 10^6$ | 3601.10 | 11.88 |
| 5000 | big-M20 | $-1.24027 \cdot 10^6$ | 3601.01 | 10.26 |
| 5000 | big-M5 | $-1.24027 \cdot 10^6$ | 3600.19 | 4.34 |
| 5000 | ours | $-1.24027 \cdot 10^6$ | **466.42** | **0.00** |
| 6000 | big-M50 | $-1.45417 \cdot 10^6$ | 3600.11 | 9.78 |
| 6000 | big-M20 | $-1.45417 \cdot 10^6$ | 3600.08 | 9.11 |
| 6000 | big-M5 | $-1.45417 \cdot 10^6$ | 3600.19 | 3.94 |
| 6000 | ours | $-1.45417 \cdot 10^6$ | **57.67** | **0.00** |
| 7000 | big-M50 | $-1.70483 \cdot 10^6$ | 3601.55 | 8.34 |
| 7000 | big-M20 | $-1.70483 \cdot 10^6$ | 3600.10 | 7.77 |
| 7000 | big-M5 | $-1.70483 \cdot 10^6$ | 3600.38 | 3.39 |
| 7000 | ours | $-1.70483 \cdot 10^6$ | **21.42** | **0.00** |

Table 94: Comparison of time and optimality gap on the synthetic datasets (p = 3000, k = 10, $\rho = 0.7$, $\lambda_2 = 0.001$).

| # of samples | method | upper bound | time(s) | gap(%) |
|---|---|---|---|---|
| 3000 | big-M50 | $-2.78521 \cdot 10^6$ | 3600.15 | 15.90 |
| 3000 | big-M20 | $-2.78521 \cdot 10^6$ | 3600.05 | 10.37 |
| 3000 | big-M5 | $-2.78521 \cdot 10^6$ | **3600.36** | **3.69** |
| 3000 | ours | $-2.78521 \cdot 10^6$ | 3600.01 | 19.60 |
| 4000 | big-M50 | $-3.75202 \cdot 10^6$ | 3600.06 | 13.33 |
| 4000 | big-M20 | $-3.75203 \cdot 10^6$ | 3600.28 | 8.64 |
| 4000 | big-M5 | $-3.75132 \cdot 10^6$ | **3600.37** | **3.18** |
| 4000 | ours | $-3.75203 \cdot 10^6$ | 3600.42 | 3.50 |
| 5000 | big-M50 | $-4.61877 \cdot 10^6$ | 3600.19 | 11.37 |
| 5000 | big-M20 | $-4.61877 \cdot 10^6$ | 3600.05 | 7.62 |
| 5000 | big-M5 | $-4.61877 \cdot 10^6$ | 3600.06 | 2.72 |
| 5000 | ours | $-4.61877 \cdot 10^6$ | **3600.57** | **0.93** |
| 6000 | big-M50 | $-5.43826 \cdot 10^6$ | 3600.06 | 9.77 |
| 6000 | big-M20 | $-5.43826 \cdot 10^6$ | 3600.07 | 6.92 |
| 6000 | big-M5 | $-5.43826 \cdot 10^6$ | 3600.15 | 2.50 |
| 6000 | ours | $-5.43826 \cdot 10^6$ | **3600.65** | **0.33** |
| 7000 | big-M50 | $-6.36767 \cdot 10^6$ | 3600.38 | 8.22 |
| 7000 | big-M20 | $-6.36767 \cdot 10^6$ | 3600.05 | 6.06 |
| 7000 | big-M5 | $-6.36767 \cdot 10^6$ | 3600.21 | 2.23 |
| 7000 | ours | $-6.36767 \cdot 10^6$ | **3600.78** | **0.11** |

Table 95: Comparison of time and optimality gap on the synthetic datasets (p = 3000, k = 10, $\rho = 0.9$, $\lambda_2 = 0.001$).

### G.3.8    Synthetic 2 Benchmark with $\lambda_2 = 0.001$ with Warmstart

| # of samples | method | upper bound | time(s) | gap(%) |
|:---:|:---:|:---:|:---:|:---:|
| 3000 | big-M50 + warm start | $-6.34585 \cdot 10^4$ | 3601.20 | 19.16 |
| 3000 | big-M20 + warm start | $-6.34585 \cdot 10^4$ | 3600.06 | 18.11 |
| 3000 | big-M5 + warm start | $-6.34585 \cdot 10^4$ | **2435.86** | **0.00** |
| 3000 | ours | $-6.34585 \cdot 10^4$ | 3600.49 | 19.45 |
| 4000 | big-M50 + warm start | $-8.64294 \cdot 10^4$ | 3600.87 | 15.32 |
| 4000 | big-M20 + warm start | $-8.64294 \cdot 10^4$ | 3600.12 | 14.37 |
| 4000 | big-M5 + warm start | $-8.64294 \cdot 10^4$ | 1618.73 | 0.00 |
| 4000 | ours | $-8.64294 \cdot 10^4$ | **20.25** | **0.00** |
| 5000 | big-M50 + warm start | $-1.09576 \cdot 10^5$ | 3601.86 | 10.25 |
| 5000 | big-M20 + warm start | $-1.09576 \cdot 10^5$ | 3601.59 | 11.72 |
| 5000 | big-M5 + warm start | $-1.09576 \cdot 10^5$ | 1530.18 | 0.00 |
| 5000 | ours | $-1.09576 \cdot 10^5$ | **20.51** | **0.00** |
| 6000 | big-M50 + warm start | $-1.27643 \cdot 10^5$ | 3601.14 | 9.93 |
| 6000 | big-M20 + warm start | $-1.27643 \cdot 10^5$ | 3600.64 | 9.82 |
| 6000 | big-M5 + warm start | $-1.27643 \cdot 10^5$ | 972.83 | 0.00 |
| 6000 | ours | $-1.27643 \cdot 10^5$ | **1.70** | **0.00** |
| 7000 | big-M50 + warm start | $-1.50614 \cdot 10^5$ | 3600.13 | 8.29 |
| 7000 | big-M20 + warm start | $-1.50614 \cdot 10^5$ | 3600.94 | 5.81 |
| 7000 | big-M5 + warm start | $-1.50614 \cdot 10^5$ | 824.89 | 0.00 |
| 7000 | ours | $-1.50614 \cdot 10^5$ | **1.67** | **0.00** |

Table 96: Comparison of time and optimality gap on the synthetic datasets (p = 3000, k = 10, $\rho = 0.1$, $\lambda_2 = 0.001$). All baselines use our beam search solution as a warm start.

| # of samples | method | upper bound | time(s) | gap(%) |
|---|---|---|---|---|
| 3000 | big-M50 + warm start | $-1.59509 \cdot 10^5$ | 3600.60 | 19.07 |
| 3000 | big-M20 + warm start | $-1.59509 \cdot 10^5$ | 3600.07 | 17.35 |
| 3000 | big-M5 + warm start | $-1.59509 \cdot 10^5$ | **3600.05** | **6.40** |
| 3000 | ours | $-1.59509 \cdot 10^5$ | 3600.67 | 19.53 |
| 4000 | big-M50 + warm start | $-2.17574 \cdot 10^5$ | 3600.10 | 14.78 |
| 4000 | big-M20 + warm start | $-2.17574 \cdot 10^5$ | 3600.68 | 14.34 |
| 4000 | big-M5 + warm start | $-2.17574 \cdot 10^5$ | 3600.04 | 4.63 |
| 4000 | ours | $-2.17574 \cdot 10^5$ | **100.13** | **0.00** |
| 5000 | big-M50 + warm start | $-2.71986 \cdot 10^5$ | 3600.06 | 11.22 |
| 5000 | big-M20 + warm start | $-2.71986 \cdot 10^5$ | 3601.05 | 11.12 |
| 5000 | big-M5 + warm start | $-2.71986 \cdot 10^5$ | 2571.91 | 0.00 |
| 5000 | ours | $-2.71986 \cdot 10^5$ | **20.80** | **0.00** |
| 6000 | big-M50 + warm start | $-3.16761 \cdot 10^5$ | 3600.07 | 9.96 |
| 6000 | big-M20 + warm start | $-3.16761 \cdot 10^5$ | 3600.65 | 9.95 |
| 6000 | big-M5 + warm start | $-3.16761 \cdot 10^5$ | 1861.02 | 0.00 |
| 6000 | ours | $-3.16761 \cdot 10^5$ | **20.70** | **0.00** |
| 7000 | big-M50 + warm start | $-3.72610 \cdot 10^5$ | 3600.08 | 8.33 |
| 7000 | big-M20 + warm start | $-3.72610 \cdot 10^5$ | 3600.41 | 8.13 |
| 7000 | big-M5 + warm start | $-3.72610 \cdot 10^5$ | 1752.62 | 0.00 |
| 7000 | ours | $-3.72610 \cdot 10^5$ | **1.98** | **0.00** |

Table 97: Comparison of time and optimality gap on the synthetic datasets (p = 3000, k = 10, $\rho = 0.3$, $\lambda_2 = 0.001$). All baselines use our beam search solution as a warm start.

| # of samples | method | upper bound | time(s) | gap(%) |
|---|---|---|---|---|
| 3000 | big-M50 + warm start | $-3.33604 \cdot 10^5$ | 3600.17 | 18.79 |
| 3000 | big-M20 + warm start | $-3.33604 \cdot 10^5$ | 3600.12 | 16.11 |
| 3000 | big-M5 + warm start | $-3.33604 \cdot 10^5$ | **3600.20** | **7.53** |
| 3000 | ours | $-3.33604 \cdot 10^5$ | 3600.58 | 19.57 |
| 4000 | big-M50 + warm start | $-4.53261 \cdot 10^5$ | 3600.17 | 14.98 |
| 4000 | big-M20 + warm start | $-4.53261 \cdot 10^5$ | 3600.77 | 13.50 |
| 4000 | big-M5 + warm start | $-4.53261 \cdot 10^5$ | 3600.23 | 6.23 |
| 4000 | ours | $-4.53261 \cdot 10^5$ | **3600.25** | **0.63** |
| 5000 | big-M50 + warm start | $-5.63007 \cdot 10^5$ | 3600.29 | 11.70 |
| 5000 | big-M20 + warm start | $-5.63007 \cdot 10^5$ | 3600.07 | 11.25 |
| 5000 | big-M5 + warm start | $-5.63007 \cdot 10^5$ | 3600.32 | 3.38 |
| 5000 | ours | $-5.63007 \cdot 10^5$ | **36.97** | **0.00** |
| 6000 | big-M50 + warm start | $-6.57799 \cdot 10^5$ | 3600.94 | 9.98 |
| 6000 | big-M20 + warm start | $-6.57799 \cdot 10^5$ | 3600.28 | 9.36 |
| 6000 | big-M5 + warm start | $-6.57799 \cdot 10^5$ | 3600.06 | 2.89 |
| 6000 | ours | $-6.57799 \cdot 10^5$ | **20.75** | **0.00** |
| 7000 | big-M50 + warm start | $-7.72267 \cdot 10^5$ | 3600.55 | 8.33 |
| 7000 | big-M20 + warm start | $-7.72267 \cdot 10^5$ | 3600.07 | 7.85 |
| 7000 | big-M5 + warm start | $-7.72267 \cdot 10^5$ | 3600.65 | 2.40 |
| 7000 | ours | $-7.72267 \cdot 10^5$ | **21.02** | **0.00** |

Table 98: Comparison of time and optimality gap on the synthetic datasets (p = 3000, k = 10, $\rho = 0.5$, $\lambda_2 = 0.001$). All baselines use our beam search solution as a warm start.

| # of samples | method | upper bound | time(s) | gap(%) |
|---|---|---|---|---|
| 3000 | big-M50 + warm start | $-7.41187 \cdot 10^5$ | 3600.76 | 18.15 |
| 3000 | big-M20 + warm start | $-7.41187 \cdot 10^5$ | 3600.29 | 14.15 |
| 3000 | big-M5 + warm start | $-7.41187 \cdot 10^5$ | **3600.04** | **5.94** |
| 3000 | ours | $-7.41187 \cdot 10^5$ | 3601.14 | 19.60 |
| 4000 | big-M50 + warm start | $-1.00287 \cdot 10^6$ | 3600.07 | 14.98 |
| 4000 | big-M20 + warm start | $-1.00287 \cdot 10^6$ | 3600.10 | 11.81 |
| 4000 | big-M5 + warm start | $-1.00287 \cdot 10^6$ | 3600.04 | 5.06 |
| 4000 | ours | $-1.00287 \cdot 10^6$ | **3600.37** | **1.88** |
| 5000 | big-M50 + warm start | $-1.24027 \cdot 10^6$ | 3600.05 | 11.88 |
| 5000 | big-M20 + warm start | $-1.24027 \cdot 10^6$ | 3600.49 | 10.26 |
| 5000 | big-M5 + warm start | $-1.24027 \cdot 10^6$ | 3600.24 | 4.34 |
| 5000 | ours | $-1.24027 \cdot 10^6$ | **466.42** | **0.00** |
| 6000 | big-M50 + warm start | $-1.45417 \cdot 10^6$ | 3600.45 | 9.78 |
| 6000 | big-M20 + warm start | $-1.45417 \cdot 10^6$ | 3600.26 | 9.06 |
| 6000 | big-M5 + warm start | $-1.45417 \cdot 10^6$ | 3600.07 | 3.94 |
| 6000 | ours | $-1.45417 \cdot 10^6$ | **57.67** | **0.00** |
| 7000 | big-M50 + warm start | $-1.70483 \cdot 10^6$ | 3601.38 | 8.34 |
| 7000 | big-M20 + warm start | $-1.70483 \cdot 10^6$ | 3600.06 | 7.77 |
| 7000 | big-M5 + warm start | $-1.70483 \cdot 10^6$ | 3600.05 | 3.39 |
| 7000 | ours | $-1.70483 \cdot 10^6$ | **21.42** | **0.00** |

Table 99: Comparison of time and optimality gap on the synthetic datasets (p = 3000, k = 10, $\rho = 0.7$, $\lambda_2 = 0.001$). All baselines use our beam search solution as a warm start.

| # of samples | method | upper bound | time(s) | gap(%) |
|---|---|---|---|---|
| 3000 | big-M50 + warm start | $-2.78521 \cdot 10^6$ | 3600.74 | 15.90 |
| 3000 | big-M20 + warm start | $-2.78521 \cdot 10^6$ | 3600.24 | 10.37 |
| 3000 | big-M5 + warm start | $-2.78521 \cdot 10^6$ | **3600.32** | **3.69** |
| 3000 | ours | $-2.78521 \cdot 10^6$ | 3600.01 | 19.60 |
| 4000 | big-M50 + warm start | $-3.75203 \cdot 10^6$ | 3600.07 | 13.33 |
| 4000 | big-M20 + warm start | $-3.75203 \cdot 10^6$ | 3600.24 | 8.64 |
| 4000 | big-M5 + warm start | $-3.75203 \cdot 10^6$ | **3600.16** | **3.16** |
| 4000 | ours | $-3.75203 \cdot 10^6$ | 3600.42 | 3.50 |
| 5000 | big-M50 + warm start | $-4.61877 \cdot 10^6$ | 3601.36 | 11.37 |
| 5000 | big-M20 + warm start | $-4.61877 \cdot 10^6$ | 3600.23 | 7.62 |
| 5000 | big-M5 + warm start | $-4.61877 \cdot 10^6$ | 3600.07 | 2.72 |
| 5000 | ours | $-4.61877 \cdot 10^6$ | **3600.57** | **0.93** |
| 6000 | big-M50 + warm start | $-5.43826 \cdot 10^6$ | 3600.22 | 9.77 |
| 6000 | big-M20 + warm start | $-5.43826 \cdot 10^6$ | 3601.07 | 6.92 |
| 6000 | big-M5 + warm start | $-5.43826 \cdot 10^6$ | 3600.43 | 2.50 |
| 6000 | ours | $-5.43826 \cdot 10^6$ | **3600.65** | **0.33** |
| 7000 | big-M50 + warm start | $-6.36767 \cdot 10^6$ | 3600.12 | 8.22 |
| 7000 | big-M20 + warm start | $-6.36767 \cdot 10^6$ | 3600.57 | 6.06 |
| 7000 | big-M5 + warm start | $-6.36767 \cdot 10^6$ | 3600.04 | 2.23 |
| 7000 | ours | $-6.36767 \cdot 10^6$ | **3600.78** | **0.11** |

Table 100: Comparison of time and optimality gap on the synthetic datasets (p = 3000, k = 10, $\rho = 0.9$, $\lambda_2 = 0.001$). All baselines use our beam search solution as a warm start.

### G.3.9 Synthetic 2 Benchmark with $\lambda_2 = 0.1$ with No Warmstart

| # of samples | method | upper bound | time(s) | gap(%) |
|---|---|---|---|---|
| 3000 | big-M50 | $-6.34575 \cdot 10^4$ | 3600.07 | 19.14 |
| 3000 | big-M20 | $-6.34575 \cdot 10^4$ | 3600.11 | 18.10 |
| 3000 | big-M5 | $-6.34575 \cdot 10^4$ | **2452.71** | **0.00** |
| 3000 | ours | $-6.34575 \cdot 10^4$ | 3600.33 | 19.31 |
| 4000 | big-M50 | $-8.64284 \cdot 10^4$ | 3600.23 | 15.33 |
| 4000 | big-M20 | $-8.64284 \cdot 10^4$ | 3600.11 | 14.04 |
| 4000 | big-M5 | $-8.64284 \cdot 10^4$ | 1675.19 | 0.00 |
| 4000 | ours | $-8.64284 \cdot 10^4$ | **20.81** | **0.00** |
| 5000 | big-M50 | $-1.09575 \cdot 10^5$ | 3600.08 | 11.60 |
| 5000 | big-M20 | $-1.09575 \cdot 10^5$ | 3600.07 | 9.94 |
| 5000 | big-M5 | $-1.09575 \cdot 10^5$ | 1316.91 | 0.00 |
| 5000 | ours | $-1.09575 \cdot 10^5$ | **20.65** | **0.00** |
| 6000 | big-M50 | $-1.27642 \cdot 10^5$ | 3601.02 | 9.92 |
| 6000 | big-M20 | $-1.27642 \cdot 10^5$ | 3600.06 | 9.92 |
| 6000 | big-M5 | $-1.27642 \cdot 10^5$ | 1040.84 | 0.00 |
| 6000 | ours | $-1.27642 \cdot 10^5$ | **1.70** | **0.00** |
| 7000 | big-M50 | $-1.50613 \cdot 10^5$ | 3600.45 | 8.32 |
| 7000 | big-M20 | $-1.50613 \cdot 10^5$ | 3600.10 | 5.74 |
| 7000 | big-M5 | $-1.50613 \cdot 10^5$ | 901.46 | 0.00 |
| 7000 | ours | $-1.50613 \cdot 10^5$ | **1.67** | **0.00** |

Table 101: Comparison of time and optimality gap on the synthetic datasets (p = 3000, k = 10, $\rho = 0.1$, $\lambda_2 = 0.1$).

| # of samples | method | upper bound | time(s) | gap(%) |
|---|---|---|---|---|
| 3000 | big-M50 | $-1.59508 \cdot 10^5$ | 3601.50 | 19.07 |
| 3000 | big-M20 | $-1.59508 \cdot 10^5$ | 3600.41 | 17.35 |
| 3000 | big-M5 | $-1.59508 \cdot 10^5$ | **3600.05** | **6.40** |
| 3000 | ours | $-1.59508 \cdot 10^5$ | 3600.27 | 19.40 |
| 4000 | big-M50 | $-2.17573 \cdot 10^5$ | 3600.23 | 14.78 |
| 4000 | big-M20 | $-2.17573 \cdot 10^5$ | 3600.11 | 14.34 |
| 4000 | big-M5 | $-2.17573 \cdot 10^5$ | 3600.36 | 4.63 |
| 4000 | ours | $-2.17573 \cdot 10^5$ | **99.53** | **0.00** |
| 5000 | big-M50 | $-2.71985 \cdot 10^5$ | 3600.09 | 11.22 |
| 5000 | big-M20 | $-2.71985 \cdot 10^5$ | 3600.06 | 11.29 |
| 5000 | big-M5 | $-2.71985 \cdot 10^5$ | 2986.61 | 0.00 |
| 5000 | ours | $-2.71985 \cdot 10^5$ | **21.01** | **0.00** |
| 6000 | big-M50 | $-3.16760 \cdot 10^5$ | 3600.07 | 9.96 |
| 6000 | big-M20 | $-3.16760 \cdot 10^5$ | 3601.06 | 9.95 |
| 6000 | big-M5 | $-3.16760 \cdot 10^5$ | 2025.19 | 0.00 |
| 6000 | ours | $-3.16760 \cdot 10^5$ | **20.91** | **0.00** |
| 7000 | big-M50 | $-3.72609 \cdot 10^5$ | 3600.80 | 8.33 |
| 7000 | big-M20 | $-3.72609 \cdot 10^5$ | 3600.81 | 8.12 |
| 7000 | big-M5 | $-3.72609 \cdot 10^5$ | 1452.72 | 0.00 |
| 7000 | ours | $-3.72609 \cdot 10^5$ | **1.73** | **0.00** |

Table 102: Comparison of time and optimality gap on the synthetic datasets (p = 3000, k = 10, $\rho = 0.3$, $\lambda_2 = 0.1$).

| # of samples | method | upper bound | time(s) | gap(%) |
|---|---|---|---|---|
| 3000 | big-M50 | $-3.33603 \cdot 10^5$ | 3601.02 | 18.78 |
| 3000 | big-M20 | $-3.33603 \cdot 10^5$ | 3600.70 | 16.11 |
| 3000 | big-M5 | $-3.33603 \cdot 10^5$ | **3600.22** | **7.55** |
| 3000 | ours | $-3.33603 \cdot 10^5$ | 3600.68 | 19.44 |
| 4000 | big-M50 | $-4.53260 \cdot 10^5$ | 3600.12 | 14.95 |
| 4000 | big-M20 | $-4.53260 \cdot 10^5$ | 3600.25 | 13.47 |
| 4000 | big-M5 | $-4.53260 \cdot 10^5$ | 3600.05 | 6.23 |
| 4000 | ours | $-4.53260 \cdot 10^5$ | **3600.41** | **0.62** |
| 5000 | big-M50 | $-5.63006 \cdot 10^5$ | 3600.07 | 11.69 |
| 5000 | big-M20 | $-5.63006 \cdot 10^5$ | 3601.43 | 11.26 |
| 5000 | big-M5 | $-5.63006 \cdot 10^5$ | 3600.31 | 3.51 |
| 5000 | ours | $-5.63006 \cdot 10^5$ | **37.26** | **0.00** |
| 6000 | big-M50 | $-6.57798 \cdot 10^5$ | 3600.53 | 9.98 |
| 6000 | big-M20 | $-6.57798 \cdot 10^5$ | 3600.62 | 9.36 |
| 6000 | big-M5 | $-6.57798 \cdot 10^5$ | 3600.05 | 2.82 |
| 6000 | ours | $-6.57798 \cdot 10^5$ | **20.64** | **0.00** |
| 7000 | big-M50 | $-7.72266 \cdot 10^5$ | 3600.96 | 8.33 |
| 7000 | big-M20 | $-7.72266 \cdot 10^5$ | 3600.57 | 7.80 |
| 7000 | big-M5 | $-7.72266 \cdot 10^5$ | 3600.14 | 2.34 |
| 7000 | ours | $-7.72266 \cdot 10^5$ | **20.91** | **0.00** |

Table 103: Comparison of time and optimality gap on the synthetic datasets (p = 3000, k = 10, $\rho = 0.5$, $\lambda_2 = 0.1$).

| # of samples | method | upper bound | time(s) | gap(%) |
|---|---|---|---|---|
| 3000 | big-M50 | $-7.41186 \cdot 10^5$ | 3600.75 | 18.15 |
| 3000 | big-M20 | $-7.41186 \cdot 10^5$ | 3600.78 | 14.15 |
| 3000 | big-M5 | $-7.41186 \cdot 10^5$ | **3600.34** | **5.92** |
| 3000 | ours | $-7.41186 \cdot 10^5$ | 3600.59 | 19.46 |
| 4000 | big-M50 | $-1.00287 \cdot 10^6$ | 3600.07 | 14.98 |
| 4000 | big-M20 | $-1.00287 \cdot 10^6$ | 3600.06 | 12.06 |
| 4000 | big-M5 | $-1.00287 \cdot 10^6$ | 3600.36 | 5.06 |
| 4000 | ours | $-1.00287 \cdot 10^6$ | **3600.08** | **1.88** |
| 5000 | big-M50 | $-1.24027 \cdot 10^6$ | 3600.07 | 11.88 |
| 5000 | big-M20 | $-1.24027 \cdot 10^6$ | 3600.08 | 10.26 |
| 5000 | big-M5 | $-1.24027 \cdot 10^6$ | 3600.22 | 4.34 |
| 5000 | ours | $-1.24027 \cdot 10^6$ | **469.43** | **0.00** |
| 6000 | big-M50 | $-1.45416 \cdot 10^6$ | 3601.17 | 9.78 |
| 6000 | big-M20 | $-1.45416 \cdot 10^6$ | 3600.60 | 9.11 |
| 6000 | big-M5 | $-1.45416 \cdot 10^6$ | 3600.20 | 3.94 |
| 6000 | ours | $-1.45416 \cdot 10^6$ | **59.16** | **0.00** |
| 7000 | big-M50 | $-1.70483 \cdot 10^6$ | 3602.06 | 8.34 |
| 7000 | big-M20 | $-1.70483 \cdot 10^6$ | 3600.61 | 7.77 |
| 7000 | big-M5 | $-1.70483 \cdot 10^6$ | 3600.22 | 3.39 |
| 7000 | ours | $-1.70483 \cdot 10^6$ | **21.12** | **0.00** |

Table 104: Comparison of time and optimality gap on the synthetic datasets (p = 3000, k = 10, $\rho = 0.7$, $\lambda_2 = 0.1$).

| # of samples | method | upper bound | time(s) | gap(%) |
|---|---|---|---|---|
| 3000 | big-M50 | $-2.78468 \cdot 10^6$ | 3600.31 | 15.93 |
| 3000 | big-M20 | $-2.78521 \cdot 10^6$ | 3600.05 | 10.59 |
| 3000 | big-M5 | $-2.78521 \cdot 10^6$ | **3600.09** | **3.69** |
| 3000 | ours | $-2.78521 \cdot 10^6$ | 3600.23 | 19.46 |
| 4000 | big-M50 | $-3.75203 \cdot 10^6$ | 3600.08 | 13.35 |
| 4000 | big-M20 | $-3.75203 \cdot 10^6$ | 3600.04 | 8.60 |
| 4000 | big-M5 | $-3.75203 \cdot 10^6$ | **3600.14** | **3.16** |
| 4000 | ours | $-3.75203 \cdot 10^6$ | 3601.10 | 3.50 |
| 5000 | big-M50 | $-4.61877 \cdot 10^6$ | 3600.08 | 11.40 |
| 5000 | big-M20 | $-4.61877 \cdot 10^6$ | 3600.05 | 7.62 |
| 5000 | big-M5 | $-4.61877 \cdot 10^6$ | 3600.16 | 2.72 |
| 5000 | ours | $-4.61877 \cdot 10^6$ | **3600.40** | **0.93** |
| 6000 | big-M50 | $-5.43826 \cdot 10^6$ | 3600.23 | 9.79 |
| 6000 | big-M20 | $-5.43826 \cdot 10^6$ | 3600.11 | 6.92 |
| 6000 | big-M5 | $-5.43826 \cdot 10^6$ | 3600.17 | 2.50 |
| 6000 | ours | $-5.43826 \cdot 10^6$ | **3600.26** | **0.33** |
| 7000 | big-M50 | $-6.36767 \cdot 10^6$ | 3600.07 | 8.22 |
| 7000 | big-M20 | $-6.36767 \cdot 10^6$ | 3600.24 | 6.06 |
| 7000 | big-M5 | $-6.36767 \cdot 10^6$ | 3600.05 | 2.25 |
| 7000 | ours | $-6.36767 \cdot 10^6$ | **3600.58** | **0.11** |

Table 105: Comparison of time and optimality gap on the synthetic datasets (p = 3000, k = 10, $\rho = 0.9$, $\lambda_2 = 0.1$).

**G.3.10 Synthetic 2 Benchmark with $\lambda_2 = 0.1$ with Warmstart**

| # of samples | method | upper bound | time(s) | gap(%) |
|---|---|---|---|---|
| 3000 | big-M50 + warm start | $-6.34575 \cdot 10^4$ | 3600.12 | 19.14 |
| 3000 | big-M20 + warm start | $-6.34575 \cdot 10^4$ | 3600.22 | 18.10 |
| 3000 | big-M5 + warm start | $-6.34575 \cdot 10^4$ | **2424.12** | **0.00** |
| 3000 | ours | $-6.34575 \cdot 10^4$ | 3600.33 | 19.31 |
| 4000 | big-M50 + warm start | $-8.64284 \cdot 10^4$ | 3600.05 | 15.33 |
| 4000 | big-M20 + warm start | $-8.64284 \cdot 10^4$ | 3600.18 | 14.04 |
| 4000 | big-M5 + warm start | $-8.64284 \cdot 10^4$ | 1702.95 | 0.00 |
| 4000 | ours | $-8.64284 \cdot 10^4$ | **20.81** | **0.00** |
| 5000 | big-M50 + warm start | $-1.09575 \cdot 10^5$ | 3601.88 | 11.60 |
| 5000 | big-M20 + warm start | $-1.09575 \cdot 10^5$ | 3600.31 | 9.94 |
| 5000 | big-M5 + warm start | $-1.09575 \cdot 10^5$ | 1284.19 | 0.00 |
| 5000 | ours | $-1.09575 \cdot 10^5$ | **20.65** | **0.00** |
| 6000 | big-M50 + warm start | $-1.27642 \cdot 10^5$ | 3600.62 | 9.93 |
| 6000 | big-M20 + warm start | $-1.27642 \cdot 10^5$ | 3600.15 | 9.92 |
| 6000 | big-M5 + warm start | $-1.27642 \cdot 10^5$ | 1004.13 | 0.00 |
| 6000 | ours | $-1.27642 \cdot 10^5$ | **1.70** | **0.00** |
| 7000 | big-M50 + warm start | $-1.50613 \cdot 10^5$ | 3600.07 | 8.32 |
| 7000 | big-M20 + warm start | $-1.50613 \cdot 10^5$ | 3600.06 | 5.74 |
| 7000 | big-M5 + warm start | $-1.50613 \cdot 10^5$ | 794.91 | 0.00 |
| 7000 | ours | $-1.50613 \cdot 10^5$ | **1.67** | **0.00** |

Table 106: Comparison of time and optimality gap on the synthetic datasets (p = 3000, k = 10, $\rho = 0.1$, $\lambda_2 = 0.1$). All baselines use our beam search solution as a warm start.

| # of samples | method | upper bound | time(s) | gap(%) |
|---|---|---|---|---|
| 3000 | big-M50 + warm start | $-1.59508 \cdot 10^5$ | 3600.22 | 19.07 |
| 3000 | big-M20 + warm start | $-1.59508 \cdot 10^5$ | 3600.22 | 17.30 |
| 3000 | big-M5 + warm start | $-1.59508 \cdot 10^5$ | **3600.05** | **6.35** |
| 3000 | ours | $-1.59508 \cdot 10^5$ | 3600.27 | 19.40 |
| 4000 | big-M50 + warm start | $-2.17573 \cdot 10^5$ | 3600.10 | 14.78 |
| 4000 | big-M20 + warm start | $-2.17573 \cdot 10^5$ | 3600.06 | 14.34 |
| 4000 | big-M5 + warm start | $-2.17573 \cdot 10^5$ | 3600.04 | 4.63 |
| 4000 | ours | $-2.17573 \cdot 10^5$ | **99.53** | **0.00** |
| 5000 | big-M50 + warm start | $-2.71985 \cdot 10^5$ | 3600.07 | 11.22 |
| 5000 | big-M20 + warm start | $-2.71985 \cdot 10^5$ | 3600.10 | 11.29 |
| 5000 | big-M5 + warm start | $-2.71985 \cdot 10^5$ | 2980.53 | 0.00 |
| 5000 | ours | $-2.71985 \cdot 10^5$ | **21.01** | **0.00** |
| 6000 | big-M50 + warm start | $-3.16760 \cdot 10^5$ | 3600.07 | 9.96 |
| 6000 | big-M20 + warm start | $-3.16760 \cdot 10^5$ | 3600.74 | 9.95 |
| 6000 | big-M5 + warm start | $-3.16760 \cdot 10^5$ | 2034.39 | 0.00 |
| 6000 | ours | $-3.16760 \cdot 10^5$ | **20.91** | **0.00** |
| 7000 | big-M50 + warm start | $-3.72609 \cdot 10^5$ | 3601.58 | 8.33 |
| 7000 | big-M20 + warm start | $-3.72609 \cdot 10^5$ | 3600.38 | 8.12 |
| 7000 | big-M5 + warm start | $-3.72609 \cdot 10^5$ | 1513.64 | 0.00 |
| 7000 | ours | $-3.72609 \cdot 10^5$ | **1.73** | **0.00** |

Table 107: Comparison of time and optimality gap on the synthetic datasets (p = 3000, k = 10, $\rho = 0.3$, $\lambda_2 = 0.1$). All baselines use our beam search solution as a warm start.

| # of samples | method | upper bound | time(s) | gap(%) |
|---|---|---|---|---|
| 3000 | big-M50 + warm start | $-3.33603 \cdot 10^5$ | 3601.53 | 18.78 |
| 3000 | big-M20 + warm start | $-3.33603 \cdot 10^5$ | 3600.81 | 16.11 |
| 3000 | big-M5 + warm start | $-3.33603 \cdot 10^5$ | **3600.22** | **7.55** |
| 3000 | ours | $-3.33603 \cdot 10^5$ | 3600.68 | 19.44 |
| 4000 | big-M50 + warm start | $-4.53260 \cdot 10^5$ | 3600.07 | 15.04 |
| 4000 | big-M20 + warm start | $-4.53260 \cdot 10^5$ | 3600.26 | 13.47 |
| 4000 | big-M5 + warm start | $-4.53260 \cdot 10^5$ | 3600.21 | 6.23 |
| 4000 | ours | $-4.53260 \cdot 10^5$ | **3600.41** | **0.62** |
| 5000 | big-M50 + warm start | $-5.63006 \cdot 10^5$ | 3601.70 | 11.69 |
| 5000 | big-M20 + warm start | $-5.63006 \cdot 10^5$ | 3600.25 | 11.26 |
| 5000 | big-M5 + warm start | $-5.63006 \cdot 10^5$ | 3600.05 | 3.43 |
| 5000 | ours | $-5.63006 \cdot 10^5$ | **37.26** | **0.00** |
| 6000 | big-M50 + warm start | $-6.57798 \cdot 10^5$ | 3600.18 | 9.98 |
| 6000 | big-M20 + warm start | $-6.57798 \cdot 10^5$ | 3600.55 | 9.36 |
| 6000 | big-M5 + warm start | $-6.57798 \cdot 10^5$ | 3600.04 | 2.89 |
| 6000 | ours | $-6.57798 \cdot 10^5$ | **20.64** | **0.00** |
| 7000 | big-M50 + warm start | $-7.72266 \cdot 10^5$ | 3600.63 | 8.33 |
| 7000 | big-M20 + warm start | $-7.72266 \cdot 10^5$ | 3600.06 | 7.80 |
| 7000 | big-M5 + warm start | $-7.72266 \cdot 10^5$ | 3600.15 | 2.46 |
| 7000 | ours | $-7.72266 \cdot 10^5$ | **20.91** | **0.00** |

Table 108: Comparison of time and optimality gap on the synthetic datasets (p = 3000, k = 10, $\rho = 0.5$, $\lambda_2 = 0.1$). All baselines use our beam search solution as a warm start.

| # of samples | method | upper bound | time(s) | gap(%) |
|---|---|---|---|---|
| 3000 | big-M50 + warm start | $-7.41186 \cdot 10^5$ | 3601.56 | 18.15 |
| 3000 | big-M20 + warm start | $-7.41186 \cdot 10^5$ | 3600.35 | 14.15 |
| 3000 | big-M5 + warm start | $-7.41186 \cdot 10^5$ | **3600.27** | **5.92** |
| 3000 | ours | $-7.41186 \cdot 10^5$ | 3600.59 | 19.46 |
| 4000 | big-M50 + warm start | $-1.00287 \cdot 10^6$ | 3600.09 | 14.98 |
| 4000 | big-M20 + warm start | $-1.00287 \cdot 10^6$ | 3600.12 | 12.06 |
| 4000 | big-M5 + warm start | $-1.00287 \cdot 10^6$ | 3600.07 | 5.06 |
| 4000 | ours | $-1.00287 \cdot 10^6$ | **3600.08** | **1.88** |
| 5000 | big-M50 + warm start | $-1.24027 \cdot 10^6$ | 3600.40 | 11.88 |
| 5000 | big-M20 + warm start | $-1.24027 \cdot 10^6$ | 3600.98 | 10.26 |
| 5000 | big-M5 + warm start | $-1.24027 \cdot 10^6$ | 3600.22 | 4.34 |
| 5000 | ours | $-1.24027 \cdot 10^6$ | **469.43** | **0.00** |
| 6000 | big-M50 + warm start | $-1.45416 \cdot 10^6$ | 3600.05 | 9.78 |
| 6000 | big-M20 + warm start | $-1.45416 \cdot 10^6$ | 3600.06 | 9.06 |
| 6000 | big-M5 + warm start | $-1.45416 \cdot 10^6$ | 3600.19 | 3.94 |
| 6000 | ours | $-1.45416 \cdot 10^6$ | **59.16** | **0.00** |
| 7000 | big-M50 + warm start | $-1.70483 \cdot 10^6$ | 3600.07 | 8.34 |
| 7000 | big-M20 + warm start | $-1.70483 \cdot 10^6$ | 3600.28 | 7.77 |
| 7000 | big-M5 + warm start | $-1.70483 \cdot 10^6$ | 3600.24 | 3.39 |
| 7000 | ours | $-1.70483 \cdot 10^6$ | **21.12** | **0.00** |

Table 109: Comparison of time and optimality gap on the synthetic datasets (p = 3000, k = 10, $\rho = 0.7$, $\lambda_2 = 0.1$). All baselines use our beam search solution as a warm start.

| # of samples | method | upper bound | time(s) | gap(%) |
|---|---|---|---|---|
| 3000 | big-M50 + warm start | $-2.78521 \cdot 10^6$ | 3600.07 | 15.91 |
| 3000 | big-M20 + warm start | $-2.78521 \cdot 10^6$ | 3600.71 | 10.37 |
| 3000 | big-M5 + warm start | $-2.78521 \cdot 10^6$ | **3600.08** | **3.69** |
| 3000 | ours | $-2.78521 \cdot 10^6$ | 3600.23 | 19.46 |
| 4000 | big-M50 + warm start | $-3.75203 \cdot 10^6$ | 3600.27 | 13.35 |
| 4000 | big-M20 + warm start | $-3.75203 \cdot 10^6$ | 3600.05 | 8.60 |
| 4000 | big-M5 + warm start | $-3.75203 \cdot 10^6$ | **3600.09** | **3.16** |
| 4000 | ours | $-3.75203 \cdot 10^6$ | 3601.10 | 3.50 |
| 5000 | big-M50 + warm start | $-4.61877 \cdot 10^6$ | 3600.12 | 11.40 |
| 5000 | big-M20 + warm start | $-4.61877 \cdot 10^6$ | 3600.06 | 7.62 |
| 5000 | big-M5 + warm start | $-4.61877 \cdot 10^6$ | 3600.09 | 2.72 |
| 5000 | ours | $-4.61877 \cdot 10^6$ | **3600.40** | **0.93** |
| 6000 | big-M50 + warm start | $-5.43826 \cdot 10^6$ | 3600.07 | 9.79 |
| 6000 | big-M20 + warm start | $-5.43826 \cdot 10^6$ | 3600.33 | 6.97 |
| 6000 | big-M5 + warm start | $-5.43826 \cdot 10^6$ | 3600.24 | 2.50 |
| 6000 | ours | $-5.43826 \cdot 10^6$ | **3600.26** | **0.33** |
| 7000 | big-M50 + warm start | $-6.36767 \cdot 10^6$ | 3600.07 | 8.22 |
| 7000 | big-M20 + warm start | $-6.36767 \cdot 10^6$ | 3600.67 | 6.06 |
| 7000 | big-M5 + warm start | $-6.36767 \cdot 10^6$ | 3600.07 | 2.25 |
| 7000 | ours | $-6.36767 \cdot 10^6$ | **3600.58** | **0.11** |

Table 110: Comparison of time and optimality gap on the synthetic datasets (p = 3000, k = 10, $\rho = 0.9$, $\lambda_2 = 0.1$). All baselines use our beam search solution as a warm start.

### G.3.11 Synthetic 2 Benchmark with $\lambda_2 = 10.0$ with No Warmstart

| # of samples | method | upper bound | time(s) | gap(%) |
|---|---|---|---|---|
| 3000 | big-M50 | $-6.33576 \cdot 10^4$ | 3600.31 | 18.47 |
| 3000 | big-M20 | $-6.33576 \cdot 10^4$ | 3600.43 | 17.82 |
| 3000 | big-M5 | $-6.33576 \cdot 10^4$ | **2349.33** | **0.00** |
| 3000 | ours | $-6.33576 \cdot 10^4$ | 3601.41 | 4.54 |
| 4000 | big-M50 | $-8.63287 \cdot 10^4$ | 3600.85 | 15.21 |
| 4000 | big-M20 | $-8.63287 \cdot 10^4$ | 3600.06 | 13.93 |
| 4000 | big-M5 | $-8.63287 \cdot 10^4$ | 1577.17 | 0.00 |
| 4000 | ours | $-8.63287 \cdot 10^4$ | **20.33** | **0.00** |
| 5000 | big-M50 | $-1.09474 \cdot 10^5$ | 3600.28 | 11.85 |
| 5000 | big-M20 | $-1.09474 \cdot 10^5$ | 3600.55 | 10.17 |
| 5000 | big-M5 | $-1.09474 \cdot 10^5$ | 1627.54 | 0.00 |
| 5000 | ours | $-1.09474 \cdot 10^5$ | **20.71** | **0.00** |
| 6000 | big-M50 | $-1.27542 \cdot 10^5$ | 3600.37 | 9.90 |
| 6000 | big-M20 | $-1.27542 \cdot 10^5$ | 3600.09 | 9.81 |
| 6000 | big-M5 | $-1.27542 \cdot 10^5$ | 1071.17 | 0.00 |
| 6000 | ours | $-1.27542 \cdot 10^5$ | **1.69** | **0.00** |
| 7000 | big-M50 | $-1.50512 \cdot 10^5$ | 3600.56 | 8.31 |
| 7000 | big-M20 | $-1.50512 \cdot 10^5$ | 3600.07 | 8.30 |
| 7000 | big-M5 | $-1.50512 \cdot 10^5$ | 856.04 | 0.00 |
| 7000 | ours | $-1.50512 \cdot 10^5$ | **1.70** | **0.00** |

Table 111: Comparison of time and optimality gap on the synthetic datasets (p = 3000, k = 10, $\rho = 0.1$, $\lambda_2 = 10.0$).

| # of samples | method | upper bound | time(s) | gap(%) |
|---|---|---|---|---|
| 3000 | big-M50 | $-1.59409 \cdot 10^5$ | 3600.72 | 18.52 |
| 3000 | big-M20 | $-1.59409 \cdot 10^5$ | 3600.07 | 17.23 |
| 3000 | big-M5 | $-1.59409 \cdot 10^5$ | 3600.44 | 5.92 |
| 3000 | ours | $-1.59409 \cdot 10^5$ | **3600.62** | **4.68** |
| 4000 | big-M50 | $-2.17472 \cdot 10^5$ | 3600.07 | 14.64 |
| 4000 | big-M20 | $-2.17472 \cdot 10^5$ | 3600.82 | 14.15 |
| 4000 | big-M5 | $-2.17472 \cdot 10^5$ | 3600.05 | 4.05 |
| 4000 | ours | $-2.17472 \cdot 10^5$ | **86.90** | **0.00** |
| 5000 | big-M50 | $-2.71885 \cdot 10^5$ | 3603.74 | 11.16 |
| 5000 | big-M20 | $-2.71885 \cdot 10^5$ | 3601.16 | 11.14 |
| 5000 | big-M5 | $-2.71885 \cdot 10^5$ | 2522.90 | 0.00 |
| 5000 | ours | $-2.71885 \cdot 10^5$ | **20.75** | **0.00** |
| 6000 | big-M50 | $-3.16660 \cdot 10^5$ | 3600.07 | 9.93 |
| 6000 | big-M20 | $-3.16660 \cdot 10^5$ | 3600.26 | 9.92 |
| 6000 | big-M5 | $-3.16660 \cdot 10^5$ | 1893.58 | 0.00 |
| 6000 | ours | $-3.16660 \cdot 10^5$ | **21.05** | **0.00** |
| 7000 | big-M50 | $-3.72509 \cdot 10^5$ | 3601.17 | 8.31 |
| 7000 | big-M20 | $-3.72509 \cdot 10^5$ | 3600.31 | 8.29 |
| 7000 | big-M5 | $-3.72509 \cdot 10^5$ | 1472.84 | 0.00 |
| 7000 | ours | $-3.72509 \cdot 10^5$ | **1.71** | **0.00** |

Table 112: Comparison of time and optimality gap on the synthetic datasets (p = 3000, k = 10, $\rho = 0.3$, $\lambda_2 = 10.0$).

| # of samples | method | upper bound | time(s) | gap(%) |
|---|---|---|---|---|
| 3000 | big-M50 | $-3.33503 \cdot 10^5$ | 3600.07 | 18.43 |
| 3000 | big-M20 | $-3.33503 \cdot 10^5$ | 3600.06 | 16.05 |
| 3000 | big-M5 | $-3.33503 \cdot 10^5$ | 3600.22 | 7.54 |
| 3000 | ours | $-3.33503 \cdot 10^5$ | **3600.25** | **6.21** |
| 4000 | big-M50 | $-4.53159 \cdot 10^5$ | 3600.16 | 14.84 |
| 4000 | big-M20 | $-4.53159 \cdot 10^5$ | 3600.90 | 13.44 |
| 4000 | big-M5 | $-4.53159 \cdot 10^5$ | 3600.21 | 6.24 |
| 4000 | ours | $-4.53159 \cdot 10^5$ | **3600.95** | **0.35** |
| 5000 | big-M50 | $-5.62906 \cdot 10^5$ | 3600.16 | 11.59 |
| 5000 | big-M20 | $-5.62906 \cdot 10^5$ | 3600.07 | 11.13 |
| 5000 | big-M5 | $-5.62906 \cdot 10^5$ | 3600.36 | 3.65 |
| 5000 | ours | $-5.62906 \cdot 10^5$ | **31.47** | **0.00** |
| 6000 | big-M50 | $-6.57698 \cdot 10^5$ | 3600.16 | 9.95 |
| 6000 | big-M20 | $-6.57698 \cdot 10^5$ | 3600.52 | 9.33 |
| 6000 | big-M5 | $-6.57698 \cdot 10^5$ | 3600.37 | 2.90 |
| 6000 | ours | $-6.57698 \cdot 10^5$ | **21.24** | **0.00** |
| 7000 | big-M50 | $-7.72166 \cdot 10^5$ | 3600.12 | 8.31 |
| 7000 | big-M20 | $-7.72166 \cdot 10^5$ | 3600.05 | 7.81 |
| 7000 | big-M5 | $-7.72166 \cdot 10^5$ | 3600.05 | 2.33 |
| 7000 | ours | $-7.72166 \cdot 10^5$ | **20.37** | **0.00** |

Table 113: Comparison of time and optimality gap on the synthetic datasets (p = 3000, k = 10, $\rho = 0.5$, $\lambda_2 = 10.0$).

| # of samples | method | upper bound | time(s) | gap(%) |
|---|---|---|---|---|
| 3000 | big-M50 | $-7.41086 \cdot 10^5$ | 3600.65 | 17.88 |
| 3000 | big-M20 | $-7.41086 \cdot 10^5$ | 3600.14 | 14.12 |
| 3000 | big-M5 | $-7.41086 \cdot 10^5$ | **3600.35** | **5.94** |
| 3000 | ours | $-7.41086 \cdot 10^5$ | 3600.48 | 8.32 |
| 4000 | big-M50 | $-1.00277 \cdot 10^6$ | 3600.09 | 14.85 |
| 4000 | big-M20 | $-1.00277 \cdot 10^6$ | 3600.47 | 11.77 |
| 4000 | big-M5 | $-1.00277 \cdot 10^6$ | 3600.05 | 5.04 |
| 4000 | ours | $-1.00277 \cdot 10^6$ | **3600.62** | **1.66** |
| 5000 | big-M50 | $-1.24017 \cdot 10^6$ | 3600.25 | 11.82 |
| 5000 | big-M20 | $-1.24017 \cdot 10^6$ | 3600.06 | 10.25 |
| 5000 | big-M5 | $-1.24017 \cdot 10^6$ | 3600.22 | 4.34 |
| 5000 | ours | $-1.24017 \cdot 10^6$ | **386.72** | **0.00** |
| 6000 | big-M50 | $-1.45406 \cdot 10^6$ | 3600.07 | 9.75 |
| 6000 | big-M20 | $-1.44919 \cdot 10^6$ | 3600.53 | 9.46 |
| 6000 | big-M5 | $-1.45406 \cdot 10^6$ | 3600.20 | 3.95 |
| 6000 | ours | $-1.45406 \cdot 10^6$ | **54.13** | **0.00** |
| 7000 | big-M50 | $-1.70473 \cdot 10^6$ | 3600.81 | 8.32 |
| 7000 | big-M20 | $-1.70473 \cdot 10^6$ | 3600.06 | 7.68 |
| 7000 | big-M5 | $-1.70473 \cdot 10^6$ | 3600.20 | 3.41 |
| 7000 | ours | $-1.70473 \cdot 10^6$ | **21.30** | **0.00** |

Table 114: Comparison of time and optimality gap on the synthetic datasets (p = 3000, k = 10, $\rho = 0.7$, $\lambda_2 = 10.0$).

| # of samples | method | upper bound | time(s) | gap(%) |
|---|---|---|---|---|
| 3000 | big-M50 | $-2.78507 \cdot 10^6$ | 3600.17 | 15.95 |
| 3000 | big-M20 | $-2.78507 \cdot 10^6$ | 3600.56 | 10.36 |
| 3000 | big-M5 | $-2.78507 \cdot 10^6$ | **3600.09** | **3.69** |
| 3000 | ours | $-2.78507 \cdot 10^6$ | 3600.78 | 9.79 |
| 4000 | big-M50 | $-3.75192 \cdot 10^6$ | 3600.06 | 13.19 |
| 4000 | big-M20 | $-3.75192 \cdot 10^6$ | 3600.05 | 8.60 |
| 4000 | big-M5 | $-3.75192 \cdot 10^6$ | **3600.41** | **3.16** |
| 4000 | ours | $-3.75192 \cdot 10^6$ | 3600.32 | 3.23 |
| 5000 | big-M50 | $-4.61867 \cdot 10^6$ | 3600.17 | 11.32 |
| 5000 | big-M20 | $-4.61867 \cdot 10^6$ | 3600.56 | 7.66 |
| 5000 | big-M5 | $-4.61867 \cdot 10^6$ | 3600.22 | 2.72 |
| 5000 | ours | $-4.61867 \cdot 10^6$ | **3600.30** | **0.89** |
| 6000 | big-M50 | $-5.43816 \cdot 10^6$ | 3600.12 | 9.75 |
| 6000 | big-M20 | $-5.43816 \cdot 10^6$ | 3600.33 | 6.91 |
| 6000 | big-M5 | $-5.43816 \cdot 10^6$ | 3600.20 | 2.50 |
| 6000 | ours | $-5.43816 \cdot 10^6$ | **3600.22** | **0.32** |
| 7000 | big-M50 | $-6.36757 \cdot 10^6$ | 3600.32 | 8.20 |
| 7000 | big-M20 | $-6.36757 \cdot 10^6$ | 3600.05 | 6.05 |
| 7000 | big-M5 | $-6.36757 \cdot 10^6$ | 3600.37 | 2.25 |
| 7000 | ours | $-6.36757 \cdot 10^6$ | **3600.06** | **0.10** |

Table 115: Comparison of time and optimality gap on the synthetic datasets (p = 3000, k = 10, $\rho = 0.9$, $\lambda_2 = 10.0$).

## G.3.12 Synthetic 2 Benchmark with $\lambda_2 = 10.0$ with Warmstart

| # of samples | method | upper bound | time(s) | gap(%) |
|:---:|:---:|:---:|:---:|:---:|
| 3000 | big-M50 + warm start | $-6.33576 \cdot 10^4$ | 3600.12 | 18.47 |
| 3000 | big-M20 + warm start | $-6.33576 \cdot 10^4$ | 3600.44 | 17.82 |
| 3000 | big-M5 + warm start | $-6.33576 \cdot 10^4$ | **2370.86** | **0.00** |
| 3000 | ours | $-6.33576 \cdot 10^4$ | 3601.41 | 4.54 |
| 4000 | big-M50 + warm start | $-8.63287 \cdot 10^4$ | 3600.29 | 15.21 |
| 4000 | big-M20 + warm start | $-8.63287 \cdot 10^4$ | 3600.09 | 13.93 |
| 4000 | big-M5 + warm start | $-8.63287 \cdot 10^4$ | 1870.35 | 0.00 |
| 4000 | ours | $-8.63287 \cdot 10^4$ | **20.33** | **0.00** |
| 5000 | big-M50 + warm start | $-1.09474 \cdot 10^5$ | 3600.15 | 11.85 |
| 5000 | big-M20 + warm start | $-1.09474 \cdot 10^5$ | 3600.43 | 10.17 |
| 5000 | big-M5 + warm start | $-1.09474 \cdot 10^5$ | 1146.28 | 0.00 |
| 5000 | ours | $-1.09474 \cdot 10^5$ | **20.71** | **0.00** |
| 6000 | big-M50 + warm start | $-1.27542 \cdot 10^5$ | 3601.39 | 9.89 |
| 6000 | big-M20 + warm start | $-1.27542 \cdot 10^5$ | 3600.08 | 9.81 |
| 6000 | big-M5 + warm start | $-1.27542 \cdot 10^5$ | 1038.32 | 0.00 |
| 6000 | ours | $-1.27542 \cdot 10^5$ | **1.69** | **0.00** |
| 7000 | big-M50 + warm start | $-1.50512 \cdot 10^5$ | 3601.02 | 8.31 |
| 7000 | big-M20 + warm start | $-1.50512 \cdot 10^5$ | 3600.99 | 7.98 |
| 7000 | big-M5 + warm start | $-1.50512 \cdot 10^5$ | 853.59 | 0.00 |
| 7000 | ours | $-1.50512 \cdot 10^5$ | **1.70** | **0.00** |

Table 116: Comparison of time and optimality gap on the synthetic datasets (p = 3000, k = 10, $\rho = 0.1$, $\lambda_2 = 10.0$). All baselines use our beam search solution as a warm start.

| # of samples | method | upper bound | time(s) | gap(%) |
|---|---|---|---|---|
| 3000 | big-M50 + warm start | $-1.59409 \cdot 10^5$ | 3600.67 | 18.52 |
| 3000 | big-M20 + warm start | $-1.59409 \cdot 10^5$ | 3601.34 | 17.23 |
| 3000 | big-M5 + warm start | $-1.59409 \cdot 10^5$ | 3600.06 | 5.92 |
| 3000 | ours | $-1.59409 \cdot 10^5$ | **3600.62** | **4.68** |
| 4000 | big-M50 + warm start | $-2.17472 \cdot 10^5$ | 3600.08 | 14.64 |
| 4000 | big-M20 + warm start | $-2.17472 \cdot 10^5$ | 3600.35 | 14.15 |
| 4000 | big-M5 + warm start | $-2.17472 \cdot 10^5$ | 3600.06 | 3.77 |
| 4000 | ours | $-2.17472 \cdot 10^5$ | **86.90** | **0.00** |
| 5000 | big-M50 + warm start | $-2.71885 \cdot 10^5$ | 3600.07 | 11.16 |
| 5000 | big-M20 + warm start | $-2.71885 \cdot 10^5$ | 3600.19 | 11.09 |
| 5000 | big-M5 + warm start | $-2.71885 \cdot 10^5$ | 2700.47 | 0.00 |
| 5000 | ours | $-2.71885 \cdot 10^5$ | **20.75** | **0.00** |
| 6000 | big-M50 + warm start | $-3.16660 \cdot 10^5$ | 3600.06 | 9.93 |
| 6000 | big-M20 + warm start | $-3.16660 \cdot 10^5$ | 3600.74 | 9.92 |
| 6000 | big-M5 + warm start | $-3.16660 \cdot 10^5$ | 1742.17 | 0.00 |
| 6000 | ours | $-3.16660 \cdot 10^5$ | **21.05** | **0.00** |
| 7000 | big-M50 + warm start | $-3.72509 \cdot 10^5$ | 3601.61 | 8.31 |
| 7000 | big-M20 + warm start | $-3.72509 \cdot 10^5$ | 3600.25 | 8.11 |
| 7000 | big-M5 + warm start | $-3.72509 \cdot 10^5$ | 1466.07 | 0.00 |
| 7000 | ours | $-3.72509 \cdot 10^5$ | **1.71** | **0.00** |

Table 117: Comparison of time and optimality gap on the synthetic datasets (p = 3000, k = 10, $\rho = 0.3$, $\lambda_2 = 10.0$). All baselines use our beam search solution as a warm start.

| # of samples | method | upper bound | time(s) | gap(%) |
|---|---|---|---|---|
| 3000 | big-M50 + warm start | $-3.33503 \cdot 10^5$ | 3600.08 | 18.43 |
| 3000 | big-M20 + warm start | $-3.33503 \cdot 10^5$ | 3600.16 | 16.05 |
| 3000 | big-M5 + warm start | $-3.33503 \cdot 10^5$ | 3600.22 | 7.54 |
| 3000 | ours | $-3.33503 \cdot 10^5$ | **3600.25** | **6.21** |
| 4000 | big-M50 + warm start | $-4.53159 \cdot 10^5$ | 3600.06 | 14.84 |
| 4000 | big-M20 + warm start | $-4.53159 \cdot 10^5$ | 3600.21 | 13.44 |
| 4000 | big-M5 + warm start | $-4.53159 \cdot 10^5$ | 3600.22 | 6.24 |
| 4000 | ours | $-4.53159 \cdot 10^5$ | **3600.95** | **0.35** |
| 5000 | big-M50 + warm start | $-5.62906 \cdot 10^5$ | 3600.08 | 11.59 |
| 5000 | big-M20 + warm start | $-5.62906 \cdot 10^5$ | 3601.71 | 11.22 |
| 5000 | big-M5 + warm start | $-5.62906 \cdot 10^5$ | 3600.13 | 3.66 |
| 5000 | ours | $-5.62906 \cdot 10^5$ | **31.47** | **0.00** |
| 6000 | big-M50 + warm start | $-6.57698 \cdot 10^5$ | 3600.14 | 9.95 |
| 6000 | big-M20 + warm start | $-6.57698 \cdot 10^5$ | 3600.97 | 9.33 |
| 6000 | big-M5 + warm start | $-6.57698 \cdot 10^5$ | 3600.14 | 2.91 |
| 6000 | ours | $-6.57698 \cdot 10^5$ | **21.24** | **0.00** |
| 7000 | big-M50 + warm start | $-7.72166 \cdot 10^5$ | 3601.52 | 8.31 |
| 7000 | big-M20 + warm start | $-7.72166 \cdot 10^5$ | 3600.30 | 7.81 |
| 7000 | big-M5 + warm start | $-7.72166 \cdot 10^5$ | 3600.46 | 2.33 |
| 7000 | ours | $-7.72166 \cdot 10^5$ | **20.37** | **0.00** |

Table 118: Comparison of time and optimality gap on the synthetic datasets (p = 3000, k = 10, $\rho = 0.5$, $\lambda_2 = 10.0$). All baselines use our beam search solution as a warm start.

| # of samples | method | upper bound | time(s) | gap(%) |
|---|---|---|---|---|
| 3000 | big-M50 + warm start | $-7.41086 \cdot 10^5$ | 3601.32 | 17.88 |
| 3000 | big-M20 + warm start | $-7.41086 \cdot 10^5$ | 3600.23 | 14.06 |
| 3000 | big-M5 + warm start | $-7.41086 \cdot 10^5$ | **3600.11** | **5.94** |
| 3000 | ours | $-7.41086 \cdot 10^5$ | 3600.48 | 8.32 |
| 4000 | big-M50 + warm start | $-1.00277 \cdot 10^6$ | 3600.44 | 14.85 |
| 4000 | big-M20 + warm start | $-1.00277 \cdot 10^6$ | 3600.09 | 11.77 |
| 4000 | big-M5 + warm start | $-1.00277 \cdot 10^6$ | 3600.26 | 5.04 |
| 4000 | ours | $-1.00277 \cdot 10^6$ | **3600.62** | **1.66** |
| 5000 | big-M50 + warm start | $-1.24017 \cdot 10^6$ | 3600.27 | 11.82 |
| 5000 | big-M20 + warm start | $-1.24017 \cdot 10^6$ | 3600.69 | 10.25 |
| 5000 | big-M5 + warm start | $-1.24017 \cdot 10^6$ | 3600.22 | 4.34 |
| 5000 | ours | $-1.24017 \cdot 10^6$ | **386.72** | **0.00** |
| 6000 | big-M50 + warm start | $-1.45406 \cdot 10^6$ | 3600.26 | 9.75 |
| 6000 | big-M20 + warm start | $-1.45406 \cdot 10^6$ | 3600.07 | 9.09 |
| 6000 | big-M5 + warm start | $-1.45406 \cdot 10^6$ | 3600.18 | 3.95 |
| 6000 | ours | $-1.45406 \cdot 10^6$ | **54.13** | **0.00** |
| 7000 | big-M50 + warm start | $-1.70473 \cdot 10^6$ | 3600.20 | 8.32 |
| 7000 | big-M20 + warm start | $-1.70473 \cdot 10^6$ | 3600.12 | 7.68 |
| 7000 | big-M5 + warm start | $-1.70473 \cdot 10^6$ | 3600.04 | 3.41 |
| 7000 | ours | $-1.70473 \cdot 10^6$ | **21.30** | **0.00** |

Table 119: Comparison of time and optimality gap on the synthetic datasets (p = 3000, k = 10, $\rho = 0.7$, $\lambda_2 = 10.0$). All baselines use our beam search solution as a warm start.

| # of samples | method | upper bound | time(s) | gap(%) |
|---|---|---|---|---|
| 3000 | big-M50 + warm start | $-2.78507 \cdot 10^6$ | 3600.07 | 15.95 |
| 3000 | big-M20 + warm start | $-2.78507 \cdot 10^6$ | 3600.04 | 10.36 |
| 3000 | big-M5 + warm start | $-2.78507 \cdot 10^6$ | **3600.09** | **3.69** |
| 3000 | ours | $-2.78507 \cdot 10^6$ | 3600.78 | 9.79 |
| 4000 | big-M50 + warm start | $-3.75192 \cdot 10^6$ | 3600.06 | 13.19 |
| 4000 | big-M20 + warm start | $-3.75192 \cdot 10^6$ | 3600.11 | 8.61 |
| 4000 | big-M5 + warm start | $-3.75192 \cdot 10^6$ | **3600.15** | **3.16** |
| 4000 | ours | $-3.75192 \cdot 10^6$ | 3600.32 | 3.23 |
| 5000 | big-M50 + warm start | $-4.61867 \cdot 10^6$ | 3600.06 | 11.32 |
| 5000 | big-M20 + warm start | $-4.61867 \cdot 10^6$ | 3600.79 | 7.64 |
| 5000 | big-M5 + warm start | $-4.61867 \cdot 10^6$ | 3600.30 | 2.72 |
| 5000 | ours | $-4.61867 \cdot 10^6$ | **3600.30** | **0.89** |
| 6000 | big-M50 + warm start | $-5.43816 \cdot 10^6$ | 3600.39 | 9.75 |
| 6000 | big-M20 + warm start | $-5.43816 \cdot 10^6$ | 3600.05 | 6.91 |
| 6000 | big-M5 + warm start | $-5.43816 \cdot 10^6$ | 3600.05 | 2.50 |
| 6000 | ours | $-5.43816 \cdot 10^6$ | **3600.22** | **0.32** |
| 7000 | big-M50 + warm start | $-6.36757 \cdot 10^6$ | 3600.10 | 8.20 |
| 7000 | big-M20 + warm start | $-6.36757 \cdot 10^6$ | 3600.28 | 6.05 |
| 7000 | big-M5 + warm start | $-6.36757 \cdot 10^6$ | 3600.37 | 2.25 |
| 7000 | ours | $-6.36757 \cdot 10^6$ | **3600.06** | **0.10** |

Table 120: Comparison of time and optimality gap on the synthetic datasets (p = 3000, k = 10, $\rho = 0.9$, $\lambda_2 = 10.0$). All baselines use our beam search solution as a warm start.

## G.4 Comparison with MOSEK Solver, SubsetSelectionCIO, and L0BNB

### G.4.1 Synthetic 1 Benchmark with $\lambda_2 = 0.001$

| # of features | method | upper bound | support size | time(s) | gap(%) |
|---|---|---|---|---|---|
| 100 | SubsetSelectCIO | $-4.79406 \cdot 10^5$ | 2 | 2.46 | 0.00 |
| 100 | SubsetSelectCIO + warm start | $-4.79406 \cdot 10^5$ | 2 | 2.72 | 0.00 |
| 100 | L0BnB | $-2.10878 \cdot 10^6$ | 10 | 1420.85 | 0.00 |
| 100 | MSK persp. | $-2.10878 \cdot 10^6$ | 10 | 932.06 | 0.00 |
| 100 | MSK persp. + warm start | $-2.10878 \cdot 10^6$ | 10 | 898.00 | 0.00 |
| 100 | Convex MSK opt. persp. | $-2.10878 \cdot 10^6$ | 10 | 0.10 | 0.00 |
| 100 | ours | $-2.10878 \cdot 10^6$ | 10 | 0.29 | 0.00 |
| 500 | SubsetSelectCIO | $-4.79201 \cdot 10^5$ | 2 | 3.91 | 0.00 |
| 500 | SubsetSelectCIO + warm start | $-4.79201 \cdot 10^5$ | 2 | 3.59 | 0.00 |
| 500 | L0BnB | $-2.12833 \cdot 10^6$ | 10 | 1754.22 | 0.00 |
| 500 | MSK persp. | $-2.12833 \cdot 10^6$ | 10 | 3606.30 | 0.09 |
| 500 | MSK persp. + warm start | $-2.12833 \cdot 10^6$ | 10 | 3600.23 | 0.09 |
| 500 | Convex MSK opt. persp. | $-2.12833 \cdot 10^6$ | 10 | 6.79 | 0.01 |
| 500 | ours | $-2.12833 \cdot 10^6$ | 10 | 0.37 | 0.00 |
| 1000 | SubsetSelectCIO | $-4.71433 \cdot 10^5$ | 2 | 4.36 | 0.00 |
| 1000 | SubsetSelectCIO + warm start | $-4.71433 \cdot 10^5$ | 2 | 4.39 | 0.00 |
| 1000 | L0BnB | $-2.11005 \cdot 10^6$ | 10 | 2114.74 | 0.00 |
| 1000 | MSK persp. | - | - | 3673.42 | - |
| 1000 | MSK persp. + warm start | $-2.11005 \cdot 10^6$ | 10 | 3608.90 | 0.00 |
| 1000 | Convex MSK opt. persp. | $-2.11005 \cdot 10^6$ | 10 | 54.02 | 0.01 |
| 1000 | ours | $-2.11005 \cdot 10^6$ | 10 | 0.58 | 0.00 |
| 3000 | SubsetSelectCIO | $-4.72136 \cdot 10^5$ | 2 | 5.51 | 0.00 |
| 3000 | SubsetSelectCIO + warm start | $-4.72136 \cdot 10^5$ | 2 | 9.19 | 0.00 |
| 3000 | L0BnB | $-2.10230 \cdot 10^6$ | 10 | 2302.34 | 1.20 |
| 3000 | MSK persp. | - | - | 3600.00 | - |
| 3000 | MSK persp. + warm start | - | - | 3600.00 | - |
| 3000 | Convex MSK opt. persp. | $-2.10230 \cdot 10^6$ | 10 | 1759.45 | 0.02 |
| 3000 | ours | $-2.10230 \cdot 10^6$ | 10 | 1.73 | 0.00 |
| 5000 | SubsetSelectCIO | $-4.62647 \cdot 10^5$ | 2 | 12.04 | 0.00 |
| 5000 | SubsetSelectCIO + warm start | $-4.62647 \cdot 10^5$ | 2 | 12.66 | 0.00 |
| 5000 | L0BnB | $-2.09635 \cdot 10^6$ | 10 | 2643.23 | 5.18 |
| 5000 | MSK persp. | - | - | 3600.00 | - |
| 5000 | MSK persp. + warm start | - | - | 3600.00 | - |
| 5000 | Convex MSK opt. persp. | $-2.09635 \cdot 10^6$ | 10 | 3760.81 | 0.05 |
| 5000 | ours | $-2.09635 \cdot 10^6$ | 10 | 4.22 | 0.00 |

Table 121: Comparison of time and optimality gap on the synthetic datasets (n = 100000, k = 10, $\rho = 0.1$, $\lambda_2 = 0.001$).

| # of features | method | upper bound | support size | time(s) | gap(%) |
|---|---|---|---|---|---|
| 100 | SubsetSelectCIO | $-2.52792 \cdot 10^6$ | 2 | 2.52 | 0.00 |
| 100 | SubsetSelectCIO + warm start | $-2.52792 \cdot 10^6$ | 2 | 2.71 | 0.00 |
| 100 | L0BnB | $-5.28726 \cdot 10^6$ | 10 | 3182.66 | 0.00 |
| 100 | MSK persp. | $-5.28726 \cdot 10^6$ | 10 | 3600.08 | 0.01 |
| 100 | MSK persp. + warm start | $-5.28726 \cdot 10^6$ | 10 | 3600.67 | 0.01 |
| 100 | Convex MSK opt. persp. | $-5.28726 \cdot 10^6$ | 10 | 0.11 | 0.00 |
| 100 | ours | $-5.28726 \cdot 10^6$ | 10 | 0.29 | 0.00 |
| 500 | SubsetSelectCIO | $-2.52985 \cdot 10^6$ | 2 | 3.43 | 0.00 |
| 500 | SubsetSelectCIO + warm start | $-2.52985 \cdot 10^6$ | 2 | 3.26 | 0.00 |
| 500 | L0BnB | $-5.31852 \cdot 10^6$ | 10 | 3977.43 | 1.65 |
| 500 | MSK persp. | $-5.31852 \cdot 10^6$ | 10 | 3631.98 | 0.10 |
| 500 | MSK persp. + warm start | $-5.31852 \cdot 10^6$ | 10 | 3607.49 | 0.10 |
| 500 | Convex MSK opt. persp. | $-5.31852 \cdot 10^6$ | 10 | 7.77 | 0.01 |
| 500 | ours | $-5.31852 \cdot 10^6$ | 10 | 0.41 | 0.00 |
| 1000 | SubsetSelectCIO | $-2.50201 \cdot 10^6$ | 2 | 4.67 | 0.00 |
| 1000 | SubsetSelectCIO + warm start | $-2.50201 \cdot 10^6$ | 2 | 4.99 | 0.00 |
| 1000 | L0BnB | $-5.26974 \cdot 10^6$ | 10 | 4224.81 | 2.82 |
| 1000 | MSK persp. | - | - | 3607.94 | - |
| 1000 | MSK persp. + warm start | $-5.26974 \cdot 10^6$ | 10 | 3601.79 | 0.20 |
| 1000 | Convex MSK opt. persp. | $-5.26974 \cdot 10^6$ | 10 | 54.48 | 0.01 |
| 1000 | ours | $-5.26974 \cdot 10^6$ | 10 | 0.52 | 0.00 |
| 3000 | SubsetSelectCIO | $-2.51861 \cdot 10^6$ | 2 | 8.73 | 0.00 |
| 3000 | SubsetSelectCIO + warm start | $-2.51861 \cdot 10^6$ | 2 | 8.50 | 0.00 |
| 3000 | L0BnB | $-5.27668 \cdot 10^6$ | 10 | 4945.04 | 0.35 |
| 3000 | MSK persp. | - | - | 3600.00 | - |
| 3000 | MSK persp. + warm start | - | - | 3600.00 | - |
| 3000 | Convex MSK opt. persp. | $-5.27668 \cdot 10^6$ | 10 | 1724.71 | 0.02 |
| 3000 | ours | $-5.27668 \cdot 10^6$ | 10 | 1.75 | 0.00 |
| 5000 | SubsetSelectCIO | $-2.49167 \cdot 10^6$ | 2 | 7.60 | 0.00 |
| 5000 | SubsetSelectCIO + warm start | $-2.49167 \cdot 10^6$ | 2 | 13.32 | 0.00 |
| 5000 | L0BnB | $-5.27126 \cdot 10^6$ | 10 | 5589.24 | 0.49 |
| 5000 | MSK persp. | - | - | 3600.00 | - |
| 5000 | MSK persp. + warm start | - | - | 3600.00 | - |
| 5000 | Convex MSK opt. persp. | $-5.27126 \cdot 10^6$ | 10 | 3660.81 | 0.05 |
| 5000 | ours | $-5.27126 \cdot 10^6$ | 10 | 4.32 | 0.00 |

Table 122: Comparison of time and optimality gap on the synthetic datasets (n = 100000, k = 10, $\rho = 0.3$, $\lambda_2 = 0.001$).

| # of features | method | upper bound | support size | time(s) | gap(%) |
|---|---|---|---|---|---|
| 100 | SubsetSelectCIO | $-7.42601 \cdot 10^6$ | 2 | 2.18 | 0.00 |
| 100 | SubsetSelectCIO + warm start | $-7.42601 \cdot 10^6$ | 2 | 2.61 | 0.00 |
| 100 | L0BnB | $-1.10084 \cdot 10^7$ | 10 | 3464.13 | 0.00 |
| 100 | MSK persp. | $-1.10084 \cdot 10^7$ | 10 | 3605.37 | 0.01 |
| 100 | MSK persp. + warm start | $-1.10084 \cdot 10^7$ | 10 | 3602.75 | 0.01 |
| 100 | Convex MSK opt. persp. | $-1.10084 \cdot 10^7$ | 10 | 0.14 | 0.00 |
| 100 | ours | $-1.10084 \cdot 10^7$ | 10 | 0.29 | 0.00 |
| 500 | SubsetSelectCIO | $-7.42485 \cdot 10^6$ | 2 | 3.33 | 0.00 |
| 500 | SubsetSelectCIO + warm start | $-7.42485 \cdot 10^6$ | 2 | 3.59 | 0.00 |
| 500 | L0BnB | $-1.10518 \cdot 10^7$ | 10 | 4143.57 | 0.77 |
| 500 | MSK persp. | $-1.10518 \cdot 10^7$ | 10 | 3606.04 | 0.10 |
| 500 | MSK persp. + warm start | $-1.10518 \cdot 10^7$ | 10 | 3600.96 | 0.10 |
| 500 | Convex MSK opt. persp. | $-1.10518 \cdot 10^7$ | 10 | 7.89 | 0.01 |
| 500 | ours | $-1.10518 \cdot 10^7$ | 10 | 0.40 | 0.00 |
| 1000 | SubsetSelectCIO | $-7.35715 \cdot 10^6$ | 2 | 3.71 | 0.00 |
| 1000 | SubsetSelectCIO + warm start | $-7.35715 \cdot 10^6$ | 2 | 4.39 | 0.00 |
| 1000 | L0BnB | $-1.09508 \cdot 10^7$ | 10 | 1931.45 | 0.65 |
| 1000 | MSK persp. | - | - | 3708.29 | - |
| 1000 | MSK persp. + warm start | $-1.09508 \cdot 10^7$ | 10 | 3692.66 | 0.00 |
| 1000 | Convex MSK opt. persp. | $-1.09508 \cdot 10^7$ | 10 | 56.04 | 0.01 |
| 1000 | ours | $-1.09508 \cdot 10^7$ | 10 | 0.59 | 0.00 |
| 3000 | SubsetSelectCIO | $-7.42176 \cdot 10^6$ | 2 | 8.32 | 0.00 |
| 3000 | SubsetSelectCIO + warm start | $-7.42176 \cdot 10^6$ | 2 | 8.23 | 0.00 |
| 3000 | L0BnB | $-1.10012 \cdot 10^7$ | 10 | 5903.24 | 1.27 |
| 3000 | MSK persp. | - | - | 3600.00 | - |
| 3000 | MSK persp. + warm start | $-1.10012 \cdot 10^7$ | 10 | 4174.40 | 0.00 |
| 3000 | Convex MSK opt. persp. | $-1.10012 \cdot 10^7$ | 10 | 1848.58 | 0.02 |
| 3000 | ours | $-1.10012 \cdot 10^7$ | 10 | 1.74 | 0.00 |
| 5000 | SubsetSelectCIO | $-7.37395 \cdot 10^6$ | 2 | 8.42 | 0.00 |
| 5000 | SubsetSelectCIO + warm start | $-7.37395 \cdot 10^6$ | 2 | 11.87 | 0.00 |
| 5000 | L0BnB | $-1.09966 \cdot 10^7$ | 10 | 6029.87 | 1.11 |
| 5000 | MSK persp. | - | - | 3600.00 | - |
| 5000 | MSK persp. + warm start | $-1.09966 \cdot 10^7$ | 10 | 6700.63 | 0.00 |
| 5000 | Convex MSK opt. persp. | $-1.09966 \cdot 10^7$ | 10 | 3657.70 | 0.06 |
| 5000 | ours | $-1.09966 \cdot 10^7$ | 10 | 4.00 | 0.00 |

Table 123: Comparison of time and optimality gap on the synthetic datasets (n = 100000, k = 10, $\rho = 0.5$, $\lambda_2 = 0.001$).

| # of features | method | upper bound | support size | time(s) | gap(%) |
|---|---|---|---|---|---|
| 100 | SubsetSelectCIO | $-2.01567 \cdot 10^7$ | 2 | 2.50 | 0.00 |
| 100 | SubsetSelectCIO + warm start | $-2.01567 \cdot 10^7$ | 2 | 2.64 | 0.00 |
| 100 | L0BnB | $-2.43575 \cdot 10^7$ | 10 | 2716.56 | 0.00 |
| 100 | MSK persp. | $-2.43575 \cdot 10^7$ | 10 | 3602.86 | 0.01 |
| 100 | MSK persp. + warm start | $-2.43575 \cdot 10^7$ | 10 | 3600.41 | 0.01 |
| 100 | Convex MSK opt. persp. | $-2.43575 \cdot 10^7$ | 10 | 0.12 | 0.00 |
| 100 | ours | $-2.43575 \cdot 10^7$ | 10 | 0.32 | 0.00 |
| 500 | SubsetSelectCIO | $-2.01420 \cdot 10^7$ | 2 | 3.26 | 0.00 |
| 500 | SubsetSelectCIO + warm start | $-2.01420 \cdot 10^7$ | 2 | 3.40 | 0.00 |
| 500 | L0BnB | $-2.44185 \cdot 10^7$ | 10 | 3747.19 | 0.00 |
| 500 | MSK persp. | $-2.24367 \cdot 10^7$ | 10 | 3608.22 | 8.94 |
| 500 | MSK persp. + warm start | $-2.44185 \cdot 10^7$ | 10 | 3623.31 | 0.10 |
| 500 | Convex MSK opt. persp. | $-2.44185 \cdot 10^7$ | 10 | 8.07 | 0.01 |
| 500 | ours | $-2.44185 \cdot 10^7$ | 10 | 0.46 | 0.00 |
| 1000 | SubsetSelectCIO | $-1.99762 \cdot 10^7$ | 2 | 3.67 | 0.00 |
| 1000 | SubsetSelectCIO + warm start | $-1.99762 \cdot 10^7$ | 2 | 4.71 | 0.00 |
| 1000 | L0BnB | $-2.41994 \cdot 10^7$ | 11 | 4179.51 | 0.01 |
| 1000 | MSK persp. | - | - | 3620.44 | - |
| 1000 | MSK persp. + warm start | $-2.41989 \cdot 10^7$ | 10 | 3686.70 | 0.20 |
| 1000 | Convex MSK opt. persp. | $-2.41989 \cdot 10^7$ | 10 | 57.29 | 0.01 |
| 1000 | ours | $-2.41989 \cdot 10^7$ | 10 | 0.52 | 0.00 |
| 3000 | SubsetSelectCIO | $-2.01741 \cdot 10^7$ | 2 | 9.04 | 0.00 |
| 3000 | SubsetSelectCIO + warm start | $-2.01741 \cdot 10^7$ | 2 | 9.24 | 0.00 |
| 3000 | L0BnB | $-2.42599 \cdot 10^7$ | 9 | 7445.35 | 0.00 |
| 3000 | MSK persp. | - | - | 3600.00 | - |
| 3000 | MSK persp. + warm start | $-2.43713 \cdot 10^7$ | 10 | 3888.61 | 0.00 |
| 3000 | Convex MSK opt. persp. | $-2.43713 \cdot 10^7$ | 10 | 1929.94 | 0.02 |
| 3000 | ours | $-2.43713 \cdot 10^7$ | 10 | 1.77 | 0.00 |
| 5000 | SubsetSelectCIO | $-2.00985 \cdot 10^7$ | 2 | 13.61 | 0.00 |
| 5000 | SubsetSelectCIO + warm start | $-2.00985 \cdot 10^7$ | 2 | 12.21 | 0.00 |
| 5000 | L0BnB | $-2.43694 \cdot 10^7$ | 10 | 4485.99 | 23.68 |
| 5000 | MSK persp. | - | - | 3600.00 | - |
| 5000 | MSK persp. + warm start | - | - | 3600.00 | - |
| 5000 | Convex MSK opt. persp. | $-2.43694 \cdot 10^7$ | 10 | 3748.60 | 0.06 |
| 5000 | ours | $-2.43694 \cdot 10^7$ | 10 | 4.15 | 0.00 |

Table 124: Comparison of time and optimality gap on the synthetic datasets (n = 100000, k = 10, $\rho = 0.7$, $\lambda_2 = 0.001$).

| # of features | method | upper bound | support size | time(s) | gap(%) |
|---|---|---|---|---|---|
| 100 | SubsetSelectCIO | $-8.64519 \cdot 10^7$ | 2 | 2.27 | 0.00 |
| 100 | SubsetSelectCIO + warm start | $-8.64519 \cdot 10^7$ | 2 | 2.79 | 0.00 |
| 100 | L0BnB | $-9.11185 \cdot 10^7$ | 93 | 6954.00 | 0.04 |
| 100 | MSK persp. | $-8.96852 \cdot 10^7$ | 10 | 3601.43 | 1.59 |
| 100 | MSK persp. + warm start | $-9.11026 \cdot 10^7$ | 10 | 3601.28 | 0.01 |
| 100 | Convex MSK opt. persp. | $-9.11026 \cdot 10^7$ | 10 | 0.13 | 0.00 |
| 100 | ours | $-9.11026 \cdot 10^7$ | 10 | 0.30 | 0.00 |
| 500 | SubsetSelectCIO | $-8.63791 \cdot 10^7$ | 2 | 2.89 | 0.00 |
| 500 | SubsetSelectCIO + warm start | $-8.63791 \cdot 10^7$ | 2 | 3.47 | 0.00 |
| 500 | L0BnB | $-9.10885 \cdot 10^7$ | 9 | 7562.05 | 0.14 |
| 500 | MSK persp. | - | - | 3605.36 | - |
| 500 | MSK persp. + warm start | $-9.12047 \cdot 10^7$ | 10 | 3630.75 | 0.10 |
| 500 | Convex MSK opt. persp. | $-9.12047 \cdot 10^7$ | 10 | 7.26 | 0.01 |
| 500 | ours | $-9.12047 \cdot 10^7$ | 10 | 0.38 | 0.00 |
| 1000 | SubsetSelectCIO | $-8.56943 \cdot 10^7$ | 2 | 4.58 | 0.00 |
| 1000 | SubsetSelectCIO + warm start | $-8.56943 \cdot 10^7$ | 2 | 4.73 | 0.00 |
| 1000 | L0BnB | $-8.23906 \cdot 10^7$ | 1 | 7300.29 | 9.17 |
| 1000 | MSK persp. | - | - | 3646.29 | - |
| 1000 | MSK persp. + warm start | $-9.04075 \cdot 10^7$ | 10 | 3613.37 | 0.00 |
| 1000 | Convex MSK opt. persp. | $-9.04075 \cdot 10^7$ | 10 | 67.12 | 0.01 |
| 1000 | ours | $-9.04075 \cdot 10^7$ | 10 | 0.52 | 0.00 |
| 3000 | SubsetSelectCIO | $-8.66243 \cdot 10^7$ | 2 | 8.50 | 0.00 |
| 3000 | SubsetSelectCIO + warm start | $-8.66243 \cdot 10^7$ | 2 | 8.12 | 0.00 |
| 3000 | L0BnB | $-9.12781 \cdot 10^7$ | 12 | 1706.18 | 7.80 |
| 3000 | MSK persp. | - | - | 3600.00 | - |
| 3000 | MSK persp. + warm start | $-9.12761 \cdot 10^7$ | 10 | 3998.60 | 0.00 |
| 3000 | Convex MSK opt. persp. | $-9.12761 \cdot 10^7$ | 10 | 2038.29 | 0.02 |
| 3000 | ours | $-9.12761 \cdot 10^7$ | 10 | 20.74 | 0.00 |
| 5000 | SubsetSelectCIO | $-8.64926 \cdot 10^7$ | 2 | 12.63 | 0.00 |
| 5000 | SubsetSelectCIO + warm start | $-8.64926 \cdot 10^7$ | 2 | 11.94 | 0.00 |
| 5000 | L0BnB | $-8.32072 \cdot 10^7$ | 1 | 7472.42 | 9.32 |
| 5000 | MSK persp. | - | - | 3600.00 | - |
| 5000 | MSK persp. + warm start | - | - | 3600.00 | - |
| 5000 | Convex MSK opt. persp. | $-9.12916 \cdot 10^7$ | 10 | 3722.78 | 3.21 |
| 5000 | ours | $-9.12916 \cdot 10^7$ | 10 | 58.29 | 0.00 |

Table 125: Comparison of time and optimality gap on the synthetic datasets (n = 100000, k = 10, $\rho = 0.9$, $\lambda_2 = 0.001$).

**G.4.2 Synthetic 1 Benchmark with $\lambda_2 = 0.1$**

| # of features | method | upper bound | support size | time(s) | gap(%) |
|---|---|---|---|---|---|
| 100 | SubsetSelectCIO | - | - | 3600.00 | - |
| 100 | SubsetSelectCIO + warm start | - | - | 3600.00 | - |
| 100 | L0BnB | $-2.10878 \cdot 10^6$ | 10 | 1415.23 | 0.00 |
| 100 | MSK persp. | $-2.10878 \cdot 10^6$ | 10 | 928.86 | 0.00 |
| 100 | MSK persp. + warm start | $-2.10878 \cdot 10^6$ | 10 | 912.78 | 0.00 |
| 100 | Convex MSK opt. persp. | $-2.10878 \cdot 10^6$ | 10 | 0.10 | 0.00 |
| 100 | ours | $-2.10878 \cdot 10^6$ | 10 | 0.36 | 0.00 |
| 500 | SubsetSelectCIO | $-1.19239 \cdot 10^5$ | 1 | 3.17 | 0.00 |
| 500 | SubsetSelectCIO + warm start | $-1.19239 \cdot 10^5$ | 1 | 3.39 | 0.00 |
| 500 | L0BnB | $-2.12833 \cdot 10^6$ | 10 | 1973.90 | 0.00 |
| 500 | MSK persp. | $-2.12830 \cdot 10^6$ | 10 | 3606.82 | 0.10 |
| 500 | MSK persp. + warm start | $-2.12833 \cdot 10^6$ | 10 | 3604.01 | 0.10 |
| 500 | Convex MSK opt. persp. | $-2.12833 \cdot 10^6$ | 10 | 6.49 | 0.01 |
| 500 | ours | $-2.12833 \cdot 10^6$ | 10 | 0.37 | 0.00 |
| 1000 | SubsetSelectCIO | $-4.71433 \cdot 10^5$ | 2 | 4.47 | 0.00 |
| 1000 | SubsetSelectCIO + warm start | $-4.71433 \cdot 10^5$ | 2 | 4.68 | 0.00 |
| 1000 | L0BnB | $-2.11005 \cdot 10^6$ | 10 | 2101.63 | 0.00 |
| 1000 | MSK persp. | - | - | 3602.15 | - |
| 1000 | MSK persp. + warm start | $-2.11005 \cdot 10^6$ | 10 | 3600.98 | 0.20 |
| 1000 | Convex MSK opt. persp. | $-2.11005 \cdot 10^6$ | 10 | 58.36 | 0.01 |
| 1000 | ours | $-2.11005 \cdot 10^6$ | 10 | 0.57 | 0.00 |
| 3000 | SubsetSelectCIO | $-4.72135 \cdot 10^5$ | 2 | 8.81 | 0.00 |
| 3000 | SubsetSelectCIO + warm start | $-4.72135 \cdot 10^5$ | 2 | 8.98 | 0.00 |
| 3000 | L0BnB | $-2.10230 \cdot 10^6$ | 10 | 2515.21 | 0.00 |
| 3000 | MSK persp. | - | - | 3600.00 | - |
| 3000 | MSK persp. + warm start | $-2.10230 \cdot 10^6$ | 10 | 3660.82 | 0.00 |
| 3000 | Convex MSK opt. persp. | $-2.10230 \cdot 10^6$ | 10 | 1765.70 | 0.02 |
| 3000 | ours | $-2.10230 \cdot 10^6$ | 10 | 1.79 | 0.00 |
| 5000 | SubsetSelectCIO | $-4.62647 \cdot 10^5$ | 2 | 12.26 | 0.00 |
| 5000 | SubsetSelectCIO + warm start | $-4.62647 \cdot 10^5$ | 2 | 12.06 | 0.00 |
| 5000 | L0BnB | $-2.09635 \cdot 10^6$ | 10 | 2666.21 | 3.42 |
| 5000 | MSK persp. | - | - | 3600.00 | - |
| 5000 | MSK persp. + warm start | - | - | 3600.00 | - |
| 5000 | Convex MSK opt. persp. | $-2.09635 \cdot 10^6$ | 10 | 3685.08 | 0.06 |
| 5000 | ours | $-2.09635 \cdot 10^6$ | 10 | 4.20 | 0.00 |

Table 126: Comparison of time and optimality gap on the synthetic datasets (n = 100000, k = 10, $\rho = 0.1$, $\lambda_2 = 0.1$).

| # of features | method | upper bound | support size | time(s) | gap(%) |
|---|---|---|---|---|---|
| 100 | SubsetSelectCIO | $-2.52792 \cdot 10^6$ | 2 | 2.32 | 0.00 |
| 100 | SubsetSelectCIO + warm start | $-2.52792 \cdot 10^6$ | 2 | 2.46 | 0.00 |
| 100 | L0BnB | $-5.28726 \cdot 10^6$ | 10 | 3182.16 | 0.00 |
| 100 | MSK persp. | $-5.28724 \cdot 10^6$ | 10 | 3601.53 | 0.01 |
| 100 | MSK persp. + warm start | $-5.28726 \cdot 10^6$ | 10 | 3600.02 | 0.01 |
| 100 | Convex MSK opt. persp. | $-5.28726 \cdot 10^6$ | 10 | 0.10 | 0.00 |
| 100 | ours | $-5.28726 \cdot 10^6$ | 10 | 0.27 | 0.00 |
| 500 | SubsetSelectCIO | $-2.52985 \cdot 10^6$ | 2 | 3.54 | 0.00 |
| 500 | SubsetSelectCIO + warm start | $-2.52985 \cdot 10^6$ | 2 | 3.26 | 0.00 |
| 500 | L0BnB | $-5.31852 \cdot 10^6$ | 10 | 3918.77 | 1.65 |
| 500 | MSK persp. | $-5.31852 \cdot 10^6$ | 10 | 3601.88 | 0.10 |
| 500 | MSK persp. + warm start | $-5.31852 \cdot 10^6$ | 10 | 3602.35 | 0.10 |
| 500 | Convex MSK opt. persp. | $-5.31852 \cdot 10^6$ | 10 | 8.26 | 0.01 |
| 500 | ours | $-5.31852 \cdot 10^6$ | 10 | 0.41 | 0.00 |
| 1000 | SubsetSelectCIO | $-2.50201 \cdot 10^6$ | 2 | 4.49 | 0.00 |
| 1000 | SubsetSelectCIO + warm start | $-2.50201 \cdot 10^6$ | 2 | 4.12 | 0.00 |
| 1000 | L0BnB | $-5.26974 \cdot 10^6$ | 10 | 4052.64 | 1.67 |
| 1000 | MSK persp. | - | - | 3600.90 | - |
| 1000 | MSK persp. + warm start | $-5.26974 \cdot 10^6$ | 10 | 3600.10 | 0.20 |
| 1000 | Convex MSK opt. persp. | $-5.26974 \cdot 10^6$ | 10 | 55.50 | 0.01 |
| 1000 | ours | $-5.26974 \cdot 10^6$ | 10 | 0.52 | 0.00 |
| 3000 | SubsetSelectCIO | $-2.51861 \cdot 10^6$ | 2 | 5.94 | 0.00 |
| 3000 | SubsetSelectCIO + warm start | $-2.51861 \cdot 10^6$ | 2 | 8.61 | 0.00 |
| 3000 | L0BnB | $-5.27668 \cdot 10^6$ | 10 | 4807.82 | 2.18 |
| 3000 | MSK persp. | - | - | 3600.00 | - |
| 3000 | MSK persp. + warm start | $-5.27668 \cdot 10^6$ | 10 | 3744.12 | 0.00 |
| 3000 | Convex MSK opt. persp. | $-5.27668 \cdot 10^6$ | 10 | 1795.54 | 0.02 |
| 3000 | ours | $-5.27668 \cdot 10^6$ | 10 | 1.84 | 0.00 |
| 5000 | SubsetSelectCIO | $-2.49167 \cdot 10^6$ | 2 | 13.36 | 0.00 |
| 5000 | SubsetSelectCIO + warm start | $-2.49167 \cdot 10^6$ | 2 | 12.61 | 0.00 |
| 5000 | L0BnB | $-5.27126 \cdot 10^6$ | 10 | 5667.77 | 0.00 |
| 5000 | MSK persp. | - | - | 3600.00 | - |
| 5000 | MSK persp. + warm start | - | - | 3600.00 | - |
| 5000 | Convex MSK opt. persp. | $-5.27126 \cdot 10^6$ | 10 | 3606.96 | 0.05 |
| 5000 | ours | $-5.27126 \cdot 10^6$ | 10 | 4.42 | 0.00 |

Table 127: Comparison of time and optimality gap on the synthetic datasets (n = 100000, k = 10, $\rho = 0.3$, $\lambda_2 = 0.1$).

| # of features | method | upper bound | support size | time(s) | gap(%) |
|---|---|---|---|---|---|
| 100 | SubsetSelectCIO | $-7.42601 \cdot 10^6$ | 2 | 2.59 | 0.00 |
| 100 | SubsetSelectCIO + warm start | $-7.42601 \cdot 10^6$ | 2 | 2.80 | 0.00 |
| 100 | L0BnB | $-1.10084 \cdot 10^7$ | 10 | 3398.37 | 0.00 |
| 100 | MSK persp. | $-1.09018 \cdot 10^7$ | 10 | 3599.98 | 0.99 |
| 100 | MSK persp. + warm start | $-1.10084 \cdot 10^7$ | 10 | 3603.80 | 0.01 |
| 100 | Convex MSK opt. persp. | $-1.10084 \cdot 10^7$ | 10 | 0.11 | 0.00 |
| 100 | ours | $-1.10084 \cdot 10^7$ | 10 | 0.32 | 0.00 |
| 500 | SubsetSelectCIO | $-7.42485 \cdot 10^6$ | 2 | 2.97 | 0.00 |
| 500 | SubsetSelectCIO + warm start | $-7.42485 \cdot 10^6$ | 2 | 3.22 | 0.00 |
| 500 | L0BnB | $-1.10518 \cdot 10^7$ | 10 | 4364.09 | 0.77 |
| 500 | MSK persp. | - | - | 3605.36 | - |
| 500 | MSK persp. + warm start | $-1.10518 \cdot 10^7$ | 10 | 3610.84 | 0.00 |
| 500 | Convex MSK opt. persp. | $-1.10518 \cdot 10^7$ | 10 | 7.25 | 0.01 |
| 500 | ours | $-1.10518 \cdot 10^7$ | 10 | 0.40 | 0.00 |
| 1000 | SubsetSelectCIO | $-7.35715 \cdot 10^6$ | 2 | 4.35 | 0.00 |
| 1000 | SubsetSelectCIO + warm start | $-7.35715 \cdot 10^6$ | 2 | 4.97 | 0.00 |
| 1000 | L0BnB | $-1.09508 \cdot 10^7$ | 10 | 1882.56 | 0.65 |
| 1000 | MSK persp. | - | - | 3609.88 | - |
| 1000 | MSK persp. + warm start | $-1.09508 \cdot 10^7$ | 10 | 3616.06 | 0.00 |
| 1000 | Convex MSK opt. persp. | $-1.09508 \cdot 10^7$ | 10 | 63.89 | 0.01 |
| 1000 | ours | $-1.09508 \cdot 10^7$ | 10 | 0.53 | 0.00 |
| 3000 | SubsetSelectCIO | $-7.42176 \cdot 10^6$ | 2 | 5.59 | 0.00 |
| 3000 | SubsetSelectCIO + warm start | $-7.42176 \cdot 10^6$ | 2 | 8.42 | 0.00 |
| 3000 | L0BnB | $-1.10012 \cdot 10^7$ | 10 | 5588.51 | 1.24 |
| 3000 | MSK persp. | - | - | 3600.00 | - |
| 3000 | MSK persp. + warm start | $-1.10012 \cdot 10^7$ | 10 | 4173.41 | 0.00 |
| 3000 | Convex MSK opt. persp. | $-1.10012 \cdot 10^7$ | 10 | 1779.87 | 0.02 |
| 3000 | ours | $-1.10012 \cdot 10^7$ | 10 | 1.67 | 0.00 |
| 5000 | SubsetSelectCIO | $-7.37395 \cdot 10^6$ | 2 | 8.72 | 0.00 |
| 5000 | SubsetSelectCIO + warm start | $-7.37395 \cdot 10^6$ | 2 | 12.24 | 0.00 |
| 5000 | L0BnB | $-1.09966 \cdot 10^7$ | 10 | 6428.17 | 1.09 |
| 5000 | MSK persp. | - | - | 3600.00 | - |
| 5000 | MSK persp. + warm start | $-1.09966 \cdot 10^7$ | 10 | 6106.85 | 0.00 |
| 5000 | Convex MSK opt. persp. | $-1.09966 \cdot 10^7$ | 10 | 3665.75 | 0.06 |
| 5000 | ours | $-1.09966 \cdot 10^7$ | 10 | 4.13 | 0.00 |

Table 128: Comparison of time and optimality gap on the synthetic datasets (n = 100000, k = 10, $\rho = 0.5$, $\lambda_2 = 0.1$).

| # of features | method | upper bound | support size | time(s) | gap(%) |
|---|---|---|---|---|---|
| 100 | SubsetSelectCIO | $-2.01567 \cdot 10^7$ | 2 | 2.23 | 0.00 |
| 100 | SubsetSelectCIO + warm start | $-2.01567 \cdot 10^7$ | 2 | 3.02 | 0.00 |
| 100 | L0BnB | $-2.43575 \cdot 10^7$ | 10 | 2673.71 | 0.00 |
| 100 | MSK persp. | $-2.41233 \cdot 10^7$ | 10 | 3601.54 | 0.98 |
| 100 | MSK persp. + warm start | $-2.43575 \cdot 10^7$ | 10 | 3600.03 | 0.01 |
| 100 | Convex MSK opt. persp. | $-2.43575 \cdot 10^7$ | 10 | 0.14 | 0.00 |
| 100 | ours | $-2.43575 \cdot 10^7$ | 10 | 0.34 | 0.00 |
| 500 | SubsetSelectCIO | $-2.01420 \cdot 10^7$ | 2 | 3.58 | 0.00 |
| 500 | SubsetSelectCIO + warm start | $-2.01420 \cdot 10^7$ | 2 | 3.34 | 0.00 |
| 500 | L0BnB | $-2.44185 \cdot 10^7$ | 10 | 3602.76 | 0.00 |
| 500 | MSK persp. | $-2.44185 \cdot 10^7$ | 10 | 3603.68 | 0.10 |
| 500 | MSK persp. + warm start | $-2.44185 \cdot 10^7$ | 10 | 3604.92 | 0.10 |
| 500 | Convex MSK opt. persp. | $-2.44185 \cdot 10^7$ | 10 | 7.37 | 0.01 |
| 500 | ours | $-2.44185 \cdot 10^7$ | 10 | 0.44 | 0.00 |
| 1000 | SubsetSelectCIO | $-1.99762 \cdot 10^7$ | 2 | 4.47 | 0.00 |
| 1000 | SubsetSelectCIO + warm start | $-1.99762 \cdot 10^7$ | 2 | 4.56 | 0.00 |
| 1000 | L0BnB | $-2.41994 \cdot 10^7$ | 11 | 4204.09 | 0.01 |
| 1000 | MSK persp. | - | - | 3610.81 | - |
| 1000 | MSK persp. + warm start | $-2.41989 \cdot 10^7$ | 10 | 3609.15 | 0.00 |
| 1000 | Convex MSK opt. persp. | $-2.41989 \cdot 10^7$ | 10 | 56.82 | 0.01 |
| 1000 | ours | $-2.41989 \cdot 10^7$ | 10 | 0.53 | 0.00 |
| 3000 | SubsetSelectCIO | $-2.01741 \cdot 10^7$ | 2 | 5.42 | 0.00 |
| 3000 | SubsetSelectCIO + warm start | $-2.01741 \cdot 10^7$ | 2 | 8.40 | 0.00 |
| 3000 | L0BnB | $-2.42599 \cdot 10^7$ | 9 | 7207.55 | 0.00 |
| 3000 | MSK persp. | - | - | 3600.00 | - |
| 3000 | MSK persp. + warm start | $-2.43713 \cdot 10^7$ | 10 | 3980.12 | 0.00 |
| 3000 | Convex MSK opt. persp. | $-2.43713 \cdot 10^7$ | 10 | 1799.08 | 0.02 |
| 3000 | ours | $-2.43713 \cdot 10^7$ | 10 | 1.75 | 0.00 |
| 5000 | SubsetSelectCIO | $-2.00985 \cdot 10^7$ | 2 | 12.64 | 0.00 |
| 5000 | SubsetSelectCIO + warm start | $-2.00985 \cdot 10^7$ | 2 | 13.96 | 0.00 |
| 5000 | L0BnB | $-2.43694 \cdot 10^7$ | 10 | 4979.94 | 21.63 |
| 5000 | MSK persp. | - | - | 3600.00 | - |
| 5000 | MSK persp. + warm start | - | - | 3600.00 | - |
| 5000 | Convex MSK opt. persp. | $-2.43694 \cdot 10^7$ | 10 | 3606.45 | 0.06 |
| 5000 | ours | $-2.43694 \cdot 10^7$ | 10 | 4.22 | 0.00 |

Table 129: Comparison of time and optimality gap on the synthetic datasets (n = 100000, k = 10, $\rho = 0.7$, $\lambda_2 = 0.1$).

| # of features | method | upper bound | support size | time(s) | gap(%) |
|---|---|---|---|---|---|
| 100 | SubsetSelectCIO | $-8.11196 \cdot 10^7$ | 1 | 2.42 | 0.00 |
| 100 | SubsetSelectCIO + warm start | $-8.11196 \cdot 10^7$ | 1 | 2.56 | 0.00 |
| 100 | L0BnB | $-9.11185 \cdot 10^7$ | 93 | 6919.82 | 0.04 |
| 100 | MSK persp. | $-8.94241 \cdot 10^7$ | 10 | 3600.44 | 1.89 |
| 100 | MSK persp. + warm start | $-9.11026 \cdot 10^7$ | 10 | 3600.46 | 0.01 |
| 100 | Convex MSK opt. persp. | $-9.11026 \cdot 10^7$ | 10 | 0.18 | 0.00 |
| 100 | ours | $-9.11026 \cdot 10^7$ | 10 | 0.31 | 0.00 |
| 500 | SubsetSelectCIO | $-8.63791 \cdot 10^7$ | 2 | 3.47 | 0.00 |
| 500 | SubsetSelectCIO + warm start | $-8.63791 \cdot 10^7$ | 2 | 3.56 | 0.00 |
| 500 | L0BnB | $-9.10885 \cdot 10^7$ | 9 | 7359.92 | 0.14 |
| 500 | MSK persp. | - | - | 3601.63 | - |
| 500 | MSK persp. + warm start | $-9.12047 \cdot 10^7$ | 10 | 3602.72 | 0.00 |
| 500 | Convex MSK opt. persp. | $-9.12047 \cdot 10^7$ | 10 | 8.27 | 0.01 |
| 500 | ours | $-9.12047 \cdot 10^7$ | 10 | 0.38 | 0.00 |
| 1000 | SubsetSelectCIO | $-8.56943 \cdot 10^7$ | 2 | 4.83 | 0.00 |
| 1000 | SubsetSelectCIO + warm start | $-8.56943 \cdot 10^7$ | 2 | 4.74 | 0.00 |
| 1000 | L0BnB | $-8.23906 \cdot 10^7$ | 1 | 7326.90 | 9.06 |
| 1000 | MSK persp. | - | - | 3641.35 | - |
| 1000 | MSK persp. + warm start | $-9.04075 \cdot 10^7$ | 10 | 3611.36 | 0.00 |
| 1000 | Convex MSK opt. persp. | $-9.04075 \cdot 10^7$ | 10 | 61.25 | 0.01 |
| 1000 | ours | $-9.04075 \cdot 10^7$ | 10 | 0.52 | 0.00 |
| 3000 | SubsetSelectCIO | $-8.66243 \cdot 10^7$ | 2 | 8.37 | 0.00 |
| 3000 | SubsetSelectCIO + warm start | $-8.66243 \cdot 10^7$ | 2 | 9.68 | 0.00 |
| 3000 | L0BnB | $-9.12781 \cdot 10^7$ | 12 | 2205.35 | 5.43 |
| 3000 | MSK persp. | - | - | 3600.00 | - |
| 3000 | MSK persp. + warm start | $-9.12761 \cdot 10^7$ | 10 | 3661.26 | 0.00 |
| 3000 | Convex MSK opt. persp. | $-9.12761 \cdot 10^7$ | 10 | 2087.06 | 0.02 |
| 3000 | ours | $-9.12761 \cdot 10^7$ | 10 | 20.88 | 0.00 |
| 5000 | SubsetSelectCIO | $-8.64926 \cdot 10^7$ | 2 | 12.48 | 0.00 |
| 5000 | SubsetSelectCIO + warm start | $-8.64926 \cdot 10^7$ | 2 | 12.73 | 0.00 |
| 5000 | L0BnB | $-8.32072 \cdot 10^7$ | 1 | 7201.18 | 9.32 |
| 5000 | MSK persp. | - | - | 3600.00 | - |
| 5000 | MSK persp. + warm start | $-9.12916 \cdot 10^7$ | 10 | 8314.78 | 0.00 |
| 5000 | Convex MSK opt. persp. | $-9.12916 \cdot 10^7$ | 10 | 3691.61 | 3.40 |
| 5000 | ours | $-9.12916 \cdot 10^7$ | 10 | 58.07 | 0.00 |

Table 130: Comparison of time and optimality gap on the synthetic datasets (n = 100000, k = 10, $\rho = 0.9$, $\lambda_2 = 0.1$).

### G.4.3 Synthetic 1 Benchmark with $\lambda_2 = 10.0$

| # of features | method | upper bound | support size | time(s) | gap(%) |
|---|---|---|---|---|---|
| 100 | SubsetSelectCIO | $-1.62628 \cdot 10^6$ | 10 | 3693.06 | 55.44 |
| 100 | SubsetSelectCIO + warm start | $-1.62806 \cdot 10^6$ | 10 | 3672.85 | 55.27 |
| 100 | L0BnB | $-2.10868 \cdot 10^6$ | 10 | 1432.58 | 0.00 |
| 100 | MSK persp. | $-2.10868 \cdot 10^6$ | 10 | 926.33 | 0.00 |
| 100 | MSK persp. + warm start | $-2.10868 \cdot 10^6$ | 10 | 919.61 | 0.00 |
| 100 | Convex MSK opt. persp. | $-2.10868 \cdot 10^6$ | 10 | 0.10 | 0.00 |
| 100 | ours | $-2.10868 \cdot 10^6$ | 10 | 0.30 | 0.00 |
| 500 | SubsetSelectCIO | $-1.33331 \cdot 10^6$ | 10 | 3776.44 | 91.62 |
| 500 | SubsetSelectCIO + warm start | $-4.87330 \cdot 10^5$ | 7 | 1654.44 | 0.00 |
| 500 | L0BnB | $-2.12823 \cdot 10^6$ | 10 | 1929.01 | 0.00 |
| 500 | MSK persp. | $-2.12822 \cdot 10^6$ | 10 | 3601.80 | 0.09 |
| 500 | MSK persp. + warm start | $-2.12823 \cdot 10^6$ | 10 | 3609.69 | 0.09 |
| 500 | Convex MSK opt. persp. | $-2.12823 \cdot 10^6$ | 10 | 7.05 | 0.01 |
| 500 | ours | $-2.12823 \cdot 10^6$ | 10 | 0.38 | 0.00 |
| 1000 | SubsetSelectCIO | $-7.92745 \cdot 10^5$ | 10 | 3709.35 | 219.40 |
| 1000 | SubsetSelectCIO + warm start | $-1.33080 \cdot 10^6$ | 10 | 3702.48 | 90.26 |
| 1000 | L0BnB | $-2.10995 \cdot 10^6$ | 10 | 2118.04 | 0.00 |
| 1000 | MSK persp. | - | - | 3601.31 | - |
| 1000 | MSK persp. + warm start | $-2.10995 \cdot 10^6$ | 10 | 3600.48 | 0.20 |
| 1000 | Convex MSK opt. persp. | $-2.10995 \cdot 10^6$ | 10 | 54.57 | 0.01 |
| 1000 | ours | $-2.10995 \cdot 10^6$ | 10 | 0.57 | 0.00 |
| 3000 | SubsetSelectCIO | $-7.89940 \cdot 10^5$ | 10 | 3633.47 | 219.21 |
| 3000 | SubsetSelectCIO + warm start | $-1.31406 \cdot 10^6$ | 10 | 4260.43 | 91.89 |
| 3000 | L0BnB | $-2.10220 \cdot 10^6$ | 10 | 2318.67 | 4.73 |
| 3000 | MSK persp. | - | - | 3600.00 | - |
| 3000 | MSK persp. + warm start | - | - | 3600.00 | - |
| 3000 | Convex MSK opt. persp. | $-2.10220 \cdot 10^6$ | 10 | 1669.36 | 0.02 |
| 3000 | ours | $-2.10220 \cdot 10^6$ | 10 | 1.68 | 0.00 |
| 5000 | SubsetSelectCIO | $-9.76808 \cdot 10^5$ | 10 | 3838.13 | 157.56 |
| 5000 | SubsetSelectCIO + warm start | $-1.30652 \cdot 10^6$ | 10 | 4063.53 | 92.56 |
| 5000 | L0BnB | $-2.09625 \cdot 10^6$ | 10 | 2700.89 | 3.42 |
| 5000 | MSK persp. | - | - | 3600.00 | - |
| 5000 | MSK persp. + warm start | - | - | 3600.00 | - |
| 5000 | Convex MSK opt. persp. | $-2.09625 \cdot 10^6$ | 10 | 3669.10 | 0.06 |
| 5000 | ours | $-2.09625 \cdot 10^6$ | 10 | 5.09 | 0.00 |

Table 131: Comparison of time and optimality gap on the synthetic datasets (n = 100000, k = 10, $\rho = 0.1$, $\lambda_2 = 10.0$).

| # of features | method | upper bound | support size | time(s) | gap(%) |
|---|---|---|---|---|---|
| 100 | SubsetSelectCIO | $-4.75666 \cdot 10^6$ | 10 | 3695.63 | 33.24 |
| 100 | SubsetSelectCIO + warm start | $-3.72965 \cdot 10^6$ | 10 | 1008.71 | 0.00 |
| 100 | L0BnB | $-5.28716 \cdot 10^6$ | 10 | 3186.48 | 0.00 |
| 100 | MSK persp. | $-5.28716 \cdot 10^6$ | 10 | 3600.15 | 0.01 |
| 100 | MSK persp. + warm start | $-5.28716 \cdot 10^6$ | 10 | 3600.07 | 0.01 |
| 100 | Convex MSK opt. persp. | $-5.28716 \cdot 10^6$ | 10 | 0.16 | 0.00 |
| 100 | ours | $-5.28716 \cdot 10^6$ | 10 | 0.30 | 0.00 |
| 500 | SubsetSelectCIO | $-4.61560 \cdot 10^6$ | 10 | 3761.41 | 38.34 |
| 500 | SubsetSelectCIO + warm start | $-4.00602 \cdot 10^6$ | 10 | 3621.18 | 59.39 |
| 500 | L0BnB | $-5.31842 \cdot 10^6$ | 10 | 3958.53 | 1.65 |
| 500 | MSK persp. | $-5.31842 \cdot 10^6$ | 10 | 3604.61 | 0.09 |
| 500 | MSK persp. + warm start | $-5.31842 \cdot 10^6$ | 10 | 3604.65 | 0.09 |
| 500 | Convex MSK opt. persp. | $-5.31842 \cdot 10^6$ | 10 | 7.05 | 0.01 |
| 500 | ours | $-5.31842 \cdot 10^6$ | 10 | 0.41 | 0.00 |
| 1000 | SubsetSelectCIO | $-3.05644 \cdot 10^6$ | 6 | 3348.60 | 0.00 |
| 1000 | SubsetSelectCIO + warm start | $-3.95760 \cdot 10^6$ | 10 | 3658.16 | 59.79 |
| 1000 | L0BnB | $-5.26964 \cdot 10^6$ | 10 | 4043.93 | 1.62 |
| 1000 | MSK persp. | - | - | 3600.44 | - |
| 1000 | MSK persp. + warm start | $-5.26964 \cdot 10^6$ | 10 | 3600.14 | 0.20 |
| 1000 | Convex MSK opt. persp. | $-5.26964 \cdot 10^6$ | 10 | 54.61 | 0.01 |
| 1000 | ours | $-5.26964 \cdot 10^6$ | 10 | 0.51 | 0.00 |
| 3000 | SubsetSelectCIO | $-4.74347 \cdot 10^6$ | 10 | 3927.16 | 33.42 |
| 3000 | SubsetSelectCIO + warm start | - | - | 3600.00 | - |
| 3000 | L0BnB | $-5.27658 \cdot 10^6$ | 10 | 4906.10 | 0.35 |
| 3000 | MSK persp. | - | - | 3600.00 | - |
| 3000 | MSK persp. + warm start | - | - | 3600.00 | - |
| 3000 | Convex MSK opt. persp. | $-5.27658 \cdot 10^6$ | 10 | 1962.60 | 0.02 |
| 3000 | ours | $-5.27658 \cdot 10^6$ | 10 | 1.76 | 0.00 |
| 5000 | SubsetSelectCIO | - | - | 3600.00 | - |
| 5000 | SubsetSelectCIO + warm start | - | - | 3600.00 | - |
| 5000 | L0BnB | - | - | 7200.00 | - |
| 5000 | MSK persp. | - | - | 3600.00 | - |
| 5000 | MSK persp. + warm start | - | - | 3600.00 | - |
| 5000 | Convex MSK opt. persp. | $-5.27116 \cdot 10^6$ | 10 | 3651.58 | 0.05 |
| 5000 | ours | $-5.27116 \cdot 10^6$ | 10 | 4.32 | 0.00 |

Table 132: Comparison of time and optimality gap on the synthetic datasets (n = 100000, k = 10, $\rho = 0.3$, $\lambda_2 = 10.0$).

| # of features | method | upper bound | support size | time(s) | gap(%) |
|---|---|---|---|---|---|
| 100 | SubsetSelectCIO | $-1.00851 \cdot 10^7$ | 10 | 3728.97 | 30.84 |
| 100 | SubsetSelectCIO + warm start | $-1.02880 \cdot 10^7$ | 10 | 3654.11 | 28.26 |
| 100 | L0BnB | $-1.10083 \cdot 10^7$ | 10 | 3417.76 | 0.26 |
| 100 | MSK persp. | $-1.10083 \cdot 10^7$ | 10 | 1286.52 | 0.00 |
| 100 | MSK persp. + warm start | $-1.10083 \cdot 10^7$ | 10 | 1786.69 | 0.00 |
| 100 | Convex MSK opt. persp. | $-1.10083 \cdot 10^7$ | 10 | 0.16 | 0.00 |
| 100 | ours | $-1.10083 \cdot 10^7$ | 10 | 0.32 | 0.00 |
| 500 | SubsetSelectCIO | $-9.42612 \cdot 10^6$ | 10 | 3685.22 | 40.77 |
| 500 | SubsetSelectCIO + warm start | $-9.92278 \cdot 10^6$ | 10 | 3685.15 | 33.72 |
| 500 | L0BnB | $-1.10517 \cdot 10^7$ | 10 | 4308.04 | 0.77 |
| 500 | MSK persp. | $-1.10517 \cdot 10^7$ | 10 | 3601.09 | 0.09 |
| 500 | MSK persp. + warm start | $-1.10517 \cdot 10^7$ | 10 | 3602.10 | 0.09 |
| 500 | Convex MSK opt. persp. | $-1.10517 \cdot 10^7$ | 10 | 7.72 | 0.01 |
| 500 | ours | $-1.10517 \cdot 10^7$ | 10 | 0.41 | 0.00 |
| 1000 | SubsetSelectCIO | $-1.00437 \cdot 10^7$ | 10 | 3649.62 | 30.84 |
| 1000 | SubsetSelectCIO + warm start | $-6.62765 \cdot 10^6$ | 2 | 3147.50 | 0.00 |
| 1000 | L0BnB | $-1.09507 \cdot 10^7$ | 10 | 1834.54 | 0.65 |
| 1000 | MSK persp. | - | - | 3600.46 | - |
| 1000 | MSK persp. + warm start | $-1.09507 \cdot 10^7$ | 10 | 3601.83 | 0.20 |
| 1000 | Convex MSK opt. persp. | $-1.09507 \cdot 10^7$ | 10 | 57.71 | 0.01 |
| 1000 | ours | $-1.09507 \cdot 10^7$ | 10 | 0.55 | 0.00 |
| 3000 | SubsetSelectCIO | - | - | 3600.00 | - |
| 3000 | SubsetSelectCIO + warm start | - | - | 3600.00 | - |
| 3000 | L0BnB | $-1.10011 \cdot 10^7$ | 10 | 5263.44 | 1.10 |
| 3000 | MSK persp. | - | - | 3600.00 | - |
| 3000 | MSK persp. + warm start | $-1.10011 \cdot 10^7$ | 10 | 4076.24 | 0.00 |
| 3000 | Convex MSK opt. persp. | $-1.10011 \cdot 10^7$ | 10 | 1538.48 | 0.02 |
| 3000 | ours | $-1.10011 \cdot 10^7$ | 10 | 1.73 | 0.00 |
| 5000 | SubsetSelectCIO | - | - | 3600.00 | - |
| 5000 | SubsetSelectCIO + warm start | - | - | 3600.00 | - |
| 5000 | L0BnB | $-1.09965 \cdot 10^7$ | 10 | 6222.74 | 1.11 |
| 5000 | MSK persp. | - | - | 3600.00 | - |
| 5000 | MSK persp. + warm start | - | - | 3600.00 | - |
| 5000 | Convex MSK opt. persp. | $-1.09965 \cdot 10^7$ | 10 | 3742.63 | 0.05 |
| 5000 | ours | $-1.09965 \cdot 10^7$ | 10 | 4.24 | 0.00 |

Table 133: Comparison of time and optimality gap on the synthetic datasets (n = 100000, k = 10, $\rho = 0.5$, $\lambda_2 = 10.0$).

| # of features | method | upper bound | support size | time(s) | gap(%) |
|---|---|---|---|---|---|
| 100 | SubsetSelectCIO | $-2.36231 \cdot 10^7$ | 10 | 3732.54 | 23.59 |
| 100 | SubsetSelectCIO + warm start | $-2.32147 \cdot 10^7$ | 10 | 3623.89 | 25.77 |
| 100 | L0BnB | $-2.43574 \cdot 10^7$ | 10 | 2671.45 | 0.00 |
| 100 | MSK persp. | $-2.39754 \cdot 10^7$ | 10 | 3601.78 | 1.61 |
| 100 | MSK persp. + warm start | $-2.43574 \cdot 10^7$ | 10 | 3604.40 | 0.01 |
| 100 | Convex MSK opt. persp. | $-2.43574 \cdot 10^7$ | 10 | 0.13 | 0.00 |
| 100 | ours | $-2.43574 \cdot 10^7$ | 10 | 0.30 | 0.00 |
| 500 | SubsetSelectCIO | $-2.27569 \cdot 10^7$ | 10 | 3601.02 | 28.83 |
| 500 | SubsetSelectCIO + warm start | $-2.32447 \cdot 10^7$ | 10 | 3624.13 | 26.12 |
| 500 | L0BnB | $-2.44184 \cdot 10^7$ | 10 | 3822.46 | 0.00 |
| 500 | MSK persp. | - | - | 3605.48 | - |
| 500 | MSK persp. + warm start | $-2.44184 \cdot 10^7$ | 10 | 3603.64 | 0.00 |
| 500 | Convex MSK opt. persp. | $-2.44184 \cdot 10^7$ | 10 | 7.40 | 0.01 |
| 500 | ours | $-2.44184 \cdot 10^7$ | 10 | 0.44 | 0.00 |
| 1000 | SubsetSelectCIO | $-2.18726 \cdot 10^7$ | 7 | 2623.29 | 0.00 |
| 1000 | SubsetSelectCIO + warm start | $-2.32752 \cdot 10^7$ | 10 | 4344.29 | 24.76 |
| 1000 | L0BnB | $-2.41993 \cdot 10^7$ | 11 | 3983.89 | 0.31 |
| 1000 | MSK persp. | - | - | 3606.66 | - |
| 1000 | MSK persp. + warm start | $-2.41988 \cdot 10^7$ | 10 | 3608.58 | 0.00 |
| 1000 | Convex MSK opt. persp. | $-2.41988 \cdot 10^7$ | 10 | 63.01 | 0.01 |
| 1000 | ours | $-2.41988 \cdot 10^7$ | 10 | 0.52 | 0.00 |
| 3000 | SubsetSelectCIO | $-2.34295 \cdot 10^7$ | 10 | 3767.28 | 24.75 |
| 3000 | SubsetSelectCIO + warm start | - | - | 3600.00 | - |
| 3000 | L0BnB | $-2.42598 \cdot 10^7$ | 9 | 7332.12 | 0.00 |
| 3000 | MSK persp. | - | - | 3600.00 | - |
| 3000 | MSK persp. + warm start | $-2.43712 \cdot 10^7$ | 10 | 4135.58 | 0.00 |
| 3000 | Convex MSK opt. persp. | $-2.43712 \cdot 10^7$ | 10 | 1968.86 | 0.02 |
| 3000 | ours | $-2.43712 \cdot 10^7$ | 10 | 1.81 | 0.00 |
| 5000 | SubsetSelectCIO | - | - | 3600.00 | - |
| 5000 | SubsetSelectCIO + warm start | - | - | 3600.00 | - |
| 5000 | L0BnB | $-2.43693 \cdot 10^7$ | 10 | 4661.71 | 23.68 |
| 5000 | MSK persp. | - | - | 3600.00 | - |
| 5000 | MSK persp. + warm start | - | - | 3600.00 | - |
| 5000 | Convex MSK opt. persp. | $-2.43693 \cdot 10^7$ | 10 | 3769.92 | 0.06 |
| 5000 | ours | $-2.43693 \cdot 10^7$ | 10 | 4.08 | 0.00 |

Table 134: Comparison of time and optimality gap on the synthetic datasets (n = 100000, k = 10, $\rho = 0.7$, $\lambda_2 = 10.0$).

| # of features | method | upper bound | support size | time(s) | gap(%) |
|---|---|---|---|---|---|
| 100 | SubsetSelectCIO | $-8.97109 \cdot 10^7$ | 10 | 3602.39 | 21.73 |
| 100 | SubsetSelectCIO + warm start | $-9.05496 \cdot 10^7$ | 10 | 3626.15 | 20.60 |
| 100 | L0BnB | $-9.11184 \cdot 10^7$ | 93 | 7102.10 | 0.04 |
| 100 | MSK persp. | $-8.99265 \cdot 10^7$ | 10 | 3600.18 | 1.32 |
| 100 | MSK persp. + warm start | $-9.11025 \cdot 10^7$ | 10 | 3600.47 | 0.01 |
| 100 | Convex MSK opt. persp. | $-9.11025 \cdot 10^7$ | 10 | 0.13 | 0.00 |
| 100 | ours | $-9.11025 \cdot 10^7$ | 10 | 0.32 | 0.00 |
| 500 | SubsetSelectCIO | $-9.00037 \cdot 10^7$ | 10 | 4088.17 | 21.67 |
| 500 | SubsetSelectCIO + warm start | $-9.02231 \cdot 10^7$ | 10 | 3817.48 | 21.37 |
| 500 | L0BnB | $-9.11085 \cdot 10^7$ | 11 | 843.53 | 6.21 |
| 500 | MSK persp. | - | - | 3604.01 | - |
| 500 | MSK persp. + warm start | $-9.12046 \cdot 10^7$ | 10 | 3604.99 | 0.00 |
| 500 | Convex MSK opt. persp. | $-9.12046 \cdot 10^7$ | 10 | 7.57 | 0.01 |
| 500 | ours | $-9.12046 \cdot 10^7$ | 10 | 0.38 | 0.00 |
| 1000 | SubsetSelectCIO | - | - | 3600.00 | - |
| 1000 | SubsetSelectCIO + warm start | - | - | 3600.00 | - |
| 1000 | L0BnB | $-8.23898 \cdot 10^7$ | 1 | 7369.81 | 9.08 |
| 1000 | MSK persp. | - | - | 3604.50 | - |
| 1000 | MSK persp. + warm start | $-9.04074 \cdot 10^7$ | 10 | 3688.88 | 0.00 |
| 1000 | Convex MSK opt. persp. | $-9.04074 \cdot 10^7$ | 10 | 62.54 | 0.01 |
| 1000 | ours | $-9.04074 \cdot 10^7$ | 10 | 0.52 | 0.00 |
| 3000 | SubsetSelectCIO | - | - | 3600.00 | - |
| 3000 | SubsetSelectCIO + warm start | - | - | 3600.00 | - |
| 3000 | L0BnB | $-9.12760 \cdot 10^7$ | 10 | 1883.50 | 6.26 |
| 3000 | MSK persp. | - | - | 3600.00 | - |
| 3000 | MSK persp. + warm start | $-9.12760 \cdot 10^7$ | 10 | 3764.99 | 0.00 |
| 3000 | Convex MSK opt. persp. | $-9.12760 \cdot 10^7$ | 10 | 1999.52 | 0.02 |
| 3000 | ours | $-9.12760 \cdot 10^7$ | 10 | 20.84 | 0.00 |
| 5000 | SubsetSelectCIO | - | - | 3600.00 | - |
| 5000 | SubsetSelectCIO + warm start | - | - | 3600.00 | - |
| 5000 | L0BnB | $-8.32064 \cdot 10^7$ | 1 | 7664.70 | 9.32 |
| 5000 | MSK persp. | - | - | 3600.00 | - |
| 5000 | MSK persp. + warm start | $-9.12915 \cdot 10^7$ | 10 | 8218.76 | 0.00 |
| 5000 | Convex MSK opt. persp. | $-9.12915 \cdot 10^7$ | 10 | 3654.46 | 3.94 |
| 5000 | ours | $-9.12915 \cdot 10^7$ | 10 | 57.41 | 0.00 |

Table 135: Comparison of time and optimality gap on the synthetic datasets (n = 100000, k = 10, $\rho = 0.9$, $\lambda_2 = 10.0$).

### G.4.4 Synthetic 2 Benchmark with $\lambda_2 = 0.001$

| # of samples | method | upper bound | support size | time(s) | gap(%) |
|---|---|---|---|---|---|
| 3000 | SubsetSelectCIO | $-1.26607 \cdot 10^4$ | 2 | 1.93 | 0.00 |
| 3000 | SubsetSelectCIO + warm start | $-1.26607 \cdot 10^4$ | 2 | 2.02 | 0.00 |
| 3000 | L0BnB | $-6.34585 \cdot 10^4$ | 10 | 1200.55 | 0.00 |
| 3000 | MSK persp. | $-5.27912 \cdot 10^4$ | 10 | 3601.00 | 43.41 |
| 3000 | MSK persp. + warm start | $-6.34585 \cdot 10^4$ | 10 | 3601.81 | 19.30 |
| 3000 | Convex MSK opt. persp. | $-4.55044 \cdot 10^4$ | 10 | 1573.44 | 66.10 |
| 3000 | ours | $-6.34585 \cdot 10^4$ | 10 | 3600.49 | 19.45 |
| 4000 | SubsetSelectCIO | $-1.88907 \cdot 10^4$ | 2 | 1.90 | 0.00 |
| 4000 | SubsetSelectCIO + warm start | $-1.88907 \cdot 10^4$ | 2 | 2.06 | 0.00 |
| 4000 | L0BnB | $-8.64294 \cdot 10^4$ | 10 | 579.39 | 0.00 |
| 4000 | MSK persp. | $-8.64294 \cdot 10^4$ | 10 | 3602.16 | 15.37 |
| 4000 | MSK persp. + warm start | $-8.64294 \cdot 10^4$ | 10 | 3600.84 | 15.37 |
| 4000 | Convex MSK opt. persp. | $-8.64294 \cdot 10^4$ | 10 | 1398.74 | 2.40 |
| 4000 | ours | $-8.64294 \cdot 10^4$ | 10 | 20.25 | 0.00 |
| 5000 | SubsetSelectCIO | $-2.49366 \cdot 10^4$ | 2 | 2.08 | 0.00 |
| 5000 | SubsetSelectCIO + warm start | $-2.49366 \cdot 10^4$ | 2 | 1.96 | 0.00 |
| 5000 | L0BnB | $-1.09576 \cdot 10^5$ | 10 | 1292.85 | 0.00 |
| 5000 | MSK persp. | $-1.09576 \cdot 10^5$ | 10 | 3601.60 | 11.91 |
| 5000 | MSK persp. + warm start | $-1.09576 \cdot 10^5$ | 10 | 3600.55 | 11.91 |
| 5000 | Convex MSK opt. persp. | $-1.09576 \cdot 10^5$ | 10 | 1496.25 | 1.03 |
| 5000 | ours | $-1.09576 \cdot 10^5$ | 10 | 20.51 | 0.00 |
| 6000 | SubsetSelectCIO | - | - | 3600.00 | - |
| 6000 | SubsetSelectCIO + warm start | $-2.83063 \cdot 10^4$ | 2 | 2.22 | 0.00 |
| 6000 | L0BnB | $-1.27643 \cdot 10^5$ | 10 | 1379.36 | 0.00 |
| 6000 | MSK persp. | - | - | 3600.07 | - |
| 6000 | MSK persp. + warm start | $-1.27643 \cdot 10^5$ | 10 | 3600.30 | 9.93 |
| 6000 | Convex MSK opt. persp. | $-1.27643 \cdot 10^5$ | 10 | 1318.81 | 0.62 |
| 6000 | ours | $-1.27643 \cdot 10^5$ | 10 | 1.70 | 0.00 |
| 7000 | SubsetSelectCIO | $-3.24781 \cdot 10^4$ | 2 | 2.06 | 0.00 |
| 7000 | SubsetSelectCIO + warm start | $-3.24781 \cdot 10^4$ | 2 | 2.22 | 0.00 |
| 7000 | L0BnB | $-1.50614 \cdot 10^5$ | 10 | 1105.58 | 0.00 |
| 7000 | MSK persp. | $-2.07297 \cdot 10^4$ | 3 | 3600.06 | 687.08 |
| 7000 | MSK persp. + warm start | $-1.50614 \cdot 10^5$ | 10 | 3599.93 | 8.33 |
| 7000 | Convex MSK opt. persp. | $-1.50614 \cdot 10^5$ | 10 | 1455.27 | 0.45 |
| 7000 | ours | $-1.50614 \cdot 10^5$ | 10 | 1.67 | 0.00 |

Table 136: Comparison of time and optimality gap on the synthetic datasets (p = 3000, k = 10, $\rho = 0.1$, $\lambda_2 = 0.001$).

| # of samples | method | upper bound | support size | time(s) | gap(%) |
|:---:|:---:|:---:|:---:|:---:|:---:|
| 3000 | SubsetSelectCIO | $-7.20755 \cdot 10^4$ | 2 | 1.99 | 0.00 |
| 3000 | SubsetSelectCIO + warm start | $-7.20755 \cdot 10^4$ | 2 | 2.08 | 0.00 |
| 3000 | L0BnB | $-1.59509 \cdot 10^5$ | 10 | 3023.58 | 0.00 |
| 3000 | MSK persp. | $-1.48826 \cdot 10^5$ | 10 | 3601.42 | 27.95 |
| 3000 | MSK persp. + warm start | $-1.59509 \cdot 10^5$ | 10 | 3601.71 | 19.38 |
| 3000 | Convex MSK opt. persp. | $-1.13813 \cdot 10^5$ | 10 | 1627.92 | 67.04 |
| 3000 | ours | $-1.59509 \cdot 10^5$ | 10 | 3600.67 | 19.53 |
| 4000 | SubsetSelectCIO | $-1.03099 \cdot 10^5$ | 2 | 2.07 | 0.00 |
| 4000 | SubsetSelectCIO + warm start | $-1.03099 \cdot 10^5$ | 2 | 1.81 | 0.00 |
| 4000 | L0BnB | $-2.17574 \cdot 10^5$ | 10 | 3396.85 | 0.00 |
| 4000 | MSK persp. | $-2.17574 \cdot 10^5$ | 10 | 3600.26 | 15.27 |
| 4000 | MSK persp. + warm start | $-2.17574 \cdot 10^5$ | 10 | 3601.91 | 15.27 |
| 4000 | Convex MSK opt. persp. | $-2.17574 \cdot 10^5$ | 10 | 1509.12 | 2.41 |
| 4000 | ours | $-2.17574 \cdot 10^5$ | 10 | 100.13 | 0.00 |
| 5000 | SubsetSelectCIO | $-1.30339 \cdot 10^5$ | 2 | 1.98 | 0.00 |
| 5000 | SubsetSelectCIO + warm start | $-1.30339 \cdot 10^5$ | 2 | 2.01 | 0.00 |
| 5000 | L0BnB | $-2.71986 \cdot 10^5$ | 10 | 531.44 | 0.00 |
| 5000 | MSK persp. | $-2.71986 \cdot 10^5$ | 10 | 3600.74 | 12.00 |
| 5000 | MSK persp. + warm start | $-2.71986 \cdot 10^5$ | 10 | 3600.26 | 12.00 |
| 5000 | Convex MSK opt. persp. | $-2.71986 \cdot 10^5$ | 10 | 1514.31 | 1.04 |
| 5000 | ours | $-2.71986 \cdot 10^5$ | 10 | 20.80 | 0.00 |
| 6000 | SubsetSelectCIO | $-1.50202 \cdot 10^5$ | 2 | 2.02 | 0.00 |
| 6000 | SubsetSelectCIO + warm start | $-1.50202 \cdot 10^5$ | 2 | 1.96 | 0.00 |
| 6000 | L0BnB | $-3.16761 \cdot 10^5$ | 10 | 3098.88 | 0.00 |
| 6000 | MSK persp. | $-7.83273 \cdot 10^4$ | 1 | 3603.79 | 344.72 |
| 6000 | MSK persp. + warm start | $-3.16761 \cdot 10^5$ | 10 | 3604.35 | 9.96 |
| 6000 | Convex MSK opt. persp. | $-3.16761 \cdot 10^5$ | 10 | 1445.46 | 0.62 |
| 6000 | ours | $-3.16761 \cdot 10^5$ | 10 | 20.70 | 0.00 |
| 7000 | SubsetSelectCIO | $-1.75217 \cdot 10^5$ | 2 | 2.12 | 0.00 |
| 7000 | SubsetSelectCIO + warm start | $-1.75217 \cdot 10^5$ | 2 | 2.12 | 0.00 |
| 7000 | L0BnB | $-3.72610 \cdot 10^5$ | 10 | 3130.56 | 0.00 |
| 7000 | MSK persp. | - | - | 3599.99 | - |
| 7000 | MSK persp. + warm start | $-3.72610 \cdot 10^5$ | 10 | 3600.24 | 8.33 |
| 7000 | Convex MSK opt. persp. | $-3.72610 \cdot 10^5$ | 10 | 1520.22 | 0.45 |
| 7000 | ours | $-3.72610 \cdot 10^5$ | 10 | 1.98 | 0.00 |

Table 137: Comparison of time and optimality gap on the synthetic datasets (p = 3000, k = 10, $\rho = 0.3$, $\lambda_2 = 0.001$).

| # of samples | method | upper bound | support size | time(s) | gap(%) |
|:---:|:---:|:---:|:---:|:---:|:---:|
| 3000 | SubsetSelectCIO | $-2.19279 \cdot 10^5$ | 2 | 1.96 | 0.00 |
| 3000 | SubsetSelectCIO + warm start | $-2.19279 \cdot 10^5$ | 2 | 1.85 | 0.00 |
| 3000 | L0BnB | $-3.27442 \cdot 10^5$ | 8 | 7379.57 | 5.16 |
| 3000 | MSK persp. | $-2.83751 \cdot 10^5$ | 10 | 3601.66 | 40.41 |
| 3000 | MSK persp. + warm start | $-3.33604 \cdot 10^5$ | 10 | 3601.38 | 19.43 |
| 3000 | Convex MSK opt. persp. | $-2.77416 \cdot 10^5$ | 10 | 1878.24 | 43.38 |
| 3000 | ours | $-3.33604 \cdot 10^5$ | 10 | 3600.58 | 19.57 |
| 4000 | SubsetSelectCIO | $-3.03679 \cdot 10^5$ | 2 | 1.87 | 0.00 |
| 4000 | SubsetSelectCIO + warm start | $-3.03679 \cdot 10^5$ | 2 | 1.89 | 0.00 |
| 4000 | L0BnB | $-4.53261 \cdot 10^5$ | 10 | 4149.90 | 32.94 |
| 4000 | MSK persp. | $-4.53261 \cdot 10^5$ | 10 | 3602.72 | 15.24 |
| 4000 | MSK persp. + warm start | $-4.53261 \cdot 10^5$ | 10 | 3599.91 | 15.24 |
| 4000 | Convex MSK opt. persp. | $-4.53261 \cdot 10^5$ | 10 | 1445.07 | 2.53 |
| 4000 | ours | $-4.53261 \cdot 10^5$ | 10 | 3600.25 | 0.63 |
| 5000 | SubsetSelectCIO | $-4.12520 \cdot 10^5$ | 3 | 1.79 | 0.00 |
| 5000 | SubsetSelectCIO + warm start | $-4.12520 \cdot 10^5$ | 3 | 2.03 | 0.00 |
| 5000 | L0BnB | $-5.63007 \cdot 10^5$ | 10 | 3522.00 | 0.00 |
| 5000 | MSK persp. | $-5.63007 \cdot 10^5$ | 10 | 3600.76 | 12.04 |
| 5000 | MSK persp. + warm start | $-5.63007 \cdot 10^5$ | 10 | 3603.42 | 12.04 |
| 5000 | Convex MSK opt. persp. | $-5.63007 \cdot 10^5$ | 10 | 1618.41 | 1.04 |
| 5000 | ours | $-5.63007 \cdot 10^5$ | 10 | 36.97 | 0.00 |
| 6000 | SubsetSelectCIO | $-4.42065 \cdot 10^5$ | 2 | 1.94 | 0.00 |
| 6000 | SubsetSelectCIO + warm start | $-4.42065 \cdot 10^5$ | 2 | 2.12 | 0.00 |
| 6000 | L0BnB | $-6.57799 \cdot 10^5$ | 10 | 3327.45 | 0.00 |
| 6000 | MSK persp. | - | - | 3605.66 | - |
| 6000 | MSK persp. + warm start | $-6.57799 \cdot 10^5$ | 10 | 3605.89 | 9.98 |
| 6000 | Convex MSK opt. persp. | $-6.57799 \cdot 10^5$ | 10 | 1523.01 | 0.62 |
| 6000 | ours | $-6.57799 \cdot 10^5$ | 10 | 20.75 | 0.00 |
| 7000 | SubsetSelectCIO | $-5.16405 \cdot 10^5$ | 2 | 2.11 | 0.00 |
| 7000 | SubsetSelectCIO + warm start | $-5.16405 \cdot 10^5$ | 2 | 2.14 | 0.00 |
| 7000 | L0BnB | $-7.72267 \cdot 10^5$ | 10 | 3567.05 | 0.00 |
| 7000 | MSK persp. | - | - | 3600.27 | - |
| 7000 | MSK persp. + warm start | $-7.72267 \cdot 10^5$ | 10 | 3600.18 | 8.33 |
| 7000 | Convex MSK opt. persp. | $-7.72267 \cdot 10^5$ | 10 | 1681.35 | 0.45 |
| 7000 | ours | $-7.72267 \cdot 10^5$ | 10 | 21.02 | 0.00 |

Table 138: Comparison of time and optimality gap on the synthetic datasets (p = 3000, k = 10, $\rho = 0.5$, $\lambda_2 = 0.001$).

| # of samples | method | upper bound | support size | time(s) | gap(%) |
|---|---|---|---|---|---|
| 3000 | SubsetSelectCIO | $-6.06571 \cdot 10^5$ | 2 | 1.88 | 0.00 |
| 3000 | SubsetSelectCIO + warm start | $-6.06571 \cdot 10^5$ | 2 | 1.88 | 0.00 |
| 3000 | L0BnB | $-5.52986 \cdot 10^5$ | 1 | 7308.45 | 34.21 |
| 3000 | MSK persp. | $-6.89430 \cdot 10^5$ | 10 | 3600.74 | 28.43 |
| 3000 | MSK persp. + warm start | $-7.41187 \cdot 10^5$ | 10 | 3601.61 | 19.46 |
| 3000 | Convex MSK opt. persp. | $-6.84471 \cdot 10^5$ | 10 | 1652.72 | 29.14 |
| 3000 | ours | $-7.41187 \cdot 10^5$ | 10 | 3601.14 | 19.60 |
| 4000 | SubsetSelectCIO | $-8.24785 \cdot 10^5$ | 2 | 1.85 | 0.00 |
| 4000 | SubsetSelectCIO + warm start | $-8.24785 \cdot 10^5$ | 2 | 1.86 | 0.00 |
| 4000 | L0BnB | $-7.53893 \cdot 10^5$ | 1 | 7293.04 | 32.01 |
| 4000 | MSK persp. | $-9.95041 \cdot 10^5$ | 10 | 3601.67 | 16.12 |
| 4000 | MSK persp. + warm start | $-1.00287 \cdot 10^6$ | 10 | 3602.00 | 15.21 |
| 4000 | Convex MSK opt. persp. | $-1.00287 \cdot 10^6$ | 10 | 1442.72 | 2.82 |
| 4000 | ours | $-1.00287 \cdot 10^6$ | 10 | 3600.37 | 1.88 |
| 5000 | SubsetSelectCIO | $-1.02490 \cdot 10^6$ | 2 | 2.05 | 0.00 |
| 5000 | SubsetSelectCIO + warm start | $-1.02490 \cdot 10^6$ | 2 | 1.92 | 0.00 |
| 5000 | L0BnB | $-1.23555 \cdot 10^6$ | 9 | 7387.59 | 4.90 |
| 5000 | MSK persp. | $-1.24027 \cdot 10^6$ | 10 | 3604.54 | 12.07 |
| 5000 | MSK persp. + warm start | $-1.24027 \cdot 10^6$ | 10 | 3600.77 | 12.07 |
| 5000 | Convex MSK opt. persp. | $-1.24027 \cdot 10^6$ | 10 | 1654.86 | 1.06 |
| 5000 | ours | $-1.24027 \cdot 10^6$ | 10 | 466.42 | 0.00 |
| 6000 | SubsetSelectCIO | $-1.20146 \cdot 10^6$ | 2 | 1.96 | 0.00 |
| 6000 | SubsetSelectCIO + warm start | $-1.20146 \cdot 10^6$ | 2 | 2.05 | 0.00 |
| 6000 | L0BnB | $-1.44860 \cdot 10^6$ | 9 | 7265.23 | 4.30 |
| 6000 | MSK persp. | $-1.14100 \cdot 10^6$ | 2 | 3603.46 | 40.19 |
| 6000 | MSK persp. + warm start | $-1.45417 \cdot 10^6$ | 10 | 3604.24 | 10.00 |
| 6000 | Convex MSK opt. persp. | $-1.45417 \cdot 10^6$ | 10 | 1667.19 | 0.63 |
| 6000 | ours | $-1.45417 \cdot 10^6$ | 10 | 57.67 | 0.00 |
| 7000 | SubsetSelectCIO | $-1.46875 \cdot 10^6$ | 3 | 1.86 | 0.00 |
| 7000 | SubsetSelectCIO + warm start | $-1.46875 \cdot 10^6$ | 3 | 2.09 | 0.00 |
| 7000 | L0BnB | $-1.69804 \cdot 10^6$ | 9 | 7320.83 | 2.92 |
| 7000 | MSK persp. | $-1.32733 \cdot 10^6$ | 2 | 3600.13 | 39.15 |
| 7000 | MSK persp. + warm start | $-1.70483 \cdot 10^6$ | 10 | 3600.05 | 8.34 |
| 7000 | Convex MSK opt. persp. | $-1.70483 \cdot 10^6$ | 10 | 1760.17 | 0.45 |
| 7000 | ours | $-1.70483 \cdot 10^6$ | 10 | 21.42 | 0.00 |

Table 139: Comparison of time and optimality gap on the synthetic datasets (p = 3000, k = 10, $\rho = 0.7$, $\lambda_2 = 0.001$).

| # of samples | method | upper bound | support size | time(s) | gap(%) |
|---|---|---|---|---|---|
| 3000 | SubsetSelectCIO | $-2.63352 \cdot 10^6$ | 2 | 1.90 | 0.00 |
| 3000 | SubsetSelectCIO + warm start | $-2.63352 \cdot 10^6$ | 2 | 2.04 | 0.00 |
| 3000 | L0BnB | $-2.55637 \cdot 10^6$ | 1 | 7236.54 | 18.84 |
| 3000 | MSK persp. | $-2.72590 \cdot 10^6$ | 10 | 3599.87 | 22.06 |
| 3000 | MSK persp. + warm start | $-2.72590 \cdot 10^6$ | 10 | 3600.80 | 22.06 |
| 3000 | Convex MSK opt. persp. | $-2.72998 \cdot 10^6$ | 10 | 2311.76 | 21.67 |
| 3000 | ours | $-2.78521 \cdot 10^6$ | 10 | 3600.01 | 19.60 |
| 4000 | SubsetSelectCIO | $-3.38912 \cdot 10^6$ | 1 | 1.79 | 0.00 |
| 4000 | SubsetSelectCIO + warm start | $-3.38912 \cdot 10^6$ | 1 | 1.89 | 0.00 |
| 4000 | L0BnB | $-3.45201 \cdot 10^6$ | 1 | 7291.93 | 15.48 |
| 4000 | MSK persp. | - | - | 3609.13 | - |
| 4000 | MSK persp. + warm start | $-3.75203 \cdot 10^6$ | 10 | 3605.54 | 15.12 |
| 4000 | Convex MSK opt. persp. | $-3.75202 \cdot 10^6$ | 10 | 1481.77 | 3.23 |
| 4000 | ours | $-3.75203 \cdot 10^6$ | 10 | 3600.42 | 3.50 |
| 5000 | SubsetSelectCIO | $-4.37679 \cdot 10^6$ | 2 | 1.99 | 0.00 |
| 5000 | SubsetSelectCIO + warm start | $-4.37679 \cdot 10^6$ | 2 | 2.07 | 0.00 |
| 5000 | L0BnB | $-4.23095 \cdot 10^6$ | 1 | 7316.72 | 13.65 |
| 5000 | MSK persp. | $-4.60534 \cdot 10^6$ | 10 | 3602.67 | 12.43 |
| 5000 | MSK persp. + warm start | $-4.61877 \cdot 10^6$ | 10 | 3601.35 | 12.11 |
| 5000 | Convex MSK opt. persp. | $-4.61877 \cdot 10^6$ | 10 | 1537.61 | 1.22 |
| 5000 | ours | $-4.61877 \cdot 10^6$ | 10 | 3600.57 | 0.93 |
| 6000 | SubsetSelectCIO | $-5.15897 \cdot 10^6$ | 2 | 2.23 | 0.00 |
| 6000 | SubsetSelectCIO + warm start | $-5.15897 \cdot 10^6$ | 2 | 2.24 | 0.00 |
| 6000 | L0BnB | $-4.97742 \cdot 10^6$ | 1 | 7330.86 | 13.05 |
| 6000 | MSK persp. | - | - | 3603.01 | - |
| 6000 | MSK persp. + warm start | $-5.43826 \cdot 10^6$ | 10 | 3602.04 | 0.00 |
| 6000 | Convex MSK opt. persp. | $-5.43826 \cdot 10^6$ | 10 | 1652.62 | 0.65 |
| 6000 | ours | $-5.43826 \cdot 10^6$ | 10 | 3600.65 | 0.33 |
| 7000 | SubsetSelectCIO | $-6.02832 \cdot 10^6$ | 2 | 1.89 | 0.00 |
| 7000 | SubsetSelectCIO + warm start | $-6.02832 \cdot 10^6$ | 2 | 2.27 | 0.00 |
| 7000 | L0BnB | $-5.82321 \cdot 10^6$ | 1 | 7356.27 | 11.85 |
| 7000 | MSK persp. | - | - | 3600.06 | - |
| 7000 | MSK persp. + warm start | $-6.36767 \cdot 10^6$ | 10 | 3600.21 | 8.35 |
| 7000 | Convex MSK opt. persp. | $-6.36767 \cdot 10^6$ | 10 | 1821.91 | 0.45 |
| 7000 | ours | $-6.36767 \cdot 10^6$ | 10 | 3600.78 | 0.11 |

Table 140: Comparison of time and optimality gap on the synthetic datasets (p = 3000, k = 10, $\rho = 0.9$, $\lambda_2 = 0.001$).

## G.4.5 Synthetic 2 Benchmark with $\lambda_2 = 0.1$

| # of samples | method | upper bound | support size | time(s) | gap(%) |
|---|---|---|---|---|---|
| 3000 | SubsetSelectCIO | $-1.83147 \cdot 10^4$ | 10 | 4229.80 | 313.93 |
| 3000 | SubsetSelectCIO + warm start | $-5.70956 \cdot 10^4$ | 10 | 3602.11 | 32.77 |
| 3000 | L0BnB | $-6.34575 \cdot 10^4$ | 10 | 1201.41 | 0.00 |
| 3000 | MSK persp. | $-6.34575 \cdot 10^4$ | 10 | 3601.57 | 17.90 |
| 3000 | MSK persp. + warm start | $-6.34575 \cdot 10^4$ | 10 | 3600.12 | 17.89 |
| 3000 | Convex MSK opt. persp. | $-6.34575 \cdot 10^4$ | 10 | 1444.07 | 17.85 |
| 3000 | ours | $-6.34575 \cdot 10^4$ | 10 | 3600.33 | 19.31 |
| 4000 | SubsetSelectCIO | $-3.10931 \cdot 10^4$ | 10 | 3872.26 | 235.85 |
| 4000 | SubsetSelectCIO + warm start | $-7.28764 \cdot 10^4$ | 10 | 4101.05 | 43.29 |
| 4000 | L0BnB | $-8.64284 \cdot 10^4$ | 10 | 566.25 | 0.00 |
| 4000 | MSK persp. | $-8.64284 \cdot 10^4$ | 10 | 3602.29 | 15.03 |
| 4000 | MSK persp. + warm start | $-8.64284 \cdot 10^4$ | 10 | 3601.14 | 15.05 |
| 4000 | Convex MSK opt. persp. | $-8.64284 \cdot 10^4$ | 10 | 1458.63 | 2.40 |
| 4000 | ours | $-8.64284 \cdot 10^4$ | 10 | 20.81 | 0.00 |
| 5000 | SubsetSelectCIO | $-4.06480 \cdot 10^4$ | 10 | 3939.75 | 222.71 |
| 5000 | SubsetSelectCIO + warm start | $-9.16380 \cdot 10^4$ | 10 | 3804.66 | 43.14 |
| 5000 | L0BnB | $-1.09575 \cdot 10^5$ | 10 | 1314.94 | 0.00 |
| 5000 | MSK persp. | $-1.09575 \cdot 10^5$ | 10 | 3603.48 | 11.78 |
| 5000 | MSK persp. + warm start | $-1.09575 \cdot 10^5$ | 10 | 3602.79 | 11.78 |
| 5000 | Convex MSK opt. persp. | $-1.09575 \cdot 10^5$ | 10 | 1478.60 | 1.03 |
| 5000 | ours | $-1.09575 \cdot 10^5$ | 10 | 20.65 | 0.00 |
| 6000 | SubsetSelectCIO | - | - | 3600.00 | - |
| 6000 | SubsetSelectCIO + warm start | $-1.21549 \cdot 10^5$ | 10 | 3850.54 | 26.10 |
| 6000 | L0BnB | $-1.27642 \cdot 10^5$ | 10 | 1536.59 | 0.00 |
| 6000 | MSK persp. | - | - | 3606.50 | - |
| 6000 | MSK persp. + warm start | $-1.27642 \cdot 10^5$ | 10 | 3603.98 | 9.85 |
| 6000 | Convex MSK opt. persp. | $-1.27642 \cdot 10^5$ | 10 | 1303.39 | 0.62 |
| 6000 | ours | $-1.27642 \cdot 10^5$ | 10 | 1.70 | 0.00 |
| 7000 | SubsetSelectCIO | $-5.54763 \cdot 10^4$ | 10 | 4067.63 | 224.61 |
| 7000 | SubsetSelectCIO + warm start | $-1.42693 \cdot 10^5$ | 10 | 3615.01 | 26.20 |
| 7000 | L0BnB | $-1.50613 \cdot 10^5$ | 10 | 1064.72 | 0.00 |
| 7000 | MSK persp. | - | - | 3599.90 | - |
| 7000 | MSK persp. + warm start | $-1.50613 \cdot 10^5$ | 10 | 3600.13 | 8.28 |
| 7000 | Convex MSK opt. persp. | $-1.50613 \cdot 10^5$ | 10 | 1405.96 | 0.45 |
| 7000 | ours | $-1.50613 \cdot 10^5$ | 10 | 1.67 | 0.00 |

Table 141: Comparison of time and optimality gap on the synthetic datasets (p = 3000, k = 10, $\rho = 0.1$, $\lambda_2 = 0.1$).

| # of samples | method | upper bound | support size | time(s) | gap(%) |
|---|---|---|---|---|---|
| 3000 | SubsetSelectCIO | $-1.13982 \cdot 10^5$ | 10 | 3755.86 | 67.28 |
| 3000 | SubsetSelectCIO + warm start | $-1.59508 \cdot 10^5$ | 10 | 3857.37 | 19.54 |
| 3000 | L0BnB | $-1.59508 \cdot 10^5$ | 10 | 1491.90 | 0.00 |
| 3000 | MSK persp. | $-1.59508 \cdot 10^5$ | 10 | 3600.12 | 17.99 |
| 3000 | MSK persp. + warm start | $-1.59508 \cdot 10^5$ | 10 | 3600.37 | 17.99 |
| 3000 | Convex MSK opt. persp. | $-1.59508 \cdot 10^5$ | 10 | 1454.84 | 17.99 |
| 3000 | ours | $-1.59508 \cdot 10^5$ | 10 | 3600.27 | 19.40 |
| 4000 | SubsetSelectCIO | - | - | 3600.00 | - |
| 4000 | SubsetSelectCIO + warm start | $-2.17573 \cdot 10^5$ | 10 | 3711.79 | 20.70 |
| 4000 | L0BnB | $-2.17573 \cdot 10^5$ | 10 | 3399.35 | 0.00 |
| 4000 | MSK persp. | $-2.17573 \cdot 10^5$ | 10 | 3601.17 | 14.95 |
| 4000 | MSK persp. + warm start | $-2.17573 \cdot 10^5$ | 10 | 3602.25 | 14.95 |
| 4000 | Convex MSK opt. persp. | $-2.17573 \cdot 10^5$ | 10 | 1510.77 | 2.40 |
| 4000 | ours | $-2.17573 \cdot 10^5$ | 10 | 99.53 | 0.00 |
| 5000 | SubsetSelectCIO | $-1.92108 \cdot 10^5$ | 10 | 3602.76 | 69.69 |
| 5000 | SubsetSelectCIO + warm start | $-2.71985 \cdot 10^5$ | 10 | 3733.49 | 19.85 |
| 5000 | L0BnB | $-2.71985 \cdot 10^5$ | 10 | 532.76 | 0.00 |
| 5000 | MSK persp. | $-2.71985 \cdot 10^5$ | 10 | 3601.74 | 11.88 |
| 5000 | MSK persp. + warm start | $-2.71985 \cdot 10^5$ | 10 | 3602.31 | 11.88 |
| 5000 | Convex MSK opt. persp. | $-2.71985 \cdot 10^5$ | 10 | 1576.92 | 1.04 |
| 5000 | ours | $-2.71985 \cdot 10^5$ | 10 | 21.01 | 0.00 |
| 6000 | SubsetSelectCIO | $-2.03206 \cdot 10^5$ | 10 | 4031.62 | 87.30 |
| 6000 | SubsetSelectCIO + warm start | $-3.10827 \cdot 10^5$ | 10 | 3780.98 | 22.45 |
| 6000 | L0BnB | $-3.16760 \cdot 10^5$ | 10 | 3082.87 | 0.00 |
| 6000 | MSK persp. | - | - | 3600.17 | - |
| 6000 | MSK persp. + warm start | $-3.16760 \cdot 10^5$ | 10 | 3599.98 | 9.90 |
| 6000 | Convex MSK opt. persp. | $-3.16760 \cdot 10^5$ | 10 | 1487.94 | 0.62 |
| 6000 | ours | $-3.16760 \cdot 10^5$ | 10 | 20.91 | 0.00 |
| 7000 | SubsetSelectCIO | $-2.44730 \cdot 10^5$ | 10 | 3992.94 | 82.05 |
| 7000 | SubsetSelectCIO + warm start | $-3.37947 \cdot 10^5$ | 10 | 4164.11 | 31.83 |
| 7000 | L0BnB | $-3.72609 \cdot 10^5$ | 10 | 3126.74 | 0.00 |
| 7000 | MSK persp. | - | - | 3600.16 | - |
| 7000 | MSK persp. + warm start | $-3.72609 \cdot 10^5$ | 10 | 3604.63 | 8.28 |
| 7000 | Convex MSK opt. persp. | $-3.72609 \cdot 10^5$ | 10 | 1555.38 | 0.45 |
| 7000 | ours | $-3.72609 \cdot 10^5$ | 10 | 1.73 | 0.00 |

Table 142: Comparison of time and optimality gap on the synthetic datasets (p = 3000, k = 10, $\rho = 0.3$, $\lambda_2 = 0.1$).

| # of samples | method | upper bound | support size | time(s) | gap(%) |
|---|---|---|---|---|---|
| 3000 | SubsetSelectCIO | $-2.79591 \cdot 10^5$ | 9 | 3686.43 | 42.67 |
| 3000 | SubsetSelectCIO + warm start | $-3.11853 \cdot 10^5$ | 10 | 3738.09 | 27.91 |
| 3000 | L0BnB | $-3.27441 \cdot 10^5$ | 8 | 7347.92 | 4.34 |
| 3000 | MSK persp. | $-3.33603 \cdot 10^5$ | 10 | 3600.83 | 18.03 |
| 3000 | MSK persp. + warm start | $-3.33603 \cdot 10^5$ | 10 | 3600.62 | 18.03 |
| 3000 | Convex MSK opt. persp. | $-3.18968 \cdot 10^5$ | 10 | 1341.73 | 23.47 |
| 3000 | ours | $-3.33603 \cdot 10^5$ | 10 | 3600.68 | 19.44 |
| 4000 | SubsetSelectCIO | - | - | 3600.00 | - |
| 4000 | SubsetSelectCIO + warm start | $-4.44307 \cdot 10^5$ | 10 | 3651.78 | 23.07 |
| 4000 | L0BnB | $-4.53260 \cdot 10^5$ | 10 | 4150.43 | 32.94 |
| 4000 | MSK persp. | $-4.53260 \cdot 10^5$ | 10 | 3602.80 | 14.94 |
| 4000 | MSK persp. + warm start | $-4.53260 \cdot 10^5$ | 10 | 3601.24 | 14.94 |
| 4000 | Convex MSK opt. persp. | $-4.53260 \cdot 10^5$ | 10 | 1444.17 | 2.53 |
| 4000 | ours | $-4.53260 \cdot 10^5$ | 10 | 3600.41 | 0.62 |
| 5000 | SubsetSelectCIO | $-4.69039 \cdot 10^5$ | 8 | 3951.25 | 43.95 |
| 5000 | SubsetSelectCIO + warm start | $-5.44684 \cdot 10^5$ | 9 | 4020.24 | 23.96 |
| 5000 | L0BnB | $-5.63006 \cdot 10^5$ | 10 | 3516.88 | 0.00 |
| 5000 | MSK persp. | $-5.63006 \cdot 10^5$ | 10 | 3603.31 | 11.92 |
| 5000 | MSK persp. + warm start | $-5.63006 \cdot 10^5$ | 10 | 3601.57 | 11.92 |
| 5000 | Convex MSK opt. persp. | $-5.63006 \cdot 10^5$ | 10 | 1572.75 | 1.04 |
| 5000 | ours | $-5.63006 \cdot 10^5$ | 10 | 37.26 | 0.00 |
| 6000 | SubsetSelectCIO | $-2.80001 \cdot 10^5$ | 1 | 2.28 | 0.00 |
| 6000 | SubsetSelectCIO + warm start | $-6.44963 \cdot 10^5$ | 10 | 3724.51 | 22.59 |
| 6000 | L0BnB | $-6.57798 \cdot 10^5$ | 10 | 3317.18 | 0.00 |
| 6000 | MSK persp. | - | - | 3604.37 | - |
| 6000 | MSK persp. + warm start | $-6.57798 \cdot 10^5$ | 10 | 3606.26 | 9.92 |
| 6000 | Convex MSK opt. persp. | $-6.57798 \cdot 10^5$ | 10 | 1657.28 | 0.62 |
| 6000 | ours | $-6.57798 \cdot 10^5$ | 10 | 20.64 | 0.00 |
| 7000 | SubsetSelectCIO | $-3.48812 \cdot 10^5$ | 1 | 2.12 | 0.00 |
| 7000 | SubsetSelectCIO + warm start | $-3.45235 \cdot 10^5$ | 1 | 2.06 | 0.00 |
| 7000 | L0BnB | $-7.72266 \cdot 10^5$ | 10 | 3589.79 | 0.00 |
| 7000 | MSK persp. | - | - | 3599.94 | - |
| 7000 | MSK persp. + warm start | $-7.72266 \cdot 10^5$ | 10 | 3600.05 | 8.29 |
| 7000 | Convex MSK opt. persp. | $-7.72266 \cdot 10^5$ | 10 | 1603.99 | 0.45 |
| 7000 | ours | $-7.72266 \cdot 10^5$ | 10 | 20.91 | 0.00 |

Table 143: Comparison of time and optimality gap on the synthetic datasets (p = 3000, k = 10, $\rho = 0.5$, $\lambda_2 = 0.1$).

| # of samples | method | upper bound | support size | time(s) | gap(%) |
|---|---|---|---|---|---|
| 3000 | SubsetSelectCIO | $-6.90645 \cdot 10^5$ | 10 | 4096.61 | 28.35 |
| 3000 | SubsetSelectCIO + warm start | $-7.31489 \cdot 10^5$ | 10 | 3621.77 | 21.19 |
| 3000 | L0BnB | $-5.52981 \cdot 10^5$ | 1 | 7310.65 | 34.21 |
| 3000 | MSK persp. | $-7.39018 \cdot 10^5$ | 10 | 3600.41 | 18.44 |
| 3000 | MSK persp. + warm start | $-7.41186 \cdot 10^5$ | 10 | 3600.40 | 18.10 |
| 3000 | Convex MSK opt. persp. | $-7.07443 \cdot 10^5$ | 10 | 1698.29 | 23.74 |
| 3000 | ours | $-7.41186 \cdot 10^5$ | 10 | 3600.59 | 19.46 |
| 4000 | SubsetSelectCIO | $-9.43345 \cdot 10^5$ | 10 | 3677.02 | 28.22 |
| 4000 | SubsetSelectCIO + warm start | $-9.93033 \cdot 10^5$ | 10 | 3684.08 | 21.81 |
| 4000 | L0BnB | $-7.53887 \cdot 10^5$ | 1 | 7333.72 | 31.99 |
| 4000 | MSK persp. | $-9.98256 \cdot 10^5$ | 10 | 3600.92 | 15.46 |
| 4000 | MSK persp. + warm start | $-1.00287 \cdot 10^6$ | 10 | 3602.18 | 14.92 |
| 4000 | Convex MSK opt. persp. | $-1.00287 \cdot 10^6$ | 10 | 1574.91 | 2.82 |
| 4000 | ours | $-1.00287 \cdot 10^6$ | 10 | 3600.08 | 1.88 |
| 5000 | SubsetSelectCIO | - | - | 3600.00 | - |
| 5000 | SubsetSelectCIO + warm start | $-1.22972 \cdot 10^6$ | 10 | 3997.81 | 21.00 |
| 5000 | L0BnB | $-1.23555 \cdot 10^6$ | 9 | 7414.89 | 5.14 |
| 5000 | MSK persp. | $-1.24027 \cdot 10^6$ | 10 | 3601.11 | 11.96 |
| 5000 | MSK persp. + warm start | $-1.24027 \cdot 10^6$ | 10 | 3601.56 | 11.96 |
| 5000 | Convex MSK opt. persp. | $-1.24027 \cdot 10^6$ | 10 | 1638.36 | 1.06 |
| 5000 | ours | $-1.24027 \cdot 10^6$ | 10 | 469.43 | 0.00 |
| 6000 | SubsetSelectCIO | $-9.41611 \cdot 10^5$ | 1 | 59.04 | 0.00 |
| 6000 | SubsetSelectCIO + warm start | $-9.78724 \cdot 10^5$ | 1 | 2.14 | 0.00 |
| 6000 | L0BnB | $-1.44860 \cdot 10^6$ | 9 | 7352.27 | 4.07 |
| 6000 | MSK persp. | - | - | 3602.12 | - |
| 6000 | MSK persp. + warm start | $-1.45416 \cdot 10^6$ | 10 | 3600.04 | 9.94 |
| 6000 | Convex MSK opt. persp. | $-1.45416 \cdot 10^6$ | 10 | 1653.65 | 0.63 |
| 6000 | ours | $-1.45416 \cdot 10^6$ | 10 | 59.16 | 0.00 |
| 7000 | SubsetSelectCIO | $-1.14528 \cdot 10^6$ | 1 | 2.13 | 0.00 |
| 7000 | SubsetSelectCIO + warm start | $-1.14528 \cdot 10^6$ | 1 | 2.11 | 0.00 |
| 7000 | L0BnB | $-1.69804 \cdot 10^6$ | 9 | 7503.37 | 3.09 |
| 7000 | MSK persp. | - | - | 3600.28 | - |
| 7000 | MSK persp. + warm start | $-1.70483 \cdot 10^6$ | 10 | 3600.18 | 8.30 |
| 7000 | Convex MSK opt. persp. | $-1.70483 \cdot 10^6$ | 10 | 1715.12 | 0.45 |
| 7000 | ours | $-1.70483 \cdot 10^6$ | 10 | 21.12 | 0.00 |

Table 144: Comparison of time and optimality gap on the synthetic datasets (p = 3000, k = 10, $\rho = 0.7$, $\lambda_2 = 0.1$).

| # of samples | method | upper bound | support size | time(s) | gap(%) |
|---|---|---|---|---|---|
| 3000 | SubsetSelectCIO | $-2.70507 \cdot 10^6$ | 5 | 3696.31 | 23.15 |
| 3000 | SubsetSelectCIO + warm start | $-2.76623 \cdot 10^6$ | 7 | 4030.96 | 20.43 |
| 3000 | L0BnB | $-2.55637 \cdot 10^6$ | 1 | 7339.86 | 18.82 |
| 3000 | MSK persp. | $-2.75384 \cdot 10^6$ | 10 | 3600.44 | 19.50 |
| 3000 | MSK persp. + warm start | $-2.75384 \cdot 10^6$ | 10 | 3600.07 | 19.50 |
| 3000 | Convex MSK opt. persp. | $-2.72647 \cdot 10^6$ | 10 | 1999.54 | 20.67 |
| 3000 | ours | $-2.78521 \cdot 10^6$ | 10 | 3600.23 | 19.46 |
| 4000 | SubsetSelectCIO | $-3.65950 \cdot 10^6$ | 9 | 3841.51 | 23.56 |
| 4000 | SubsetSelectCIO + warm start | $-3.74433 \cdot 10^6$ | 10 | 3649.94 | 20.76 |
| 4000 | L0BnB | $-3.45200 \cdot 10^6$ | 1 | 7481.26 | 15.51 |
| 4000 | MSK persp. | $-3.72627 \cdot 10^6$ | 10 | 3601.44 | 15.63 |
| 4000 | MSK persp. + warm start | $-3.75203 \cdot 10^6$ | 10 | 3601.81 | 14.84 |
| 4000 | Convex MSK opt. persp. | $-3.75202 \cdot 10^6$ | 10 | 1543.65 | 3.23 |
| 4000 | ours | $-3.75203 \cdot 10^6$ | 10 | 3601.10 | 3.50 |
| 5000 | SubsetSelectCIO | - | - | 3600.00 | - |
| 5000 | SubsetSelectCIO + warm start | $-4.61877 \cdot 10^6$ | 10 | 3681.64 | 20.02 |
| 5000 | L0BnB | $-4.23094 \cdot 10^6$ | 1 | 7531.59 | 13.78 |
| 5000 | MSK persp. | $-4.60533 \cdot 10^6$ | 10 | 3605.23 | 12.32 |
| 5000 | MSK persp. + warm start | $-4.61877 \cdot 10^6$ | 10 | 3601.43 | 11.99 |
| 5000 | Convex MSK opt. persp. | $-4.61877 \cdot 10^6$ | 10 | 1633.96 | 1.22 |
| 5000 | ours | $-4.61877 \cdot 10^6$ | 10 | 3600.40 | 0.93 |
| 6000 | SubsetSelectCIO | $-5.38482 \cdot 10^6$ | 10 | 3762.10 | 21.45 |
| 6000 | SubsetSelectCIO + warm start | $-5.42848 \cdot 10^6$ | 10 | 3891.96 | 20.47 |
| 6000 | L0BnB | $-4.97741 \cdot 10^6$ | 1 | 7211.19 | 12.81 |
| 6000 | MSK persp. | $-4.81401 \cdot 10^6$ | 1 | 3601.53 | 24.21 |
| 6000 | MSK persp. + warm start | $-5.43826 \cdot 10^6$ | 10 | 3601.18 | 9.95 |
| 6000 | Convex MSK opt. persp. | $-5.43826 \cdot 10^6$ | 10 | 1535.09 | 0.65 |
| 6000 | ours | $-5.43826 \cdot 10^6$ | 10 | 3600.26 | 0.33 |
| 7000 | SubsetSelectCIO | $-5.66568 \cdot 10^6$ | 1 | 2.34 | 0.00 |
| 7000 | SubsetSelectCIO + warm start | $-5.66568 \cdot 10^6$ | 1 | 2.16 | 0.00 |
| 7000 | L0BnB | $-5.82321 \cdot 10^6$ | 1 | 7336.27 | 11.85 |
| 7000 | MSK persp. | - | - | 3600.04 | - |
| 7000 | MSK persp. + warm start | $-6.36767 \cdot 10^6$ | 10 | 3599.94 | 8.31 |
| 7000 | Convex MSK opt. persp. | $-6.36767 \cdot 10^6$ | 10 | 1779.25 | 0.45 |
| 7000 | ours | $-6.36767 \cdot 10^6$ | 10 | 3600.58 | 0.11 |

Table 145: Comparison of time and optimality gap on the synthetic datasets (p = 3000, k = 10, $\rho = 0.9$, $\lambda_2 = 0.1$).

### G.4.6 Synthetic 2 Benchmark with $\lambda_2 = 10.0$

| # of samples | method | upper bound | support size | time(s) | gap(%) |
|---|---|---|---|---|---|
| 3000 | SubsetSelectCIO | $-4.92601 \cdot 10^4$ | 10 | 3915.82 | 53.90 |
| 3000 | SubsetSelectCIO + warm start | $-3.58868 \cdot 10^4$ | 10 | 3913.17 | 111.25 |
| 3000 | L0BnB | $-6.33576 \cdot 10^4$ | 10 | 1198.54 | 0.00 |
| 3000 | MSK persp. | $-6.33576 \cdot 10^4$ | 10 | 3601.44 | 8.58 |
| 3000 | MSK persp. + warm start | $-6.33576 \cdot 10^4$ | 10 | 3600.20 | 8.58 |
| 3000 | Convex MSK opt. persp. | $-6.33576 \cdot 10^4$ | 10 | 1334.48 | 9.04 |
| 3000 | ours | $-6.33576 \cdot 10^4$ | 10 | 3601.41 | 4.54 |
| 4000 | SubsetSelectCIO | $-7.36488 \cdot 10^4$ | 10 | 4165.15 | 41.79 |
| 4000 | SubsetSelectCIO + warm start | $-5.42345 \cdot 10^4$ | 10 | 3789.71 | 92.55 |
| 4000 | L0BnB | $-8.63287 \cdot 10^4$ | 10 | 1579.27 | 0.00 |
| 4000 | MSK persp. | $-8.63287 \cdot 10^4$ | 10 | 3600.31 | 7.64 |
| 4000 | MSK persp. + warm start | $-8.63287 \cdot 10^4$ | 10 | 3601.55 | 7.64 |
| 4000 | Convex MSK opt. persp. | $-8.63287 \cdot 10^4$ | 10 | 1478.13 | 2.24 |
| 4000 | ours | $-8.63287 \cdot 10^4$ | 10 | 20.33 | 0.00 |
| 5000 | SubsetSelectCIO | - | - | 3600.00 | - |
| 5000 | SubsetSelectCIO + warm start | $-6.16010 \cdot 10^4$ | 10 | 3623.84 | 112.94 |
| 5000 | L0BnB | $-1.09474 \cdot 10^5$ | 10 | 1296.29 | 0.00 |
| 5000 | MSK persp. | $-1.09474 \cdot 10^5$ | 10 | 3601.04 | 6.75 |
| 5000 | MSK persp. + warm start | $-1.09474 \cdot 10^5$ | 10 | 3602.25 | 6.75 |
| 5000 | Convex MSK opt. persp. | $-1.09474 \cdot 10^5$ | 10 | 1532.63 | 1.00 |
| 5000 | ours | $-1.09474 \cdot 10^5$ | 10 | 20.71 | 0.00 |
| 6000 | SubsetSelectCIO | $-9.91460 \cdot 10^4$ | 10 | 3945.11 | 54.60 |
| 6000 | SubsetSelectCIO + warm start | $-8.28165 \cdot 10^4$ | 10 | 3702.54 | 85.08 |
| 6000 | L0BnB | $-1.27542 \cdot 10^5$ | 10 | 1268.61 | 0.00 |
| 6000 | MSK persp. | $-1.27542 \cdot 10^5$ | 10 | 3602.43 | 6.47 |
| 6000 | MSK persp. + warm start | $-1.27542 \cdot 10^5$ | 10 | 3607.80 | 6.47 |
| 6000 | Convex MSK opt. persp. | $-1.27542 \cdot 10^5$ | 10 | 1314.11 | 0.61 |
| 6000 | ours | $-1.27542 \cdot 10^5$ | 10 | 1.69 | 0.00 |
| 7000 | SubsetSelectCIO | $-1.17343 \cdot 10^5$ | 10 | 3661.02 | 53.46 |
| 7000 | SubsetSelectCIO + warm start | $-9.54749 \cdot 10^4$ | 10 | 3758.42 | 88.62 |
| 7000 | L0BnB | $-1.50512 \cdot 10^5$ | 10 | 1063.19 | 0.00 |
| 7000 | MSK persp. | $-1.50512 \cdot 10^5$ | 10 | 3600.60 | 5.68 |
| 7000 | MSK persp. + warm start | $-1.50512 \cdot 10^5$ | 10 | 3600.79 | 5.68 |
| 7000 | Convex MSK opt. persp. | $-1.50512 \cdot 10^5$ | 10 | 1480.64 | 0.45 |
| 7000 | ours | $-1.50512 \cdot 10^5$ | 10 | 1.70 | 0.00 |

Table 146: Comparison of time and optimality gap on the synthetic datasets (p = 3000, k = 10, $\rho = 0.1$, $\lambda_2 = 10.0$).

| # of samples | method | upper bound | support size | time(s) | gap(%) |
|---|---|---|---|---|---|
| 3000 | SubsetSelectCIO | $-1.39974 \cdot 10^5$ | 10 | 3764.95 | 36.22 |
| 3000 | SubsetSelectCIO + warm start | $-1.44268 \cdot 10^5$ | 10 | 3937.05 | 32.16 |
| 3000 | L0BnB | $-1.59409 \cdot 10^5$ | 10 | 1563.82 | 0.00 |
| 3000 | MSK persp. | $-1.59409 \cdot 10^5$ | 10 | 3601.01 | 9.02 |
| 3000 | MSK persp. + warm start | $-1.59409 \cdot 10^5$ | 10 | 3600.73 | 9.02 |
| 3000 | Convex MSK opt. persp. | $-1.59409 \cdot 10^5$ | 10 | 1511.63 | 9.58 |
| 3000 | ours | $-1.59409 \cdot 10^5$ | 10 | 3600.62 | 4.68 |
| 4000 | SubsetSelectCIO | $-1.82403 \cdot 10^5$ | 10 | 3763.38 | 43.97 |
| 4000 | SubsetSelectCIO + warm start | $-2.08718 \cdot 10^5$ | 10 | 4023.55 | 25.82 |
| 4000 | L0BnB | $-2.17472 \cdot 10^5$ | 10 | 3126.91 | 0.00 |
| 4000 | MSK persp. | $-2.17472 \cdot 10^5$ | 10 | 3605.75 | 7.91 |
| 4000 | MSK persp. + warm start | $-2.17472 \cdot 10^5$ | 10 | 3601.48 | 7.91 |
| 4000 | Convex MSK opt. persp. | $-2.17472 \cdot 10^5$ | 10 | 1478.15 | 2.24 |
| 4000 | ours | $-2.17472 \cdot 10^5$ | 10 | 86.90 | 0.00 |
| 5000 | SubsetSelectCIO | $-2.60617 \cdot 10^5$ | 10 | 3976.98 | 25.08 |
| 5000 | SubsetSelectCIO + warm start | $-2.19315 \cdot 10^5$ | 10 | 3615.40 | 48.64 |
| 5000 | L0BnB | $-2.71885 \cdot 10^5$ | 10 | 1571.26 | 0.00 |
| 5000 | MSK persp. | $-2.71885 \cdot 10^5$ | 10 | 3602.81 | 7.26 |
| 5000 | MSK persp. + warm start | $-2.71885 \cdot 10^5$ | 10 | 3603.13 | 7.26 |
| 5000 | Convex MSK opt. persp. | $-2.71885 \cdot 10^5$ | 10 | 1509.28 | 1.01 |
| 5000 | ours | $-2.71885 \cdot 10^5$ | 10 | 20.75 | 0.00 |
| 6000 | SubsetSelectCIO | $-2.62606 \cdot 10^5$ | 10 | 3828.76 | 44.93 |
| 6000 | SubsetSelectCIO + warm start | $-2.84305 \cdot 10^5$ | 10 | 3662.07 | 33.87 |
| 6000 | L0BnB | $-3.16660 \cdot 10^5$ | 10 | 3085.01 | 0.00 |
| 6000 | MSK persp. | $-3.16660 \cdot 10^5$ | 10 | 3601.61 | 6.73 |
| 6000 | MSK persp. + warm start | $-3.16660 \cdot 10^5$ | 10 | 3605.65 | 6.73 |
| 6000 | Convex MSK opt. persp. | $-3.16660 \cdot 10^5$ | 10 | 1471.81 | 0.61 |
| 6000 | ours | $-3.16660 \cdot 10^5$ | 10 | 21.05 | 0.00 |
| 7000 | SubsetSelectCIO | $-2.96322 \cdot 10^5$ | 10 | 3700.24 | 50.35 |
| 7000 | SubsetSelectCIO + warm start | $-2.97673 \cdot 10^5$ | 10 | 3747.76 | 49.67 |
| 7000 | L0BnB | $-3.72509 \cdot 10^5$ | 10 | 3125.79 | 0.00 |
| 7000 | MSK persp. | $-3.72509 \cdot 10^5$ | 10 | 3605.45 | 5.87 |
| 7000 | MSK persp. + warm start | $-3.72509 \cdot 10^5$ | 10 | 3601.55 | 5.87 |
| 7000 | Convex MSK opt. persp. | $-3.72509 \cdot 10^5$ | 10 | 1549.05 | 0.45 |
| 7000 | ours | $-3.72509 \cdot 10^5$ | 10 | 1.71 | 0.00 |

Table 147: Comparison of time and optimality gap on the synthetic datasets (p = 3000, k = 10, $\rho = 0.3$, $\lambda_2 = 10.0$).

| # of samples | method | upper bound | support size | time(s) | gap(%) |
|---|---|---|---|---|---|
| 3000 | SubsetSelectCIO | $-3.12174 \cdot 10^5$ | 10 | 3636.16 | 27.78 |
| 3000 | SubsetSelectCIO + warm start | $-3.27648 \cdot 10^5$ | 10 | 3759.74 | 21.75 |
| 3000 | L0BnB | $-3.27321 \cdot 10^5$ | 8 | 7307.65 | 4.14 |
| 3000 | MSK persp. | $-3.33503 \cdot 10^5$ | 10 | 3600.22 | 9.25 |
| 3000 | MSK persp. + warm start | $-3.33503 \cdot 10^5$ | 10 | 3600.89 | 9.25 |
| 3000 | Convex MSK opt. persp. | $-3.33503 \cdot 10^5$ | 10 | 1475.73 | 9.93 |
| 3000 | ours | $-3.33503 \cdot 10^5$ | 10 | 3600.25 | 6.21 |
| 4000 | SubsetSelectCIO | $-3.96358 \cdot 10^5$ | 10 | 3655.16 | 37.96 |
| 4000 | SubsetSelectCIO + warm start | $-4.44111 \cdot 10^5$ | 10 | 3679.12 | 23.13 |
| 4000 | L0BnB | $-4.53159 \cdot 10^5$ | 10 | 4152.36 | 32.87 |
| 4000 | MSK persp. | $-4.53159 \cdot 10^5$ | 10 | 3602.35 | 8.06 |
| 4000 | MSK persp. + warm start | $-4.53159 \cdot 10^5$ | 10 | 3600.09 | 8.06 |
| 4000 | Convex MSK opt. persp. | $-4.53159 \cdot 10^5$ | 10 | 1513.22 | 2.34 |
| 4000 | ours | $-4.53159 \cdot 10^5$ | 10 | 3600.95 | 0.35 |
| 5000 | SubsetSelectCIO | $-4.93471 \cdot 10^5$ | 10 | 3709.44 | 36.82 |
| 5000 | SubsetSelectCIO + warm start | $-5.27817 \cdot 10^5$ | 10 | 3730.92 | 27.92 |
| 5000 | L0BnB | $-5.62906 \cdot 10^5$ | 10 | 3521.21 | 0.00 |
| 5000 | MSK persp. | $-5.62906 \cdot 10^5$ | 10 | 3602.11 | 7.22 |
| 5000 | MSK persp. + warm start | $-5.62906 \cdot 10^5$ | 10 | 3603.85 | 7.44 |
| 5000 | Convex MSK opt. persp. | $-5.62906 \cdot 10^5$ | 10 | 1530.90 | 1.02 |
| 5000 | ours | $-5.62906 \cdot 10^5$ | 10 | 31.47 | 0.00 |
| 6000 | SubsetSelectCIO | $-5.71812 \cdot 10^5$ | 10 | 3620.92 | 38.27 |
| 6000 | SubsetSelectCIO + warm start | $-6.34165 \cdot 10^5$ | 10 | 3675.07 | 24.67 |
| 6000 | L0BnB | $-6.57698 \cdot 10^5$ | 10 | 3313.82 | 0.00 |
| 6000 | MSK persp. | $-6.57698 \cdot 10^5$ | 10 | 3605.13 | 6.87 |
| 6000 | MSK persp. + warm start | $-6.57698 \cdot 10^5$ | 10 | 3608.94 | 6.87 |
| 6000 | Convex MSK opt. persp. | $-6.57698 \cdot 10^5$ | 10 | 1642.87 | 0.62 |
| 6000 | ours | $-6.57698 \cdot 10^5$ | 10 | 21.24 | 0.00 |
| 7000 | SubsetSelectCIO | $-7.32796 \cdot 10^5$ | 10 | 3784.82 | 26.02 |
| 7000 | SubsetSelectCIO + warm start | $-7.45541 \cdot 10^5$ | 10 | 3662.49 | 23.87 |
| 7000 | L0BnB | $-7.72166 \cdot 10^5$ | 10 | 3081.93 | 0.68 |
| 7000 | MSK persp. | $-7.72166 \cdot 10^5$ | 10 | 3604.29 | 5.98 |
| 7000 | MSK persp. + warm start | $-7.72166 \cdot 10^5$ | 10 | 3606.92 | 5.98 |
| 7000 | Convex MSK opt. persp. | $-7.72166 \cdot 10^5$ | 10 | 1526.71 | 0.45 |
| 7000 | ours | $-7.72166 \cdot 10^5$ | 10 | 20.37 | 0.00 |

Table 148: Comparison of time and optimality gap on the synthetic datasets (p = 3000, k = 10, $\rho = 0.5$, $\lambda_2 = 10.0$).

| # of samples | method | upper bound | support size | time(s) | gap(%) |
|---|---|---|---|---|---|
| 3000 | SubsetSelectCIO | $-6.97357 \cdot 10^5$ | 10 | 3727.41 | 27.12 |
| 3000 | SubsetSelectCIO + warm start | $-7.35099 \cdot 10^5$ | 10 | 3678.19 | 20.59 |
| 3000 | L0BnB | $-5.52435 \cdot 10^5$ | 1 | 7287.00 | 33.44 |
| 3000 | MSK persp. | $-7.41086 \cdot 10^5$ | 10 | 3602.90 | 9.56 |
| 3000 | MSK persp. + warm start | $-7.41086 \cdot 10^5$ | 10 | 3600.62 | 9.56 |
| 3000 | Convex MSK opt. persp. | $-7.41086 \cdot 10^5$ | 10 | 1525.68 | 10.22 |
| 3000 | ours | $-7.41086 \cdot 10^5$ | 10 | 3600.48 | 8.32 |
| 4000 | SubsetSelectCIO | $-9.65426 \cdot 10^5$ | 10 | 3756.83 | 25.29 |
| 4000 | SubsetSelectCIO + warm start | $-9.86629 \cdot 10^5$ | 10 | 3731.48 | 22.60 |
| 4000 | L0BnB | $-7.53327 \cdot 10^5$ | 1 | 7269.26 | 31.51 |
| 4000 | MSK persp. | $-1.00277 \cdot 10^6$ | 10 | 3600.55 | 8.17 |
| 4000 | MSK persp. + warm start | $-1.00277 \cdot 10^6$ | 10 | 3601.59 | 8.17 |
| 4000 | Convex MSK opt. persp. | $-1.00277 \cdot 10^6$ | 10 | 1555.56 | 2.60 |
| 4000 | ours | $-1.00277 \cdot 10^6$ | 10 | 3600.62 | 1.66 |
| 5000 | SubsetSelectCIO | $-1.14948 \cdot 10^6$ | 10 | 3626.73 | 29.45 |
| 5000 | SubsetSelectCIO + warm start | $-1.23402 \cdot 10^6$ | 10 | 3607.12 | 20.58 |
| 5000 | L0BnB | $-1.23544 \cdot 10^6$ | 9 | 7434.60 | 4.95 |
| 5000 | MSK persp. | $-1.24017 \cdot 10^6$ | 10 | 3602.18 | 7.35 |
| 5000 | MSK persp. + warm start | $-1.24017 \cdot 10^6$ | 10 | 3604.42 | 7.35 |
| 5000 | Convex MSK opt. persp. | $-1.24017 \cdot 10^6$ | 10 | 1606.64 | 1.03 |
| 5000 | ours | $-1.24017 \cdot 10^6$ | 10 | 386.72 | 0.00 |
| 6000 | SubsetSelectCIO | $-1.36967 \cdot 10^6$ | 10 | 3743.26 | 27.64 |
| 6000 | SubsetSelectCIO + warm start | $-1.43335 \cdot 10^6$ | 10 | 3802.62 | 21.97 |
| 6000 | L0BnB | $-1.44849 \cdot 10^6$ | 9 | 7430.07 | 4.18 |
| 6000 | MSK persp. | $-1.45406 \cdot 10^6$ | 10 | 3608.33 | 6.98 |
| 6000 | MSK persp. + warm start | $-1.45406 \cdot 10^6$ | 10 | 3604.65 | 6.98 |
| 6000 | Convex MSK opt. persp. | $-1.45406 \cdot 10^6$ | 10 | 1665.16 | 0.62 |
| 6000 | ours | $-1.45406 \cdot 10^6$ | 10 | 54.13 | 0.00 |
| 7000 | SubsetSelectCIO | $-1.60283 \cdot 10^6$ | 10 | 3661.63 | 27.20 |
| 7000 | SubsetSelectCIO + warm start | $-1.68426 \cdot 10^6$ | 10 | 3897.62 | 21.05 |
| 7000 | L0BnB | $-1.69016 \cdot 10^6$ | 8 | 7280.69 | 3.21 |
| 7000 | MSK persp. | $-1.70473 \cdot 10^6$ | 10 | 3603.26 | 6.07 |
| 7000 | MSK persp. + warm start | $-1.70473 \cdot 10^6$ | 10 | 3603.98 | 6.07 |
| 7000 | Convex MSK opt. persp. | $-1.70473 \cdot 10^6$ | 10 | 1677.45 | 0.45 |
| 7000 | ours | $-1.70473 \cdot 10^6$ | 10 | 21.30 | 0.00 |

Table 149: Comparison of time and optimality gap on the synthetic datasets (p = 3000, k = 10, $\rho = 0.7$, $\lambda_2 = 10.0$).

| # of samples | method | upper bound | support size | time(s) | gap(%) |
|---|---|---|---|---|---|
| 3000 | SubsetSelectCIO | $-2.76095 \cdot 10^6$ | 10 | 3644.99 | 20.66 |
| 3000 | SubsetSelectCIO + warm start | $-2.78489 \cdot 10^6$ | 10 | 3685.28 | 19.62 |
| 3000 | L0BnB | $-2.55553 \cdot 10^6$ | 1 | 7230.61 | 17.51 |
| 3000 | MSK persp. | $-2.78334 \cdot 10^6$ | 10 | 3601.82 | 10.13 |
| 3000 | MSK persp. + warm start | $-2.78334 \cdot 10^6$ | 10 | 3600.26 | 10.13 |
| 3000 | Convex MSK opt. persp. | $-2.77971 \cdot 10^6$ | 10 | 1638.17 | 10.70 |
| 3000 | ours | $-2.78507 \cdot 10^6$ | 10 | 3600.78 | 9.79 |
| 4000 | SubsetSelectCIO | $-3.71695 \cdot 10^6$ | 10 | 3601.35 | 21.65 |
| 4000 | SubsetSelectCIO + warm start | $-3.74577 \cdot 10^6$ | 10 | 3638.03 | 20.71 |
| 4000 | L0BnB | $-3.45116 \cdot 10^6$ | 1 | 7514.37 | 15.03 |
| 4000 | MSK persp. | $-3.74752 \cdot 10^6$ | 10 | 3602.22 | 8.70 |
| 4000 | MSK persp. + warm start | $-3.75192 \cdot 10^6$ | 10 | 3602.31 | 8.57 |
| 4000 | Convex MSK opt. persp. | $-3.75191 \cdot 10^6$ | 10 | 1564.01 | 3.01 |
| 4000 | ours | $-3.75192 \cdot 10^6$ | 10 | 3600.32 | 3.23 |
| 5000 | SubsetSelectCIO | $-4.57617 \cdot 10^6$ | 10 | 3864.92 | 21.14 |
| 5000 | SubsetSelectCIO + warm start | $-4.60856 \cdot 10^6$ | 10 | 3764.48 | 20.29 |
| 5000 | L0BnB | $-4.23012 \cdot 10^6$ | 1 | 7233.86 | 13.08 |
| 5000 | MSK persp. | $-4.61866 \cdot 10^6$ | 10 | 3603.72 | 7.81 |
| 5000 | MSK persp. + warm start | $-4.61867 \cdot 10^6$ | 10 | 3603.77 | 7.90 |
| 5000 | Convex MSK opt. persp. | $-4.61867 \cdot 10^6$ | 10 | 1492.40 | 1.19 |
| 5000 | ours | $-4.61867 \cdot 10^6$ | 10 | 3600.30 | 0.89 |
| 6000 | SubsetSelectCIO | $-5.31865 \cdot 10^6$ | 10 | 3718.93 | 22.96 |
| 6000 | SubsetSelectCIO + warm start | $-5.42880 \cdot 10^6$ | 10 | 3662.21 | 20.47 |
| 6000 | L0BnB | $-4.97660 \cdot 10^6$ | 1 | 7210.21 | 12.38 |
| 6000 | MSK persp. | $-5.43816 \cdot 10^6$ | 10 | 3606.21 | 7.09 |
| 6000 | MSK persp. + warm start | $-5.43816 \cdot 10^6$ | 10 | 3602.03 | 7.09 |
| 6000 | Convex MSK opt. persp. | $-5.43816 \cdot 10^6$ | 10 | 1527.55 | 0.64 |
| 6000 | ours | $-5.43816 \cdot 10^6$ | 10 | 3600.22 | 0.32 |
| 7000 | SubsetSelectCIO | $-6.30345 \cdot 10^6$ | 10 | 3670.08 | 20.83 |
| 7000 | SubsetSelectCIO + warm start | $-6.34815 \cdot 10^6$ | 10 | 3714.08 | 19.98 |
| 7000 | L0BnB | $-5.82238 \cdot 10^6$ | 1 | 7236.55 | 11.40 |
| 7000 | MSK persp. | - | - | 3604.74 | - |
| 7000 | MSK persp. + warm start | $-6.36757 \cdot 10^6$ | 10 | 3607.94 | 6.15 |
| 7000 | Convex MSK opt. persp. | $-6.36757 \cdot 10^6$ | 10 | 1747.23 | 0.45 |
| 7000 | ours | $-6.36757 \cdot 10^6$ | 10 | 3600.06 | 0.10 |

Table 150: Comparison of time and optimality gap on the synthetic datasets (p = 3000, k = 10, $\rho = 0.9$, $\lambda_2 = 10.0$).

## G.5 Experiments on the Case $p >> n$

| # of samples | method | upper bound | support size | time(s) | gap(%) |
|---|---|---|---|---|---|
| 100 | SOS1 + warm start | $-2.76902 \cdot 10^3$ | 10 | 3600.94 | 13.02 |
| 100 | big-M50 + warm start | $-2.76902 \cdot 10^3$ | 10 | 3600.58 | 13.02 |
| 100 | big-M20 + warm start | $-2.76902 \cdot 10^3$ | 10 | 3600.47 | 13.02 |
| 100 | big-M5 + warm start | $-2.76902 \cdot 10^3$ | 10 | 3600.17 | 13.02 |
| 100 | persp. + warm start | $-2.76902 \cdot 10^3$ | 10 | 3602.22 | 13.02 |
| 100 | eig. persp. + warm start | $-2.76902 \cdot 10^3$ | 10 | 3600.65 | 13.02 |
| 100 | Convex opt. persp. + warm start | $-2.51480 \cdot 10^3$ | 10 | 1706.56 | 24.44 |
| 100 | MSK persp. + warm start | $-2.71927 \cdot 10^3$ | 10 | 3599.92 | 15.08 |
| 100 | SubsetSelectCIO + warm start | $-1.68157 \cdot 10^3$ | 10 | 4350.20 | 86.11 |
| 100 | L0BnB | $-7.78758 \cdot 10^2$ | 1 | 7217.66 | 255.64 |
| 100 | ours | $-2.76902 \cdot 10^3$ | 10 | 3600.22 | 13.02 |
| 200 | SOS1 + warm start | $-4.15945 \cdot 10^3$ | 10 | 3600.08 | 17.20 |
| 200 | big-M50 + warm start | $-4.15945 \cdot 10^3$ | 10 | 3600.05 | 17.20 |
| 200 | big-M20 + warm start | $-4.15945 \cdot 10^3$ | 10 | 3600.11 | 17.20 |
| 200 | big-M5 + warm start | $-4.15945 \cdot 10^3$ | 10 | 3600.22 | 17.20 |
| 200 | persp. + warm start | $-4.15945 \cdot 10^3$ | 10 | 3600.20 | 17.20 |
| 200 | eig. persp. + warm start | $-4.15945 \cdot 10^3$ | 10 | 3600.24 | 17.20 |
| 200 | Convex opt. persp. + warm start | $-4.11140 \cdot 10^3$ | 10 | 1800.00 | 18.57 |
| 200 | MSK persp. + warm start | $-4.15945 \cdot 10^3$ | 10 | 3599.97 | 17.20 |
| 200 | SubsetSelectCIO + warm start | $-2.74363 \cdot 10^3$ | 10 | 3819.10 | 77.69 |
| 200 | L0BnB | $-1.20299 \cdot 10^3$ | 1 | 7208.65 | 240.10 |
| 200 | ours | $-4.15945 \cdot 10^3$ | 10 | 3601.81 | 17.20 |
| 500 | SOS1 + warm start | $-1.00615 \cdot 10^4$ | 10 | 3600.51 | 20.46 |
| 500 | big-M50 + warm start | $-1.00615 \cdot 10^4$ | 10 | 3600.28 | 20.46 |
| 500 | big-M20 + warm start | $-1.00615 \cdot 10^4$ | 10 | 3600.07 | 20.46 |
| 500 | big-M5 + warm start | $-1.00615 \cdot 10^4$ | 10 | 3600.11 | 20.46 |
| 500 | persp. + warm start | $-1.00615 \cdot 10^4$ | 10 | 3600.37 | 20.46 |
| 500 | eig. persp. + warm start | $-1.00615 \cdot 10^4$ | 10 | 3600.10 | 20.46 |
| 500 | Convex opt. persp. + warm start | $-1.00615 \cdot 10^4$ | 10 | 1744.05 | 20.46 |
| 500 | MSK persp. + warm start | $-1.00615 \cdot 10^4$ | 10 | 3600.21 | 20.46 |
| 500 | SubsetSelectCIO + warm start | - | - | 2.29 | - |
| 500 | L0BnB | $-1.00615 \cdot 10^4$ | 10 | 1834.75 | 0.00 |
| 500 | ours | $-1.00615 \cdot 10^4$ | 10 | 3600.23 | 20.46 |

Table 151: Comparison of time and optimality gap on the synthetic datasets (p = 3000, k = 10, $\rho = 0.1$, $\lambda_2 = 0.001$). All baselines use our beam search solution as a warm start whenever possible. The results in the case of $p >> n$ show that OKRidge is competitive with other MIP methods. l0bnb can solve the problem when $n = 500$ but has difficulty in finding the desired sparsity level in the case of $n = 100$ and $n = 200$. MOSEK sometimes gives the best optimality gap. However, as we have shown in the previous subsections, MOSEK is not scalable to high dimensions. Additionally, the results in differential equations show that MOSEK and l0bnb are not competitive in discovering differential equations (datasets have $n \geq p$), which is the main focus of this paper. In the future, we plan to extend OKRidge to develop a more scalable algorithm for the case $p >> n$.

| # of samples | method | upper bound | support size | time(s) | gap(%) |
|---|---|---|---|---|---|
| 100 | SOS1 + warm start | $-2.76721 \cdot 10^3$ | 10 | 3601.59 | 13.09 |
| 100 | big-M50 + warm start | $-2.76721 \cdot 10^3$ | 10 | 3600.66 | 13.09 |
| 100 | big-M20 + warm start | $-2.76721 \cdot 10^3$ | 10 | 3600.32 | 13.09 |
| 100 | big-M5 + warm start | $-2.76721 \cdot 10^3$ | 10 | 3602.22 | 13.09 |
| 100 | persp. + warm start | $-2.76721 \cdot 10^3$ | 10 | 3600.09 | 13.09 |
| 100 | eig. persp. + warm start | $-2.76721 \cdot 10^3$ | 10 | 3600.03 | 13.09 |
| 100 | Convex opt. persp. + warm start | $-2.51378 \cdot 10^3$ | 10 | 2128.32 | 24.44 |
| 100 | MSK persp. + warm start | $-2.71018 \cdot 10^3$ | 10 | 3599.92 | 15.36 |
| 100 | SubsetSelectCIO + warm start | $-7.59815 \cdot 10^2$ | 10 | 3651.10 | 311.88 |
| 100 | L0BnB | $-7.78035 \cdot 10^2$ | 1 | 7218.43 | 255.36 |
| 100 | ours | $-2.76721 \cdot 10^3$ | 10 | 3600.35 | 13.08 |
| 200 | SOS1 + warm start | $-4.15848 \cdot 10^3$ | 10 | 3600.46 | 17.23 |
| 200 | big-M50 + warm start | $-4.15848 \cdot 10^3$ | 10 | 3600.42 | 17.23 |
| 200 | big-M20 + warm start | $-4.15848 \cdot 10^3$ | 10 | 3600.30 | 17.23 |
| 200 | big-M5 + warm start | $-4.15848 \cdot 10^3$ | 10 | 3600.10 | 17.23 |
| 200 | persp. + warm start | $-4.15848 \cdot 10^3$ | 10 | 3600.12 | 17.23 |
| 200 | eig. persp. + warm start | $-4.15848 \cdot 10^3$ | 10 | 3600.47 | 17.23 |
| 200 | Convex opt. persp. + warm start | $-4.11040 \cdot 10^3$ | 10 | 2159.71 | 18.55 |
| 200 | MSK persp. + warm start | $-4.15848 \cdot 10^3$ | 10 | 3599.93 | 17.11 |
| 200 | SubsetSelectCIO + warm start | $-3.14066 \cdot 10^3$ | 10 | 3632.89 | 55.23 |
| 200 | L0BnB | $-1.20245 \cdot 10^3$ | 1 | 7208.71 | 237.38 |
| 200 | ours | $-4.15848 \cdot 10^3$ | 10 | 3600.13 | 17.22 |
| 500 | SOS1 + warm start | $-1.00605 \cdot 10^4$ | 10 | 3600.08 | 20.47 |
| 500 | big-M50 + warm start | $-1.00605 \cdot 10^4$ | 10 | 3600.08 | 20.47 |
| 500 | big-M20 + warm start | $-1.00605 \cdot 10^4$ | 10 | 3600.80 | 20.47 |
| 500 | big-M5 + warm start | $-1.00605 \cdot 10^4$ | 10 | 3600.52 | 20.47 |
| 500 | persp. + warm start | $-1.00605 \cdot 10^4$ | 10 | 3600.04 | 20.47 |
| 500 | eig. persp. + warm start | $-1.00605 \cdot 10^4$ | 10 | 3600.06 | 20.47 |
| 500 | Convex opt. persp. + warm start | $-1.00605 \cdot 10^4$ | 10 | 2059.28 | 20.41 |
| 500 | MSK persp. + warm start | $-1.00605 \cdot 10^4$ | 10 | 3600.08 | 20.33 |
| 500 | SubsetSelectCIO + warm start | $-7.47048 \cdot 10^3$ | 10 | 3748.65 | 62.24 |
| 500 | L0BnB | $-1.00605 \cdot 10^4$ | 10 | 1835.06 | 0.00 |
| 500 | ours | $-1.00605 \cdot 10^4$ | 10 | 3601.58 | 20.46 |

Table 152: Comparison of time and optimality gap on the synthetic datasets (p = 3000, k = 10, $\rho = 0.1$, $\lambda_2 = 0.1$). All baselines use our beam search solution as a warm start whenever possible. The results in the case of $p >> n$ show that OKRidge is competitive with other MIP methods. l0bnb can solve the problem when $n = 500$ but has difficulty in finding the desired sparsity level in the case of $n = 100$ and $n = 200$. MOSEK sometimes gives the best optimality gap. However, as we have shown in the previous subsections, MOSEK is not scalable to high dimensions. Additionally, the results in differential equations show that MOSEK and l0bnb are not competitive in discovering differential equations (datasets have $n \geq p$), which is the main focus of this paper. In the future, we plan to extend OKRidge to develop a more scalable algorithm for the case $p >> n$.

| # of samples | method | upper bound | support size | time(s) | gap(%) |
|---|---|---|---|---|---|
| 100 | SOS1 + warm start | $-2.61800 \cdot 10^3$ | 10 | 3600.36 | 19.31 |
| 100 | big-M50 + warm start | $-2.61800 \cdot 10^3$ | 10 | 3600.18 | 19.31 |
| 100 | big-M20 + warm start | $-2.61800 \cdot 10^3$ | 10 | 3600.40 | 19.31 |
| 100 | big-M5 + warm start | $-2.61800 \cdot 10^3$ | 10 | 3602.50 | 19.31 |
| 100 | persp. + warm start | $-2.61800 \cdot 10^3$ | 10 | 3600.47 | 19.54 |
| 100 | eig. persp. + warm start | $-2.61800 \cdot 10^3$ | 10 | 3600.19 | 19.54 |
| 100 | Convex opt. persp. + warm start | $-2.40846 \cdot 10^3$ | 10 | 1968.34 | 25.24 |
| 100 | MSK persp. + warm start | $-2.61800 \cdot 10^3$ | 10 | 3599.93 | 10.09 |
| 100 | SubsetSelectCIO + warm start | $-1.79729 \cdot 10^3$ | 10 | 3646.28 | 74.12 |
| 100 | L0BnB | $-7.11936 \cdot 10^2$ | 1 | 7212.43 | 236.77 |
| 100 | ours | $-2.61806 \cdot 10^3$ | 10 | 3600.78 | 19.09 |
| 200 | SOS1 + warm start | $-4.06441 \cdot 10^3$ | 10 | 3600.07 | 19.69 |
| 200 | big-M50 + warm start | $-4.06441 \cdot 10^3$ | 10 | 3600.92 | 19.69 |
| 200 | big-M20 + warm start | $-4.06441 \cdot 10^3$ | 10 | 3600.61 | 19.69 |
| 200 | big-M5 + warm start | $-4.06441 \cdot 10^3$ | 10 | 3600.07 | 19.69 |
| 200 | persp. + warm start | $-4.06441 \cdot 10^3$ | 10 | 3600.03 | 11.80 |
| 200 | eig. persp. + warm start | $-4.06441 \cdot 10^3$ | 10 | 3600.79 | 11.80 |
| 200 | Convex opt. persp. + warm start | $-4.01212 \cdot 10^3$ | 10 | 2175.61 | 16.91 |
| 200 | MSK persp. + warm start | $-4.06441 \cdot 10^3$ | 10 | 3599.92 | 9.05 |
| 200 | SubsetSelectCIO + warm start | $-2.68289 \cdot 10^3$ | 10 | 3704.01 | 81.71 |
| 200 | L0BnB | $-1.15079 \cdot 10^3$ | 1 | 7207.25 | 219.55 |
| 200 | ours | $-4.06441 \cdot 10^3$ | 10 | 3600.92 | 19.45 |
| 500 | SOS1 + warm start | $-9.96316 \cdot 10^3$ | 10 | 3601.58 | 21.39 |
| 500 | big-M50 + warm start | $-9.96316 \cdot 10^3$ | 10 | 3602.06 | 21.39 |
| 500 | big-M20 + warm start | $-9.96316 \cdot 10^3$ | 10 | 3602.58 | 21.39 |
| 500 | big-M5 + warm start | $-9.96316 \cdot 10^3$ | 10 | 3600.48 | 21.28 |
| 500 | persp. + warm start | $-9.96316 \cdot 10^3$ | 10 | 3600.14 | 21.65 |
| 500 | eig. persp. + warm start | $-9.96316 \cdot 10^3$ | 10 | 3600.12 | 13.16 |
| 500 | Convex opt. persp. + warm start | $-9.96316 \cdot 10^3$ | 10 | 2145.98 | 16.65 |
| 500 | MSK persp. + warm start | $-9.96316 \cdot 10^3$ | 10 | 3600.03 | 11.45 |
| 500 | SubsetSelectCIO + warm start | $-7.48575 \cdot 10^3$ | 10 | 3793.20 | 61.91 |
| 500 | L0BnB | $-9.96316 \cdot 10^3$ | 10 | 1483.07 | 0.00 |
| 500 | ours | $-9.96316 \cdot 10^3$ | 10 | 3600.96 | 21.14 |

Table 153: Comparison of time and optimality gap on the synthetic datasets (p = 3000, k = 10, $\rho = 0.1$, $\lambda_2 = 10.0$). All baselines use our beam search solution as a warm start whenever possible. The results in the case of $p >> n$ show that OKRidge is competitive with other MIP methods. When the number of samples is 100, OKRidge is able to find a slightly better solution through the branch-and-bound process, but other solvers are stuck with the warm start solution. l0bnb can solve the problem when $n = 500$ but has difficulty in finding the desired sparsity level in the case of $n = 100$ and $n = 200$. MOSEK sometimes gives the best optimality gap. However, as we have shown in the previous subsections, MOSEK is not scalable to high dimensions. Additionally, the results in differential equations show that MOSEK and l0bnb are not competitive in discovering differential equations (datasets have $n \geq p$), which is the main focus of this paper. In the future, we plan to extend OKRidge to develop a more scalable algorithm for the case $p >> n$.

## G.6 Comparison between Fast Solve and ADMM

In this subsection, we provide the empirical evidence that the ADMM method is more effective than the fast solve method in terms of the computing a tighter lower bound, which helps speed up the certification process.

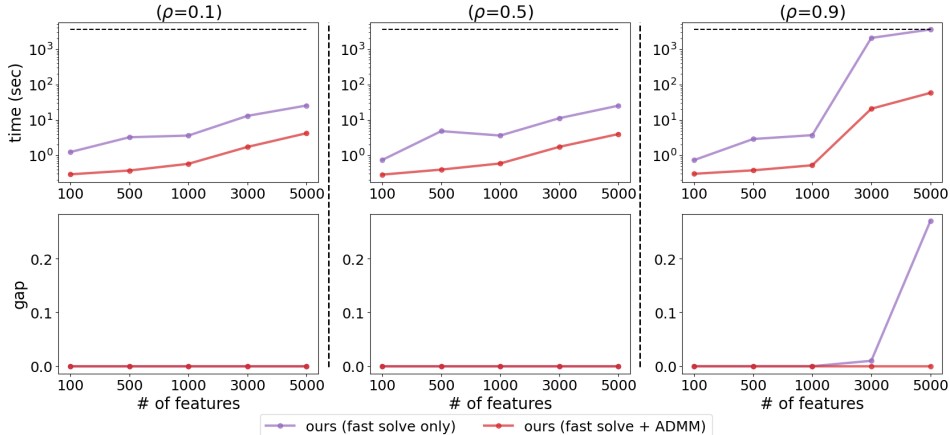

Figure 7: Comparison of running time (top row) and optimality gap (bottom row) between using fast solve (Equation (16)) alone and using fast solve and ADMM (Equations (19)- (22)) to calculate the lower bounds in the BnB tree with three correlation levels $\rho = 0.1, 0.5, 0.9$ ($n = 10000, k = 10$).

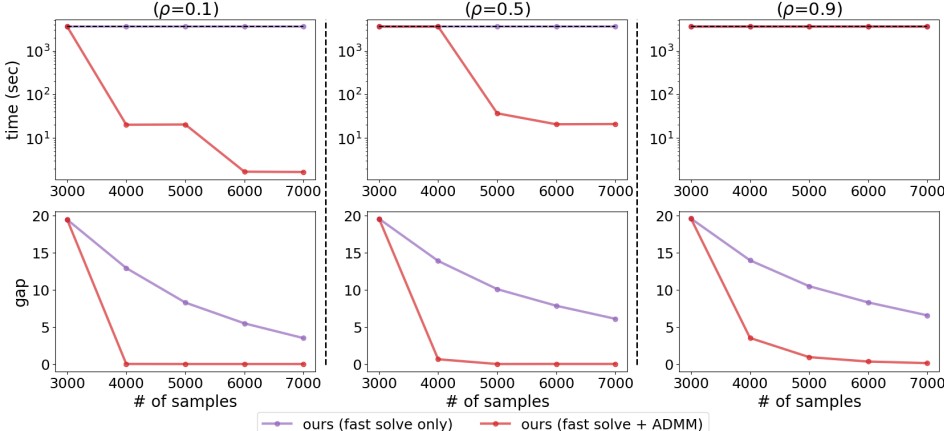

Figure 8: Comparison of running time (top row) and optimality gap (bottom row) between using fast solve (Equation (16)) alone and using fast solve and ADMM (Equations (19)- (22)) to calculate the lower bounds in the BnB tree with three correlation levels $\rho = 0.1, 0.5, 0.9$ ($p = 3000, k = 10$).

# H More Experimental Results on Dynamical Systems

## H.1 Results Summarization

**Comparison with MIOSR** We first show a direct comparison between OKRidge and MIOSR (based on SOS1 formulation). The results are in Appendix H.2, where we show OKRidge outperforms MIOSR in terms of both solution quality and running time.

**Comparison with Different MIP Formualtions** Next, we compare OKRidge with other MIP formulations (SOS1, big-M, perspective, and eigen-perspective) as done on the synthetic benchmarks. These formulations have never been applied to the differential equation experiments. Similar to the synthetic benchmarks, we show that OKRidge obtain better solution quality and smaller running time. The results are shown in Appendix H.3.

**Comparison between Beam Search and More Heuristics** In addition, we also compare our beam search method with many more heuristic methods from the sparse regression literature. Beam search outperforms all methods in terms of solution qualities in presence of highly correlated features. Block Decomposition algorithm also achieves high-quality solutions, but is much slower than beam search. The results are shown in Appendix H.4

**Comparison with Different MIP Formulations Warm Started by Beam Search** Due to the superiority of beam search, we give beam search solutions as warm starts to Gurobi with different MIP formulations and repeat the experiments. Our warm start improve the solution quality and running time, but OKRidge still outperforms other MIP formulations in terms of running time. The results are shown in Appendix H.5.

**Comparison with Other MIP Solvers and Formulations** Lastly, we solve the same problem using the MOSEK solver, l0bnb, and SubsetSelectionCIO. For this application, l0bnb has a time limit of 10 seconds for each $\ell_0$ regularization and 60 seconds for the total time limit. All other algorithms have a total time limit of 30 seconds as mentioned in the experimental setup subsection. Although they are not as competitive as Gurobi as we have shown on the synthetic benchmarks, we still conduct the experiments for the purpose of thorough comparison. Results in Appendix H.6 show that OKRidge outperforms these MIP solvers/formulations.

## H.2 Direct Comparison of OKRidge with MIOSR

In the appendix, we directly compare MIOSR (based on the SOS1 formulation) and our method on the differential equation experiments. The results are shown in Figure 9.

Although MIOSR is faster than OKRidge on the Lorenz system, OKRidge is able to find better solutions. See Appendix H.3 and Figure 10 where the eigen-perspective formulation doesn't have this problem and can produce high-quality solutions.

On the Hopf system, MIOSR is faster than OKRidge. However, the number of features is small ($p = 21$), and the running time is negligible compared to Lorenz and MHD systems.

On the MHD system ($p = 462$), OKRidge outperforms MIOSR in terms of both solution qualities and running times. This shows that OKRidge is much more scalable to high dimensions than MIOSR.

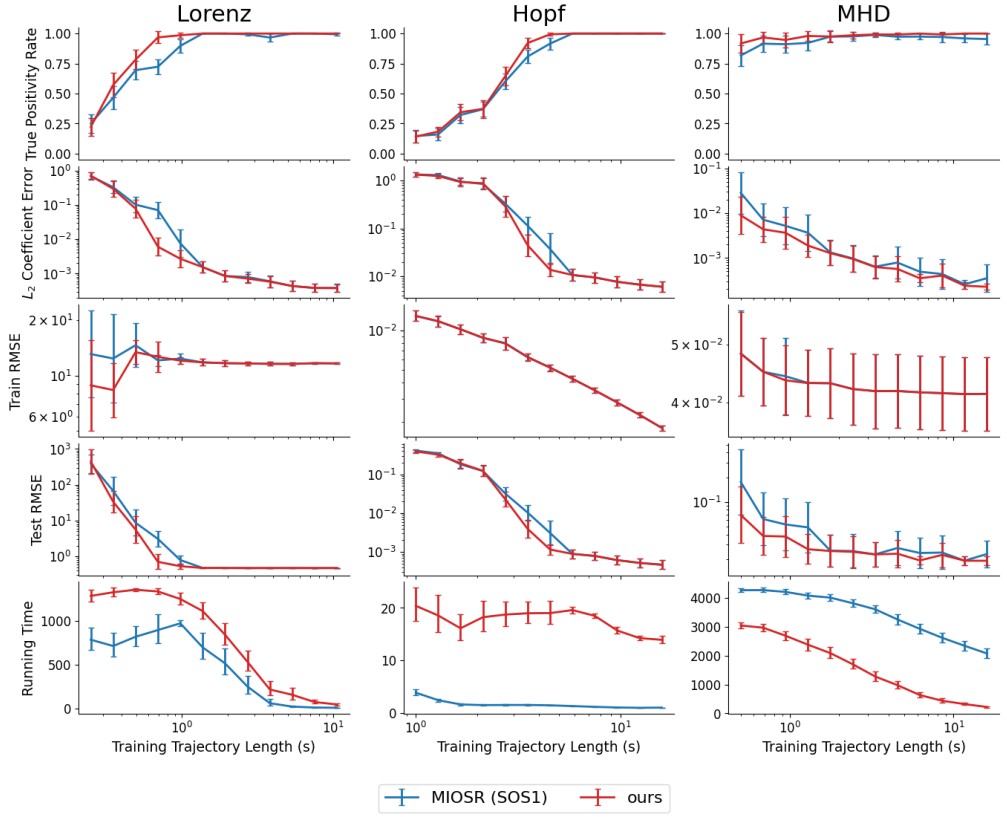

Figure 9: Comparison between our method and MIOSR on the experiments of discovering sparse differential equations. We obtain better performance on the true positivity rate, $L_2$ coefficient error, and test RMSE.

### H.3 Comparison of OKRidge with SOS1, Big-M, Perspective, and Eigen-Perspective Formulations

Besides the MIOSR baseline (which is based on the SOS1 formualtion), we also compare with other MIP formulations, including Big-M ($M = 50$), perspective, and eigen-perspective formulations. The results are shown in Figure 10. The big-M method produces errors on the Lorenz system, so its result is not shown on the left column. We have discussed about the SOS1 formulation in the last subsection. On the Lorenz system, OKRidge is faster than the perspective and eigen-perspective formulations. On the Hopf system ($p = 21$), OKRidge is slower than Gurobi, but OKRidge takes only 20 seconds. On the high-dimensional MHD system, OKRidge is significantly faster than all MIP formulations.

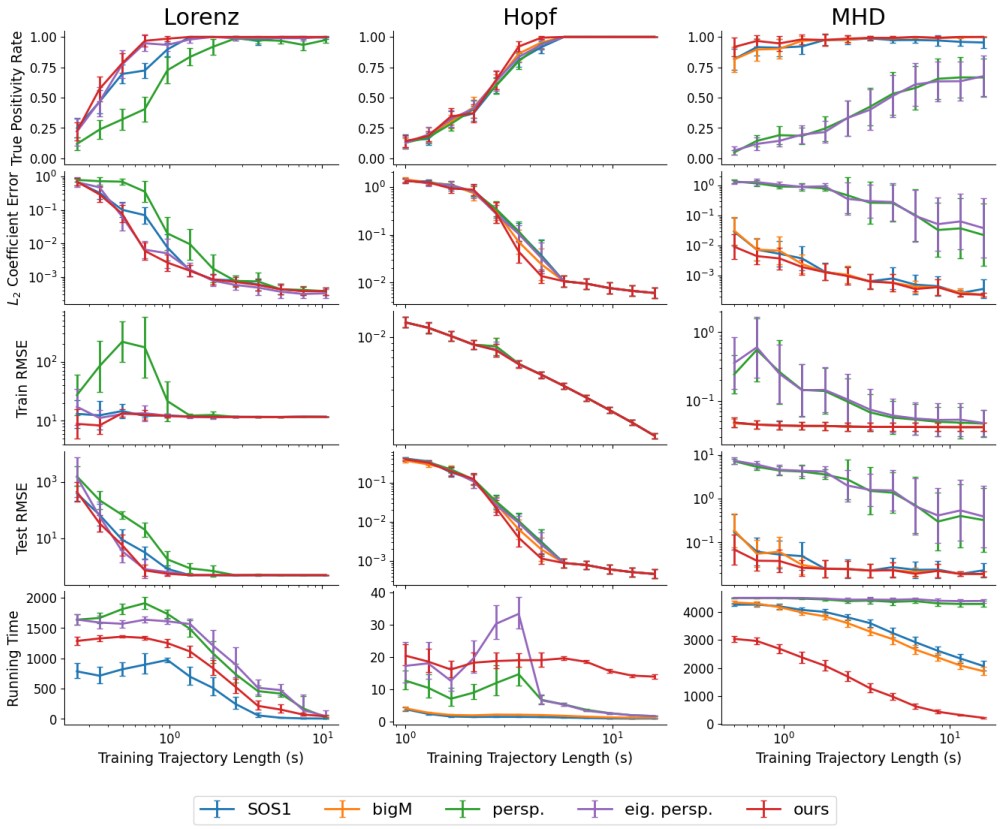

Figure 10: Comparison between our method and other MIP formulations solved by Gurobi on the experiments of discovering sparse differential equations. We obtain better performance on the true positivity rate, $L_2$ coefficient error, and test RMSE. SOS1 is faster than ours on the Lorenz system, but the solution quality is worse. OKRidge is slower than Gurobi on the Hopf system (p=21), but the time difference is not significant. On the high-dimensional MHD system, OKRidge is significantly faster than Gurobi with all exisiting MIP formulations.

### H.4 Comparison of Our Beam Search with More Heuristic Baselines

Next, we compare our proposed beam search method with more heuristic methods from the sparse regression literature. The results are shown in Figure 11. Since many heuristic method implementations do not consider the ridge regression, we set $\ell_2$ regularization for all algorithms to be 0 for a fair comparison. Since the features are highly correlated, most of the existing heuristic methods do not produce high-quality solutions except the Block Decomposition algorithm. However, our beam search method is significantly faster than the Block Decomposition algorithm.

Moreover, for the purpose of completeness, we also compare with LASSO, the classical sparse learning method. Other researchers have found that LASSO performs poorly in this context. We confirm this claim experimentally. The result is shown in Figure 12.

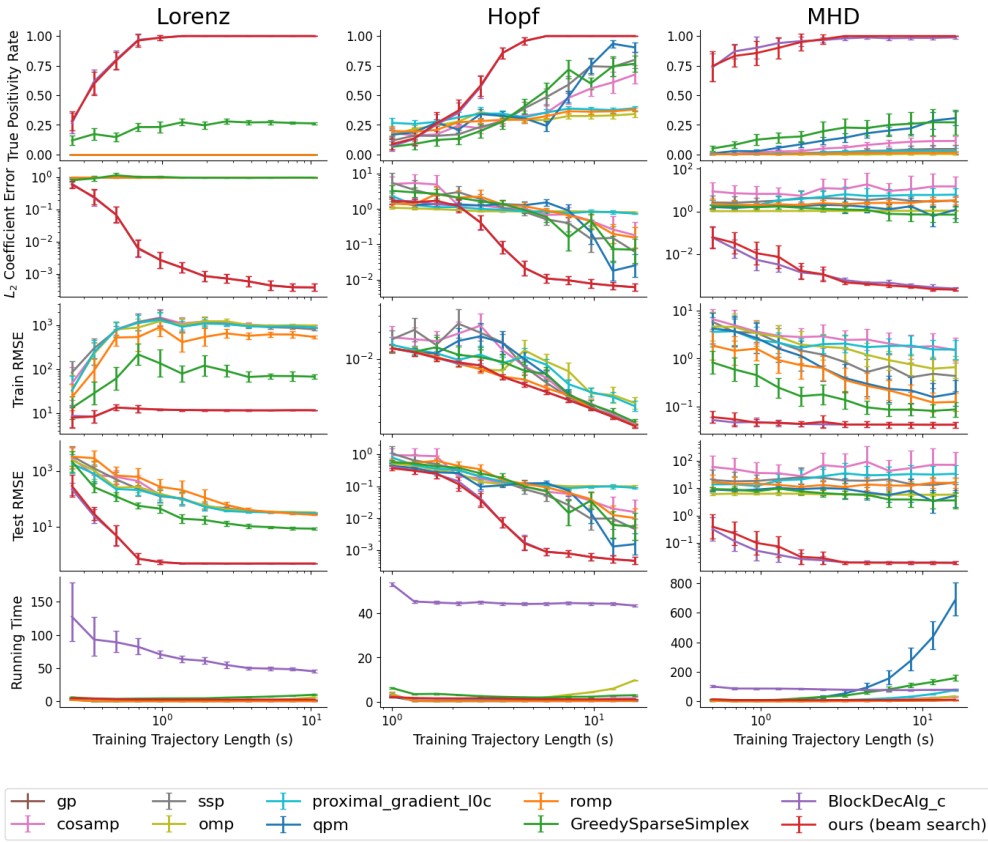

Figure 11: Comparison between our method and other heuristic methods on the experiments of discovering sparse differential equations. We obtain better performance on the true positivity rate, $L_2$ coefficient error, and test RMSE. The Block Decomposition algorithm also produces high-quality solutions but is significantly slower than our beam search.

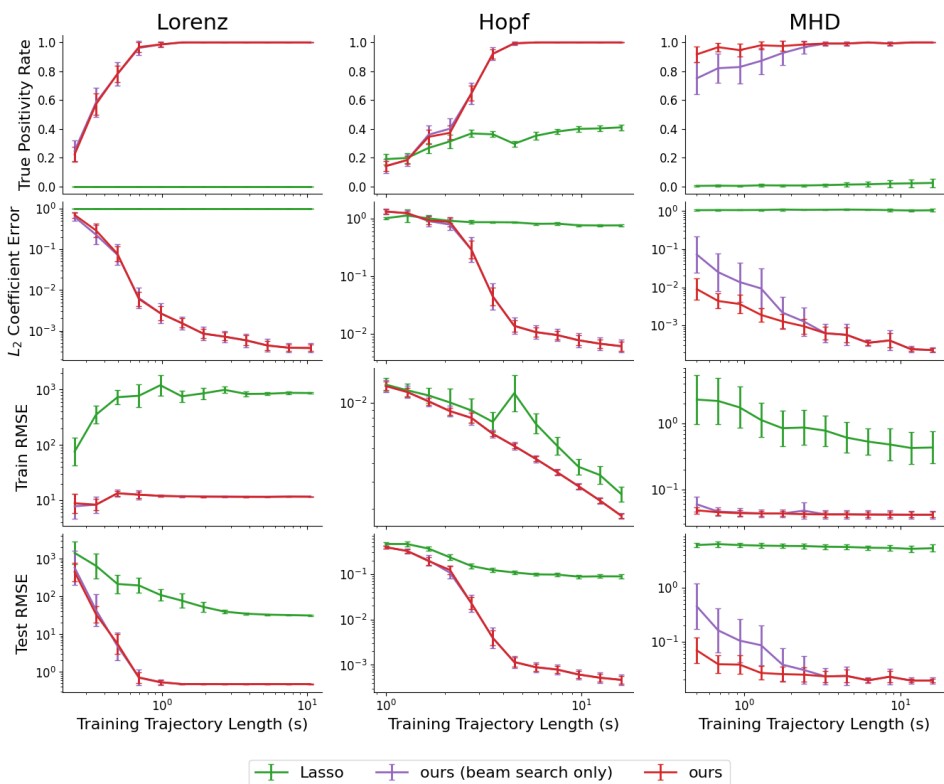

Figure 12: Results on discovering sparse differential equations. Lasso does not do well in this application. Beam search lags behind OKRidge on the MHD system. This shows that beam search alone is not sufficient, and it is necessary to use BnB to obtain the optimal solution.

### H.5 Comparison of OKRidge with MIPs Warmstarted by Our Beam Search

A natural question to ask is whether Gurobi would be faster than OKRidge in terms of running speed if we give the high-quality solutions from last subsection as warm starts to Gurobi . Here, we compare with other formulations warmstarted by our beam search solutions. The results are shown in Figure 13. By comparing Figure 10 with Figure 13, we see that our beam search solutions, if used as warm starts, improve Gurobi's speed. However, OKRidge is still significantly faster than all MIP formulations on the high-dimensional MHD system.

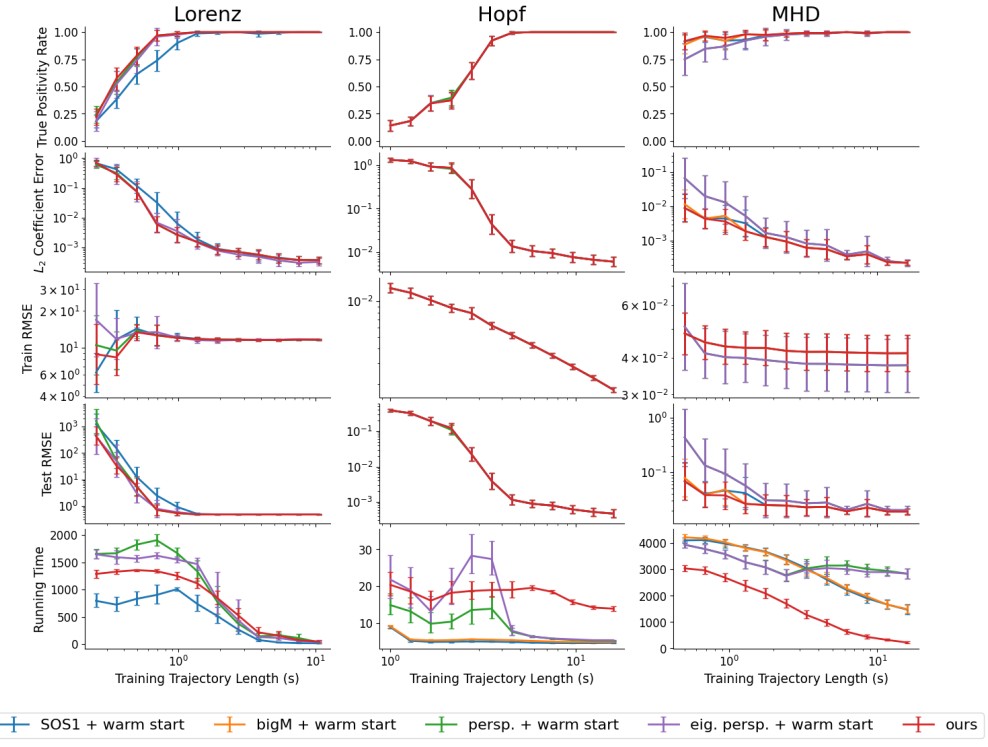

Figure 13: Comparison between our method and other MIP formulations solved by Gurobi, which was warmstarted by our beam search solutions on the experiments of discovering sparse differential equations. OKRidge is slower than Gurobi on the Hopf system ($p = 21$), but the running time is only 20 seconds. On the Lorenz and MHD systems, OKRidge is faster than other MIP formulations, especially on the high-dimensional MHD system ($p = 462$).

### H.6 Comparison of OKRidge with MOSEK solver, SubsetSelectionCIO, and L0BNB

Lastly, we compare with other MIP solvers/formulations, including the MOSEK solver, SubsetSelectionCIO, and l0bnb. The results are shown in Figure 14.

MOSEK (perspective formualtion) produces good solutions on the Hopf dataset, but could not produce optimal solutions on the Lorenz system. MOSEK (optimal formulation) produes suboptimal solutions on the Lorenz and Hopf systems. However, we are using MOSEK to solve the relaxed convex SDP problem. This demonstrates why we need to tackle the NP hard problem instead of the relaxed convex problem. MOSEK runs out of memory on the MHD dataset, so we don't show the results on the MHD dataset.

SubsetSelectionCIO doesn't do well on the Lorenz and MHD systems. On many instances, SubsetSelectionCIO would have errors.

l0bnb performs poorly on these datasets. One major reason is that l0bnb could not retrieve the exact support size ($k = 1, 2, 3, 4, 5$) because the sparsity is controlled implicitly through $\ell_0$ regularization. It takes a long time to run these differential experiments. l0bnb couldn't find a good $\ell_0$ regularization which produces the right sparsity level within the time limit.

Please see Appendix G.1 for a detailed discussion of the drawbacks on using MOSEK, SubsetSelectionCIO, and l0bnb.

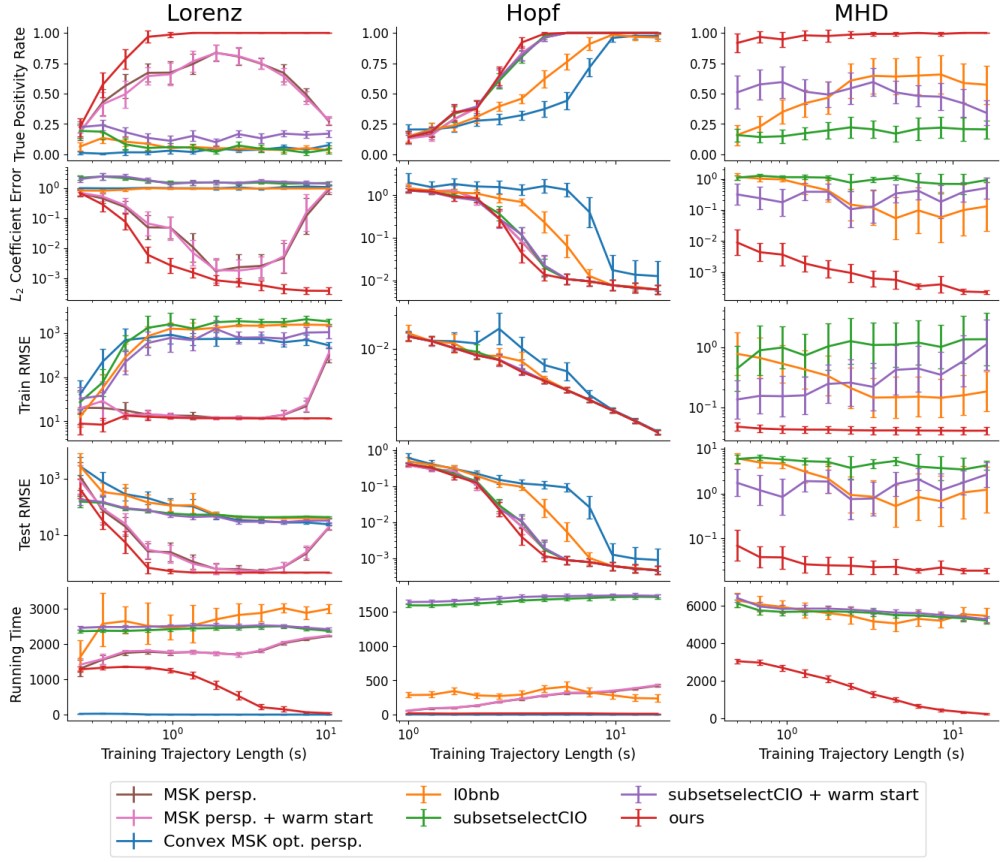

Figure 14: Comparison between our method and other MIP solvers or formulations on the experiments of discovering sparse differential equations. We obtain better performance on the true positivity rate, $L_2$ coefficient error, and test RMSE.

