# OpenReview forum: "OKRidge: Scalable Optimal k-Sparse Ridge Regression"
_NeurIPS.cc/2023/Conference — NeurIPS 2023 spotlight_

### Official Review · Reviewer_Nrfj · 2023-06-13

**Soundness:** 3 good
**Presentation:** 3 good
**Contribution:** 3 good
**Rating:** 7
**Confidence:** 4

**Summary:**

This paper proposes a novel branch-and-bound technique called OKRidge to solve sparse Ridge regression problems to learn differential equations. The paper makes 3 contributions:
1) A saddle point formulation of the optimization problem at hand which allows the use of a branch-and-bound algorithm leveraging the special structure (sparsity) of the problem.
2) This efficient branch-and-bound algorithm constructs tight lower-bounds thanks to the saddle-point formulation.
3) A beam search that leverages previous searches for past nodes to speed up the search process of the current node.

**Strengths:**

The paper is well-written and easy to follow. Apart from some details (see question section below) that might be unclear to the non-expert readers, the context of the paper is clear and the literature review is comprehensive.

Besides, several extensive experiments have been made. Given the number of hyperparameters in the method ($\lambda$, $\lambda_2$) and the hidden parameters that might impact the convergence speed of the proposed methodology (correlation $\rho$, setup size $n$ and $p$, $\textrm{SNR}$), the sensitivity analysis carried out in the experiments section is welcome.

**Weaknesses:**

The paper does not have major weaknesses.

Besides, as a suggestion, since your experiments assess the convergence speed as well as the quality of the sub-optimality gaps yielded by the solvers, you can use the Benchopt framework for more impactful figures: https://github.com/benchopt/benchopt . For instance, figure 1 could be summarized into a one-row figure.

**Questions:**

- As a non-expert in the field of differential equation learning, it is unclear to me why $z_j \in \{0, 1\}$. Is it a generic formulation or does this integer constraint arise often in this field? A clarification in the intro would help the reader understand why it is necessary and why you compare yourselves to MIP solvers.

- How does OKRidge scale with respect to the number of features?

- "Our algorithms are written in Python, Any improvements we see over commercial MIP solvers, which are coded in C/C++, are solely due to our specialized algorithms" (l. 204): how can a Python routine (even very efficient) can beat an algorithm written in C/C++? Are you using a JIT compiler like Numba to compile critical parts of the code? If not, have you tried? It could significantly increase the runtime speed.

**Limitations:**

I see one unaddressed limitation to the method: while OKRidge scales easily to hundreds of thousands of samples, it does not seem to scale as well on the number of features. For instance, as stated in l.132, deriving the lower bound requires inverting a $p\times p$ matrix, which does not scale well for $p \gg 10000$.

---

> ### Author Rebuttal · Authors · 2023-08-09
>
> $\textbf{Use Benchopt framework to do plotting}$ Thank you for this suggestion!
> Looking at Benchopt, we find that none of the available benchmarks support non-relaxed sparsity constraints.
> This is a helpful library that could be useful for the broad ML community.
> We can cite this work during revision.
> Actually, when we extend this work to some other applications, this is a good benchmark library to build upon, and we can make a pull request on GitHub for this repository so that Benchopt includes a cardinality-constrained ridge regression benchmark.
>
> $\textbf{Why}$ $z_j \in \\{0, 1\\}$ $\textbf{?}$ $z_j$ is a binary variable to indicate whether we choose to select feature $j$ or not (i.e., $z_j = 1$ vs. $z_j = 0$).
> These binary variables are used to enforce the sparsity constraint.
> There are approaches other than the MIP approach to produce sparse models (such as the baselines in Figure 3 on page 8 of the main paper or Figure 9 on page 177 in the appendix), but they are not competitive.
> Getting to know the true dynamics is important for scientific discovery.
> This is why we focus on the MIP approach and try to come up with novel lower bound formulation and calculation.
>
> $\textbf{How scalable is OKRidge with respect to the number of features?}$
> In the submitted manuscript, the largest number of features we have tried is 7000.
> As the number of features increases, solving the linear system (Equation 23) will take longer.
> However, if you look at results in the main paper and appendix, we are already much faster than the commercial solver Gurobi with all existing MIP formulations.
> Hence, for our current code base, we enjoy either better or comparable scalability than commercial solver Gurobi.
> See below on our plan to make OKRidge more scalable.
> Note that licenses for Gurobi software are extremely expensive.
>
> $\textbf{How to make OKRidge more scalable with respect to the number of features?}$
> As we have mentioned above, solving the linear system (Equation 23) will take longer time as the feature dimension increases.
> One promising solution to further stretch the scalability of our method is to apply Joel A. Tropp's randomized linear algebra method to solve the linear system.
> Note, we are already making a Cholesky factorization at the beginning of the ADMM to save some computational time according to Stephen P. Boyd's book.
> We hope randomized linear algebra could give us another boost of performance.
>
> $\textbf{Why can we outperform an algorithm implemented in C/C++?}$
> The most important reason is that our proposed algorithm exploits the special mathematical structure (our fast solve and ADMM methods) of this problem. In contrast, the commercial solver relies on a generic MIP algorithm, which is proprietary and closed source.
> The paper finds a new way of calculating the lower bound in order to prune the search space more effectively, specifically for sparse ridge regression.
> We use numpy, which uses BLAS and LAPACK to do the heavy linear algebra computation.
> Python is slow when there are loops (our ADMM method is an iterative method and involves loops).
> Python is also slow because many of its functions have wrappers and needs to do lots of checking for types and bounds.
> However, because our algorithm is efficient, these two disadvantages are reduced to a minimal negative effect.
> Of course, coding everything in C++ or substituting part of the code in Numba (which we haven't tried) could make our code go even faster.
> This current work is to first demonstrate that our idea works even without resorting to C++ or Numba compiled code.

---

> > ### Comment · Reviewer_Nrfj · 2023-08-16
> >
> > I would like to thank the authors for their detailed answer. Keeping my grade unchanged as it is already positive.

---

### Official Review · Reviewer_kX5s · 2023-07-10

**Soundness:** 2 fair
**Presentation:** 2 fair
**Contribution:** 2 fair
**Rating:** 6
**Confidence:** 2

**Summary:**

The paper proposes a lower bound calculation method and warm start method for sparse ridge regression via a branch-and-bound approach. Starting from a saddle point formulation, the lower bound on the original objective can be given by either a linear system or ADMM problem which can be efficiently solved. The warm-start is given by the beam search method. Experiments are provided to demonstrate the better efficiency of the proposed approach over baselines.

**Strengths:**

1. propose a scalable method for sparse ridge regression.
2. demonstrate the feasibility of the proposed method in practice.

**Weaknesses:**

1. The main contribution lies in efficient certifiable lower bound which however is just part of the whole algorithm. It would be better if the whole algorithm can be explicitly given in the main text.
2. There is no analysis that gives the complexity of the whole algorithm to achieve a acceptable optimality gap.

**Questions:**

See above

**Limitations:**

Limitations are addressed in the paper.

---

> ### Author Rebuttal · Authors · 2023-08-09
>
> $\textbf{Whole algorithm should be explicitly given in the main text}$ Detailed algorithms are already given in the original manuscript.
> Please see Algorithm 1-8 on page 34-40 in Appendix E.
> The whole algorithm doesn't fit in 9 pages in the main paper, but we tried to organize it well in the Appendix.
> NeurIPS often gives an extra page that will help.
> During revision, we can put the overall algorithm (Algorithm 1) in the main paper.
>
> $\textbf{No complexity analysis of the algorithm to achieve an acceptable optimality gap}$
> Since the problem is NP-hard, one cannot easily talk about convergence rate to the optimal solution in the same way as in the context of convex optimization, such as $O(1/k)$ or $O(1/k^2)$ convergence rate.
> We have already shown that most existing heuristic methods for producing sparse linear models failed to find the correct features (see Figure 9 on page 177 of the Appendix).
> These heuristic methods may enjoy fast convergence rate to a locally stationary solution, but they cannot be used reliably in discovering true dynamical systems.
> Therefore, using branch-and-bound to certify global optimality is really needed in this context.
>
> The speed of BnB depends heavily on how efficiently we find the upper bound and lower bound.
> The branch-and-bound framework focuses heavily on intelligent enumeration.
> If the lower bound is tight, we can prune most of the search space, which can greatly increase our certification process.
> Computing a tight lower bound efficiently is a major focus of our work.

---

> > ### Comment · Reviewer_kX5s · 2023-08-16
> > **Comment by kX5s**
> >
> > Thanks for the clarification. I have adjusted my score.

---

### Official Review · Reviewer_ono9 · 2023-07-12

**Soundness:** 4 excellent
**Presentation:** 4 excellent
**Contribution:** 4 excellent
**Rating:** 7
**Confidence:** 4

**Summary:**

This paper proposes a method, namely OKRidge, to solve spare ridge regression problems and identify terms of differential equations of nonlinear dynamical systems. In particular, OKRidge is formulated as a branch-and-bound algorithm, and this paper provides a lower bound calculation.

**Strengths:**

$\mathbf{1}.$ The proposed method is novel, mainly on the formulation of the optimization problem as prescribed by Eq (12), as well as the approach to solve the problem.

$\mathbf{2}.$ The empirical analysis is thorough and results are promising compared with various solvers/algorithms, including commercial solver, e.g., GuRoBi. The experiments very well supports the proposal of the algorithm with adequate evidence.

$\mathbf{3}.$ The proposed method is very complete, which covers a major branch-and-bound algorithm with novel lower bound approach and a heuristic to search in a tree.

**Weaknesses:**

$\mathbf{1}.$ Despite the novel formulation of the optimization problem (as well as the lower bound derivation), it might be still combination of known approach to derive the package of approach, including BnB, dynamic programming and heuristics.

**Questions:**

$\mathbf{1}.$ Please see $\mathbf{1}.$ in $\mathbf{Weakness}$ section.

**Limitations:**

This paper includes a limitation statement, and I do not have any concerns regarding a potential negative societal impact.

---

> ### Author Rebuttal · Authors · 2023-08-09
>
> $\textbf{Our method is not novel due to the use of branch-and-bound}$ Branch-and-bound is the foundation of all commercial MIP solvers (one variant is branch-and-cut, but it is still based on top of branch-and-bound).
> Without using branch-and-bound, we cannot certify optimality for this NP-hard problem.
> We hope the reviewer sees the effectiveness of our novel lower bound formulation and calculation and not use the fact of us relying upon branch-and-bound as a reason for rejection.
>  (One wouldn't, for instance, reject a paper with a novel neural architecture because neural networks have been used before.)
>
> $\textbf{Novelty of dynamic programming}$ Dynamic programming is a general method.
> However, we believe dynamic programming has not been applied in combination of beam search in a branch-and-bound framework for sparse ridge regression.

---

### Official Review · Reviewer_X2ir · 2023-07-26

**Soundness:** 4 excellent
**Presentation:** 4 excellent
**Contribution:** 4 excellent
**Rating:** 8
**Confidence:** 3

**Summary:**

This paper proposes a novel algorithm to solve sparse ridge regression problems -- which are standard ridge regression problems with an additional l0 sparsity constraint (so this problem is NP Hard in general). The algorithm is a branch and bound based approach that uses a novel, tighter, lower bound computed by solving a saddle point reformulation of the problem using ADMM. The reformulation is done using a perspective transform approach along with a reparameterization trick. Additionally, beam search over the chosen non-zero coordinates is used as a heuristic to obtain better upper bounds while expanding the branch and bound tree. A provable guarantee on the optimality gap of the solution obtained from this algorithm is also provided. Experiments on synthetic data as well as data from dynamical systems like the Hopf bifurcation, the Lorenz system and a non-linear magnetohydrodynamical model demonstrate the efficacy of this algorithm compared to existing open source and commericial benchmarks.

**Strengths:**

1. The paper is well very written and clear.
2. I greatly appreciate the detailed and helpful annotations in each step of the proofs as well as the algorithms.
3. The main result is obtained by combining a novel reformulation of the sparse ridge regression problem with ADMM to obtain a tighter lower bound that can be used in branch and bound.
4. Thorough experiments were conducted to study and analyze the performance of the described algorithm.


**Weaknesses:**

1. Perhaps the authors could include experiments on more applications that arise from challenging physical systems beyond the three presented in the paper, since the beam search alone was sufficient for the synthetic problem.
2. The improvement over MIOSR in the case of the sparse differential equations like Lorenz, Hopf and MHD is not very significant (Figure 3), however it still matches the performance of using a proprietary solver.

**Questions:**

1. How does this method compare to relaxing the sparsity constraint to a lasso regularization, and approximating a sparse solution from the solution of the lasso?
2. Have the authors conducted experiments using the simpler (but not as tight) lower bound obtained by solving the ridge regression linear system? If so, were there significant differences between this and the baseline? Further, are there any theoretical guarantees that can be shown for this simpler bound?

**Limitations:**

Yes, the authors addressed limitations of OKRidge in the case of smaller feature dimensions.

---

> ### Author Rebuttal · Authors · 2023-08-09
>
> $\textbf{Is beam search sufficient}$ Please see Figure 2 in our rebuttal pdf.
> For the MHD system, beam search alone is not sufficient for discovering the correct dynamical systems.
> Thus, it is necessary to use BnB to get the optimal solution in this high-dimensional and highly correlated setting.
>
> $\textbf{Improvement over MIOSR is not significant}$ If we look at the training and test losses, we agree with you sometimes the improvement over MIOSR is not significant.
> However, if we look at the metric of the true positive rate, then our method is doing better than MIOSR.
> True positive rate measures how often we recover the true dynamics (true features).
> Even though MIOSR might achieve a small training loss, the loss is not optimal, which results in MIOSR not discovering the true dynamics.
> In practice, this means MIOSR cannot help us discover the true physics from the data.
> (Again, note that MIOSR has a major disadvantage of being proprietary software with an expensive license.)
>
> $\textbf{Results on Lasso}$ Please see Figure 2 in our rebuttal pdf.
> Lasso does not do well in the application of discovering sparse differential equations because the features are highly correlated.
> The poor performance of Lasso has been discussed in previous SINDy literature: 1. "A unified sparse optimization framework to learn parsimonious physics-informed models from data" (Section 2.1) by Kathleen Champion, Peng Zheng, Aleksandr Y. Aravkin, Steven L. Brunton, J. Nathan Kutz. 2. "Benchmarking sparse system identification with low-dimensional chaos." by Alan A. Kaptanoglu, Lanyue Zhang, Zachary G. Nicolaou, Urban Fasel, and Steven L. Brunton.
>
> $\textbf{Comparison between fast solve and ADMM}$ Please see Figure 1 in our rebuttal pdf.
> The results show that using the simpler method (fast solve) cannot calculate a tight lower bound, which results in longer running time and larger optimality gap.
> Also, we would like to clarify one point in our OKRidge framework: we use fast solve and ADMM together.
> If you look at Algorithm 1 on page 34 in the Appendix, we run ADMM to tighten the lower bound if the fast solve cannot help us prune the current node.
>
> $\textbf{Theory behind the simpler method}$ We are not very sure what you mean by "theoretical guarantee". In the main paper, we made some mathematical connections by showing that the simpler method of calculating the lower bound is actually trying to maximize another lower bound of Problem 12 (See Theorem 3.2).
> The simpler method can also be intuitively understood from the perspective of strong convexity (See Line 122-134).
> Does this answer your question?
> If not, could you please clarify?

---

> > ### Comment · Reviewer_X2ir · 2023-08-18
> >
> > Thank you for your response, it addresses most of my questions. Regarding the theoretical guarantee, I meant if you could should a provable guarantee on the optimality gap using only the fast solver instead of the ADMM solver (similar to Theorem 3.4). However, this is just out of curiosity and does not really affect the merits of your main results, so I will maintain my positive score.

---

### Official Review · Reviewer_GTWb · 2023-07-26

**Soundness:** 4 excellent
**Presentation:** 4 excellent
**Contribution:** 3 good
**Rating:** 8
**Confidence:** 3

**Summary:**

The authors propose OKRidge, a fast algorithm to solve k-sparse ridge regression problems with provable optimality.
The proposed method is a customized branch-and-bound approach, leveraging the specific structure of the ridge regression problem in contrast to black-box solvers such as Gurobi.
Namely, OKRidge first uses a new saddle point formulation specifically for efficiently computing lower bounds (either "roughly" by solving a rather simple linear system of more tightly with an ADMM scheme.
Second, a beam search heuristic is proposed to find a near-optimal solution, with the global BnB structure in mind. The near-optimality of such solutions is provably guaranteed.
Third, relatively straightforward branching and queuing strategies are also proposed to "complete the package", building a full custom solver.
Last, the authors benchmark OKRidge against a wide variety of other approaches, not only limited to MIP formulations. Both performance of the solutions, and compute time are evaluated on synthetic benchmarks and on sparse identification of nonlinear dynamical systems.

**Strengths:**

I think that the significance of this work lies in the careful and smart design and integration of multiple components, building into what seems to be a very efficient custom solver for a relatively broad class of problems.
While each of this components, taken individually might not be ground-breaking, the method as a whole seems to build into a tool that has important significance.
Although I am mostly an outsider and may very well miss some important relevant references, the authors are doing a very good job at placing their contribution in the context of a very abundant existing literature.
The clarity of the paper is also to be praised. The appendices, although extremely voluminous (in part due to the hundreds of tables reported from the experiments), are completely self-contained.
The proofs of the different theorems read well and are somewhat insightful. The Figures and algorithms summarized in Appendix E are also very helpful to bring all the pieces together.
For the quality, although I am not familiar with large parts of the work, I was not able to find mistakes in the proofs and derivations. To be completely fair (maybe due to the very good quality of the presentation), some derivations can be a bit tedious, but seem relatively straightforward (which is not meant to downplay the significance of the contributions in any way).

**Weaknesses:**

As stated in the previous section, it could be argued that none of the contributions, taken in isolation are "novel enough". But I don't believe that would be a fair argument.
On the other hand, while the authors did a great job at presenting their method clearly and carefully, I find that it could have been helpful or interesting to include a little more comments on their results.
For instance, it is great that provable guarantees are derived for the heuristic solution found by their beam-search (section 3.2.1) but it could have been nice to understand a bit more how tight those bounds are (or if they are insightful at all).
More importantly, although the experimental results look good, do the authors consider this heuristic as an important limitation? Are there problems (or datasets) for which the sub-optimality of the beam-search is problematic?
Also, it would be great to understand, in practice, where the compute time is being spent as it might be a good way to understand where the computational bottlenecks of OKRidge are and where there is potential room for further improvement.

**Questions:**

I already included a few questions in the previous section.
To clarify, I have a few more questions:
1) How do you estimate \lambda_min and \lambda_max, to determine \rho for ADMM? Is it critical to estimate it carefully?
2) It might be a silly question but do you have any insight on how to pick the optimality gap \epsilon in practice? Would it make sense to pick it adaptatively?
3) What about the scenario where you don't "know" k a priori but would like to find which level of sparsity offers a good trade-off? Can OKRidge be easily adapted to solve the problem for different levels and not have to rebuild the solution from scratch for every k?
4) In the end, while learning dynamical systems is an important problem of its own, OKRidge should be useful for other problems which can be cast as a sparse ridge regression. Why so much emphasis on dynamical systems?

**Limitations:**

Some limitations of OKRidge are clearly identified and pinpointed towards the end of the manuscript.
Potential negative societal impacts were not discussed, but I don't foresee any.

---

> ### Author Rebuttal · Authors · 2023-08-09
>
> $\textbf{Tightness of the bound for the beam search solution}$
> The bound is not as tight as one might desire because the bound does not incorporate the beam size.
> We spent lots of time on improving the bound but made no breakthrough.
> Therefore, the bound in our theorem has potential for improvement by incorporating the beam size $B$.
> However, at this point, we are unsure how to achieve this enhancement (and it remains an open question whether it is feasible to incorporate the beam size into the bound).
>
> $\textbf{When might beam search fail}$
> Beam search is a heuristic method and is not guaranteed to produce optimal solutions.
> This is why we need the branch-and-bound framework.
> Beam search is already doing very well in the regime of highly-correlated features as shown in the synthetic benchmarks and differential experiments.
> However, since you ask when beam search might fail, we conjecture that beam search might fail in finding the optimal solution in the regime of super-correlated features.
> As a pathological example, suppose we have lots of features, which can be divided into different groups based on their correlation levels.
> Inside each group, features are super-correlated and are almost the same and only differed in several samples.
> In this case, beam search may have a hard time because there exist many near-optimal solutions.
> (This is also the case where MIOSR will be extremely slow.)
> For most practical applications, beam search should work well.
> Note, we have already compared with many existing heuristics in Figure 9 (page 177) of the Appendix.
>
> $\textbf{Where to improve}$
> Beam search is doing OK with respect to time according to our profiling.
> Currently, lower bound calculation through ADMM is still taking more time than beam search.
> Since ADMM involves repeatedly solving linear systems (See Equation 23), we are thinking about using Joel A. Tropp's randomized linear algebra method to solve the large-scale linear system approximately to save time.
> Note, we are already making a Cholesky factorization at the beginning of the ADMM to save some computational time according to Stephen P. Boyd's book.
> We hope randomized linear algebra could give us another boost of performance.
>
> $\lambda_{\text{min}}$  $\textbf{and}$  $\lambda_{\text{max}}$  $\textbf{estimation}$
> We use numpy to calculate all eigenvalues and pick $\lambda_{\text{min}}$ and $\lambda_{\text{max}}$ afterwards.
> However, this eigenvalue calculation is only performed at the root node in the BnB tree.
> After we calculate the step size for ADMM at the root node, we use this step size for ADMM for all the nodes in the BnB tree.
> To the rest of the nodes, such a step size is an approximation, but it works well in all our experiments.
> For the smallest eigenvalue used in $Q_{\lambda}$, please look at Appendix E7 on the Interlacing Eigenvalue Property.
>
> $\textbf{How to choose tolerance level $\epsilon$?}$
> It is hard to provide a definitive answer or guidance.
> Any reasonable small number (such as $10^{-4}$ used in all our experiments; this is also the default value used in Gurobi) is probably fine.
>
> $\textbf{What to do when we don't know $k$ a priori?}$
> In our differential equation experiments, $k$ is not known a priori and is treated as a hyperparameter chosen via cross-validation. We opt to use the hyperparameters that minimize $AIC_c$ developed specifically for sparse regression. Please see the reference [46] and details in Appendix F.3. This is the approach we recommend.
>
> $\textbf{Why emphasis on dynamical systems?}$
> You are right that our method can be applied broadly to other applications involving sparse ridge regression.
> The reason, as we explained in the introduction, is that (Line 23) selection of the correct features is key to discovering the true dynamics.
> Empirically, due to the highly-correlated features formed by high-order polynomials, most existing heuristic methods failed to find the correct features (see Figure 9 on page 177 of the Appendix).
> Therefore, finding optimal sparsity-constrained solution is really needed in this context and may have an impact in real scientific discovery.
> Besides, we know the true dynamics so that it is easy to verify the correctness of the solution to make sure our code base is solid.
> Furthermore, sparse ridge regression is used extensively in the SINDy literature, making it a solid choice for application.
> Nevertheless, we plan to extend this work and branch out to other applications.

---

> > ### Comment · Reviewer_GTWb · 2023-08-11
> > **My rating of the paper won't change as it was already very high but thanks a lot for the detailed answers.**
> >
> > For the tightness of the beam search solution bound, I had not even realized that $B$ was not even incorporated. It looks like a difficult problem. Maybe it would be worth mentioning it in the discussion as future work as it seems important and probably relevant to other bounds / problems too. Same comment could hold for the comment on the use of randomized linear algebra tools for speed-up.
> >
> > For my last comment on the emphasis on dynamical systems, I think the motivation was already clearly stated. I guess that my remark eluded more to the points that two other reviewers made, and I agree with them, that apart from potential "strategical reasons" (i.e. make sure that you reach the dynamical systems crowd), not much justifies to have it in the title as nothing in the algorithm or its analysis leverages anything that is specific to it. I agree that it should be removed. (And also, to be clear, I'm not going to leave the decision to you and it won't affect my evaluation / rating of the submission.)

---

### Official Review · Reviewer_MGha · 2023-07-26

**Soundness:** 3 good
**Presentation:** 4 excellent
**Contribution:** 3 good
**Rating:** 7
**Confidence:** 4

**Summary:**

In this manuscript the authors consider the ridge regression with a hard sparsity constraints. Since the problem is posed in the fixed design setting, the paper is on the optimization theory, while the main motivation for its study authors find in the applications related to data-driven methods for dynamical systems. Using the work of Atamturk and Gomez 2020 on max-min relaxation of the original problem, novel lower bounds are developed (one in closed form and the other using ADMM to improve the tightness). These results coupled with Beam-Search of Wiseman and Rush 2016 are used within Bound-and Branch framework to propose an algorithm with a theoretical bound on the optimality gap. Experiments are run on synthetic data to compare with SOTA methods for the studied problem, as well as on datasets from Brunton, Proctor and Kutz 2016 on toy problems in dynamical systems to showcase the application in the sparse identification of (continuous-time deterministic) equations of motion from data under the (strong) assumptions that the finite-dimensional basis of a function space is know and that dataset contains states and their time derivatives.

**Strengths:**

The paper is very well written, apart of issues stated below. All the important concepts are properly introduced in the main body and extensively elaborated upon in the appendix. Main contribution well presented and the novelty of the new ideas w.r.t. existing work is made very clear. Up to my knowledge, relevant references are cited. The theoretical results are correct, except minor issues explained below. Numerical experiments are very detailed.

**Weaknesses:**

In the following I give the main weaknesses of this paper:

 __(A) Title vs content:__ The paper is content wise purely work on optimization. Namely, the link to ML and to learning of dynamical systems is only through the problem which is an empirical risk minimization for linear model with $\ell_2$ regularization and $\ell_0$ constraints. Hence, there is no really any learning of dynamical systems as the title suggests. Therefore, since I find that the paper is in its current form of interest to ML community, I propose the authors to remove from the title “for Learning Dynamical Systems”.  If not, then much more than elaborated discussion needs to be done to justify the title. In particular:

1. Assumptions under which the system is learned from data providing explaining when they are met in practice. My opinion is that relying just on SINDy approach is very limited since this aspect is quite vague.

2. Comparison with other SOTA methods to learn dynamical systems from data is needed (meaning those solving the true problem and not an empirical one) with the discussion on benefits and limitations of the current proposal. Also some insights in how statistical learning theory of the proposed method might be developed.

3. At least one experiment on the performance of the method on the real dataset of challenging dynamical system (unknown non-polynomial nonlinearities in ODE).


__(B) Error in Thm. 3.1.:__ When $Q_\lambda$ is singular, i.e. when $\lambda = \lambda_{\min}(X^\top X)$, not every $c$ is in the range of $X^\top$, and, hence, reparametrization doesn’t hold, yielding that (12) is not equivalent to (10). Consequently, when $\sigma_{\min}(X)=0$, e.g. when number of features exceeds number of samples,  Thm. 3.1. on which all else is based doesn’t hold. This aspect needs to be checked and clarified.

__(C) Notion of optimality:__ In the paper, starting from the title and abstract, the word _optimal_ is used. I believe that many of such uses are not entirely clear. To me it seams that sometimes it means just feasible k-sparse solution. Although upper bound and lower bound guarantees are shown, in the paper I do not find theoretical guarantee for the convergence of the proposed method to the optimal solution stated. Can the authors, please, elaborate on this?


To conclude, the current evaluation score I have given reflect the aforementioned issues. Depending on the authors reply, I am willing to reconsider it.

**Questions:**

Please elaborate on the claim in the line 57/58. What is the meaning of outperforms? Seams that on the validation set it performs similarly as MIOSR.

**Limitations:**

Apart of currently discussed limitations, if the title is kept as is, the authors need more discussion as noted above.

---

> ### Author Rebuttal · Authors · 2023-08-09
>
> $\textbf{Review Correction}$ There is one minor correction we want to point out with regards to your summary section.
> In the differential equation experiments, only states are given in the dataset, not the time derivatives.
> Derivatives were approximated using a smoothing finite difference with a window size of 9 points.
> The derivatives were calculated automatically by the PySINDy library.
> You can find this information in Appendix F "More Experimental Setup Details".
>
> $\textbf{Title vs content}$
> We will listen to your suggestion and remove from the title "for Learning Dynamical Systems".
> We still plan to use applications in learning dynamical systems as a strong motivation for this paper and also use the experimental results to demonstrate the effectiveness and potential scientific impact of our method.
> The reason, as we explained in the introduction, is that (Line 23) selection of the correct features is key to discovering the true dynamics.
> Empirically, due to the highly-correlated features formed by high-order polynomials, most existing heuristic methods failed to find the correct features (see Figure 9 on page 177 of the Appendix).
> Therefore, finding optimal sparsity-constrained solution is really needed in this context.
> Furthermore, sparse ridge regression is used extensively in the SINDy literature, making it a solid choice for application.
>
> We agree with you that some discussion should be given upon the assumption of the SINDy approach.
> During the revision, we can add that the SINDy approach assumes that the true dynamics exist in the finite-dimensional candidate functions/features.
> As the reviewer commented, our work focuses on the optimization side of sparse ridge regression.
> One benefit of OKRidge is that it can be used to push the limit of the SINDy approach.
> Comparison with methods that do not go under the SINDy approach (sparse linear regression) does not align with our central focus and requires a complete new work in a different direction.
>
> $\textbf{Theorem 3.1}$
> Please look at our answer in the general response.
> We believe even when $Q_{\lambda}$ is singular, Problem 10 and Problem 12 are still equivalent in the sense that they have the same optimal values.
> The insight is that the optimal solution $c$ for Problem 10 must be reparametrized by some $\beta$ with Equation (11) $c = \frac{2}{\lambda_2 + \lambda} (X^T y - Q_{\lambda} \beta)$.
> Please read our proof in the general response and see if you agree with us.
> If the proof looks ok, then we will insert this lemma into our main paper during revision.
>
> $\textbf{Notion of optimality}$
> We apologize for the potential confusion. There is a difference between 0 optimality gap in a mathematical sense and in a more practical and engineering sense.
> Strictly speaking, your second interpretation is correct.
> Since the problem is nonconvex and NP-hard, the solution is only certifiably optimal when the upper bound equals to the lower bound.
> In practice, however, commercial solvers will terminate the algorithm once the optimality gap is within some tolerance level (like the one we used $10^{-4}$ in all experiments for fair comparison; this is also the default value in Gurobi).
> We loosely interpret such solutions as optimal solutions when a very small tolerance level is used.
> Another case to consider is that when the problem is too hard, the optimality gap may not go below the tolerance level within a reasonable amount of time.
> In this case, we have to stop the algorithm after reaching some pre-defined time limit.
> The final solution obtained in this scenario can only be called a feasible solution but we should not call this solution optimal.
> Maybe the reviewer has the second scenario in mind.
> We will revise the manuscript to give more background on practical MIP terminology and stopping criteria in the appendix to clarify the notion of optimality and stopping rules.
> Our contribution is that we have developed a customized branch-and-bound framework that either certifies the optimal solution much faster or provides a much smaller or comparable optimality gap.
> These two contributions (faster certification and smaller optimality gap) are the standard judging criteria of an algorithm in the MIP literature.
>
> $\textbf{Convergence of BnB}$
> Since the problem is NP-hard, we cannot talk about convergence rate to the optimal solution in the same way as in the context of convex optimization, such as $O(1/k)$ or $O(1/k^2)$ convergence rate.
> The speed of the algorithm depends heavily on how efficiently we find the upper bound and lower bound.
> However, it is not hard to see that the generic branch-and-bound framework will terminate in a finite number of nodes (although in the worst case, the number of nodes will grow exponentially with respect to the problem size).
> A naive approach to prove this is through exhaustive enumeration.
> In this case, we need to consider all $\binom{p}{k}$ feature combinations to select the best support.
> Therefore, $\binom{p}{k}$ is the maximum number of nodes we need to process.
> The branch-and-bound framework focuses heavily on intelligent enumeration.
> If the lower bound is tight, we can prune most of the search space, which can greatly speed up our certification process.
> Computing a tight lower bound efficiently is a major focus of our work.
>
> $\textbf{Meaning of``outperform''}$
> In Figure 3, if we look at the training and test losses, we agree with you sometimes the improvement over MIOSR is not significant.
> However, if we look at the metric of the true positive rate, then our method is doing better than MIOSR.
> True positive rate measures how often we recover the true dynamics (true features).
> Even though MIOSR might achieve a small training/test loss, the loss is still not optimal, which results in MIOSR not discovering the true dynamics.
> In practice, this means OKRidge does a better job in discovering the true physics than MIOSR.
> Note also that MIOSR requires an expensive license to access whereas ours is free.

---

> > ### Comment · Reviewer_MGha · 2023-08-15
> > **Rising my score**
> >
> > I thank the authors for their detailed replies. After reading other reviews as well as the authors replies to them, I am confident that the the promised steps of the revision will be implemented appropriately and that the paper’s quality will be improved. Hence, I rise my score to accepted.

---

### Official Review · Reviewer_fUoX · 2023-07-27

**Soundness:** 2 fair
**Presentation:** 2 fair
**Contribution:** 4 excellent
**Rating:** 7
**Confidence:** 2

**Summary:**

The authors propose a new framework for computing the k-sparse ridge regression. Specifically, the framework considers the Lagrangian dual and relaxes the lower bound to an ADMM problem with an isotonic proximal operator, which has a linear time solution. Combined with beam search and branch-and-bound, the authors demonstrate the framework's effectiveness in the experiment. Further, they claim to prove a tighter approximation bound based on the beam-search result.

**Strengths:**

The paper's main contribution is the relation between the k-sparse problem with an isotonic proximal operator, which can be computed in linear time ($O(n log k)$ for sorting the coefficients.) I believe this technique is new and may have a significant impact on other works with k-sparse constraints. Actually, the framework can be generalized to any strongly convex and Lipschitz C^1 optimization with k-sparse constraints. The proposed Lagrangian dual lower-bound and isolation of the binary variables only depends on the k-sparse constraints, not the ridge regression itself. In addition, the proofs are clearly written with annotations on every inequality, which helps a lot in verification.

**Weaknesses:**

Despite the proofs being clearly annotated, there are still major bugs in the main theorems. However, I think these bugs may be corrected by weakening the theoretical results, and these errors may not impact the practical value of the work. Specifically, the $L_{\text{ridge-}\lambda}^\text{saddle}$ problem should be a lower bound of equation (12), not an equivalent problem. To prove equivalence, we need to prove that the set of optimality conditions are the same, and the error can be easily checked by taking stationary condition on c in eq (10), which leads to the missing slackness condition $2\beta_j=c_jz_j$ which doesn't appear in the stationary condition of the new formulation in eq (12) unless other assumptions are made. And I think the subtle error in the derivation is that the proof of Theorem 3.1 is not rigorous, i.e., not every step in the L569 derivation is an equivalence relation. The equation (40) may not be one-to-one, and the step of taking $\beta=\gamma$ with additional constraint eq (40), reparameterization of $c$, and moving $\gamma$ from the minimum operator to the maximum operator may not be rigorous. There may be a way to prove it more rigorously, and in the worst case, the derivation becomes a relaxation, which is still helpful. However, we will need to see the more formal derivation first.

Second, there are also bugs in the authors' proof for the beam search in Theorem 3.4. In Section 3.2, the authors select the new support by the Gauss-Southwell rule of largest improvement. However, when combined with beam search, it is required to select the candidates with the lowest objective value, which is a conflict. As we see in Algorithm 1 and Algorithm 7 in the Appendix, the actual implementation generates the next step using the B largest improvements for each candidate and chooses the B best candidates with the best objective value, which differs from the proof in Theorem 3.4. On the other hand, the theorem and proofs on the beam search solution are independent to the number of candidates $B$ in the authors' proof, so there must be some error or hidden assumption (e.g., $B\geq p^k$) here.

However, I still consider this draft as a good paper and recommend it for acceptance. Just that these bugs need to be addressed, or the Theorems need to be weakened. However, please address these main issues in the rebuttal.

**Questions:**

1. The proposed method is an algorithm for general sparse ridge regression with experiments on the dynamic system. There is no other utilization of the property of the dynamic system here other than the data itself. So I don't think the title is appropriate.
2. Appendix L562: missing z.
3. Appendix L567: The page overflows, so part of the equations are hidden.
4. Appendix L588: The conclusion (Therefore $\gamma=$) is your assumption.

---

> ### Author Rebuttal · Authors · 2023-08-09
>
> $\textbf{Theorem 3.1}$
> We have answered this question in the general response.
> We give an alternative proof for the equivalence of the optimal values between Problem (10) and Problem (12) by showing the two-way inequalities.
> Let us know if you have further questions on Theorem 3.1.
>
> $\textbf{Beam Search Description}$
> You are right that we first propose to select the support by the Gauss-Southwell rule of largest improvement.
> However, if you keep reading to the next paragraph (Line 167-170), we pointed out the limitation of this approach and introduced beam search.
> We think the confusion arose due to the absence of a mathematical formula after Line 170, which might have led readers to focus solely on Equation (25).
> For revision, maybe we can insert after Line 170 an extra sentence and another formula like the one below:
>
> For each solution $\beta$ from the previous stage of support expansion, we select $B$ new features as potential support expansion candidates
>
> \begin{align*}
>     j^* \in \arg \text{BottomB}_j   \left( \min\_\alpha  L\_{\text{ridge}} ( \beta + \alpha e_j ) \right),
> \end{align*}
>
> where $j^* \in \arg \text{BottomB}_j$ means $j^*$ belongs to the set of solutions whose loss is one of the B smallest losses.
> Afterwards, we finetune the solution on the newly expanded support and choose the top B solutions for the next stage of support expansion.
>
> $\textbf{Theorem 3.4}$
> We apologize for the confusion.
> Theorem 3.4 indeed provides a bound for the beam search solution.
> We think the confusion was caused by the sentence inside the parentheses on Line 187 ("starting from 0 and adding a variable k times to get support k").
> We were trying to describe beam search in a simplified way.
> However, you are right in that this description does not capture the full picture of beam search.
> Adding one variable at a time is part of the beam search algorithm, but we also need to select the best solution after each stage of support expansion.
> It is possible that the best solutions of two neighboring stages could have supports that differ by 2 features instead of 1 feature.
> Therefore, the simplified description inside the parentheses is not precise.
> We will delete this sentence during revision.
>
> This has no impact on the correctness of the theorem for the beam search algorithm.
> To show you that we are indeed proving for the beam search algorithm, please look at Line 661-664 and Equation (52) in Appendix 52.
> The crucial argument in that proof is that the decrease of loss between best solutions of two successive stages (stage $t$ and stage $t+1$) of beam search is greater than that of minimizing a single coordinate on the best solution of the early stage (stage $t$).
>
> With regards to the question on beam size, you are right that our bound does not involve the beam size.
> We spent lots of time trying to incorporate the beam size into the formula but made no breakthrough.
> Although the beam size is not part of the theoretical bound, we do observe empirically that larger beam size leads to better solution quality.
> The reason we are able to achieve a tighter theoretical bound than [29] is largely due to step 2 (Line 672-684) in our proof, where we show we can actually bound the term using $M_1$ instead of $M_{2k}$.
> To summarize, the bound in our theorem has potential for improvement by incorporating the beam size $B$.
> However, at this point, we are unsure how to achieve this enhancement (and it remains an open question whether it is feasible to incorporate the beam size into the bound).
>
> $\textbf{Title Change}$
> We will listen to your suggestion and remove from the title "for Learning Dynamical Systems".
> We still plan to use applications in learning dynamical systems as a strong motivation for this paper and also use the experimental results to demonstrate the effectiveness and potential scientific impact of our method.
> The reason, as we explained in the introduction, is that (Line 23) selection of the correct features is key to discovering the true dynamics.
> Empirically, due to the highly-correlated features formed by high-order polynomials, most existing heuristic methods failed to find the correct features (see Figure 9 on page 177 of the Appendix).
> Therefore, finding optimal sparsity-constrained solution is really needed in this context.
> Furthermore, sparse ridge regression is used extensively in the SINDy literature, making it a solid choice for application.
>
> $\textbf{Appendix L562}$ Thank you, we will fix this typo: $(\beta_j c_j - \frac{c_j^2}{4}) \rightarrow (\beta_j c_j - \frac{c_j^2}{4}z_j)$
>
> $\textbf{Appendix L567}$ Thank you, we will make this more clear by rewriting "reparameterize $c$ with $\gamma$" $\rightarrow$ "reparameterize $c$ with $\gamma$ by having $c = \frac{2}{\lambda_2 + \lambda} [X^T y - (X^T X - \lambda I)\gamma]$".
>
> $\textbf{Appendix 588}$ Thank you, this is an accidental writing error. We will fix this by rewriting "we have $\hat{\gamma} = \arg\min_{\gamma} L_{\text{ridge}}(\gamma)$" $\rightarrow$ "we have $X^T y - Q_{\lambda} \hat{\gamma} = (\lambda_2 + \lambda) \hat{\gamma}$ when $\hat{\gamma} = \arg\min_{\gamma} L_{\text{ridge}}(\gamma)$".

---

> > ### Comment · Reviewer_fUoX · 2023-08-16
> >
> > The reviewer is okay with the explanation and will keep the rating.

---

### Author Rebuttal · Authors · 2023-08-09

First of all, we just want to thank the reviewers for their thoughtful comments.
We will use this general response to address an important theoretical question and will address specific issues to each reviewer.
Requested experimental results on 1) comparison between fast solve and ADMM and 2) comparison between Lasso, beam search, and OKRidge are also included as a PDF document as part of this general response.

$\textbf{Correctness of Theorem 3.1}$
We still believe that Theorem 3.1 should hold.

Reviewer MGha asks a very good question: what will happen if $Q_{\lambda}$ is singular?
If $Q_{\lambda}$ is singular, our reparametrization trick could miss some subspace and therefore Problem (12) may become a relaxation of Problem (10), as Reviewer fUoX worries.
Fortunately, during revision, we can insert the following lemma: for an optimal solution $c^*$ to Problem (10), there exists some $\beta^*$ so that $c^*=\frac{2}{\lambda_2 + \lambda}(X^Ty - Q_{\lambda}\beta^*)$.
This lemma implies that the subspace we miss will not prevent Problem (12) from achieving the same optimal value as Problem (10).

We prove this lemma by the method of contradiction.
Suppose the optimal solution $c$ to Problem (10) cannot be written by Equation (11).
Through the rank-nullity theorem in linear algebra, we can find $d_1 \in col(Q_{\lambda})$ and $d_2 \in ker(Q_{\lambda}), d_2 \neq 0$ so that $c = \frac{2}{\lambda_2 + \lambda}(X^Ty-d_1 + d_2)$, where $col(\cdot)$ denotes the column space of a matrix and $ker(\cdot)$ denotes the kernel of a matrix.
If we let $\beta = \alpha d_2$ for some real value $\alpha$, then the inner optimization of Problem (10) (ignore the last term involving $z_j$'s for now since $\beta$ and $z$ are separable in the objective function) becomes
\begin{align*}
    L_{\text{ridge}-\lambda}^{\text{Fenchel}}(\beta, z, c) &= \beta^T Q_{\lambda} \beta - ((\lambda_2+\lambda)c - 2 X^Ty)^T \beta \\\\
    &= \beta^T Q_{\lambda} \beta - ((\lambda_2+\lambda)\frac{2}{\lambda_2 + \lambda} (X^T y-d_1 +d_2) - 2 X^Ty)^T \beta\\\\
    &= \beta^T Q_{\lambda} \beta - (2 (X^T y-d_1 +d_2) - 2 X^Ty)^T \beta \\\\
    &= \beta^T Q_{\lambda} \beta + 2 (d_1 -d_2)^T \beta \\\\
    &= \beta^T Q_{\lambda} \beta + 2 d_1^T \beta -2 d_2^T \beta.
\end{align*}
Because $\beta =\alpha d_2$, we have $\beta \in ker(Q_{\lambda})$ and $\beta \perp d_1$.
Thus, the first two terms become $0$, and we are left with $L_{\text{ridge}-\lambda}^{\text{Fenchel}} = - 2d_2^T \beta = - 2\alpha \lVert d_2\rVert_2^2$.
For the inner optimization of Problem (10), because we want to minimize over $\beta$, we can let $\alpha \rightarrow \infty$, and we will get $L_{\text{ridge}-\lambda}^{\text{Fenchel}} \rightarrow - \infty$.
This is obviously not the optimal value for Problem (10) (a simple counter-example is to let $c = \frac{2}{\lambda_2 + \lambda} X^T y$, and we can easily show that $\min_{\beta, z} L_{\text{ridge}-\lambda}^{\text{Fenchel}}(\beta, z, c)$ is finite).
Therefore, $d_2$ must be 0, and the optimal solution $c$ to Problem (10) must be represented by some $\beta$ through the equation $c=\frac{2}{\lambda_2 + \lambda}(X^Ty - Q_{\lambda}\beta)$.

Now that we have this lemma, to clear the questions of Reviewer Nrfj, we can prove Theorem 3.1 with an alternative approach.
We will show $\max_{c} \min_{\beta, z} L_{\text{ridge}-\lambda}^{\text{Fenchel}}(\beta, z, c) \leq \max_{\gamma} \min_{z} L_{\text{ridge}-\lambda}^{\text{saddle}}(\gamma, z)$ and $\max_{c} \min_{\beta, z} L_{\text{ridge}-\lambda}^{\text{Fenchel}}(\beta, z, c) \geq \max_{\gamma} \min_{z} L_{\text{ridge}-\lambda}^{\text{saddle}}(\gamma, z)$ (both under the constrains $\sum_{j=1}^p z_j \leq k, z_j \in [0, 1]$) and therefore establish the equivalence of optimal values.

For the first inequality, let $\beta^*, c^*$ be the optimal solution to Problem 10 with the relation $c^*=\frac{2}{\lambda_2 + \lambda}(X^Ty - Q_{\lambda}\beta^*)$.
Furthermore, let $\gamma^* = \beta^*$.
Then we have
\begin{align*}
    \max_{c} \min_{\beta, z} L_{\text{ridge}-\lambda}^{\text{Fenchel}}(\beta, z, c) = \min_{z} L_{\text{ridge}-\lambda}^{\text{Fenchel}}(\beta^*, z, c^*) = \min_{z} L_{\text{ridge}-\lambda}^{\text{saddle}}(\gamma^*, z) \leq \max_{\gamma} \min_{z} L_{\text{ridge}-\lambda}^{\text{saddle}}(\gamma, z)
\end{align*}
The derivation for the second equality roughly follows the derivation in Appendix C1 on page 18-19.

For the second inequality, let $\hat{\gamma}$ be the optimal solution to Problem 12.
Let $\hat{c} = \frac{2}{\lambda_2 + \lambda}(X^Ty - Q_{\lambda}\hat{\gamma})$
Furthermore, let $\hat{\beta} = \hat{\gamma}$ (notice that this $\hat{\beta}$ satisfies the optimality condition of Problem 10).
Then we have
\begin{align*}
    \max_{\gamma} \min_{z} L_{\text{ridge}-\lambda}^{\text{saddle}}(\gamma, z) &=
    \min_{z} L_{\text{ridge}-\lambda}^{\text{saddle}}(\hat{\gamma}, z) \\\\
    &= \min_{z} L_{\text{ridge}-\lambda}^{\text{Fenchel}}(\hat{\beta}, z, \hat{c}) \\\\
    &= \min_{\beta, z} L_{\text{ridge}-\lambda}^{\text{Fenchel}}(\beta, z, \hat{c}) \\\\
    &\leq \max_{c} \min_{\beta, z} L_{\text{ridge}-\lambda}^{\text{Fenchel}}(\beta, z, c)
\end{align*}
Similarly, the derivation for the second equality roughly follows the derivation (but this time in reverse order) in Appendix C1 on page 18-19.

Using these two inequalities, we can establish the equivalence of optimal values between Problem 10 and Problem 12, $\textit{i.e.}$, $\max_{c} \min_{\beta, z} L_{\text{ridge}-\lambda}^{\text{Fenchel}}(\beta, z, c) = \max_{\gamma} \min_{z} L_{\text{ridge}-\lambda}^{\text{saddle}}(\gamma, z)$.

---

### Decision · Program_Chairs · 2023-09-21

**Decision:**

Accept (spotlight)

**Comment:**

The authors develops a fast algorithm for solving k-sparse ridge regression problems. The proposed method is based on branch-and-bound approach and leverages the specific structure of the ridge regression problem. The paper is well written, the experiments are illuminating, and the ideas developed have many use cases in data analysis and scientific discovery.